

# Microclimatic and ecophysiological conditions experienced by epiphytic bryophytes in an Amazonian rain forest

Nina Löbs[1]*, David Walter[1], Cybelli G. G. Barbosa[1], Sebastian Brill[1], Gabriela R. Cerqueira[2], Marta de Oliveira Sá[2], Alessandro C. de Araújo[3], Leonardo R. de Oliveira[2], Florian Ditas[1], Daniel Moran-Zuloaga[1], Ana Paula Pires Florentino[1], Stefan Wolff[1], Ricardo H. M. Godoi[4], Jürgen Kesselmeier[1], Sylvia Mota de Oliveira[5], Meinrat O. Andreae[1,6], Christopher Pöhlker[1], Bettina Weber[1]*

1 Multiphase Chemistry and Biogeochemistry Departments, Max Planck Institute for Chemistry, Mainz, 55128, Germany

2 Large Scale Biosphere-Atmosphere Experiment in Amazonia (LBA), Instituto Nacional de Pesquisas da Amazonia (INPA), Manaus-AM, CEP 69067-375, Brazil

3 Empresa Brasileira de Pesquisa Agropecuária (EMBRAPA), Belém-PA, CEP 66095-100, Brazil

4 Environmental Engineering Department, Federal University of Parana, Curitiba, PR, Brazil

5 Biodiversity Discovery Group, Naturalis Biodiversity Center, Leiden, 2333 CR, The Netherlands

6 Scripps Institution of Oceanography, University of California San Diego, La Jolla, CA 92037, US

*Correspondence to: Nina Löbs (n.loebs@mpic.de) and Bettina Weber (b.weber@mpic.de)

To be submitted in journal *Biogeosciences, ATTO special issue*



**Abstract**. In the Amazonian rain forest, major parts of trees and shrubs are covered by epiphytic cryptogams of great taxonomic variety, but their relevance in biosphere-atmosphere exchange, climate processes, and nutrient cycling are largely unknown. As cryptogams are poikilohydric organisms, they are physiologically active only under moist conditions. Thus, information on their water content, as well as temperature and light conditions experienced by them are essential to analyzing their impact on local, regional, and even global biogeochemical processes.

In this study, we present data on the microclimatic and ecophysiological conditions of epiphytic bryophytes along a vertical gradient and combine these with mesoclimate data collected at the *Amazon Tall Tower Observatory* (*ATTO*) in the Amazonian rain forest between October 2014 and December 2016. While the monthly average mesoclimatic ambient light intensities above the canopy revealed only minor variations, the light intensities incident on the bryophytes showed different patterns at different heights, probably depending on individual shading by vegetation. At 1.5 m height, monthly average light intensities were similar throughout the year and individual values were extremely low, exceeding 5 µmol m$^{-2}$ s$^{-1}$ photosynthetic photon flux density only during 8 % of the time. Temperatures showed only minor variations throughout the year with higher values and larger height-dependent differences during the dry season. Water contents of bryophytes varied depending on precipitation and air humidity. Whereas bryophytes at higher levels were affected by frequent wetting and drying events, those close to the forest floor remained wet over longer time spans during the wet seasons. Based on estimates of the potential duration of net photosynthesis and dark respiration, our data suggest that water contents are decisive for overall physiological activity, and light intensities determine whether net photosynthesis or dark respiration occurs, whereas temperature variations are only of minor relevance in this environment. In general, bryophytes growing close to the forest floor are limited by light availability, while those growing in the canopy must withstand larger variations in microclimatic conditions, especially in the dry season. Measurements of $CO_2$ gas exchange are essential to elucidate their physiological activity patterns in greater detail.

## 1    Introduction

The Amazon rain forest covers 5 821 800 km² on the South American continent, thus forming the second largest terrestrial vegetation type after the boreal forests of the Taiga (Melack and Hess, 2010). Tropical rain forests are characterized by humid conditions, high temperatures, minor annual fluctuations of temperature, and an immense species diversity of flora and fauna. They have been described to play important roles in the water cycle as well as for carbon, nitrogen, and phosphor fluxes on regional and global scales (Andreae et al., 2015). Tropical rain forests are known to absorb large amounts of solar radiation and convert it to latent heat, thereby cooling and stabilizing temperatures, and to carry moisture into the atmosphere, thus helping to generate rainfall (Lawrence and Vandecar, 2015). Consequently, rain forests are a key player in regional and global nutrient cycling and climate. However,



the rain forests are endangered by human activities, such as clear-cutting of primary forests for plantations, live-stock, and settlement of residential and industrial areas, but also by atmospheric pollution (Koren et al., 2014; Rosenfeld et al., 2008; ter Steege et al., 2015). Up to now, ~ 16 000 tree species have been estimated for the Amazon (ter Steege et al., 2013), but the impact of anthropogenic activities on these numbers is highly uncertain.

Similarly, it is also hard to predict, to which extent the ongoing and envisioned changes will still ensure its eco-logical services as "green lung" and carbon sink of planet Earth (Soepadmo, 1993).

Apart from vascular plants, forming a predominant fraction of the biomass within this biome, there are also cryp-togamic photoautotrophs comprising bryophytes, algae, lichens, and cyanobacteria, which form communities to-gether with heterotrophic fungi, other bacteria, and archaea. These communities can colonize different substrates,

such as soil, rock, and plant surfaces in almost all habitats throughout the world (Büdel, 2002; Elbert et al., 2012; Freiberg, 1999). In the Amazon rain forest, cryptogamic communities mainly occur epiphytically on the stems, branches, and even leaves of trees, and in open forest fractions they may also occur on the soil. In 2013, 800 species of mosses and liverworts, 250 lichens species, and 1 800 fungal species have been reported for the Amazon region (Komposch and Hafellner, 2000; Mota de Oliveira and ter Steege, 2013; Normann et al., 2010; Piepenbring, 2007).

The epiphytic bryophytes in the tropics play a prominent role in environmental nutrient cycling (Coxson et al., 1992; Zotz et al., 1997) and also influence the microclimate within the forest, thus contributing to the overall fitness of the host plants and the surrounding vegetation (Zartman, 2003). However, they are equally affected by defor-estation and an increasing fragmentation (Zartman, 2003; Zotz et al., 1997).

Physiologically, cryptogamic organisms are characterized by their poikilohydric nature, as they do not actively

regulate their water status, but passively follow the water conditions of their surrounding environment (Walter and Stadelmann, 1968). In a dry state they can outlast extreme weather conditions, being reactivated by water (Oliver et al., 2005; Proctor, 2000; Proctor et al., 2007; Seel et al., 1992), and even fog and dew can serve as a source of water (Lancaster et al., 1984; Lange et al., 2006; Lange and Kilian, 1985; Reiter et al., 2008). In contrast, high water contents may cause a supersaturation, where gas diffusion is restrained, causing a reduced $CO_2$ gas exchange

rates (Cowan et al., 1992; Lange and Tenhunen, 1981; Snelgar et al., 1981) and even ethanolic fermentation (Wilske et al., 2001). Accordingly, their physiological activity is primarily regulated by the presence of water and only secondarily by light and temperature (Lange et al., 1996, 1998, 2000; Rodriguez-Iturbe et al., 1999). Depend-ing on their habitat, they may be active for only a minor fraction of their entire life, as, e.g., described for deserts (Raggio et al., 2017), where water is only rarely available, and for arctic/alpine regions, where frost limits the water

availability (Colesie et al., 2016; Reiter et al., 2008; Schroeter et al., 2011). During nighttime, when only respiration takes place in active organisms, the temperature is highly relevant, as high temperatures may lead to major carbon loss (Lüttge, 2008; Zotz et al., 1997).

It has been shown that despite their inconspicuous growth, cryptogamic communities play a significant role in regional and even global nutrient cycles, as they were estimated to fix ~ 4 Pg carbon per year, corresponding to




about 7 % of the annual net primary productivity of vascular vegetation (Elbert et al., 2012). It has also been calculated that they may play a highly relevant role in the global nitrogen cycle by fixing ~ 49 Tg of nitrogen per year and releasing reactive nitrogen compounds as well as the greenhouse gas $N_2O$ into the atmosphere (Elbert et al., 2012; Lenhart et al., 2015). Furthermore, they may contribute to the uptake of carbonyl sulfide (OCS) (Kuhn

et al., 2000; Kuhn and Kesselmeier, 2000), the bidirectional exchange of volatile organic acids and aldehydes (Wilske and Kesselmeier, 1999), and other volatile organic compounds (Kesselmeier et al., 1999).

In the current study, we present the ecophysiological conditions and activity patterns of bryophyte communities in response to annual and seasonal variations, as well as along a vertical gradient from the understory to the canopy of the forest. These data are essential to analyze the spatio-temporal effects of cryptogamic communities on particle

and trace gas emission patterns.

## 2    Material and Methods

### 2.1    Study site

The measurements were conducted at the research site *ATTO* (*Amazon Tall Tower Observatory*; S 02° 08.602', W 59° 00.033', 130 m a. s. l.) in the Amazonian rain forest in Brazil, which has been described by Andreae and co-

authors (2015). It comprises one walk-up tower and one mast of 80 m each, which have been operational since 2012, and a 325 m tower, which has been erected in 2015. The ATTO research platform has been established to investigate the functioning of tropical forests within the Earth system. It is operated to conduct basic research on greenhouse gas as well as reactive gas exchange between forests and the atmosphere and contributes to our under-standing of climate interactions driven by carbon exchange, atmospheric chemistry, aerosol production, and cloud

condensation. ATTO is located on a *terra firme* forest area, approx. 150 km northeast of Manaus, Brazil. The average annual rainfall is 2 540 mm year$^{-1}$ (de Ribeiro, 1984), reaching its monthly maximum of ~ 335 mm in the wet (February to May) and its minimum of ~ 47 mm in the dry season (August to November) (Pöhlker et al., 2018). These main seasons are linked by transitional periods covering June and July after the wet and December and January after the dry season (Andreae et al., 2015; Martin et al., 2010; Pöhlker et al., 2016). The site is located on

a plateau of yellow clayey ferralsols (latosols, oxisols) deposited on top of sedimentary layers of the Miocene Barreiras formation (Chauvel et al., 1987).

The *terra firme* (plateau forest) has an average growth height of ~ 21 meters, a tree density of around 598 trees ha$^{-1}$, and harbors around 4 590 tree species on an area of ~ 3 784 000 km$^2$, thus comprising a very high species richness compared to other forest types (McWilliam et al., 1993; ter Steege et al., 2013). Other forest types in the

Amazon, such as the *igapó* (floodplain) and *campina/campinarana* (white sand areas), host around 824 and around 1 179 tree species on areas of ~ 100 000 km² and ~ 81 000 km², respectively (Adeney et al., 2016; Andreae et al., 2015; Melack and Hess, 2010; Montero et al., 2014; Sears, 2018; ter Steege et al., 2013, 2015).



### 2.2 Microclimatic conditions within epiphytic habitat

The microclimatic parameters temperature and light within/on top of the bryophytes and the ecophysiological water content of bryophytes are being measured with a microclimate station installed in September 2014 (Fig. S1). The sensors were placed along a vertical gradient at ~ 1.5, 8, 18, and 23 m above the ground, corresponding to the zones 1 to 4 described by Mota de Oliveira and ter Steege (2015). At each height level, six water content, two temperature, and two light sensors (except for 1.5 m with only one light sensor) were installed in different bryophyte species (Table S1b). The temperature and the water content sensors were installed within the epiphytic bryophytes, while the light sensors were fixed on ~ 5 cm long sticks and installed next to the bryophyte cushions (Fig. S1). Whereas bryophytes were selected as cryptogamic exemplary organisms to be measured, similar microclimatic conditions and activity patterns are expected for all cryptogamic organisms due to their poikilohydric nature (Raggio et al., 2017). Since the installation, automatic measurements at 5-minute intervals have been taken with a data logger (CR1000; Campbell Scientific, Logan, Utah, USA) equipped with a relay multiplexer (AM16/32; Campbell Scientific, Bremen, Germany) and two interfaces. For weather protection the logger was placed in a waterproof enclosure (Enc12/14, Campbell Scientific, Logan, Utah, USA), which is additionally sheltered by an external housing.

The water content sensors, initially developed for biological soil crust research (Tucker et al., 2017; Weber et al., 2016), were optimized for measurements in epiphytic bryophyte communities by a straight-lined construction (without a 90° angle) and with outer pins of 25 mm length, serving as an effective holdfast. The WC values are subject to a fluctuation corresponding to approximately 15 % dry weight (DW). For the temperature measurement, thermocouples (Conatex, St. Wendel, Germany) with a tip length of 80 mm and a measurement accuracy of ± 0.5 °C were used. For the light sensors, GaAsP-photodiodes (G1118, Hamamatsu Photonics Deutschland GmbH, Herrsching, Germany) were placed in a housing covered by a convex translucent polytetrafluoroethylene (PTFE) cap and calibrated against a PAR (photosynthetically active radiation) quantum sensor (SKP215; Skye Instruments, Llandrindod Wells, Powys, UK). The average daily PAR values were calculated from the data collected during daytime, i.e., 6:00 to 18:00, while $PAR_{max}$ represents the daily maximum value. The values obtained from the light sensors fluctuated by approximately ± 10 µmol m$^{-2}$ s$^{-1}$ photosynthetic photon flux density (PPFD), thus an averaging of 30-minute intervals allowed a smoothening of the data (Fig. S2). The smoothened data were used for detailed illustrations of seasonal variability (Fig. 2 and S4) and data calculations.

### 2.3 Calculation of the water content

The water content sensors measure the electrical conductivity in the field ($EC_t$), which is influenced by temperature; consequently, a temperature correction was performed according to Eq. (1), analogous to Weber et al. (2016):

$$EC_{25} = f_T * EC_t \tag{1}$$



with $EC_{25}$ as $EC$ at 25°C, $T$ as bryophyte temperature [°C] and the temperature conversion factor $f_T$:

$$f_T = 0.447 + 1.4034 \, e^{-T/26.815} \tag{2}$$

The calibration described below was then applied to the temperature-corrected electrical conductivity values to obtain final water contents as percentage of the dry weight values and precipitation equivalents.

## 2.4   Calibration of the water content sensors

A calibration of the water content sensors was required for each of the different bryophyte species. Thus, the same bryophyte species as those where sensors had been installed in (i.e., *Leucobryum martianum* (Hornsch.) Hampe, *Sematophyllum subsimplex* (Hedw.) Mitt., *Symbiezidium barbiflorum* (Lindenb. & Gottsche) A. Evans, and *Octoblepharum cocuiense* Mitt.), were collected in the forest area surrounding the ATTO site. For the sensors at the upper three height levels the bryophyte taxa could not be securely determined. Thus, the bryophyte taxon with the highest abundance in the canopy communities, i.e., the liverwort *Symbiezidium barbiflorum* was used (Gradstein and Allen, 1992; Mota de Oliveira et al., 2009; Mota de Oliveira and ter Steege, 2015; Pardow et al., 2012; Romanski et al., 2011; Sporn et al., 2010). The bryophytes were removed from the stem with a pocket knife and stored in paper bags in an air conditioned lab container until calibration (few hours to few days after collection). Prior to the calibration, the samples were cleaned from adhering material using forceps.

Subsequently, the water content sensors were calibrated for the different bryophyte species, as, besides the water content, also nutrient content, salinity, thallus structure, and temperature have an impact on the assessed electrical conductivity (Weber et al., 2016). The calibration procedure largely followed Weber et al. (2016), being conducted in a large metal box (60 x 60 x 41 cm; Zarges, Weilheim, Germany) placed in an air conditioned laboratory container at ~ 26 to 28 °C. The samples were placed on a small Styrodur C block (BASF SE, Ludwigshafen, Germany), which was positioned centrally on a balance (ME403, Mettler Toledo, Gießen, Germany). The sensor was fixed in the Styrodur C block, simultaneously fixing the bryophyte sample on the block. In the beginning of each calibration the sample was wetted to the water holding capacity and during drying of the sample, the values of the balance and sensor were recorded at 60-second-intervals. Between the measurement points, a fan placed approx. 10 cm away from the sample was started automatically to cause circulation of the air, thus speeding up the drying process. The measurements were automated by a self-designed program (Docklight scripting) and were continued until the samples reached a constant weight.

Following the measurements, a digital image was taken of each sample and the surface area (*SA*) was determined by means of the software ImageJ (Wayne Rasband, National Institutes of Health, Bethesda, Maryland, USA). The dry weight (*DW*) was determined after drying at 60 °C. The water content (*WC*) of the samples was calculated according to the formula in Weber et al. (2016):

$$WC \, [\% \, DW] = \frac{(SW - (SensW + DW))}{DW} * 100 \, \% \tag{3}$$



with *SW* as sample weight [g], *SensW* as sensor weight [g], and *DW* as sample dry weight [g].

The amount of water within the sample can also be calculated as the precipitation equivalent (*PE*), representing the amount of water per surface area stored within the sample:

$$PE \ [\text{mm } H_2O] = \frac{SW-(SensW+DW)}{SA*\rho H2O} * 10 \qquad (4)$$

with *SA* as surface area [cm²] and *ρH₂O* as the density of water [1 g cm⁻³].

For each of the four bryophyte taxa three replicates (except for *Sematophyllum subsimplex*: four, and *Octoblepharum cocuiense:* two replicates) were investigated and for each replicate four drying cycles were performed. The first cycle of each replicate was excluded from further calculations, as this was needed for the sensor setup to adjust itself. The resulting nine (twelve for *Sematophyllum subsimplex* and six for *Octoblepharum cocuiense*) cal-

ibration curves of each bryophyte species were fit either to a linear, a linear with exponential correction, or a quadratic function. As the variations between the replicates of each taxon tended to be large, the determination of the best fit was not trivial (Fig. S3).

The linear fit was calculated according to the following Eq. (5):

$$WC = a_1 * C + a_2 \qquad (5)$$

The linear fit with exponential correction was calculated as follows:

$$WC = \exp(a_1 * C) * a_2 * C + a_3 \qquad (6)$$

The quadratic fit was calculated as follows:

$$WC = a_1 * C + a_2 + a_3 * C^2 \qquad (7)$$

with *WC* as water content, *C* as electrical conductivity, and $a_0, a_1,$ and $a_2$ as coefficients.

The linear fit was used for the bryophyte species *Symbiezidium barbiflorum* and *Octoblepharum cocuiense*, the linear fit with exponential correction for *Sematophyllum subsimplex,* and the quadratic fit for *Leucobryum martianum* (Fig. S3).

The decision for these fits was based on the fits used in Weber et al. (2016) and on the maximum water contents reached during the calibration. In cases where the electrical conductivity values of field measurements were in a

similar range as during calibration, we applied the fit of highest accuracy. If, however, values of field measurements largely exceeded those of calibration, we applied a quadratic fit or linear fit with exponential correction to avoid unrealistically high water content values in the extrapolated data range. Generally, we observed the fits to have a high accuracy in the range of medium water contents, whereas at high and low water contents the accuracy tended to be lower (Fig. S3).

The quality of each fit was determined by calculating the RMSE of Pearson's R, comparing the predicted versus observed WC (*WCpred*, *WCobs*) with: (Table S1).

$$RMSE = \sqrt{\frac{\sum(WCobs-WCpred)^2}{N}} \qquad (8)$$



The electrical conductivity showed short-time oscillations, which could be removed with a 30-minute smoothing algorithm (Fig. S2). The smoothened data were used for figures and calculations as stated in section 2.2. The electrical conductivity data of replicate samples at the same height (and of the same species) were combined to obtain average values for each height. As microclimatic conditions within one height level differed between samples, depending on their exact position and micromorphology, the average values reflect mean moisture conditions at the respective height levels.

## 2.5 Meteorology

The meteorological parameters have been measured within the ATTO project since 2012. In our study we utilized rainfall data collected at 81 m [mm min$^{-1}$] (Rain gauge TB4, Hydrological Services Pty. Ltd., Australia), relative humidity (RH) at 26 m [%], air temperature at 26 m [°C] (Termohygrometer CS215, Rotronic Measurement Solutions,UK), and photosynthetically active radiation (PAR) data at 75 m [µmol m$^{-2}$ s$^{-1}$ PPFD] (Quantum sensor PAR LITE, Kipp & Zonen, Netherlands). All data were recorded with data loggers (CR3000 and CR1000, Campbell Scientific, Logan, Utah, USA) on the walk-up tower at 1-minute intervals (Andreae et al., 2015).

For the calculation of the average light intensities per month, season or year (PAR$_{avg}$ month, PAR$_{avg}$ season, PAR$_{avg}$ year) only values during daytime (6:00 – 18:00 local time) were considered. The rainfall is presented as accumulated value in millimeters per month, season, or year, which was calculated by an integration of 5-minute intervals. As there were gaps in the readings of the rain gauge, additional information from the water content sensors was used to calculate the number of days with rain events. The sensors at 1.5 m height were found to react reliably to rain events. Thus, the gaps in rain gauge readings were corrected with the information received from these sensors. Furthermore, the amount of rain within each month was corrected by assuming that during the missing days there were the same amounts as during the rest of the month. Overall, a malfunction of the rain detection was observed on only 6 % of the days (Table S3).

The information on fog events was provided by visibility measurements using an optical fog sensor installed at 50 m height (OFS, Eigenbrodt GmbH, Königsmoor, Germany). Fog was defined to occur at visibility values below 2 000 m.

The time is always presented as UTC values, except for diurnal cycles, where local time (LT, i.e., UTC-4) is shown, as labeled in the figures.

## 2.6 Potential physiological activity of bryophytes

The physiological activity of bryophytes – and of cryptogams in general – is primarily controlled by water and light, whereas temperature plays a secondary role – at least in the environment of the central Amazon (Lange et al., 2000; León-Vargas et al., 2006; Romero et al., 2006; Wagner et al., 2014). While the availability of water determines the overall time of physiological activity, the light intensity regulates between net photosynthesis (NP)



and dark respiration (DR). Furthermore, high nighttime temperatures cause increased carbon losses due to high respiration rates (Lange et al., 1998, 2000).

To assess the potential physiological activity of bryophyte communities, the water and light conditions as major drivers of the metabolism need to be investigated. The lower water compensation point ($WCP_l$) presents the minimum WC that allows physiological activity. For tropical species, values in the range between ~ 30 and ~ 225 % have been determined (Romero et al., 2006; Wagner et al., 2013; Zotz et al., 1997, 2003) (Table S2). The water saturation point (WSP) presents the level of WC at which 95 % of the maximum NP rate is reached; at a higher water content a supersaturation may limit the gas diffusion and cause a decrease of NP. The WSP has been determined to range between ~ 349 and ~ 1 053 % for tropical bryophyte species (Romero et al., 2006; Wagner et al., 2013).

The lower light compensation point ($LCP_l$) represents the minimum light intensity that allows a positive photosynthesis; it ranges between ~ 5 and ~ 69 µmol m$^{-2}$ s$^{-1}$ PPFD for tropical cryptogams (Lange et al., 2000; Romero et al., 2006; Wagner et al., 2013). At light intensities below the compensation point and water contents above $WCP_l$ respiration takes place. The light saturation point (LSP), where 95 % of the maximum NP rate are reached, was determined to range between ~110 and ~400 µmol m$^{-2}$ s$^{-1}$ PPFD (León-Vargas et al., 2006; Zotz et al., 1997).

With regard to temperature, a range for optimal NP ($T_{opt}$; 95 % of maximum NP rate reached) and an upper compensation point, where NP equals DR, ($TCP_u$), can be defined. For tropical bryophytes, $T_{opt}$ ranges between 16.0 and 27.5 °C and the $TCP_u$ between 30.0 and 36.0 °C (Frahm, 1990; Wagner et al., 2013). For long-term survival and growth, the bryophytes need to be predominantly exposed to temperatures below the upper compensation point, at least under humid conditions.

Based on the literature data, we calculated ranges of timespans when these key points were passed. A water content above the compensation point allows NP if both light intensity and temperature are above the compensation point. If water contents are above the compensation point but light intensities are too low, or if temperatures are above the upper compensation point, net DR occurs. There is also a narrow span of low water contents, when samples are activated already but despite sufficient light intensities only net respiration can be proceeded. As this span of water contents is narrow and respiration rates are low, it has been neglected in the current calculations. The compensation points for the different parameters are also to some extent interrelated, e.g., the water compensation point of lichens has been shown to slightly increase with increasing temperature (Lange, 1980), but we found that this can be neglected in such a first qualitative approach. Finally, also inter- and intraspecific variation of compensation points could not be considered in the current study.

## 2.7    Statistical analysis

All data processing steps and analyses were performed with the software IGOR Pro (Igor Pro 6.37, WaveMetrics. Inc, Lake Oswego, Oregon, USA). Statistical tests for normal distribution and variance homogeneity of data sets,





as well as Mann-Whitney U tests were performed with the software OriginPro (Version 8.6; OriginLab Corporation, Northampton, Massachusetts, USA), the Kruskal-Wallis-test with subsequent post-hoc test was performed with Statistica (Version 13.3; StatSoft GmbH, Hamburg, Germany).

## 3  Results

### 3.1  Microclimatic conditions

#### 3.1.1  Annual fluctuation of monthly mean values

Over the course of the two years of measurements, the monthly mean values of (micro-) climatic and ecophysiological parameters varied depending on seasons and years. Comparing the two consecutive years, the effect of an El Niño event was clearly detectable, as rainfall amounts were 35 % (525 mm versus 805 mm) and relative air
humidity 11 % lower (81 % versus 92 %) between October 2015 and February 2016 than in the previous year (Fig. 1, Table 1).

The monthly mean values of mesoclimatic ambient PAR (ambient $PAR_{avg}$) were rather stable throughout the years and did not differ significantly between the years 2015 and 2016, ranging between 315 and 570 µmol m$^{-2}$ s$^{-1}$ PPFD in the daytime (Fig. 1, Table 1). Within the canopy, the $PAR_{avg}$ values at 1.5 m also showed only minor seasonal
variation, whereas those at higher levels had larger variations (Table S4). At 23 m height, $PAR_{avg}$ values tended to be higher during the dry seasons. Comparing the two subsequent years, the annual mean values of the monthly $PAR_{avg}$ were significantly higher at 1.5, 8, and 18 m, whereas at 23 m they were lower in 2015 compared to 2016 (Table 1).

Over the course of the years, the monthly mean temperatures at all heights as well as mesoclimatic ambient tem-
peratures showed a parallel behavior (Fig. 1). The temperatures decreased in a stepwise manner from the canopy to the understory, and temperatures within bryophytes at 23 m height were frequently higher than the mesoclimatic ambient temperatures measured (Fig. 1, Table 1, Fig. S5). Overall, temperatures at all height levels were lower and more similar during the wet than the dry seasons. Maximum differences of monthly mean temperatures between the wet and dry season were 5.0 °C at 23 m height, 3.0 °C at 1.5 m height, and 4.0 °C for mesoclimatic ambient
values.

The monthly rain, RH, and water contents of epiphytic bryophytes showed similar patterns over the course of the years. During the dry season 2015, it rained on 25 % of the days per month, while in the previous and subsequent years rain occurred at a higher frequency (58 % and 31 % of the days per month, respectively; Fig. 1, Table S3). Monthly rain amounts varied from 9 mm during the dry to 340 mm during the wet season. In 2016 the rain in-
creased from January to March and decreased from March to August, while in 2015 the monthly rain amounts were more variable but still lower throughout the year.



The RH values were characterized by a similar behavior, with an increase from January to March 2016 and a subsequent decrease, while in the end of 2014 and the first half of 2015 the RH values showed minor fluctuations compared to those of the rain amounts. The lowest monthly average of the RH was detected during the dry season 2015 with $74 \pm 15$ %.

The WC values of epiphytic bryophytes at different height levels were the highest at 1.5 m and decreased towards 23 m and 18 m, being the lowest at 8 m height (Table 1). The highest monthly averages of the WC values were reached from January to May 2015 and from February to April 2016, whereas the lowest contents were measured from September 2015 to January 2016. Furthermore, the bryophytes at 8 and 18 m height showed particularly high WC values in November and December 2016.

**3.1.2      Seasonal changes between wet and dry season**

The wet seasons were characterized by a high frequency of precipitation events, large amounts of rain per event, frequent appearance of fog, and high RH values, ranging mostly above 70 %. In contrast, during the dry season the precipitation events were much rarer and smaller, there was hardly any occurrence of fog, and the RH values were regularly below 60 % (Fig. 2). The temperature and light conditions within and on top of the epiphytic bryo-

phytes followed the mesoclimatic ambient conditions, modified by the canopy shading.

The mesoclimatic ambient light intensity above the canopy was on average higher in the dry season than the wet season ($970 \pm 650$ vs. $740 \pm 570$ µmol m$^{-2}$ s$^{-1}$ PPFD) and the values showed stronger fluctuations. During both main seasons the average light intensity decreased from the canopy towards the understory. During the dry season this happened in a regular stepwise manner, whereas in the wet season there were some irregularities, with values

at 23 m being lower than at 8 m or 18 m height (Fig. 2, Table 2).

The temperatures showed larger diel amplitudes in the dry compared to the wet season. Temperatures reflected a decreasing gradient from the canopy towards the understory and differences among heights were more pronounced during the dry season (Fig. 2, Table 2). At 23 m height, temperatures within bryophytes were frequently higher than the mesoclimatic ambient values, and during the dry season even the average seasonal value was higher.

During 2015 and 2016, rain occurred in the wet season on 81 and 87 % of the days and in the dry season on 25 and 31 % of the days, respectively. During the dry season the RH reached on average $87 \pm 14$ %, while in the wet season the average RH was $95 \pm 9$ % and frequently even full saturation was reached. Fog was recorded on 56 and 67 % of the days during the wet seasons of 2015 and 2016 and on 27 and 16 % of the days during the dry seasons, respectively (Fig. 2, Table 2).

The WC of bryophytes at 1.5 m height responded consistently to rain events, while at 8, 18, and 23 m height not in all cases an immediate response was observed. The epiphytes at 1.5 m height had a high WC over several days or even weeks during the wet season, while in the dry season they had lower WC values. The bryophytes at the upper three heights showed a regular nightly increase of WC, especially during the dry season (Fig. 2).



### 3.1.3    Diel cycles (in different seasons and years along the vertical gradient)

The diel cycles of micrometeorological and ecophysiological conditions showed varying characteristics during the wet and the dry season (Fig. 3). The diel variability of light and temperature experienced by the bryophytes was larger in the canopy than in the understory, while the variation of the water content (WC) of bryophytes was the largest at the lowest level. Comparing the seasons, the diel amplitudes of light, temperature, and RH were larger in the dry compared to the wet season, while for the WC of bryophytes the results do not present a clear pattern. The average daily mesoclimatic ambient light intensities ($PAR_{avg}$) were higher during the dry than in the wet season, and also the $PAR_{avg}$ on top of the epiphytic bryophytes at different height levels predominantly reached higher values during the dry season (Fig. 3). This mostly corresponds well with the daily maximum and amplitude values measured by the mesoclimatic ambient and the microclimatic sensors, as these mostly were also higher during the dry seasons. Exceptions from that were the lower mesoclimatic ambient values during the dry season 2015 and relatively low values at 8 m and 1.5 m height during the dry season 2016 (Table S6). The variability and the diel amplitudes tended to be higher for the epiphytes in the canopy than for the organisms in the understory.

The mesoclimatic ambient temperatures showed larger diel amplitudes and higher values in the dry compared to the wet seasons (Fig. 3). Also mean daily maxima were higher with $33.5 \pm 2.0$ and $32.5 \pm 2.0$ °C during the dry compared to $29.0 \pm 2.5$ and $30.5 \pm 2.0$ °C reached during the wet seasons of 2015 and 2016, respectively. The microclimatic mean temperatures measured within the epiphytic bryophytes showed an increasing daily amplitude and increasingly higher maximal temperatures from the understory to the canopy. Daily maxima, minima, and amplitudes were larger in the dry than the wet seasons.

The mean RH values showed larger daily amplitudes in the dry compared to the wet seasons with particularly large amplitudes during the dry season 2015 (Fig. 3). Also the mean daily maxima of RH reached only 96 % in the dry season 2015, whereas in all other seasons (i.e., dry season 2016 and both wet seasons) values above 99 % were reached. The diel mean WC of epiphytic bryophytes was the highest at 1.5 m and also daily maxima, minima, and amplitudes were the highest at this level. At 1.5 m height, the daily amplitudes were significantly higher during the wet compared to the dry seasons, whereas at the higher levels were less clear. The WC of the epiphytic bryophytes at 8 and 18 m height were relatively constant in the year 2015, while values were more variable during the dry season 2016.

### 3.2    Potential physiological activity of bryophytes

In the tropical rain forest environment, the physiological activity of cryptogams, including bryophytes, is predominantly controlled by water and light, whereas the temperature plays a minor role (Lange et al., 2000; León-Vargas et al., 2006; Romero et al., 2006; Wagner et al., 2014). While the availability of water determines the overall time



of physiological activity, light regulates between net photosynthesis and dark respiration. Furthermore, a high nighttime temperature causes increased carbon losses due to high respiration rates.

Whereas overall light intensities at the upper three height levels were similar, with values mostly ranging between 0 and 100 µmol m$^{-2}$ s$^{-1}$ PPFD and maximum light intensities of 1 500 (8 m), 1 040 (18 m), and 950 µmol m$^{-2}$ s$^{-1}$

(23 m), intensities at 1.5 m height were extremely low, mostly reaching 0 – 10 µmol m$^{-2}$ s$^{-1}$ PPFD, although maximum values of 1 550 µmol m$^{-2}$ s$^{-1}$ PPFD were measured (Fig. 4). The light intensities in the understory (1.5 m) reached the lower compensation points (LCP$_l$) described in the literature (Wagner et al., 2013) only during 0 – 8% of the reported time, suggesting that during most times respiration exceeded photosynthesis. At higher canopy height levels, the bryophytes reached these values during 4 – 45 % of the time (Table 3). Light saturation points

(LSP), which have been reported to range between 110 and 400 µmol m$^{-2}$ s$^{-1}$ PPFD (León-Vargas et al., 2006; Zotz et al., 1997), were hardly ever reached in the understory and only rarely in the canopy. Light intensities above the saturation point were mostly reached when sunspots briefly reached the bryophytes.

The microclimatic temperatures were fairly similar throughout the canopy, mainly ranging between 23.0 and 33.0 °C (Fig. 4). In tropical lowland regions, the optimum temperatures for bryophytes range between 25.0 and

27.5 °C (Frahm, 1990; Wagner et al., 2013). In our studies this optimum was matched during 6 – 32 % of the time (Table 3). The upper temperature compensation point (TCP$_u$) of 30.0 – 36.0 °C (Wagner et al., 2013), above which respiration exceeds photosynthesis, was surpassed during 0 – 11 % of the time in the understory and 0 – 17 % in the upper three canopy levels.

The water content of bryophytes differed along the vertical profile, with substantially higher values reached in the

understory than at the upper three height levels, where a stepwise increase in water contents was observed. The lower water compensation point (WCP$_l$), ranging between 30 and 225 % according to the literature (Romero et al., 2006; Wagner et al., 2013), was reached during 25 – 100 % of the time in the understory and 0 – 100 % at the upper three height levels (Fig. 4; Table 3). The water saturation point (WSP) was reached during 1 – 22 % of the time by understory bryophytes, while at the upper three canopy levels water saturation was never reached.

Recapitulating our findings about the compensation points of water, light, and temperature, one can make rough estimates of the potential time fractions of physiological activity, i.e., NP and DR, of the bryophytes at different heights (Table 3). As the upper end of the WCP$_l$ range is reached during 100% of the time at all height levels, one can expect bryophytes to be active during major fractions of their lifetime, although this fraction potentially is larger at lower heights of growth. Also the microclimatic temperatures are mostly favorable and only rarely above

the upper compensation points at all canopy height levels, thus in theory mostly facilitating NP. Light intensities, however, vary widely with height and are decisive for the overall theoretical duration of NP taking place. Considering the lower end of the range of the LCP$_l$ (i.e., 5 µmol m$^{-2}$ s$^{-1}$ PPFD) and the WCP$_l$ (i.e., 30 %), and the upper end of the TCP$_u$ (i.e., 36.0 °C), the NP is reached during 47 – 59 % of the time at higher canopy levels, while in the understory (1.5 m) it is only achieved during 8 % of the time. Assuming the upper end of the LCP$_l$ (69 µmol



m$^{-2}$ s$^{-1}$ PPFD) and the WCP$_l$ (225 %), and the lower end of TCP$_u$ (i.e., 30.0 °C), NP would not occur at any of the different height levels. The time fractions representing theoretical duration of DR are similar at the three canopy levels, ranging between 0 – 43 %, whereas in the understory DR occurs during 50 – 96 % of the time (Table 3).

## 4    Discussion

### 5    4.1    Microclimatic conditions

The microclimatic conditions experienced by bryophytes along an altitudinal gradient at the ATTO site follow the meteorological characteristics to some extent, but they also reveal microsite-specific properties regarding annual, seasonal, and diel microclimate patterns.

Over the course of two years, the mesoclimatic ambient average monthly light conditions (PAR$_{avg}$) were rather 10   stable. In previous studies, increased biomass burning activities during El Niño in 2015 were reported to cause an increase of smoke and soot particles in the atmosphere (Saturno et al., 2017), and our data also suggest a slight reduction of monthly PAR$_{max}$ during the dry season 2015 (Table S3). Within the canopy, the monthly PAR$_{avg}$ values at 23 m height tended to be higher during the dry seasons, whereas patterns were less clear at 18 and 8 m height and there was hardly any seasonal variation at 1.5 m height. This was most probably an effect of the canopy 15   structure and cushion orientation, as the sensors were installed with the following orientations: at 1.5 and 8 m vertically along the trunk, at 18 m at the upper side of a slightly sloped branch, and at 23 m at the upper side of a vertical branch. As the light sensors on top of the bryophytes at 23 m height were within the canopy, newly growing leaves may have periodically shaded the organisms. This can explain lower monthly PAR$_{avg}$ values in the canopy compared to the values at the lower height levels.

20   The diel patterns of PAR$_{avg}$ are expected to show a decreasing gradient from canopy to understory, as the canopy receives most solar radiation, while the understory vegetation is expected to be shaded by foliage and branches. During the dry season this general pattern was indeed observed, whereas during the wet season mean light intensities were often higher at 8 than at 18 and 23 m, probably also caused by canopy shading effects at the upper two height levels. The diel amplitudes of PAR$_{avg}$ were larger in the dry than the wet season and larger in the canopy 25   than the understory. High light intensities occurred in the understory only as small light spots of short duration. Bryophyte taxa in the understory are known to be adapted to these low light conditions and are able to make efficient use of the rather short periods of high light intensities (Lakatos et al., 2006; Lange et al., 2000; Wagner et al., 2014).

The microclimatic temperatures measured within bryophyte cushions followed the mesoclimatic ambient temper- 30   ature at all height levels, with an increasing gradient from the understory towards the canopy, probably caused by a reduced shading effect towards the canopy. At the uppermost height level, mean temperatures within bryophytes



often were even higher than the mean mesoclimatic ambient temperatures. This pattern was mainly observed during the dry seasons, when canopy shading only played a minor role and reduced wind velocity resulted in a stronger surface heating effect. During the wet season, the overall temperature conditions were more buffered due to reduced incoming radiation caused by clouds and a frequent mixing of the air masses during rain events (von Arx et al., 2012; Gaudio et al., 2017; Thompson and Pinker, 1975).

The microclimatic mean temperature differences between the understory (1.5 m) and the canopy (23 m) were 1.5 °C in the dry and only 0.5 °C in the wet season. Compared to these results, a temperature difference of 4.0 °C was determined for the dry season in a tropical evergreen forest in Thailand, while in the wet season it was <1.0 °C, thus corresponding quite well to our results (Thompson and Pinker, 1975) (Table 2). The diurnal and seasonal temperatures were the most stable in the understory, whereas the largest variations were observed in the canopy. This could be expected due to exposure to the strong solar radiation and higher wind velocity in the canopy compared to the sheltered understory (Kruijt et al., 2000).

The two consecutive years 2015 and 2016 were by no means identical, as rainfall amounts and relative air humidity values were considerably higher between October 2014 and February 2015 as compared to the following year. The dry season of 2015/2016 was affected by an El Niño event, causing air humidity and WC of bryophytes to be significantly lower compared to the previous dry season (Fig. 1, Table 1). The WC measurements at 18 and 23 m height were unexpectedly high in the end of 2016. This can be explained by a reinstallation of some sensors, which previously had fallen out of the moss cushions. Sensor displacement or complete removal from the cushions might have been caused by mechanical disturbance, like heavy rain events, movement of branches, growth of epiphytic vascular plants, or animal activity. A necessary reinstallation of the sensors unfortunately affected the measured values, as electrical conductivity values vary depending on the bryophyte cushion properties. This variability of data, depending on the exact placement of the sensors, illustrates that calculated WC contents could only be considered as approximate values.

As expected, the response of the water contents of bryophytes upon rain, fog, and high RH differed between seasons. During the wet season, the RH and the bryophyte WC was significantly higher compared to the dry season (Table 3), when the RH values showed a stronger decrease in the daytime. During the wet season, the frequency of rain was much higher, and thus the bryophytes in the understory (1.5 m) were often still wet when the next rain event started, while bryophytes in the canopy had already dried out.

Furthermore, the angle of the stem or branch colonized by the investigated bryophytes played a crucial role for rainwater absorption and the subsequent drying process (Table 2). The bryophytes at 1.5 and 8 m height were oriented vertically, those at 18 m were placed on the upper side of a slightly sloping branch, and those at 23 m were located on the upper side of a nearly horizontally oriented branch. At 8 m height, the bryophytes were exposed to the west, and thus were only sometimes directly influenced by precipitation, as in most cases, due to the predominant wind directions, north-, east-, and south-oriented tree fractions received the largest precipitation amounts.




Long-term climate data have shown that the winds during the wet season predominantly originated from north and north-eastern directions, while during the dry season south- and south-easterly winds prevailed (Pöhlker et al., 2018). In contrast to that, the bryophytes at 18 m height showed a clear response to precipitation events. Here, the bryophyte cushions were exposed to the south, which is more frequently influenced by rain events. Thus, the shift

of the main wind direction from northeasterly to southeasterly might explain the fact that the bryophytes at 18 and 23 m height responded more strongly to rain events in the dry season than they did in the wet season. Moreover, the tree foliation and epiphytic vascular plants might shield the sensors from direct precipitation during the wet season.

During the dry season, the drying of the samples located in the canopy occurred quite rapidly after rain. Most rain

events in the Central Amazon occur in the early afternoon (12:00 – 14:00 LT) and more than 75 % of them are weak events of less than 10 mm (Cuartas et al., 2007), which facilitates fast drying of the cryptogams inhabiting the canopy. Besides the solar radiation, probably also the higher wind velocities accelerated the desiccation of the epiphytic cryptogams in the canopy (Oliver, 1971). The diel mesoclimatic ambient RH amplitudes were larger and reached lower values during the dry season, thus also promoting quicker drying of samples.

Furthermore, condensation and stemflow water need to be considered as a potential additional sources of water for epiphytic covers as well as for near-stem vegetation at the forest floor (Lakatos et al., 2012; van Stan and Gordon, 2018). It has been shown that in tropical forests stemflow water could provide up to 4 % of the annual rainfall amount (van Stan and Gordon, 2018), corresponding to maximum the values of 68 and75 mm in the years 2015 and 2016 at the ATTO site. The WC of bryophytes in the understory showed a high variability during the wet

season, indicating that large amounts of water were taken up during prolonged rain events, which were subsequently lost again in a stepwise manner, with bryophytes often staying wet and active over long time spans (Fig. 2). The particularly large water holding capacity of the organisms in the understory may partly also be explained by the different bryophyte species growing (and measured) in the understory, which are especially adapted to long-term water storage (Lakatos et al., 2006; Romero et al., 2006; Williams and Flanagan, 1996).

## 4.2   Potential physiological activity of bryophytes

The microenvironmental conditions influence the WC of epiphytic bryophytes, but the ability to deal with these conditions differs among species, being determined by their morphological and physiological features. Apart from the long-term adaptation of the metabolic properties, the performance of species under differing microenvironmental conditions can also be modulated by acclimation processes (Cornelissen et al., 2007; Pardow et al., 2010).

These two aspects help to understand the occurrence of bryophytes under widely varying microclimatic conditions within the canopy. It was recently demonstrated that a prediction of the physiological activity patterns of cryptogamic organisms and communities was possible alone on the basis of climatic conditions (Raggio et al., 2017). During our study, we also observed bryophyte taxa to vary depending on the microenvironmental conditions.



Whereas at the stem bases close to the ground *Sematophyllum subsimplex* and *Leucobryum martianum* were dominating, *Octoblepharum cocuiense* and the liverwort *Symbiezidium barbiflorum* were the main species occurring at higher levels along tree stems at the ATTO site, obviously in close adaption to the specific locations. These species have also been reported as being frequent at other tropical rain forest sites (Campos et al., 2015; Dislich et

al., 2018; Gradstein and Allen, 1992; Mota de Oliveira et al., 2009; Pinheiro da Costa, 1999).

Epiphytic organisms are known to be adapted to environmental light conditions, with $LCP_l$ and LSP being lower under low-light conditions in the understory as compared to the canopy (Lakatos et al., 2006). In the canopy it is essential for the cryptogams to be adapted to high light conditions and UV radiation in order to avoid cell damage by radiation (Green et al., 2005; Sinha and Häder, 2008; Westberg and Kärnefelt, 1998). As reference we included

the ranges of $LCP_l$ given by Romero et al. (2006), who reported 5 µmol $m^{-2}$ $s^{-1}$ PPFD as the lowest light compensation point, which was reached during 45 % of the time in the canopy and 8 % of the time in the understory (Fig. 4, Table 3). In such cases, the organisms need a very rapidly and efficiently operating photosystem to reach overall positive net photosynthesis rates, and it has been reported that understory mosses indeed show higher rates of net photosynthesis than canopy species (Kangas et al., 2014; Lakatos et al., 2006; Wagner et al., 2013). However, for

other habitats, light compensation points as low as 1 µmol $m^{-2}$ $s^{-1}$ PPFD have been defined for lichens (Green et al., 1991), and thus we could imagine that the understory organisms at the ATTO site exhibit similarly low $LCP_l$ values.

The temperature regulates the velocity of metabolic processes. Whereas it has a strong impact on respiration, the photosynthetic light reactions are far less sensitive. As the measured net photosynthesis rates are the sum of sim-

ultaneously occurring photosynthesis and respiration processes. Thus, positive net photosynthesis rates may still be reached at higher temperatures in the light, as long as the photosynthetic capacity is high enough, whereas during the night, high temperatures could cause a major loss of carbon due to high respiration rates (Lange et al., 2000).

The optimum temperatures for net photosynthesis ($T_{opt}$) range from 25.0 to 27.5 °C for tropical bryophytes (Wag-

ner et al., 2013), and these values were reached during 6 to 32 % of the time at all four height levels with no major differences among them. The upper temperature compensation point ($TCP_u$) between 30.0 and 36.0 °C (Wagner et al., 2013) was only rarely reached during our study (i.e., up to 17 % of the time). The lowest temperatures predominantly occurred during the night, contributing to lower respiration rates. Thus, the temperature does not seem to play a relevant role as limiting factor for the physiological activity of epiphytic bryophytes in this environment.

The WC of bryophytes has been shown to be higher in the understory than in the canopy. In the understory, the $WCP_l$ was reached between 25 and 100 % of the time, depending on the literature value being considered, and the WSP was reached during 1 – 22 % of the time. In the canopy, the $WCP_l$ was reached between 0 to 100 % of the time, whereas the WSP was almost never reached.



Investigating six tropical bryophyte species, Zotz et al. (1997) pointed out that the WC of bryophytes varied between 310 and 2 000 %, depending on the species. During their investigation in September/October 1993, the WC of the bryophyte species *Leucobryum antillarum* varied between 1 200 and 1 400 %, while in our investigation the WC of *Leucobryum martianum* ranged between 200 and 1 950 % during the same time of the year. Thus, both

ranges of WC are comparable and the larger range recorded in our study might result from the longer observation period (September 2015, 2016, October 2014, 2015, and 2016).

Furthermore, water from fog might trigger a physiological activity of bryophytes (León-Vargas et al., 2006), and also condensation needs to be considered as a water source for cryptogams (Lakatos et al., 2012). In their study on corticolous epiphytic lichens in a tropical lowland cloud forest they showed that lichens benefit from dew formation

on the thallus surface during noon, and we can assume that similar processes occur quite regularly on epiphytic cryptogams.

In the understory, the WC of cryptogams seems to be predominantly regulated by rain events. Additionally, the vegetation reduces the evaporation by its shadowing effect, whereas in the canopy, rain events, fog, and condensation seem to be equally important water sources for cryptogams. Our data have shown that in the understory the

bryophytes stay wet for most of the time, and consequently they might be more sensitive to drought than canopy species, which has also been observed by Pardow and Lakatos (2013).

As bryophytes are poikilohydric organisms, water availability controls their overall physiological activity, while light regulates the photosynthetic activity in an active organism state. Temperatures largely affect respiration of the organisms, whereas photosynthesis rates are affected to a minor extent (Green and Proctor, 2016; Lange et al.,

1998; Weber et al., 2012). In the understory, the major limiting factor for photosynthesis was light, as during most of the day its intensity was very low, i.e., during 92 % of the time light intensity was below the potential $LCP_1$ of 5 µmol m$^{-2}$ s$^{-1}$ PPFD (Table 3). Combining light, temperature, and water compensation points, the results suggest that understory bryophytes perform NP during 0 – 8 % and DR during 50 – 96 % of the time. It has been shown that understory organisms are adapted to low light conditions by rather high photosynthesis and low respiration

rates (Pardow and Lakatos, 2013), which would facilitate their existence under the given environmental conditions. In the canopy, one may expect water to represent the limiting factor, as bryophytes were frequently exposed to higher light intensities and warmer temperatures, but according to $WCP_1$ ranging between 30 and 225 %, bryophytes at the upper three height levels perform positive NP during 0 – 59 % and DR during 0 – 43 % of the time. Generally, lower WCP and LCP contribute to larger rates of NP, whereas high compensation points cause respira-

tion and inactivity. The adaption of the organisms to the environmental conditions is the crucial point of survival in this environment. All these time ranges of metabolically activity are only rough estimates, which predominantly depend on the actual compensation points being influenced by inter- and intraspecific variation.

There are also some differences between groups, as, e.g., lichens tend to perform photosynthesis at lower water contents than bryophytes, and chlorolichens (with green algae as photobionts) may utilize high air humidity



whereas cyanolichens (cyanobacteria as photobiont) need liquid water (Green et al., 2011; Lange and Kilian, 1985; Raggio et al., 2017). The wide ranges of potential activity and the scarcity of literature data illustrate the necessity of $CO_2$ gas exchange measurements to assess the actual diel and seasonal physiological activity and productivity of rainforest cryptogams under varying environmental conditions.

## 5 Conclusions

The microclimatic conditions experienced by bryophytes are being assessed in long-term measurements at the ATTO site since October 2014. These measurements provide a unique data set of the micrometeorological conditions within the understory and the inner canopy of tropical rain forests and facilitate a rough estimation of the physiological activity patterns of epiphytic bryophytes along a vertical gradient. Within this tropical rain forest habitat, the water content has turned out to be the key parameter controlling the overall physiological activity of the organisms with major differences between organisms of the canopy and the understory. While in the understory the bryophyte water contents vary widely between seasons and the organisms may stay wet over several days or even weeks during the wet season, those in the canopy are exposed to frequent wetting and drying cycles. The light intensity during periods of physiological activity mainly determines whether NP takes place or carbon is lost by respiration. As the temperature shows only minor spatial, diel, and seasonal variation, it is only of minor physiological relevance within the given habitat. Data on the potential physiological activity of bryophytes and cryptogamic organisms in general are not only relevant for their potential role in carbon cycling, but may also provide new insights into their relevance as sources of bioaerosols and different trace gases. Thus, these data may form a baseline for studies investigating the overall relevance of cryptogams in biogeochemical cycling in tropical habitats.

**Data availability**

All data are deposited in a local database at the Max Planck Institute for Chemistry and unrestricted access is provided upon request.

**Supplement link**

**Author contribution**

BW, CP, and NL designed the measurement setup. NL, CGGB, SB, and APPF conducted the practical measurements. NL, DW, GRC, MS, AA, LRO, FD, and SMO compiled the data and conducted the analyses. All authors discussed the results. NL and BW prepared the manuscript with contributions from all co-authors.





**Disclaimer**

The authors declare that they have no conflict of interest.

**Special issue statement**

**Acknowledgement**

We would like to acknowledge the German Federal Ministry of Education and Research (BMBF contracts 01LB1001A and 01LK1602B), supporting this project as well as the construction and operation of the ATTO site. We also acknowledge the support of the Brazilian Ministério da Ciência, Tecnologia e Inovação (MCTI/FINEP contract 01.11.01248.00) as well as the Amazon State University (UEA), FAPEAM, LBA/INPA and SDS/CEUC/RDS-Uatumã for their support during construction and operation of the ATTO site. Furthermore, we

would like to thank Ulrich Pöschl for provision of the scientific infrastructure and the possibility to work in the labs. We would like to thank Reiner Ditz, Susan Trumbore, Alberto Quesada, Thomas Disper, Andrew Crozier, Hermes Braga Xavier, Feliciano de Souza Coelho, Josué Ferreira de Souza, Roberta Pereira de Souza, Holger Ritter, Henno Heintz, and Henning Braß for technical, logistical, and scientific support within the ATTO project. NL would like to thank the Max Planck Graduate Center (MPGC) for its support. DW, CGGB, SB, FD, DMZ,

APPF, SW, JK, CP and BW appreciate the support by the Max Planck Society. GRC would like to thank for the support of the Instituto Nacional de Pesquisas da Amazônia (INPA) provided by the Programa de Pós-graduação em Botânica. MS and LRO appreciate the support of INPA. AA would like to thank the Empresa Brasileira de Pesquisa Agropecuária (EMBRAPA), RHMG expresses his thanks to the Federal University of Parana, SMO would like to thank Stichting Het Kronendak, and MOA appreciates the support of the Max Planck Society and

the University of San Diego. This paper contains results of research conducted under the Technical/Scientific Cooperation Agreement between the National Institute for Amazonian Research, the State University of Amazonas, and the Max Planck Society. The opinions expressed are the entire responsibility of the authors and not of the participating institutions.



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



**Tables**

**Table 1:** Annual mean values, standard deviation (± SD), and statistical significance of photosynthetically active radiation (PAR$_{avg}$), daily maxima of photosynthetically active radiation (PAR$_{max}$), temperature, relative humidity (RH), and water content of bryophytes at four the height levels and of ambient conditions (a). Annual sum of rain and fog days as well as the annual sum of rain (b). Mean values were calculated from 5-minute intervals. Due to data gaps in the measured rain data (shown in brackets) values were also extrapolated from existing data as described in methods section (values behind data in brackets). Due to absence of normal distribution and variance homogeneity, a non-parametric Mann-Whitney U test was performed to compare values obtained for the two years.

(a)

| Height | 2015 | | 2016 | | Statistical test |
|---|---|---|---|---|---|
| [m] | Mean | ± SD | Mean | ± SD | p |
| **PAR$_{avg}$ daytime[µmol m$^{-2}$ s$^{-1}$ PPFD]** | | | | | |
| ambient, 75 | 911 | 678 | 841 | 653 | 0.633 |
| 23 | 34 | 1 | 58 | 8 | ≤ 0.001 |
| 18 | 45 | 15 | 34 | 11 | ≤ 0.001 |
| 8 | 35 | 19 | 17 | 10 | ≤ 0.001 |
| 1.5 | 5 | 35 | 4 | 20 | ≤ 0.001 |
| **PAR$_{max}$ [µmol m$^{-2}$ s$^{-1}$ PPFD]** | | | | | |
| ambient, 75 | 2043 | 579 | 2153 | 433 | 0.005 |
| 23 | 320 | 24 | 497 | 51 | ≤ 0.001 |
| 18 | 310 | 38 | 331 | 26 | 0.095 |
| 8 | 322 | 236 | 116 | 86 | ≤ 0.001 |
| 1.5 | 172 | 0 | 99 | 140 | 0.033 |
| **Temperature [°C]** | | | | | |
| ambient, 26 | 26.6 | 3.4 | 26.4 | 3.1 | ≤ 0.001 |
| 23 | 25.9 | 1.0 | 26.5 | 0.5 | ≤ 0.001 |
| 18 | 26.2 | 0.0 | 26.3 | 0.0 | ≤ 0.001 |
| 8 | 25.8 | 0.2 | 25.8 | 0.2 | ≤ 0.001 |
| 1.5 | 25.4 | 0.0 | 25.5 | 0.1 | ≤ 0.001 |
| **Water content [%]** | | | | | |
| ambient RH, 26 | 86 | 15 | 90 | 13 | ≤ 0.001 |
| 23 | 88 | 7 | 94 | 14 | ≤ 0.001 |
| 18 | 65 | 5 | 102 | 36 | ≤ 0.001 |
| 8 | 46 | 9 | 63 | 16 | ≤ 0.001 |
| 1.5 | 303 | 87 | 273 | 25 | ≤ 0.001 |

(b)

| Parameter | 2015 | 2016 |
|---|---|---|
| | Sum | Sum |
| **Rain (days)** | (199) 202 | (197) 215 |
| **(mm)** | (1680) 1693 | (1702) 1863 |
| **Fog (days)** | 21* | 28* |

*: Gaps in the data record due to malfunction of fog sensor during time window of 31.05.-20.10.2015, 30.04.-06.07.2016, and 01.09.-31.12.2016.



**Table 2:** Seasonal mean values, standard deviation (± SD), and statistically significant difference between seasons for the parameters photosynthetically active radiation (PAR$_{avg}$), daily maximum of photosynthetically active radiation (PAR$_{max}$), temperature, and ambient relative humidity/water content (WC). Values measured as ambient conditions and within/on top of bryophytes at four height levels. Mean values for the respective seasons were calculated from 5-minute intervals of the years 2015 and 2016. Due to the absence of normal distribution and variance homogeneity, a non-parametric Kruskal-Wallis test with post hoc test was performed to compare values obtained for different seasons. The statistical comparison among height levels for the different seasons is shown in Table S5.

| Season | PAR$_{avg}$ daytime [µmol m$^{-2}$ s$^{-1}$ PPFD] Mean | ± SD | sig. | PAR$_{max}$ [µmol m$^{-2}$ s$^{-1}$ PPFD] Mean | ± SD | sig. | Temperature [°C] Mean | ± | sig. | WC; ambient RH [%] Mean | ± | sig. |
|---|---|---|---|---|---|---|---|---|---|---|---|---|
| **ambient** | | | | | | | | | | | | |
| Wet | 738 | 566 | a | 2086 | 515 | a | 25.6 | 2.5 | ab | 94 | 9 | a |
| Trans Wet-Dry | 861 | 649 | a | 2227 | 182 | a | 25.8 | 3.0 | b | 91 | 11 | b |
| Dry | 973 | 647 | a | 2100 | 609 | a | 26.7 | 3.4 | c | 87 | 14 | c |
| Trans Dry-Wet | 785 | 617 | a | 1988 | 509 | b | 26.5 | 3.3 | a | 85 | 15 | d |
| Statistical test, p | 1.000 | | | ≤ 0.001 | | | ≤ 0.001 | | | ≤ 0.001 | | |
| **23 m** | | | | | | | | | | | | |
| Wet | 30 | 3 | a | 248 | 194 | a | 25.3 | 2.0 | a | 92 | 11 | a |
| Trans Wet-Dry | 41 | 72 | a | 414 | 252 | b | 25.7 | 2.8 | b | 93 | 10 | b |
| Dry | 55 | 9 | a | 503 | 231 | c | 27.2 | 3.5 | c | 90 | 13 | c |
| Trans Dry-Wet | 55 | 91 | a | 530 | 297 | c | 27.2 | 3.7 | d | 89 | 17 | d |
| Statistical test, p | 1.000 | | | ≤ 0.001 | | | ≤ 0.001 | | | ≤ 0.001 | | |
| **18 m** | | | | | | | | | | | | |
| Wet | 39 | 12 | a | 282 | 175 | a | 25.2 | 1.9 | a | 65 | 9 | a |
| Trans Wet-Dry | 44 | 54 | a | 351 | 123 | b | 25.4 | 2.3 | b | 64 | 4 | b |
| Dry | 41 | 13 | a | 412 | 190 | b | 26.5 | 2.9 | c | 63 | 17 | c |
| Trans Dry-Wet | 37 | 28 | a | 185 | 109 | c | 26.6 | 3.0 | d | 77 | 45 | d |
| Statistical test, p | 1.000 | | | ≤ 0.001 | | | ≤ 0.001 | | | ≤ 0.001 | | |
| **8 m** | | | | | | | | | | | | |
| Wet | 31 | 26 | a | 144 | 194 | a | 24.9 | 1.1 | a | 51 | 12 | a |
| Trans Wet-Dry | 66 | 88 | a | 165 | 218 | a | 24.9 | 1.4 | b | 47 | 12 | b |
| Dry | 23 | 16 | a | 295 | 268 | b | 26.0 | 2.1 | c | 45 | 28 | c |
| Trans Dry-Wet | 21 | 47 | a | 269 | 178 | b | 26.3 | 2.5 | d | 61 | 55 | d |
| Statistical test, p | 1.000 | | | ≤ 0.001 | | | ≤ 0.001 | | | ≤ 0.001 | | |
| **1.5 m** | | | | | | | | | | | | |
| Wet | 4 | 15 | a | 114 | 224 | a | 24.9 | 1.0 | a | 403 | 203 | a |
| Trans Wet-Dry | 2 | 12 | a | 61 | 102 | a | 24.6 | 1.1 | b | 227 | 61 | b |
| Dry | 6 | 25 | a | 209 | 299 | b | 25.5 | 1.7 | c | 202 | 68 | c |
| Trans Dry-Wet | 4 | 20 | a | 107 | 113 | b | 26.0 | 2.1 | d | 299 | 238 | d |
| Statistical test, p | 1.000 | | | ≤ 0.001 | | | ≤ 0.001 | | | ≤ 0.001 | | |





**Table 3:** The potential time ranges [%], during which the epiphytic bryophytes reached the compensation and saturation points of light (PAR), the optimal temperature for net photosynthesis ($T_{opt}$), the upper compensation point for temperature ($TCP_u$), the compensation and saturation points of water content (WC), and the potential time range, during which NP and DR occurs (see methods section for details on calculation of NP and DR. Values

5    are given for the different height levels. Five-minute averages of measurements during the entire measurement period from October 2014 to December 2016 were considered. The ranges of the compensation (CP), saturation (SP) point and optimal ranges (opt) were reported in Frahm (1990), Zotz et al. (1997), Leon-Vargas et al. (2006), Romero et al. (2006), and Wagner et al. (2013) (see Table S2).

| Height | Time when cardinal points are reached [%] | | | | | | | |
|---|---|---|---|---|---|---|---|---|
| **[m]** | **$LCP_l$** | **LSP** | **$T_{opt}$** | **$TCP_u$** | **$WCP_l$** | **WSP** | **NP** $WC > WCP_l$ & $PAR > LCP_l$ & $T < TCP_u$ | **DR** $WC > WCP_l$ & $PAR < LCP_l$ or $WC > WCP_l$ & $T > TCP_u$ |
| | 5-69 $\mu mol\ m^{-2}\ s^{-1}$ | 110-400 $\mu mol\ m^{-2}\ s^{-1}$ | 25.0-27.3 ° C | 30.0-36.0 ° C | 30-225 % | 349-1053 % | | |
| **23** | 8-43 | 1-4 | 6-30 | 1-16 | 0-100 | 0 | 0-55 | 0-43 |
| **18** | 6-45 | 0-3 | 6-32 | 0-13 | 0-100 | 0 | 0-59 | 0-41 |
| **8** | 4-37 | 1-2 | 13-29 | 0-17 | 0-100 | 0 | 0-47 | 0-40 |
| **1.5** | 0-8 | 0 | 14-30 | 0-11 | 25-100 | 1-22 | 0-8 | 50-96 |





**Figures**

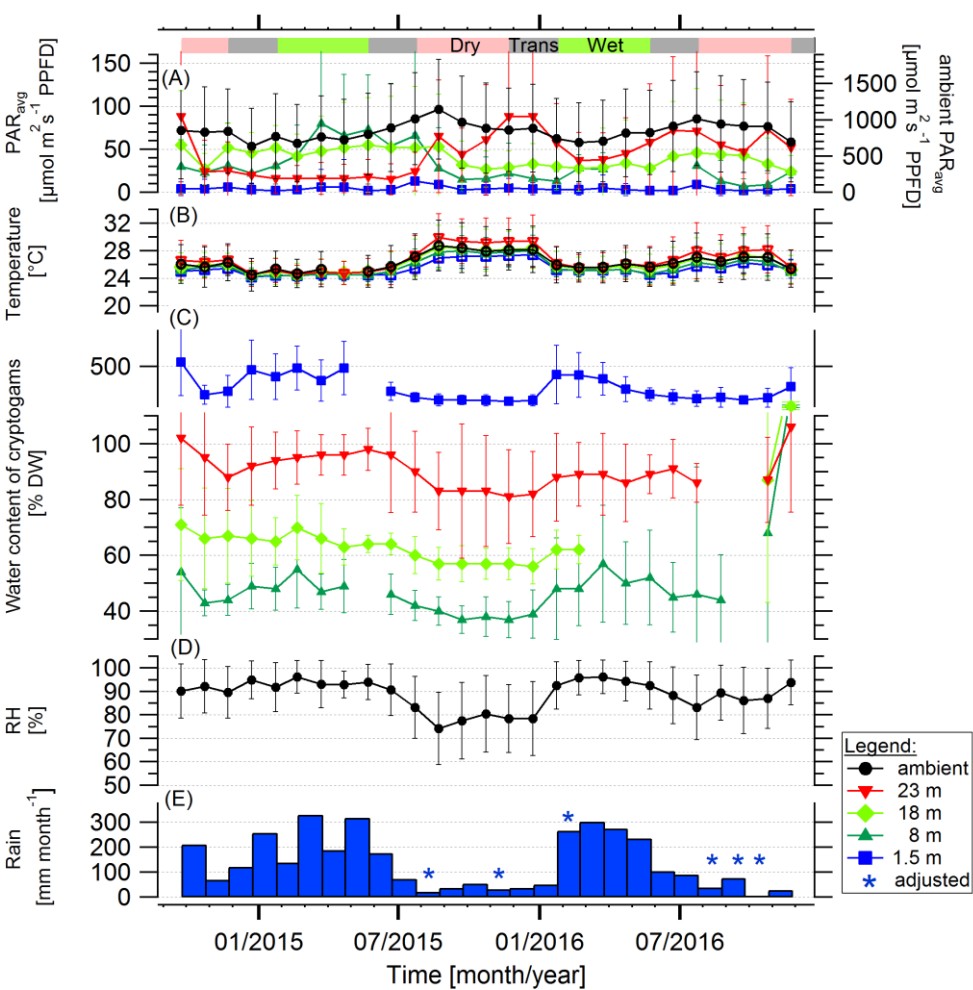

**Figure 1:** Ecophysiological, micrometeorological, and ambient meteorological conditions experienced by epi-
phytic bryophytes in the Amazonian rain forest. The ecophysiological and micrometeorological parameters on

5   top/within epiphytic the cryptogamic communities represent monthly mean values ± SD of (A) average by day
(06:00 – 18:00 LT) of photosynthetically active radiation (PAR$_{avg}$) on top, (B) temperature within, and (C) water
content of cryptogamic communities. The ambient meteorological parameters comprise (A) the monthly mean
value of the average by day(06:00 – 18:00 LT) of ambient photosynthetically active radiation (PAR$_{avg}$ at 75 m),
(B) monthly mean value of ambient temperature (at 26 m), (D) monthly mean value of relative air humidity (RH

10  at 26 m height), and (E) monthly amount of precipitation (rain). Data of replicate sensors installed within commu-
nities at the same height level were pooled, while ambient parameters were measured with one sensor each. Colored
horizontal bars in the upper part of the figure indicate the seasons. Exact values and additional data are presented
in Tables S3 and S4.





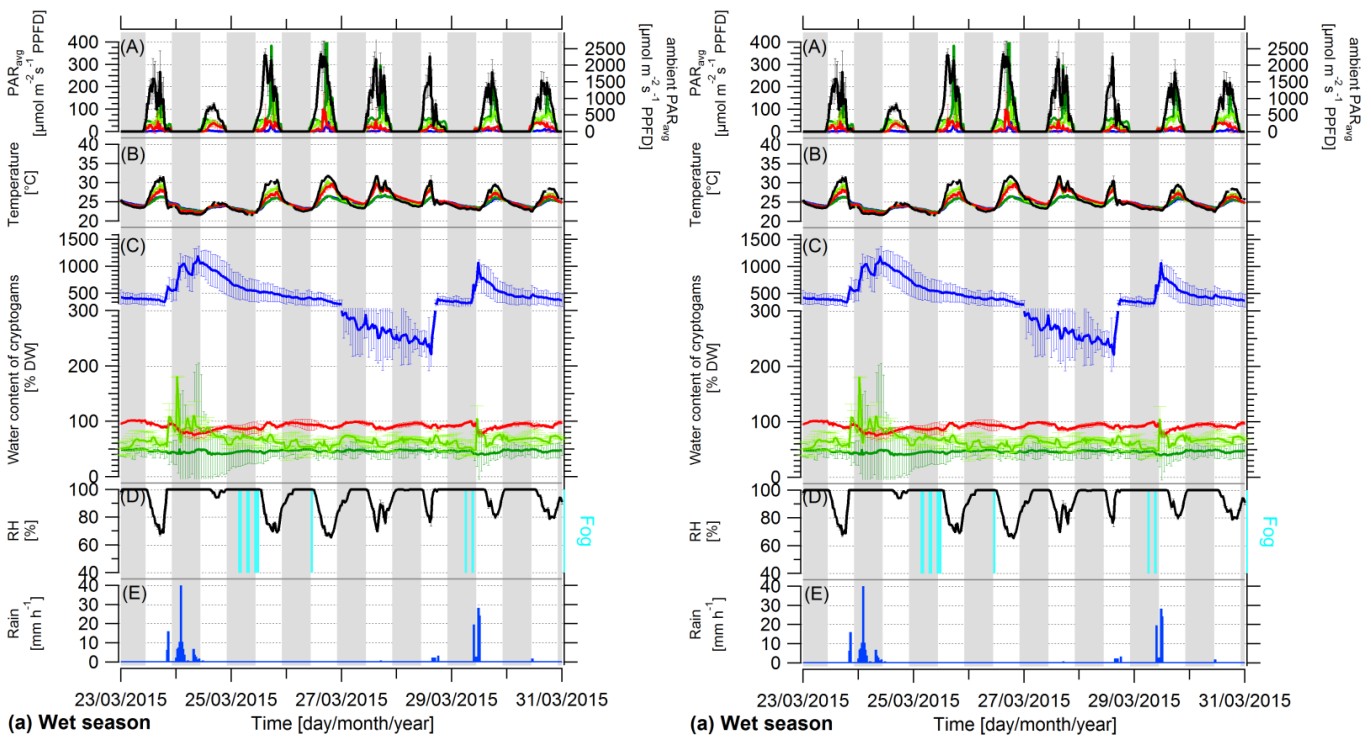

**Figure 2**: Representative periods during wet and dry season under average conditions showing ecophysiological, micrometeorological, and ambient meteorological conditions experienced by epiphytic cryptogamic communities in the Amazonian rain forest. Shown are 8-day periods during (a) the wet season 2015 and (b) the dry season 2016. Ecophysiological and micrometeorological parameters on top/within epiphytic cryptogamic communities represent (A) the photosynthetically active radiation (PAR$_{avg}$) on top, (B) the temperature within, and (C) the water content of cryptogamic communities. The ambient meteorological parameters comprise (A) ambient photosynthetically active radiation (PAR$_{avg}$ at 75 m), (B) ambient temperature (at 26 m), (D) relative air humidity (RH at 26 m height), presence of fog events (turquoise bars), and (E) precipitation (rain). The data show 30-minute averages ± SD except for rain. Data of replicate sensors installed within communities at the same height level were pooled, while ambient parameters were measured with one sensor each. The night time is shaded in grey (06:00 – 18:00 LT).



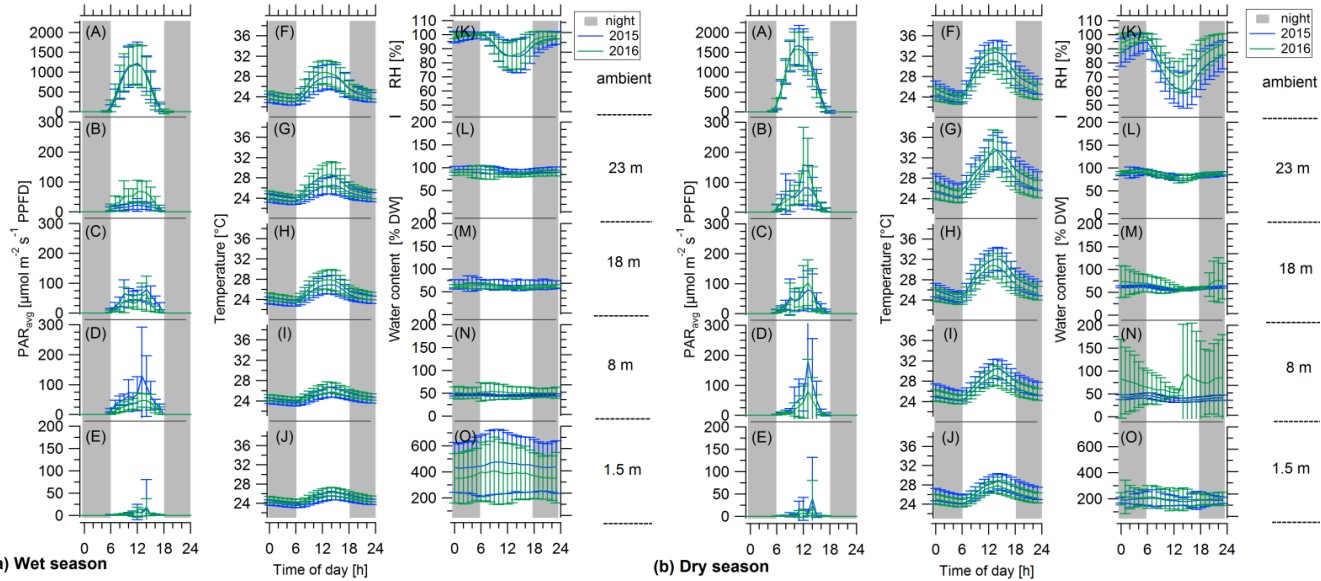

**Figure 3**: Mean diurnal cycles of ecophysiological, micrometeorological, and ambient meteorological parameters during (a) wet season and (b) dry season of the years 2015 (blue lines) and 2016 (green lines) based on 30-minute intervals. The ambient meteorological parameters comprise (A) the ambient photosynthetically active radiation (PAR$_{avg}$ at 75 m), (F) the ambient temperature (at 26 m), and (K) the relative air humidity (RH at 26 m height). Ecophysiological and micrometeorological parameters measured on top/within epiphytic cryptogamic communities comprise (B − E) the photosynthetically active radiation (PAR) on top, (G − J) the temperature within, and (L − O) the water content of cryptogamic communities at different height levels. Diel cycles were calculated from whole seasons and show hourly mean values ± SD. Data of the sensors installed at the same height level were pooled, while the ambient parameters were measured with one sensor each. Nighttime is shaded in grey (06:00 − 18:00 LT). Statistical comparisons of maximum and minimum values and diel amplitudes of light, temperature, and humidity between seasons are shown in Table S6 − S8. Comparisons among height levels are presented in Table S9 − Table S11.



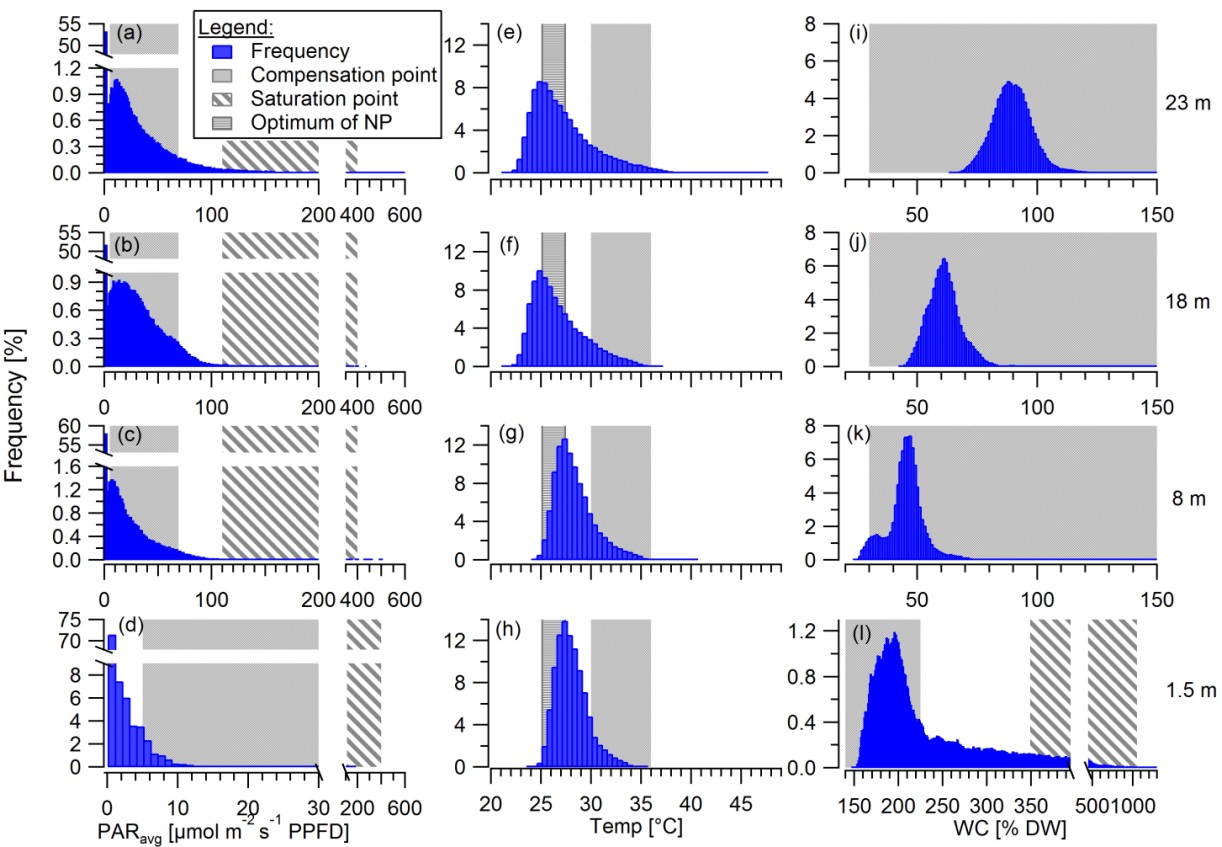

**Figure 4:** Frequency of photosynthetically active radiation (PAR$_{avg}$; a – d), temperature (e – h), and water content (i – l) measured on top/within bryophytes at 1.5, 8, 18, and 23 m height within the canopy based on 30-minute intervals. Grey areas represent the ranges of lower (PAR, water content) and upper (temperature) compensation, optimum net photosynthesis is shown with horizontal and saturation with inclined hachure. Value ranges are adopted from Zotz et al., (1997), Leon-Vargas et al., (2006), Romero et al., (2006), and Wagner et al., (2013) (Table S2). Bin sizes: PAR: 1 µmol m$^{-2}$ s$^{-1}$ PPFD; temperature: 0.5 °C; WC: 1 %.