# Peer review of "Microclimatic conditions and water content fluctuations experienced by epiphytic bryophytes in an Amazonian rain forest"

_Biogeosciences, 2018_

## Author Comment (AC1) · 23 Jan 2019

Dear readers,

in the current discussion version of the manuscript Fig. 1 and Fig. 2 are incorrect. Please considere the here uploaded versions of these two figures as reference in the manuscript.

Many thanks for your understanding. We highly apologize this inconvenience. Kind regards, Nina Löbs

[Figure]

[Figure]

Fig. 1.

[Figure]

**Fig. 2.**

Legend:
- ambient (black)
- 23 m (red)
- 18 m (light green)
- 8 m (dark green)
- 1.5 m (blue)
- fog (cyan)
- rain (blue)
- night (grey)

(a) Wet season — Time [day/month/year]

(b) Dry season — Time [day/month/year]

---

## Referee Comment (RC1) · Maaike Bader (Referee) · 10 Feb 2019

1. Does the paper address relevant scientific questions within the scope of BG? yes

2. Does the paper present novel concepts, ideas, tools, or data? Yes (data)

3. Are substantial conclusions reached? No, at least no supported ones.

4. Are the scientific methods and assumptions valid and clearly outlined? They may be valid but should be outlined better (uncertainty evaluation, parameter selection, statistical methods)

5. Are the results sufficient to support the interpretations and conclusions? No, con-

clusions needs to be downtuned.

6. Is the description of experiments and calculations sufficiently complete and precise to allow their reproduction by fellow scientists (traceability of results)? Of the experiments yes, of the calculations (statistics) no.

7. Do the authors give proper credit to related work and clearly indicate their own new/original contribution? Literature citations can be improved.

8. Does the title clearly reflect the contents of the paper? Almost. . .

9. Does the abstract provide a concise and complete summary? Yes.

10. Is the overall presentation well structured and clear? Yes.

11. Is the language fluent and precise? Fluent yes, could be more precise.

12. Are mathematical formulae, symbols, abbreviations, and units correctly defined and used? Yes, but some are unnecessary.

13. Should any parts of the paper (text, formulae, figures, tables) be clarified, reduced, combined, or eliminated? Yes.

14. Are the number and quality of references appropriate? Yes, but some need to be cited differently in the text.

15. Is the amount and quality of supplementary material appropriate? Yes.

Dear authors,

The manuscript "Microclimatic and ecophysiological conditions experienced by epiphytic bryophytes in an Amazonian rain forest" presents interesting data about the microclimate experienced by epiphytic bryophytes in a tropical rainforest, as well as unique measurements of the time these organisms stay wet. Such data is indeed very valuable for understanding the distribution and ecophysiological behaviour of such mosses and liverworts. The data are well-presented graphically at different time scales,

showing seasonal and diel patterns. There are some issues about the presentation of the interpretation that need addressing though, as explained below.

General issues:

It is clear that is a great effort to measure such data in a rain forest environment and the difficulty of canopy access. Because of this, and because of the absence of comparable data, the lack of replication (all samples were located close together on one stem or branch section per height on the tree) can be 'forgiven', but it should be mentioned and evaluated in the text!

I am also very aware of the almost complete lack of basic ecophysiological data on gas exchange in tropical lowland bryophytes, data being available for only 6 species, presented in Wagner et al 2013. However, I do not think that this justifies using data from tropical montane forest species, especially not for temperature responses, which differ along elevation (as shown in the cited paper by Wagner et al), but also not for water content responses, because montane species experience very different water regimes and are likely to employ different strategies concerning the preservation and use of their water contents – that is to say, the 'community weighted mean' of the strategies is likely to be different. I do think that it is a valuable exercise to estimate activity times for net photosynthesis and net respiration, but I think the lack of physiological data to base this estimation on needs to be dealt with differently. Some of the cited parameters (which are from montane species) are so unlikely (like a lower activity level for water content of 225%...) or uncertain (note that in Wagner et al it is explicitly mentioned that the absolute carbon exchange values should be treated with caution because of uncertainty in the absolute carbon exchange rates measured. This is not a problem for the optimum ranges (T and WC), but it is a problem for the compensation points, to which your calculation is highly sensitive. I would recommend to use only the lowland data and to use these data more loosely, using them combined with your common sense to estimate (or select) likely parameter values and presenting only theoretical calculations like " if we assume that the LCP is 6 umol/m2/s, the total A and Rd times

would be x and x% of the time, whereas a LCP of 1 umol/m2/s would aloe net A x% of the time". This is not fundamentally different from your current presentation, but you could avoid having to present estimations of 0-100%, which are not very helpful, and it would acknowledge the fact that gas exchange data for lowland species are simply not sufficiently available to really allow the type of estimates you would like to make at this point.

- Considering my previous point this one may be obsolete now, but it is not clear how the parameters in table 3 and S2 or those presented in L17-18 P9 were selected from Wagner et al 2013. Also, a 'water content compensation point' was not presented in Wagner et al although the paper is cited for it.

- Also, a lot of the statements about 'tropical bryophytes' are supported by literature from montane forests, and a lot of the statements about 'epiphytic cryptogams' are based on literature on lichens. This is not wrong but it is a bit deceiving. There would be nothing wrong with emphasizing, not only at the end of the discussion but right up front, that very little data is available for tropical lowland bryophytes and that therefore you need to rely on quite a bit of rough guessing and extrapolation of results from other areas and other organisms. As long as you make clear what your limitations are, they can be dealt with.

- So: make clear what literature is about lichens and what is about mosses – although these organisms have ecophysiological similarities, they are not the same in all respects! For example, enthanolic fermentation and bioaerosols have been observed for lichens but not for bryophytes, or am I wrong?

- And: be very careful, and be explicit about it, with using parameters and process knowledge based on montane forests and on lichens.

- Water content can hardly be called 'ecophysiological conditions', I would recommend removing this term from the title. To make sure that the innovative data on water content are in the title, you could consider changing it to "Microclimatic conditions and watercontent fluctuations experienced by epiphytic bryophytes in an Amazonian rain forest"

- The statement "Our data suggest that water contents are decisive for overall physiological activity, and light intensities determine whether net photosynthesis or dark respiration occurs, whereas temperature variations are only of minor relevance in this environment." In the abstract, and the statement that 'water content has turned out to be key' is not justified by your results. It is probably the case, but this is not suggested by your data – it could not be and was not addressed in your study, as realistic data about gas exchange is missing.

- There is a lot of information in the methods section that is superfluous or irrelevant, whereas other information is missing. Superfluous/irrelevant: P4 L 24-26, 29-32; P5 L13-15; Equations 5-8; P6 L20 brand name of styrodur.

- There is basically no information about the statistical analyses other than in what software they were performed. . . Please explain what was tested, what were your units of replications, etc.

- I am a bit afraid that you have used days as replications to compare climatic variables between years – is 26.6° really different from 26.4 °C, or even 25.8° is different from 25.8° (Table 1)?? With enough (pseudo)replication any tiny difference can become 'significant', but that does not make it real. . .

- Please present your experimental design (what species, what positions, justification for the pseudoreplication), preferably early in the methods section.

- It was not clear whether you used the 5-minute resolution data for calculating the times for A and Rd, or whether you only used the half-hour smoothed data. The smoothed data are fine for studying seasonal differences, but for the activity times and for quantifying the frequency of sun flecks (which would be interesting to do!) I would recommend using the 5-minute data. You mention that the conductivity showed 'short-time oscillations' - could these be explained physically? Were they regular fluctuations or

just general instability?

- Limitations should not only be acknowledged for the availability of gas-exchange parameters, but also, and early in the manuscript, for the measurements themselves. In particular, the quality of the WC calibration curves could be a problem. The calibration graphs show that there is indeed great variation between samples and between measurements, and that the models do not reflect the water contents very well even for the calibration data. As an example for the variability, the curves show that a conductivity of 800 mV (why is conductivity expressed in mV?? Should this not be in Ohm?) in Symbiezidium could be caused by a water content anywhere between 300 and 1700 %. What is the effect of this uncertainty on your results? For Octoblepharum the model underestimates the WC over much of the range (can this explain the low WC at 8 m?). For Sematophyllum the maximum conductivity measured in the field greatly surpasses the maximum values measured during calibration, which will, by the looks of it, results in a very high estimated water content even with the exponential correction. Why are these models not drawn for the whole range of measured conductivities? For example, the quadratic function for Leucobryum would mean that a very high conductivity, like the 1000 observed in the field, would indicate a lower WC than intermediate values. If you do not draw the whole curve, this potential artifact cannot be evaluated well.

- Also, the observation that water saturation was never reached at the 3 higher levels seems to suggest that something was wrong either with your WC measurements or the literature parameters used... BUT, this statement (P13, L24) cannot be true based on your data, because Symbiezidium is present only in these three higher levels, and in the calibration curves you show that observed values go up to 1500% WC, which is well above the WSPs cited....

- It was unclear to me what "upper three height levels the bryophyte taxa could not be securely determined. Thus, the bryophyte taxon with the highest abundance in the canopy communities, i.e., the liverwort Symbiezidium barbiflorum was used" means exactly. Did you install sensors only in this species, or did you do the calibration curve

only for this species and then use if for all the different (unidentified) species sampled at the higher height levels? This should be made clearer. I could imagine that you installed sensors in other liverworts looking similar to Symbiezidium and then assumed that the relationship between electrical conductivity and water content should not be more different between species than within species, due to the similar life form. This seems a reasonable assumption, but should be made explicit, and in table S1b the species should not be named if you do not know the real name. Indicating if it was a moss or a liverwort, or the family it belongs to, would be useful though!

- The use of different species at the different heights is a problem that also needs to be discussed earlier and more prominently and included in the analysis. It reads all through the manuscript as though differences in water content between height zones were caused by microclimatic differences, but of course a Leucobryum (cushion moss with specialised water-holding cells) is going to have very different water content dynamics that a Symbiziedium (prostrate leafy liverwort), even under the same environmental conditions. This is also obvious from your own data in the calibration curves, the points for Leucobryum being much closer together, inciating that the drying was much slower than e.g. for Symbiezidium. For Octoblepharum the two (! Looks like they were only two though you write they were three) samples dried at quite different speeds, it looks like the slow sample was denser and thus had higher conductivity at similar water contents. At the moment, the whole manuscript reads a bit as though you consider all cryptogams are expected to respond more or less the same, but we know that there are big differences between species, in particular in terms of water-content dynamics as well as the responses to this water content. Although you do mention this briefly, I think it deserves a few more words at least.

- It would be really cool if you could detect a dew signal in the WC data, did you look for this? Mention this in the discussion to but the dew remarks into the context of your data.

- It would also be cool if you could detect relationship between cryptogam activity patterns and measured trace gas emissions – this tall canopy site would be one of the few places in the world where the needed data might be available, assuming that trace gases above the canopy are also monitored?

- The literature cited needs to be revised! Only few bryophyte papers are cited and often they are not the correct ones (see below)! Some examples:

p. 3, lines 15-16: Zotz et al 1997 is cited a lot but refers to a montane forest, and not to nutrient cycling, as suggested on this occasion.

p.8, lines 30-31: 'at least in the environment of the central Amazon' is followed by references out of which none are from the central Amazon, most are from cloud forest... (by the way, this sentence is more or less repeated on page 12, L 29-31)

p. 9, lines 5-6: 'For tropical species, values (of WCPl) in the range 5 between $\sim$ 30 and $\sim$ 225 % have been determined (Romero et al., 2006; Wagner et al., 2013; Zotz et al., 1997, 2003)' Again, these references are all from montane species or do not mention WCPl values at all.

p. 3, lines 10 - 12: ...."Thus, the bryophyte taxon with the highest abundance in the canopy communities, i.e., the liverwort Symbiezidium barbiflorum was used (Gradstein and Allen, 1992; Mota de Oliveira et al., 2009; Mota de Oliveira and ter Steege, 2015; Pardow et al., 2012; Romanski et al., 2011; Sporn et al., 2010)." Of the 6 references cited here, B. barbiflorum is only mentioned in Gradstein and Allen (1992), the other 5 references do not cite this species at all! (one of the papers cited, Sporn et al. 2010, even deals with Asia even though S. barbiflorum does not occur there, being restricted to America...). Interestingly, Gradstein and Allen (1992) state that S. barbiflorum is a characteristc shade epiphyte of forest understory communities, not canopy communities. Not-cited more recent publications on the habitat of S. barbiflorum, however, show that the species also occurs in the forest canopy (Gradstein et al. 2001, Gradstein 2006, Gradstein & Ilkiu-Borges 2009, Gehrig et al. 2013). These recent papers show that S. barbiflorum is actually an ecological generalist, occurring in understory
communities as well as in canopy communities. None of these non-cited papers document highest abundance of the species in canopy communities. Thus, the sentence on p. 3, lines 10-12, is rather wrong.

p. 3, line 12-13: "In 2013, 800 species of mosses and liverworts ..,... have been reported for the Amazon region" (..... Mota de Oliveira & ter Steege 2013). The reference cited here is quite wrong, Mota de Oliveira & ter Steege did not provide this number at all, instead they took it from Gradstein et al. (2001; correctly cited by Mota de Oliveira & ter Steege) who calculated 800 species in the Amazon region in their book based on a full-scale analysis of the bryophyte flora of the Neotropics. Thus, the correct reference here is Gradstein et al. (2001) and not Mota de Oliveira & ter Steege.

Data availability: does this local database assure future data maintenance and retrieval? Please provide more details.

Detailed suggestions, including a few technical but also many conceptual points:

General: rather than 'mesoclimate', 'above-canopy climate' would be a more intuitive name for those measurements. P3 L 9: instead of 'these' write 'such' (this is an example of the confusing mix of literature and statements about cryptogam communities in general (often based on soil crusts..) and on tropical lowland epiphytes. P3 L 21: careful, not all bryophytes are desiccation tolerant, even if they are poikilohydric P4 L4-6 Add that most of this info is based on data from soil crusts and from temperate zones and that very little is known about biomass and functions of epiphytic cryptogam in tropical forests, especially in the lowlands. P4 L 8: seasonal variation in what? P5 L2: why 'ecophysiological' water content? What other water content is there? P5 L3: use 'were' rather than 'are being', even if the measurements are continuing, because you are here presenting results of a specific period in the past. Same for P5 L 11: were taken (not have been taken) P5 L 5: instead of 'described by' use 'used by', because 'described' suggests that these zones were the output of a study, but it was the sampling design. P5 L8: a cushion is a specific bryophyte life form, seeing your species

the samples probably were not cushions in most cases... You could use 'bryophyte samples' P5 L 19: what do you mean with 'fluctuations'? P6 L17: are nutrient content and temperature species-specific? P7 L1: what is the sensor weight? P7 L12: rather that presenting the models, which are very standard (except maybe for the exponential correction; if you want you could show the models in the appendix), a discussion about uncertainty propagation would be fitting here. P8 L16: rainfall amounts would usually not be calculated by integration but by adding the rain amount (e.g. number of tipping events) per time period... L8 L26: explain 'UTC values'; and where are such times presented, and why not always use local tiem? P9 L4-5: this WCPI is not what you describe it to be (this would not be a compensation point), it is the point below which the WC is so low that photosynthesis cannot compensate respiration, respiration ceasing at lower WCs than photosynthesis. P9 L28: with 'we found' you mean 'we assumed'? P10 L17: report the statistical results (test and test statistics)! This goes for all 'significant' (or non-significant) results. P11 L1: 'The RH..'What RH? It is generally not always clear in the text what parameter you are talking about: daily means, monthly means, something else? P12 L25: word missing P13 L16-18: it would be relevant to mention whether such high temperatures were ever reached in wet bryophytes; I would expect that they would only occur while samples were dry. P13 L 27: I guess you mean the LOWER end of the WCPI range? P14 L6: you mean 'height', not 'altitude' here. P14 L6-7: 'The microclimatic conditions experienced by bryophytes along an altitudinal gradient at the ATTO site follow the meteorological characteristics to some extent' - this needs some reference to time... P14 L15-17: mention in methods P14 L18: ' may have periodically shaded the organisms': it seems to me that you can have observed whether this was the case or not: were any leaves situated close to these sensors? (Same for P16 L7-8) P14 L20: was PARavg not the monthly average? Do you mean the monthly averages of the daily patterns? P15 L9-15: is could indeed be expected and is not very exciting. Your contribution here should be discussing the differences in temperature fluctuations quantitatively. P15 L17-18: mention this reinstallation in the methods too. P15 L21-22: mention and discuss this earlier on. P15 L33-34: Is the

canopy so open that the wind direction is noticed at 8 m height? Why did you choose the west side, I would expect you to select a side with goo moss cover. Interesting if this happened to be the west side if this side receives less moisture. Can you explain this? P16 L11: why does a light rain facilitate drying?? P16 L17: this has at best been estimated, and please specify what you mean by 4%: 4% of water input for bryophytes (or other epiphytes?), or just comprising (thus not 'providing') 4% of total precipitation? P16 L22: the water holding capacity is not what you have been measuring... Otherwise, this sentence is very true: the high water contents may be due to the high water-holding capacities of these species. P17 L13-14: be careful with your wording: understorey species are probably more efficient at low light (lower LCP), but it would be weird if they had a higher potential photosynthesis. P17 L19-20: words missing P17 L22: It may be worth mentioning that Wagner et al 2013 concluded that, although respiration losses may be high, this in itself does not explain low bryophyte growth in tropical lowlands, because respiration rates are adapted or acclimatized to the prevailing temperature conditions: in mosses growing at higher elevations the respiration rates are higher at the same temperatures, but still epiphytic bryophyte biomass is much higher here. P18 L4: another example of a mismatch between cited literature and interpretation: you suggest that it is relevant that water contents in Zotz et al 1997 were measured during the same time of the year, but as this was a different region and a very different forest type, this temporal coincidence has no meaning whatsoever! P18 L13-14 ' whereas in the canopy, rain events, fog, and condensation seem to be equally important water sources for cryptogams.' What do you base this conclusion on?? P18 L16: what does 'which' refer to? The reference seems strange here. (Figure 2: the wet season data are shown twice, the dry season data are missing! A legend is also missing.) → already corrected by authors Figure S2: in what way are these integrals? Do you mean interpolations? Supplement: P4 L7: looks like 2 replicates for Octoblepharum

References: Gradstein, S.R., Churchill, S.P. & Salazar A., N. 2001. Guide to the Bryophytes of Tropical America. Memoirs New York Bot. Garden 86: 1-577. Gradstein, S.R. 2006. The lowland cloud forest of French Guiana – a liverwort hotspot. Cryptogamie, Bryol. 27: 141-152. Gradstein, S.R. & Ilkiu-Borges, A.-L. 2009. Guide to the Plants of Central French Guiana. Part IV. Liverworts and Hornworts. Memoirs of the New York Botanical Garden 76, 4: 1- 140, 83 plates. Gehrig-Downie C., Obregón A., Bendix J., Gradstein S.R. 2013. Diversity and vertical distribution of epiphytic liverworts in lowland rain forest and lowland cloud forest of French Guiana. Journal of Bryology 35: 243-254.

---

## Referee Comment (RC2) · Anonymous Referee #2 · 15 Feb 2019

The authors provide a description of bryophyte occurrence and microclimate in a tropical forest canopy. These data are scarce and therefore crucial for a variety of applications that the authors list at various times in the manuscript. However, at present, the manuscript suffers from several issues. First amongst these are poor organization and a general lack of coherence. Facts about bryophytes (such as they are poikilohydric) are repeated often. No clear hypotheses or research questions are outlined. The introduction tells us that bryophytes are 'cool' and important to study but doesn't do a good job of setting up the study itself. Until the end of the methods section, I didn't realize that gas exchange measurements were not performed (something that is mentioned in abstract- If gas exchange in epiphytes is essential, why did the authors not make these

measurements?).

While I am quite satisfied by the measurement protocols and methodology (and that the epiphyte wetness-drying data are novel and important) the study ends up being merely a data reporting exercise with conclusions that often seem unsubstantiated by the data that are presented. Other times conclusions are trivial. For instance, Pg 18, lines 18-23 it is suggested that it is dark in the understory and therefore photosynthesis is light limited. I do not think that today one needs to go to the Amazon to make this conclusion, as this has been known for decades (for e.g. read classic reviews by Chazdon and Fetcher, 1984; Mooney et al., 1984). I seem facetious here, but the authors could use the same data to build upon these earlier findings, and find some nuance and/or insights. What is the knowledge gap that you are trying to fill with your measurements?

I want to be clear that I do not think that this work is unpublishable, rather a considerable amount of work needs to be done, especially in the writing, to ensure that it is. The advantage of the study is that the authors have collected a vast amount of important data, and there are several questions that can be formulated and answered. For instance, Fig S.5 is very interesting, and one could speculate about the significance of Tair -TCryptogram relationship in different parts of the canopy, and its significance to physiology. Another question could be the importance of light flecks, since you have carefully measured PPFD within the canopy. Fog is also measured but these data seem largely ignored (I wonder if you had leaf wetness sensors, those data could bolster the study tremendously). I would recommend the corresponding author to read some of the classic literature on epiphyte distribution and abundance (e.g. Benzing, 1984). With some more data exploration and thought I think this could be a very significant contribution. In its current form however, the manuscript reads like an early draft of a thesis or a dissertation chapter, and I do not see it fit for publication in Biogeosciences, or a journal of similar repute.

Finally, authors should provide data access via a link with a doi to a data repository. I

wonder If this is required of papers that are submitted to Biogeosciences.

Specific/Minor comments below:

The abstract is a bit long with too many technical or field specific terms that should be introduced (in the introduction), since it makes it difficult to comprehend for the general reader. An example is "While the monthly average mesoclimatic ambient light intensities above the canopy revealed only minor variations..." This is a well written but complicated sentence for the average reader. Please simplify.

Line 12: 1.5 m relative to what (i.e, please include canopy height). For the abstract something general, like 'near-surface' or 'in the understory' is more appropriate.

Line 13: instead of saying "low, exceeding less than 8%..." you could say low, remaining below 5 umol... more than 92% of the time.

Lines 18-19: Dark respiration should occur independent of light (and unless temperatures are very low, which seems unlikely at your site). The references to photosynthesis and respiration are repeatedly incorrect. Photosynthesis and respiration are co-occurring biological processes (in the light), and therefore one may dominate over the other.

Introduction

The first paragraph is a well written introduction to tropical forests, but has little do with the study. Either you should reframe it in the context of epiphytes or omit. Overall, the introduction does not set up the study satisfactorily.

Pg 3 Line 13: 'By' not 'In' 2013.

Pg 4 Line 5: Update references to carbonyl sulfide: (Gimeno et al., 2017; Rastogi et al., 2018).

Methods

Sec. 2.1. A greater description of the site is required. I would recommended starting with site characteristic and then describe the tower and measurements, not the other way around.

Pg 4 line 13: Remove "The".

Pg 5 line 2: "were measured" not "are being measured"

Line 5: this seems important for your study and you should describe why sensors were placed where they were placed, in addition to citing the Mota de Oliveira (2013) study.

Pg 6: line 30: Why 60 C? Is this a temperature that these communities experience?

Sec 2.5. Again, some information (a figure ideally) describing the vertical profile of the forest is necessary. That helps put the various sensor heights in perspective.

Lines 15-21: Why were rainfall values gap-filled? Also, isn't the sensor at 1.5 m the least well placed to record rain event. For instance, a small rain event might not even be recorded at 1.5m, as interception must be high in a high LAI forest. Alternatively, there could be time lags between when rainfall occurs at the canopy top and when it is measured by the 1.5 m ht rain gauge.

Line 32- Pg 9, line 1: Rephrase sentence. Light intensity regulates the balance i.e. the Net exchange between photosynthesis and respiration.

Line 14: Respiration takes place at all light levels. IT IS NOT A LIGHT DEPENDANT PROCESS (there can be significant inhibition of respiration at high light levels). Please check this basic tenet of biology.

Line 21: Based on literature, not literature data. Also, please cite the relevant literature.

Results: Overall, I do not have issues with the content per se, but as I have stated before this section needs to be majorly revised/expanded. Some minor comments below.

Pg 10

Line 8: Micromet did not depend on years but varied amongst years.

Line 18: please define mesoclimate the first time you use this term.

Pg 11

Line 30-31: What does this mean? Please elaborate.

Pg 12 Sec 3.1.3 header: remove parenthesis

Line 13: which 'organisms. Please specify.

Sec. 3.2. The scope of your inference is limited since you do not have replicates on different trees. I do not see this as limitation, but somewhere (probably in the discussion) you need to talk about heterogeneity in the microclimatic environment.

Lines 13-14. This has been mentioned previously (in Sec. 3.1.).

Lines 25 to Pg 14 line 3: This is a well written paragraph but belongs in the discussion.

Discussion: I am going to stop commenting here, since I think this section regurgitates a lot of information that is already presented. It is well-written, but I think that based on my previous comments, I anticipate this section to be revised extensively.

Tables:

Table 1: Why are these annual means presented? Why are light levels higher at 23 m than at 18m (again discuss heterogeneity)? I still don't have a good grasp of the canopy structure. I do not understand the sensor placement with respect to the canopy structure. Is the 23 m sensor above the canopy top ($\sim$ 21m)? Probably not, since light levels seem too low. Also, there seems to be some confusion between relative humidity and water content.

Table 2. There are no significant differences between seasons for some variables (for e.g., temperature), even though this is alluded to in the results (Pg. 11, line 24).

Figures: Generally, the figures need to be clearer, and larger, since you have several subplots.

Fig 2. PAR and Temperature at different heights are very hard to see. Either summarize differently, or show a mean in this figure and direct to a figure in the supplemental with data from all heights.

Figure 3. This also has too many sub-panels crammed in one figure. In the caption, why do you say ecophysiological, micrometeorological and ambient parameters (the same is actually true of Fig. 1 as well). Which ones are which? Why are they called parameters? What are you trying to parametrize? I make a point about this, because this is one of several instances where words are not chosen carefully. Was humidity not measured at all heights?

Figure 4. The histograms are informative but the information provided in the various shaded regions is extremely hard to follow. In the end, I do not understand what the authors are trying to convey. Why is the y-axis broken in the histograms in the left most panel?

References mentioned in the review:

Benzing, D. H.: Epiphytic vegetation: A profile and suggestions for future inquiries, in Physiological ecology of plants of the wet tropics, pp. 155–171, Springer., 1984. Chazdon, R. L. and Fetcher, N.: Light environments of tropical forests, in Physiological ecology of plants of the wet tropics, pp. 27–36, Springer., 1984.

Gimeno, T. E., Ogée, J., Royles, J., Gibon, Y., West, J. B., Burlett, R., Jones, S. P., Sauze, J., Wohl, S., Benard, C., Genty, B. and Wingate, L.: Bryophyte gas-exchange dynamics along varying hydration status reveal a significant carbonyl sulphide (COS) sink in the dark and COS source in the light, New Phytol., 215(3), 965–976, doi:10.1111/nph.14584, 2017.

Mooney, H. A., Field, C. and Vazquez-Yanes, C.: Photosynthetic characteristics of wet

tropical forest plants, in Physiological ecology of plants of the wet tropics, pp. 113–128, Springer., 1984.

Rastogi, B., Berkelhammer, M., Wharton, S., Whelan, M. E., Itter, M. S., Leen, J. B., Gupta, M. X., Noone, D. and Still, C. J.: Large uptake of atmospheric OCS observed at a moist old growth forest: Controls and implications for carbon cycle applications, J. Geophys. Res. Biogeosciences, 0(ja), doi:10.1029/2018JG004430, 2018.

Referee rubric evaluation:

1. Does the paper address relevant scientific questions within the scope of BG?: Not currently, but with significant revision.

2. Does the paper present novel concepts, ideas, tools, or data? Data yes, ideas and tools or concepts: no.

3. Are substantial conclusions reached? No.

4. Are the scientific methods and assumptions valid and clearly outlined? Yes.

5. Are the results sufficient to support the interpretations and conclusions? No.

6. Is the description of experiments and calculations sufficiently complete and precise to allow their reproduction by fellow scientists (traceability of results)? Yes.

7. Do the authors give proper credit to related work and clearly indicate their own new/original contribution? Literature needs to be expanded.

8. Does the title clearly reflect the contents of the paper? Yes.

9. Does the abstract provide a concise and complete summary? No.

10. Is the overall presentation well-structured and clear? No.

11. Is the language fluent and precise? No.

12. Are mathematical formulae, symbols, abbreviations, and units correctly defined

and used? Yes.

13. Should any parts of the paper (text, formulae, figures, tables) be clarified, reduced, combined, or eliminated? Yes (see detailed comments).

14. Are the number and quality of references appropriate? No.

15. Is the amount and quality of supplementary material appropriate? Yes.
* * *

---

## Referee Comment (RC3) · Maaike Bader (Referee) · 19 Feb 2019

An additional consideration: an alternative way to use the electrical resistance measurements

Dear authors,

After some more thought and discussion with some colleagues, with whom we will be installing a similar system to measure moss wetness, I would like to suggest using more caution in the translation of the electrical resistance to moss water content and to propose an alternative way of interpreting the measurements. This is giving away the

method we intend to use ourselves, which I think may be a good alternative for your study also. You are welcome to cite me for the idea if you think it appropriate.

It is clear that there is a very wide range of moss water-content (WC) values that may be indicated by any electrical resistance value measured. The values are more constrained for the cushion species (Leucobryum), which makes sense seeing that such a life form is denser and more homogenous than the other species, which are prostrate or consist of loosely scattered turf, if I am not mistaken. With such inhomogenous substrates, with different amounts of air and tissue between the probes for each sample, it is no wonder that the measured conductance is widely scattered within species. I think you should reconsider whether you should really try to deduct an absolute value of WC from these measurements. It looks like this is not really possible for most species.

It seems that the points within each calibration curves are nicely ordered, however. Therefore an alternative approach would be to only look at the changes in electrical conductivity, which should reliably indicate changes in water content. With this, you can deduct for any time period whether the samples were drying out or being wetted. When stable at low conductivity, this indicates that the samples are dry (in equilibrium with air humidity), when stable at high conductivity they must be completely wet during rain or fog events. If you have good data about the maximum water content of the species, you might even be able to interpolate between the stable low and the stable high, considering that drying tends to follow relatively smooth extinction curves, as you will see when plotting your calibration curves against time.

I hope this suggestion is of use.

---

## Referee Comment (RC4) · Anonymous Referee #3 · 25 Feb 2019

Dear Editor, dear authors

I have read with interest the manuscript entitled "Microclimatic and ecophysiological conditions experienced by epiphytic bryophytes in an Amazonian rain forest" by Löbs et al. submitted to Biogeosciences. Please find my comments related to it below:

I appreciate a strong point in this manuscript, that is to contribute to raise the data availability regarding cryptogamic covers functional performance in tropical regions, and going further, the lack of data available in Central and South America. It seems that almost all the literature regarding this issue has been focused in Polar Regions some years ago and in drylands at the present. I also appreciate the novelty and the

[Figure]

effort made to provide microclimatic data sets at those heights at the tree trunks. If we want to understand properly the relevance of these organisms in global cycles and their response under environmental changes a huge and very different biome as the tropics can not be ignored. I think that authors do a complete revision of the literature available and try to contribute from there with their data. Mosses dominate cryptogamic covers in tropical regions in biodiversity, so the target organisms in the study seems to be quite correct

But, at the same time, my opinion is that this lack of data availability in the region is an intrinsic weakness of the manuscript. My point here is that the manuscript is based in a double assumption rather than in strictly measured data sets. The first assumption would be the water content of the bryophytes through conductivity sensors. I appreciate the effort made by the authors calibrating this methodology in the lab and this experimental testing gives higher credibility to the measurements. But then we see the big second assumption, that is to extrapolate data taken from the literature to understand the functional performance of the bryophytes in the altitudinal gradient. I think that it is likely that possible inaccuracies could arise in this sense. Data available in the literature is little, so, it must be difficult to find similar experimental designs that could help providing reliable extrapolations. I am not talking about finding same species with data available in the literature, but it would be interesting, in order to trust the ecophysiological data provided, to have data from a similar habitat following at least the light adaptation patterns of the species included in this work.

As I supose that these data sets are very difficult to get, but I think that this manuscript is interesting and useful to the scientific community, I would make a proposal to the authors: What about to include in your manuscript a few gas exchange checkpoints in the lab including relevant species inside the gradient. For example, one representative species in the understory and another one at the closer point of the canopy could serve as cardinal points to calibrate authors' predictions about net photosynthesis availability, time and amount of respiration and possible C losses, light cardinal points, adaptation

strategies. This would improve the discussion substantially from my point of view

I am not asking for a complete gas exchange profile of the species included in the study because I know how time consuming this technique is, just a few replicated checkpoints in the lab to see how close predictions are from reality. If they were far from each other, the real gas exchange parameters measured could work as a more reliable source of predictions than a very likely imprecise literature for the aim targeted. I would welcome further assumptions at this point, but based in some real measured values (I said in the lab because conditions are easier to control, but some field gas exchange data sound good for me also). I think that this could improve the manuscript and put it as a reference text in tropical epiphytic bryophytes functional performance due to the low amount of literature available

Some minor points also to comment:

INTRO:

-Page 3, Ls 20-25: I would focus in bryophytes functional properties rather than in general physiological features of cryptogamic covers because only bryophytes are included in the experimental design

METHODOLOGY:

-Section 2.5. Could you please explain in more detail why some meteorological parameters are measured at 26m and light is measured at 75m?

-Section 2.6. I would establish the possible ranges for each ecophysiological parameter analyzed focusing more in tropical epiphytic bryophytes functional performance

RESULTS:

-Section 3.1. 2 consecutive years of microclimatic data availability is a good and interesting output provided by authors

-All sections in general. I see that the headings do not correspond too much with what

is written at each of the sections. Authors mix concepts in the same paragraphs such as microclimate, mesoclimate, water content, seasonal and daily analyses….Would it be possible to rethink the headings of the sections and write text more focused to each of the headings?

-Page 10 L9, I think that authors missed a word after "35%", maybe "lower"?

-How did authors compared climate statistically between years/seasons? Did you use a monthly basis? Daily basis?

-Page 10 Ls 25-26. If I understood ok, the idea is that the microclimatic T value at the moss level was higher than ambient T, and that this is a frequent pattern. What about the shading effect of the tree canopy over microclimatic T?

-Fig 1, legend. I would say estimated water content of the bryophytes rather than "ecophysiological conditions"

DISCUSSION:

-Page 14 Ls 22-24. I think that these patterns observed reinforces that measuring some gas exchange control points might be useful

-Page 17 Ls 19-23. I do not understand this point properly

---

## Author Comment (AC2) · 12 Jun 2019

**Response to referee comments and suggestions on bg-2018-521 by N. Löbs et al.: "Microclimatic and ecophysiological conditions experienced by epiphytic bryophytes in an Amazonian rain forest"**

**Manuscript format description:**

Black text shows the original referee comment, and blue text shows the response of the authors and the explicit changes in the text. The figure and table numbers refer to the revised manuscript.

**Maaike Bader as Referee submitted the comments RC1 and RC3**

**Maaike Bader RC1:**

**General Referee comment:**

Dear authors,

The manuscript "Microclimatic and ecophysiological conditions experienced by epiphytic bryophytes in an Amazonian rain forest" presents interesting data about the microclimate experienced by epiphytic bryophytes in a tropical rainforest, as well as unique measurements of the time these organisms stay wet. Such data is indeed very valuable for understanding the distribution and ecophysiological behavior of such mosses and liverworts. The data are well-presented graphically at different time scales, showing seasonal and diel patterns. There are some issues about the presentation of the interpretation that need addressing though, as explained below.

General author response:

We would like to thank Maaike Bader for the very constructive review and the helpful comments, which helped us to identify the critical aspects and to improve our manuscript.

Referee comment 1:

It is clear that is a great effort to measure such data in a rain forest environment and the difficulty of canopy access. Because of this, and because of the absence of comparable data, the lack of replication (all samples were located close together on one stem or branch section per height on the tree) can be 'forgiven', but it should be mentioned and evaluated in the text!

Author comment 1:

We fully agree that it would be preferable to install sensors on several different trees in order to have fully independent true replicates. However, as you stated correctly, it is a great effort to install and run microclimate measurements in such a rainforest environment. Thus, we installed several sensors at each height in order to cover the variability at least to some extent. We added some more information about the incomplete replications in the methods section to explain the limitation of the measurement setup.

Author changes in the text 1:

P 4 L22: " At each height level, six water content, two temperature, and two light sensors (except for 1.5 m with only one light sensor) were installed in different bryophyte species (Table S1a). Due to constraints in accessibility, all sensors had to be installed on one tree. Thus, we expect that the replicate sensors at each height are not be fully independent and thus the variability could only partly be covered by our current setup. "

Referee comment 2:

I am also very aware of the almost complete lack of basic ecophysiological data on gas exchange in tropical lowland bryophytes, data being available for only 6 species, presented in Wagner et al 2013. However, I do not think that this justifies using data from tropical montane forest species, especially not for temperature responses, which differ along elevation (as shown in the cited paper by Wagner et al), but also not for water content responses, because montane species experience very different water regimes and are likely to employ different strategies concerning the preservation and use of their water contents – that is to say, the 'community weighted mean' of the strategies is likely to be different. I do think that it is a valuable exercise to estimate activity times for net photosynthesis and net respiration, but I think the lack of physiological data to base this estimation on needs to be dealt with differently. Some of the cited parameters (which are from montane species) are so unlikely (like a lower activity level for water content of 225%...) or uncertain (note that in Wagner et al it is explicitly mentioned that the absolute carbon exchange values should be treated with caution because of uncertainty in the absolute carbon exchange rates measured. This is not a problem for the optimum ranges (T and WC), but it is a problem for the compensation points, to which your calculation is highly sensitive. I would recommend to use only the lowland data and to use these data more loosely, using them combined with your common sense to estimate (or select) likely parameter values and presenting only theoretical calculations like " if we assume that the LCP is 6 µmol/m2/s, the total A and Rd times would be x and x% of the time, whereas a LCP of 1 µmol/m2/s would allow net A x% of the time". This is not fundamentally different from your current presentation, but you could avoid having to present estimations of 0-100%, which are not very helpful, and it would acknowledge the fact that gas exchange data for lowland species are simply not sufficiently available to really allow the type of estimates you would like to make at this point.

Author comment 2:

Thank you very much for this helpful comment! It is correct that the data collected at montane rainforest sites are not suited for a comparison and thus we now refrain to the lowland forest data (location BT) given in Wagner et al. (2013).

For the light compensation point (LCP) we include another reference for bryophytes in lowland bamboo forests with values of 3-12 µmol s²s$^{-1}$ for the LCP (Lösch et al., 1994).

Furthermore, for the estimation of the duration of NP and DR we assume different values for the compensation points of the organisms at different height levels and we now also distinguish between mosses and liverworts, as we find that these organism groups respond differently. For the current manuscript we omitted the information on saturation points, as we found that the current data are not well-suited for an inclusion of these data.

Author changes in the text 2:

All tables, figures, and the values in the text have been adapted according to the revised calculations.

Referee comment 3:

Considering my previous point this one may be obsolete now, but it is not clear how the parameters in table 3 and S2 or those presented in L17-18 P9 were selected from Wagner et al 2013. Also, a 'water content compensation point' was not presented in Wagner et al although the paper is cited for it.

Author comment 3:

The data were extracted in the following way:

P9 L31-32: The lowest Topt for tropical bryophytes of 16.1°C was reported in Wagner et al. 2013 in Tab. 3, while the highest Topt for tropical bryophytes of 27.3°C was reported in the Supplement of Wagner et al. 2013. Unfortunately, we just now found out that the species listed in Tab 3 and the ones listed in the Supplement are not containing data for the same elevation levels.

As described above, we now only consider the BT site located at sea level (see Tab. 3 and Tab S3 below). Accordingly, we cite the optimum temperature range with 24°C limiting the lower and 27°C limiting the upper end of the range as reported in Tab. 3 of Wagner et al. (2013).

The lower and upper temperature compensation points "TCP" of 30.0 and 36 °C were reported in Wagner et al. 2013 Tab. 3 as the temperatures when "$T_{NP=DR}$" for the site BT at sea level

In Table S2, we found a typo where the higher TCP value was accidentally written as 33 instead of 36°C.

Data on the water compensation point were extracted from Fig. 1 in Wagner et al. 2013.

Author changes in the text 3:

The text and tables (Tab 3, Tab, S3) were adapted accordingly.

Referee comment 4:

Also, a lot of the statements about 'tropical bryophytes' are supported by literature from montane forests, and a lot of the statements about 'epiphytic cryptogams' are based on literature on lichens. This is not wrong but it is a bit deceiving. There would be nothing wrong with emphasizing, not only at the end of the discussion but right up front, that very little data is available for tropical lowland bryophytes and that therefore you need to rely on quite

5      a bit of rough guessing and extrapolation of results from other areas and other organisms. As long as you make clear what your limitations are, they can be dealt with.

- So: make clear what literature is about lichens and what is about mosses – although these organisms have eco-physiological similarities, they are not the same in all respects! For example, enthanolic fermentation and bioaerosols have been observed for lichens but not for bryophytes, or am I wrong?

10      - And: be very careful, and be explicit about it, with using parameters and process knowledge based on montane forests and on lichens.

Author comment 4:

Thank you for your comment; we are aware that lichens and bryophytes do not behave identically in all respects, although there are quite some similarities. Thus, in the revised version we stress whenever we use information on

15      lichens. When comparing our results with those from other studies, we specify the rainforest habitat and organism group.

Referee comment 5:

Water content can hardly be called 'ecophysiological conditions', I would recommend removing this term from

20      the title. To make sure that the innovative data on water content are in the title, you could consider changing it to "Microclimatic conditions and water content fluctuations experienced by epiphytic bryophytes in an Amazonian rain forest"

Author comment 5:

We agree with your comment on the 'ecophysiological conditions'. Your recommended change of the title is a

25      good solution and thus we adapted it.

Author changes in the text 5:

Title: "Microclimatic conditions and water content fluctuations experienced by epiphytic bryophytes in an Amazonian rain forest"

30      Referee comment 6:

The statement "Our data suggest that water contents are decisive for overall physiological activity, and light intensities determine whether net photosynthesis or dark respiration occurs, whereas temperature variations are only of minor relevance in this environment." In the abstract, and the statement that 'water content has turned out to be

key' is not justified by your results. It is probably the case, but this is not suggested by your data – it could not be and was not addressed in your study, as realistic data about gas exchange is missing.

Author comment 6:

Thank you for your comment. Yes, we indeed do not have $CO_2$ gas exchange data in this study. Nevertheless, we think that already the microclimate data by themselves and the calculations on potential activity patterns support our statement that water contents are highly relevant whereas temperatures are of minor importance for physiological activity. Nevertheless, we try to clarify these issues in the revised version of the manuscript.

Author change in the text 6:

Changes were made in the abstract and the discussion sections to clarify these issues.

Referee comment 7:

There is a lot of information in the methods section that is superfluous or irrelevant, whereas other information is missing. Superfluous/irrelevant: P4 L 24-26, 29-32; P5 L13-15; Equations 5-8; P6 L20 brand name of styrodur.

Author comment 7:

We agree to delete some information on the study site (P4 L 25-27), on the neighboring forest types (P4 L 30 - P5 L2), and the styrodur brand (P7 L2). On the other hand, we would like to keep the setup information on the logger enclosure (P5 L27-28), as we consider it as relevant for the reader. Regarding the equations 5-8, as the calibration was restructured, the Eq. 8 was now moved to the supplemental section.

Referee comment 8:

There is basically no information about the statistical analyses other than in what software they were performed… Please explain what was tested, what were your units of replications, etc.

Author comment 8:

We agree with the comment, there was not enough information regarding statistical analyses. This information will be provided in more detail in section 2.6 and in all tables presenting test results.

Author changes in the text 8:

P10 L11:" For all the statistical tests 5- or 30-minute averages of the data have been considered, as indicated in the respective results sections. Furthermore, for the average of height levels the data of the individual sensors were pooled. For the statistical tests of the diel maxima, minima, and amplitudes, the daily values of all sensors at the given height level were pooled for the season to be tested."

Referee comment 9:

I am a bit afraid that you have used days as replications to compare climatic variables between years – is 26.6° really different from 26.4° C, or even 25.8° is different from 25.8° (Table 1)?? With enough (pseudo)replication any tiny difference can become 'significant', but that does not make it real…

Author comment 9:

5 Thank you very much for that good advice. We thought about this test again and figured that one probably always obtains statistically significant differences if the 5-minute-values of two subsequent years are compared (as these are never the same in two subsequent years) and a comparison of the annual mean values does not make sense as we only have two years. Thus we decided to skip this test here."

10 Referee comment 10:

Please present your experimental design (what species, what positions, justification for the pseudoreplication), preferably early in the methods section.

Author comment 10:

More details (that have been included in the Supplement material) were now included in the text, section "2.2 15 Microclimatic conditions within epiphytic habitat". Furthermore we now add a Figure S2 to the Supplement showing the distribution of all the sensors along the vertical gradient.

Author change in the text 10:

P4 L32: "Generally, the water content sensors have been placed in four different bryophyte types being heterogeneously distributed along the four height levels. At the height level of 1.5 m the water content sensors were installed 20 in the moss species *Sematophyllum subsimplex* (5 sensors) and *Leucobryum martianum* (1 sensor), at 8 m in the species *Octoblepharum cocuiense* (3 sensors) and cf. *Symbiezidium barbiflorum* (3 sensors), at 18 and 23 m in the species cf. *Symbiezidium barbiflorum* (all 6 sensors; Fig. S2 and Fig. S3). The temperature sensors were installed in the same species at each height, and the light sensors were installed just next to the measured species. For the higher levels at 18 and 23 m, one species identification could not be verified (Tab. S1a, S1b).

25 Supplement: Figure S2 and Figure S3 (see below) were added to the supplement.

[Figure]

**Figure S2:** Schematic overview on the sensors installed at different height levels below, within, and above the canopy. The parameters water content (WC) and temperature (Temp) were measured within the bryophyte samples, the light sensors (PAR) were installed directly on top of the thalli. The average tree height of 21 m was determined for the plateau forest in general.

[Figure]

**Figure S3:** The four bryophyte species being used for installation of the sensors of the microclimate station. (A, D, G, J, K) overview, (B, H, L) leaf, (C, F I) cell form, and (E, M) cross section of a leaf.

Referee comment 11:

It was not clear whether you used the 5-minute resolution data for calculating the times for A and Rd, or whether you only used the half-hour smoothed data. The smoothed data are fine for studying seasonal differences, but for the activity times and for quantifying the frequency of sun flecks (which would be interesting to do!) I would recommend using the 5-minute data.

Author comment 11:

For a calculation of A and Rd, the 5-minute data were used, as written in Table 3. We additionally provide this information in the methods section on P9 L28.

Author change in the text 11:

P9 L28:"Based on the literature values, we utilized the *5-minute data to calculate the* ranges of timespans when these cardinal points were passed. "

Referee comment 12:

You mention that the conductivity showed 'short-time oscillations' - could these be explained physically? Were they regular fluctuations or just general instability?

Author comment 12:

The oscillations of the sensors represent a general instability of the system, as the measured values oscillated around the actual values. Accordingly, the 5-minute data set was only used for an estimation of the physiological activity, as here the information on short-term events (as e.g. light flecks) is needed. For all other calculations, the 30-minute averages were used.

Author changes in the text 12:

P8 L5: "The measured electrical conductivity showed short-time oscillations, which could be removed with a 30-minute smoothing algorithm (Fig. S4). *Thus, for all calculations the 30-minute averages have been considered, except for the estimates of physiological activity.*"

[Figure]

Figure S4: Comparison of 5-minute (dots) and 30-minute (lines) averages of exemplary sensors at each height level over a period of approx. one day in December 2016.

Referee comment 13:

Limitations should not only be acknowledged for the availability of gas-exchange parameters, but also, and early in the manuscript, for the measurements themselves. In particular, the quality of the WC calibration curves could be a problem. The calibration graphs show that there is indeed great variation between samples and between measurements, and that the models do not reflect the water contents very well even for the calibration data. As an example for the variability, the curves show that a conductivity of 800 mV (why is conductivity expressed in mV?? Should this not be in Ohm?) in Symbiezidium could be caused by a water content anywhere between 300 and 1700 %. What is the effect of this uncertainty on your results? For Octoblepharum the model underestimates the WC over much of the range (can this explain the low WC at 8 m?). For Sematophyllum the maximum conductivity measured in the field greatly surpasses the maximum values measured during calibration, which will, by the looks of it, results in a very high estimated water content even with the exponential correction. Why are these models not drawn for the whole range of measured conductivities? For example, the quadratic function for Leucobryum would mean that a very high conductivity, like the 1000 observed in the field, would indicate a lower WC than intermediate values. If you do not draw the whole curve, this potential artifact cannot be evaluated well.

The calibrations were conducted for the samples starting with full water saturation and values were recorded every 60 seconds until weight constancy was reached, indicating that the sample were dry, as explained on P7 L4-6 ("In the beginning of each calibration the sample was wetted to the water holding capacity and during drying of the sample, the values of the balance and sensor were recorded at 60-second-intervals."). In the case of *Leucobryum* and *Sematophyllum* the values of conductivity measured in the laboratory unfortunately were not been higher at full water saturation.

We are aware of the fact, that this represents a problem and we thought a lot about this issue. These measurements illustrate that there are large differences between bryophyte specimens and even within one specimen we observed differences between measurements. This is probably caused by the fact, that the structure of the thalli and e.g. dead material collected within them strongly influences the calibration values. However, as measurements continued after this study, the measured specimens could not be removed from the tree for the lab calibrations. Thus, also based on your reviewer comment 3, we decided to test and use an alternative approach for the calibration of the water content. For this, the maximum and minimum values of electrical conductivity reached in the field were assumed to be assessed at the maximum and minimum water contents reached by the samples, whereas the amplitude of the water contents was determined based on the laboratory measurements. In the new approach we assume, that the maximum electrical conductivity in the field is achieved at the maximum water content, as determined in

the laboratory. The measurements of the electrical conductivity in the laboratory are kept in the supplement to discuss their drawbacks.

Author change in the text 13:

P7 L21: "The calibration of the water content was performed, based on the maximum and minimum values of electrical conductivity reached in the field and the amplitude of the water contents reached in the laboratory measurements. We assume, that the samples at least once got fully wetted in the field and thus maximum electrical conductivity values achieved in the field were reached at the maximum water content, which has been determined in the laboratory (where the samples were wetted to full saturation). We also assume (and are quite sure), that the samples dried out at least once in the field, giving the minimum values of electrical conductivity. Accordingly, the water content (WC) was calculated as follows:

$$WC \ [\% \ DW] = \frac{(EC_i - EC_{min})}{(EC_{max} - EC_{min})} * (WC_{max} - WC_{min}) \qquad \text{Eq. (4)}$$

with $EC_i$ as electrical conductivity, $EC_{min}$ as minimum electrical conductivity, $EC_{max}$ as maximum electrical conductivity, $WC_{max}$ as the maximum water content as determined in the laboratory, and $WC_{min}$ as the minimal water content in the laboratory.

The measurements of the electrical conductivity in the laboratory will be kept in the supplement to evaluate the quality of the calibration curves, by the linear fitting of the individual measurements of the different samples resulting a $R^2$ ranging from 0.79 to 0.89 and a root mean square error (RMSE) ranging from 68 to 229 % DW (Eq. S1, Fig. S5, Tab. S1b). As a consequence, the calculated values for the WC should not be considered as absolute but as approximate values. Due to this uncertainty, no statistical tests have been performed on the bryophyte WC."

**Equation S1: The calculation of the root mean square error (RMSE)**

$$RMSE = \sqrt{\frac{\sum(WCobs - WCpred)^2}{N}} \qquad \text{(Eq. S1)}$$

*where WCobs* is the observed water content, *WCpred* is the predicted water content, and *N* is the sample size.

[Figure]

Figure S5: Calibration curves of water content sensors installed within different bryophyte species. The water content [% DW] is plotted against the electrical conductivity [mV] for the species Sematophyllum subsimplex (four replicates), Octoblepharum cocuiense (two replicates), Leucobryum martianum (tree replicates), and Symbiezidium barbiflorum (three replicates). Of each sample three subsequent wetting and drying cycles were measured. The dots show the measured data points, the lines represent the linear fit. For each fit the RMSE and the R² value is given in the graphic.

**Table S1a:** Height of installation and minimum and maximum values of the individual sensors of the microclimate station measuring water content, temperature, and light (PAR). For the water content sensors, also the bryophyte species are given. Based on 30-minute integrals.*) The species name cannot be verified without any doubts. The information of the calibration done with this species was considered for further data analysis, due to morphological similarity.

| Water content | Height [m] | WC [% DW] min | WC [% DW] max | Bryophyte species | Temperature | Height [m] | Temperature [°C] min | Temperature [°C] max |
|---|---|---|---|---|---|---|---|---|
| Sensor 01 | 1.5 | 0 | 598 | *Sematophyllum subsimplex* | Sensor 01 | 1.5 | 21.1 | 36.3 |
| Sensor 02 | 1.5 | 0 | 598 | *Sematophyllum subsimplex* | Sensor 02 | 1.5 | 21.4 | 39.4 |
| Sensor 03 | 1.5 | 0 | 598 | *Sematophyllum subsimplex* | Sensor 03 | 8 | 21.6 | 34.7 |
| Sensor 04 | 1.5 | 0 | 1202 | *Leucobryum martianum* | Sensor 04 | 8 | 20.9 | 46.3 |
| Sensor 05 | 1.5 | 0 | 598 | *Sematophyllum subsimplex* | Sensor 05 | 18 | 20.3 | 38.0 |
| Sensor 06 | 1.5 | 0 | 598 | *Sematophyllum subsimplex* | Sensor 06 | 18 | 20.3 | 37.5 |
| Sensor 07 | 8 | 0 | 1725 | *Symbiezidium barbiflorum** | Sensor 07 | 23 | 20.8 | 41.2 |
| Sensor 08 | 8 | 0 | 834 | *Octoblepharum cocuiense* | Sensor 08 | 23 | 20.3 | 48.7 |

| Light | Height [m] | PAR [μmol m$^{-2}$ s$^{-1}$] min | PAR [μmol m$^{-2}$ s$^{-1}$] max |
|---|---|---|---|
| Sensor 01 | 1.5 | 0 | 1546 |
| Sensor 02 | 8 | 0 | 1461 |
| Sensor 03 | 8 | 0 | 1502 |
| Sensor 04 | 18 | 0 | 1386 |
| Sensor 05 | 18 | 0 | 1080 |
| Sensor 06 | 23 | 0 | 1326 |
| Sensor 07 | 23 | 0 | 1351 |

Remaining water content sensors:

| Water content | Height [m] | WC [% DW] min | WC [% DW] max | Bryophyte species |
|---|---|---|---|---|
| Sensor 09 | 8 | 0 | 834 | *Octoblepharum cocuiense* |
| Sensor 10 | 8 | 0 | 834 | *Octoblepharum cocuiense* |
| Sensor 11 | 8 | 0 | 1725 | *Symbiezidium barbiflorum** |
| Sensor 12 | 8 | 0 | 1725 | *Symbiezidium barbiflorum** |
| Sensor 13 | 18 | 0 | 1725 | *Symbiezidium barbiflorum** |
| Sensor 14 | 18 | 0 | 1725 | *Symbiezidium barbiflorum** |
| Sensor 15 | 18 | 0 | 1725 | *Symbiezidium barbiflorum** |
| Sensor 16 | 18 | 0 | 1725 | *Symbiezidium barbiflorum** |
| Sensor 17 | 18 | 0 | 1725 | *Symbiezidium barbiflorum** |
| Sensor 18 | 18 | 0 | 1725 | *Symbiezidium barbiflorum** |
| Sensor 19 | 23 | 0 | 1725 | *Symbiezidium barbiflorum** |
| Sensor 20 | 23 | -- | -- | *Symbiezidium barbiflorum** |
| Sensor 21 | 23 | 0 | 1725 | *Symbiezidium barbiflorum** |
| Sensor 22 | 23 | 0 | 1725 | *Symbiezidium barbiflorum** |
| Sensor 23 | 23 | 0 | 1725 | *Symbiezidium barbiflorum** |
| Sensor 24 | 23 | 0 | 1725 | *Symbiezidium barbiflorum** |

**Table S1b:** Bryophyte species and calibration data of the water content sensors. Listed are the bryophyte species with their division (moss or liverwort), their height and height zone of installation, the root mean square error (RMSE), and the determination coefficient R². *) The species name cannot be verified without any doubts. The information of the calibration done with this species was considered for further data analysis, due to morphological similarity.

| Bryophyte species | Division | Height [m] | Height zone | RMSE [% DW] | R² |
|---|---|---|---|---|---|
| *Sematophyllum subsimplex* | Moss | | | 44 | 0.95 |
| *Sematophyllum subsimplex* | Moss | | | 125 | 0.84 |
| *Sematophyllum subsimplex* | Moss | | | 35 | 0.95 |
| *Sematophyllum subsimplex* | Moss | | | 66 | 0.81 |
| *Octoblepharum cocuiense* | Moss | | | 66 | 0.89 |
| *Octoblepharum cocuiense* | Moss | | | 103 | 0.8 |
| *Leucobryum martianum* | Moss | | | 162 | 0.89 |
| *Leucobryum martianum* | Moss | | | 135 | 0.72 |
| *Leucobryum martianum* | Moss | | | 159 | 0.76 |
| *Symbiezidium barbiflorum\** | Liverwort | | | 146 | 0.89 |
| *Symbiezidium barbiflorum\** | Liverwort | | | 306 | 0.78 |
| *Symbiezidium barbiflorum\** | Liverwort | | | 235 | 0.8 |
| ***Sematophyllum subsimplex*** | **Moss** | **1.5** | **1** | **68** | **0.89** |
| ***Octoblepharum cocuiense*** | **Moss** | **8** | **2** | **85** | **0.84** |
| ***Leucobryum martianum*** | **Moss** | **1.5** | **1** | **152** | **0.79** |
| ***Symbiezidium barbiflorum\**** | **Liverwort** | **8, 18, 23** | **2, 3, 4** | **229** | **0.82** |

Referee comment 14:

Also, the observation that water saturation was never reached at the 3 higher levels seems to suggest that something was wrong either with your WC measurements or the literature parameters used… BUT, this statement (P13, L24) cannot be true based on your data, because *Symbiezidium* is present only in these three higher levels, and in the calibration curves you show that observed values go up to 1500% WC, which is well above the WSPs cited…

Author comment 14:

You are right, according to the original calibration the WSP of 349 % WC was almost never reached in the canopy (at 23, 18, and 8 m), whereas it was reached during 22% of the time at 1.5 m height. As described above, the original calibration has been replaced and we also restrict the calculations to the compensation points, which are more relevant than the saturation points in the current context.

Referee comment 15:

It was unclear to me what "upper three height levels the bryophyte taxa could not be securely determined. Thus, the bryophyte taxon with the highest abundance in the canopy communities, i.e., the liverwort *Symbiezidium barbiflorum* was used" means exactly. Did you install sensors only in this species, or did you do the calibration curve only for this species and then use if for all the different (unidentified) species sampled at the higher height levels?

5    This should be made clearer. I could imagine that you installed sensors in other liverworts looking similar to Symbiezidium and then assumed that the relationship between electrical conductivity and water content should not be more different between species than within species, due to the similar life form. This seems a reasonable assumption, but should be made explicit, and in table S1b the species should not be named if you do not know the real name. Indicating if it was a moss or a liverwort, or the family it belongs to, would be useful though!

10    Author comment 15:

The sensors were installed in bryophytes morphologically similar to *Symbiezidium barbiflorum*. However, as the sensors were installed by a climber, it could not be completely reassured that always the same species was used. Nevertheless, we know that *Symbiezidium barbiflorum* was the most dominant species in the canopy of this tree, and from all the information we have for each sensor, the identification of these samples should be correct. This

15    was corrected accordingly in the text.

Author change in the text 15:

P6 L24: "Thus, the bryophyte taxon with *a high abundance in the canopy of this tree*, i.e., the liverwort Symbiezidium sp. was *considered for all the further calculations in the course of the calibration, due to its morphological characteristics.*"

20    Table S1a and Table S1b: "*) The species name cannot be verified without any doubts. The information of the calibration done with this genus was considered for further data analysis, due to morphological similarity."

Referee comment 16:

The use of different species at the different heights is a problem that also needs to be discussed earlier and more

25    prominently and included in the analysis. It reads all through the manuscript as though differences in water content between height zones were caused by microclimatic differences, but of course a Leucobryum (cushion moss with specialized water-holding cells) is going to have very different water content dynamics that a *Symbiziedium* (prostrate leafy liverwort), even under the same environmental conditions. This is also obvious from your own data in the calibration curves, the points for Leucobryum being much closer together, indicating that the drying was much

30    slower than e.g. for *Symbiezidium*. For Oc*t*oblepharum the two (! Looks like they were only two though you write they were three) samples dried at quite different speeds, it looks like the slow sample was denser and thus had higher conductivity at similar water contents. At the moment, the whole manuscript reads a bit as though you consider all cryptogams are expected to respond more or less the same, but we know that there are big differences

between species, in particular in terms of water-content dynamics as well as the responses to this water content. Although you do mention this briefly, I think it deserves a few more words at least.

Author comment 16:

Indeed, we mixed up the replicate numbers in the Supplement Figure S5, and has been corrected.

Regarding the behavior of different species during the drying process, this section was extended with more information.

Author change in the text 16:

P5 L8: *"As the morphology of different species affects their overall WC, different maximum WC and patterns of the drying process were observed. Whereas the individual samples of Octoblepharum c. reached quite similar maximum WC values (706-1109 % DW), the individual samples of cf. Symbiezidium sp., Leucobryum m., and Sematophyllum s. reached rather different values (1313-2383, 845-1862 and 428-1128 % DW; Fig. S5)."*

Supplement Figure S5: "The water content [% DW] is plotted against the electrical conductivity [mV] for the species *Sematophyllum subsimplex* (four replicates), *Octoblepharum cocuiense* (two replicates), *Leucobryum martianum* (three replicates), and cf. *Symbiezidium barbiflorum (three replicates*. Of each sample three subsequent wetting and drying cycles were measured."

Referee comment 17:

It would be really cool if you could detect a dew signal in the WC data, did you look for this? Mention this in the discussion to but the dew remarks into the context of your data.

Author comment 17:

This indeed is a relevant aspect, and we also considered if we could calculated these values. However, in order to calculate this we would need the temperature below the bark. Due to the lack of this information we cannot calculate dew formation.

Referee comment 18:

It would also be cool if you could detect relationship between cryptogam activity patterns and measured trace gas emissions – this tall canopy site would be one of the few places in the world where the needed data might be available, assuming that trace gases above the canopy are also monitored?

Author comment 18:

Yes indeed, different trace gases are monitored at this study site and investigations on this are planned for the near future. This, however, is beyond the scope of the current manuscript.

Referee comment 19:

The literature cited needs to be revised! Only few bryophyte papers are cited and often they are not the correct ones (see below)! Some examples:

p. 3, lines 15-16: Zotz et al 1997 is cited a lot but refers to a montane forest, and not to nutrient cycling, as suggested on this occasion.

p.8, lines 30-31: 'at least in the environment of the central Amazon' is followed by references out of which none are from the central Amazon, most are from cloud forest…(by the way, this sentence is more or less repeated on page 12, L 29-31)

p. 9, lines 5-6: 'For tropical species, values (of WCPl) in the range 5 between ~ 30 and ~ 225 % have been determined (Romero et al., 2006; Wagner et al., 2013; Zotz et al., 1997, 2003)' Again, these references are all from montane species or do not mention WCPl values at all.

p. 6, lines 10 - 12: "Thus, the bryophyte taxon with the highest abundance in the canopy communities, i.e., the liverwort *Symbiezidium barbiflorum* was used (Gradstein and Allen, 1992; Mota de Oliveira et al., 2009; Mota de Oliveira and ter Steege, 2015; Pardow et al., 2012; Romanski et al., 2011; Sporn et al., 2010)." Of the 6 references cited here, *S. barbiflorum* is only mentioned in Gradstein and Allen (1992), the other 5 references do not cite this species at all! (one of the papers cited, Sporn et al. 2010, even deals with Asia even though S. barbiflorum does not occur there, being restricted to America…). Interestingly, Gradstein and Allen (1992) state that S. barbiflorum is a characteristic shade epiphyte of forest understory communities, not canopy communities. Not-cited more recent publications on the habitat of *S. barbiflorum*, however, show that the species also occurs in the forest canopy (Gradstein et al. 2001, Gradstein 2006, Gradstein & Ilkiu-Borges 2009, Gehrig et al. 2013). These recent papers show that S. barbiflorum is actually an ecological generalist, occurring in understory communities as well as in canopy communities. None of these non-cited papers document highest abundance of the species in canopy communities. Thus, the sentence on p. 6, lines 10-12, is rather wrong.

p. 3, line 12-13: "In 2013, 800 species of mosses and liverworts …,… have been reported for the Amazon region" (Mota de Oliveira & ter Steege 2013). The reference cited here is quite wrong, Mota de Oliveira & ter Steege did not provide this number at all, instead they took it from Gradstein et al. (2001; correctly cited by Mota de Oliveira & ter Steege) who calculated 800 species in the Amazon region in their book based on a full-scale analysis of the bryophyte flora of the Neotropics. Thus, the correct reference here is Gradstein et al. (2001) and not Mota de Oliveira & ter Steege.

Author comment 19:

Regarding P3 L 15-16, (Now P3 L16): reference was removed

Regarding P8 L30-31, (Now P10 L11-13): references exchanged; the repeated sentence was removed from P13 L24-26

Regarding P9 L5-6, (Now P 9 L23-25): The references of montane cloud forests (Romero et al. 2006, Zotz et al. 1997, Zotz et al. 2003) were removed. Only the information related to a research site at sea level in Wagner et al.

2013 was considered. The information for the WCP there is was extracted from figure 1. Perhaps we could obtain the exact value from the reviewer.

Regarding P6 L10-12, (Now P6 L23-28): Indeed, this sentence in its final version was wrong. Initially, this sentence intended to tell that in general in lowland rainforests liverworts are more abundant in the canopy than in the understory, which was observed by Pardow et al. 2012. However, due to internal revisions and changes in the text, the sense of this sentence changed, unfortunately resulting in a wrong statement. The sentence was changed to its initial meaning and the references for montane sites were removed.

Regarding P3 L12-13, (Now P3 L14-15): We corrected the reference according to your advice.

Author change in the text 19:

P9 L4: "The physiological activity of bryophytes – and of cryptogams in general – is primarily controlled by water and light, whereas temperature plays a secondary role – at least in the environment of the central Amazon (Lösch et al., 1994; Wagner et al., 2013)."

P9 L17: "For tropical species, values in the range between ~30 and ~*80 %* have been determined (Wagner et al., 2013; Table S3)."

P6 L 24: "Thus, the bryophyte taxon with a high abundance in the canopy of this tree, i.e., the liverwort *Symbiezidium barbiflorum* was considered for all further calculations in the course of the calibration, due to its morphological similarity. Overall, for tropical lowland rain forests in Panama and French Guyana it was shown that liverworts have a higher abundance and higher biomass at the upper trunk and in the canopy than in the understory."

P3 L2: "By 2013, 800 species of mosses and liverworts...have been reported for the Amazon region (*Gradstein et al. 2001*)."

Referee comment 20:

Data availability: does this local database assure future data maintenance and retrieval? Please provide more details.

Author comment 20:

Yes, this is a long term monitoring project and the database on the water content, temperature, and light conditions of epiphytes is uploaded to the ATTO data portal (www.attoproject.org/).The data thus are maintained, obtain a doi and can be retrieved from that site.

Referee comment 21:

General: rather than 'mesoclimate', 'above-canopy climate' would be a more intuitive name for those measurements.

Author comment 21:

We agree with your advice to rename the "mesoclimate" to "above-canopy climate". Accordingly, this expression was changed throughout the text. Furthermore, the expression "ambient" was changed into "above-canopy".

Referee comment 22:

5  P3 L 9: instead of 'these' write 'such' (this is an example of the confusing mix of literature and statements about cryptogam communities in general (often based on soil crusts…) and on tropical lowland epiphytes.

Author comment 22:

We agree to substitute "These" by "Such".

Author change in the text 22:

10  P2 L26: "*Such* communities can colonize different substrates, such as soil,…"

Referee comment 23:

P3 L 21: careful, not all bryophytes are desiccation tolerant, even if they are poikilohydric

Author comment 23:

15  Yes, we agree on that and added the expression "most species" and "for many species".

Author change in the text 23:

P 3 L14: "In a dry state most species can outlast extreme weather conditions, being reactivated by water (Oliver et al., 2005; Proctor, 2000; Proctor et al., 2007; Seel et al., 1992), and for many of them even fog and dew can serve as a source of water (Lancaster et al., 1984; Lange et al., 2006; Lange and Kilian, 1985; Reiter et al., 2008)."

Referee comment 24:

P4 L4-6: Add that most of this info is based on data from soil crusts and from temperate zones and that very little is known about biomass and functions of epiphytic cryptogam in tropical forests, especially in the lowlands.

Author comment 24:

25  That is right, that most of the fluxes were detected for soil communities. However, the information on VOC and aldehydes was performed on epiphytic lichens as well (Kesselmeier et al., 1999; Kuhn et al., 2000; Kuhn and Kesselmeier, 2000; Wilske and Kesselmeier, 1999).

This information was omit in the meantime, due to reorganization of the whole section.

30  Referee comment 25:

P4 L 8: seasonal variation in what?

Author comment 25:

…the seasonal variation of climatic conditions.

But the whole section was revised, and this sentence was removed.

Referee comment 26:

P5 L2: why 'ecophysiological' water content? What other water content is there?

Author comment 26:

It is the "normal" water content of bryophytes, thus the word "ecophysiological" can be deleted.

Author change in the text 26:

P4 L19: "The parameters temperature and light within/on top of the bryophytes *and the water content* of bryophytes are being measured with a microclimate station installed in September 2014 (Fig. S1)."

Referee comment 27:

P5 L3: use 'were' rather than 'are being', even if the measurements are continuing, because you are here presenting results of a specific period in the past. Same for P5 L 11: were taken (not have been taken)

Author comment 27:

We agree on that and changed these the tenses accordingly.

Author change in the text 27:

P4 L20: "The sensors *were* placed along a vertical gradient…"

P5 L18: "Since the installation, automatic measurements at 5-minute intervals *were taken* with a data logger…"

Referee comment 28:

P5 L 5: instead of 'described by' use 'used by', because 'described' suggests that these zones were the output of a study, but it was the sampling design.

Author comment 28:

Done accordingly.

Author change in the text 28:

P4 L21: "…, corresponding to the zones 1 to 4 *used by* Mota de Oliveira and ter Steege (2015)."

Referee comment 29:

P5 L8: a cushion is a specific bryophyte life form, seeing your species the samples probably were not cushions in most cases…You could use 'bryophyte samples'.

Author comment 29:

Done.

Author change in the text 29:

P5 L15: "…, while the light sensors were fixed on ~ 5 cm long sticks and installed next to the bryophyte *samples* (Fig. S1)."

Referee comment 30:

P5 L 19: what do you mean with 'fluctuations'?

Author comment 30:

With "fluctuation" we meant to describe the oscillations of the measurement. This was changed accordingly.

Author change in the text 30:

P5 L27: "The WC values are *oscillating, causing an inaccuracy of* approximately 15 % dry weight (DW)."

Referee comment 31:

P6 L17: are nutrient content and temperature species-specific?

Author comment 31:

Yes, the nutrient content is species-specific. But the temperature cannot be actively regulated by the species, thus it is not species-specific, but dependent on the environment. However, both parameters influence the measurements of electrical conductivity, hence it is recommended to include an assessment of the species-specific nutrient contents and to do a correction by temperature, to receive the most accurate values.

Referee comment 32:

P7 L1: what is the sensor weight?

Author comment 32:

During the calibration the sensor is fixed in the bryophyte sample and both are lying on the balance. Accordingly, the balance always reads and logs the total weight of 'sample plus sensor'. But as the weight of the sensor varies slightly depending on the tension of its wire, the weight of the sensor is not the same for each 'set of calibration measurements' the same. Thus, the weight of the dry' sample plus sensor' is recorded at the end of the measurement, as soon as weight consistence is given. Afterwards the sensor with its wire is removed from the sample and then only the weight of the sample can be recorded.

Referee comment 33:

P7 L12: rather that presenting the models, which are very standard (except maybe for the exponential correction; if you want you could show the models in the appendix), a discussion about uncertainty propagation would be fitting here.

Author comment 33:

Due to recalculation of the WC the fits are omitted from the new version of the manuscript. A discussion on the inaccuracy of the measurements and the uncertainty of the resulting values is included in the material and methods section.

Author changes in the text 33:

Equations of fits are moved to the supplement, as already described in comment 7.

P7 L30: "The electrical conductivity measurements conducted in the laboratory were used to evaluate their variability. A linear fit was placed through the individual measurements of the different samples resulted in an $R^2$ ranging from 0.79 to 0.89 and a root mean square error (RMSE) ranging from 68 to 229 % DW (Eq. S1, Fig. S5, Tab. S1b). These results demonstrate the rather high variability of the water content during measurement of different samples or repeated measurements of the same one. As a consequence, the calculated values for the WC should not be considered as absolute but as approximate values. Because of this, no statistical tests have been performed on the bryophyte WCs."

Referee comment 34:

P8 L16: rainfall amounts would usually not be calculated by integration but by adding the rain amount (e.g. number of tipping events) per time period…

Author comment 34:

Yes, you are right, the rainfall amount should be summed up for certain time periods. However, sometimes rain detection was interrupted and data of short time gaps were missing. For these time periods, we decided to integrate the data not to underestimate the amount to much. We are aware of the fact, that these data gaps and the subsequent calculations may be a source of over- or underestimation. However, this to our knowledge, is the best way to deal with this problem.

Referee comment 35:

P8 L26: explain 'UTC values'; and where are such times presented, and why not always use local time?

Author comment 35:

The UTC is the abbreviation of the universal coordinated time and it is used throughout this study. It allows the synchronization with other data sets. The local time (LT= UTC-4) is only used for the calculation and presentation of diurnal cycles, where it is explicitly marked. The UTC time is used for long data ranges, as monthly and seasonal data.

Author change in the text 35:

P9 L1:" The time for seasonal periods is always taken and presented as UTC *(universal coordinated time)*, except for diurnal cycles, where local time (LT, i.e., UTC-4) is shown, as labeled in the figures."

Referee comment 36:

P9 L23: this WCPl is not what you describe it to be (this would not be a compensation point), it is the point below which the WC is so low that photosynthesis cannot compensate respiration, respiration ceasing at lower WCs than photosynthesis.

Author comment 36:

Here indeed was a mistake. With the WCPl we wanted to explain the point, when net photosynthesis equals respiration, due to limited water availability. This was now changed in the text.

Author change in the text 36:

P9 L16: "The lower water compensation point (WCP$_l$) presents the minimum WC above which positive *net photosynthesis is reached*."

Referee comment 37:

P9 L28: with 'we found' you mean 'we assumed'?

Author comment 37:

Yes, indeed. We changed the text according to this.

Author change in the text 37:

P10 L2:" The compensation points for the different parameters are also to some extent interrelated, e.g., the water compensation point of lichens has been shown to slightly increase with increasing temperature (Lange, 1980), but *we assumed* that this can be neglected in such a first qualitative approach."

Referee comment 38:

P10 L17: report the statistical results (test and test statistics)! This goes for all 'significant' (or non-significant) results.

Author comment 38:

Indeed, we missed to provide the information of the statistical test result. Now they are provided throughout the text.

Referee comment 39:

P11 L1: 'The RH..'What RH? It is generally not always clear in the text what parameter you are talking about: daily means, monthly means, something else?

Author comment 39:

This section deals with the 'annual fluctuations of monthly mean values'. Accordingly the RH should be understood as the monthly mean.

Author change in the text 39:

P11 L13: "*The monthly average of the above-canopy RH was* characterized by a similar behavior,…"

Referee comment 40:

P12 L25: word missing

Author comment 40:

5    Done.

Author change in the text 40:

P13 L13: "At 23 m height, the daily amplitudes tended to be higher during the dry compared to the wet seasons, whereas for the mosses at the lowest height levels the amplitudes tended to be higher during the wet season. For the bryophytes at the other height levels the difference of *amplitudes between seasons* was less clear.

Referee comment 41:

P13 L16-18: it would be relevant to mention whether such high temperatures were ever reached in wet bryophytes; I would expect that they would only occur while samples were dry.

Author comment 41: Your assumption indeed is right. We included Figure S9 showing the relation of temperature

15   and water content at different heights along the tree and mentioned the temp/WC relation in the text.

Author change in the text 41:

P14 L1: "Overall, the highest temperatures were reached, when the bryophytes were rather dry (Fig. S9)."

[Figure]

Figure S9: Temperature condition of bryophytes related to their water content. The temperature weremeasured in bryophytes at different height levels along the tree. Data presented as 30-minute averages.

Referee comment 42:

P13 L 27: I guess you mean the LOWER end of the WCPl range?

Author comment 42:

Yes, indeed.

Author change in the text 42:

P17 L25:" As the *lower* end of the WCP$_1$ range (30 % DW) is reached during 100% of the time for liverworts…

Referee comment 43:

P14 L6: you mean 'height', not 'altitude' here.

Author comment 43:

Yes, it should mean 'height'.

Author change in the text 43:

P14 L10: "The microclimatic conditions experienced by bryophytes along *a height* gradient at the ATTO site follow…"

Referee comment 44:

P14 L6-7: 'The microclimatic conditions experienced by bryophytes along an altitudinal gradient at the ATTO site follow the meteorological characteristics to some extent' - this needs some reference to time…

Author comment 44:

This sentence was restructured by adding the missing information.

Author change in the text 44:

P14 L10: "The microclimatic conditions experienced by bryophytes along a height gradient at the ATTO site follow the *seasonal and diel characteristics of the meteorological parameters* to some extent…"

Referee comment 45:

P14 L15-17: mention in methods

Author comment 45:

It is a good idea to provide this information in the methods section. We also rephrased this sentence in the discussion part.

Author change in the text 45:

P5 L11: "*Furthermore, the sensors were installed at the following orientation: at 1.5 and 8 m vertically along the trunk, at 18 m at the upper side of a slightly sloped branch, and at 23 m at the upper side of a vertical branch..*"

P14 L18: "This was most probably an effect of the canopy structure and cushion orientation, as the sensors were installed *vertically along the trunk at 1.5 and 8 m, at the upper side of a slightly sloped branch at 18 m, and at the upper side of a vertical branch at 23 m.*"

Referee comment 46:

P14 L18: 'may have periodically shaded the organisms': it seems to me that you can have observed whether this was the case or not: were any leaves situated close to these sensors? (Same for P16 L7-8)

Author comment 46:

As the sensors were located at 8, 18, and 36 m height, they were out of direct sight for us. One would have needed to install cameras to explore this over time. Maybe it is better to use the expression "could" instead "can".

Author change in the text 46:

P 14 L22: "This *could* explain lower monthly PAR$_{avg}$ values…"

Referee comment 47:

P14 L20: was PARavg not the monthly average? Do you mean the monthly averages of the daily patterns?

Author comment 47:

We intended to differentiate between *PARavg* and *PARmax*. While PARavg is the average of a certain period, could be month or hour, the PARmax is the maximum PAR value reached per day. In the cited context it is the hourly average presented for the diel cycle.

Referee comment 48:

P15 L9-15: this could indeed be expected and is not very exciting. Your contribution here should be discussing the differences in temperature fluctuations quantitatively.

Author comment 48:

Yes, this is a good point and was considered by insertion of some more detail on this difference.

Author change in the text 48:

P15 L17: "*The daily amplitude of the temperature was about twice as large in the canopy as compared to the understory (Tab. S8).*"

Referee comment 49:

P15 L17-18: mention this reinstallation in the methods too.

Author comment 49:

The reinstallation is mentioned in the methods part according to your advice.

Author changes in the text 49:

P5 L23: "*However, from time to time the water content and temperature sensors fell out from the moss samples, which required a reinstallation. Accordingly, the WC sensor number 6 (1.5 m) was repositioned in 01/2015, WC sensor number 1 (1.5 m) in 11/2015, WC sensor number 1, number 6 to24 and all temperature sensors in 11/2016. The data of the periods when the sensors were not in the bryophyte samples were excluded from further calculations.*"

P15 L21-22: mention and discuss this earlier on.

Author comment 50:

This is a good point, and thus, a description of the uncertainty of the WC is now already provided in the material section.

Author change in the text 50:

P5 L28: "*Furthermore, besides the specific position in the substrate the water content depends on the texture of the sample material, its ion concentration, and the temperature. As all these factors modify the sensor readings, the provided values of the water content should rather be considered as the best possible estimate than as exact values.*"

Referee comment 51:

P15 L33-34: Is the canopy so open that the wind direction is noticed at 8 m height? Why did you choose the west side, I would expect you to select a side with good moss cover. Interesting if this happened to be the west side if this side receives less moisture. Can you explain this?

Author comment 51:

Yes, the west side indeed was chosen as we found the best bryophyte cover there. Although the wind intensity is weaker inside the canopy, we still experienced wind below the canopy and think that this could have an impact on water and habitat conditions at the different expositions, which then could have an effect on the differential growth of bryophytes.

Author change in the text 51:

P16 L6: "At 8 m height, the bryophytes *measured were oriented in western direction. Although wind intensities are clearly weaker below the canopy, we still expect that winds*, predominantly originating from north and north-eastern directions during the wet season (Pöhlker et al., 2018), could have an impact on water and habitat conditions at the different expositions."

Referee comment 52:

P16 L11: why does a light rain facilitate drying??

Author comment 52:

Maybe we expressed it in the wrong way. But we intended to say, that after a light rain event the bryophyte samples dried quicker again, as they got not completely saturated with water. We rephrased this sentences for clarification.

Author changes in the text 52:

P16 L15: "Most rain events in the Central Amazon occur in the early afternoon (12:00 – 14:00 LT) and more than 75 % of them are weak events of less than 10 mm (Cuartas et al., 2007), *which induce only a partial water saturation of the bryophytes. Consequently, the organisms likely dry within a shorter time span as compared to strong rain event, which causes full saturation of the thalli.*"

Referee comment 53:

P16 L17: this has at best been estimated, and please specify what you mean by 4%: 4% of water input for bryophytes (or other epiphytes?), or just comprising (thus not 'providing') 4% of total precipitation?

Author comment 53:

This means, it was estimated that approximately 4 % of the total precipitation will reach the ground as stemflow water. Thus, 4 % of the rain water is directly available for epiphytic organisms. By our calculations, this means that 68 to 75 mm per year are available as stemflow water. We rephrased this sentence accordingly.

Author change in the text 53:

P16 L23: "It has been *estimated* that in tropical forests stemflow water could provide up to 4 % of the annual rainfall amount (van Stan and Gordon, 2018), corresponding to values of  68 and 75 mm for the years 2015 and 2016 at the ATTO site."

Referee comment 54:

P16 L22: the water holding capacity is not what you have been measuring…Otherwise, this sentence is very true: the high water contents may be due to the high water-holding capacities of these species.

Author comment 54:

Indeed we did not measure the water holding capacity or only indirectly during the calibrations. Instead, we found high water contents over prolonged times, which we wanted to describe here. We rephrased this sentence for clarification.

Author change in the text 54:

P16 L28: "The high WC of the organisms in the understory might be partly explained by the different water holding capacity of different bryophyte species growing (and measured) there, as understory species of lichen and bryophytes are known to be adapted to long-term water storage (Lakatos et al., 2006; Romero et al., 2006; Williams and Flanagan, 1996)."

Referee comment 55:

P17 L13-14: be careful with your wording: understorey species are probably more efficient at low light (lower LCP), but it would be weird if they had a higher potential photosynthesis.

Author comment 55:

We meant to say that understory species reach higher net photosynthesis rates at low light conditions. We changed this sentence accordingly.

Author change in the text 55:

P18 L4: "…and it has been reported that understory mosses and lichens indeed show higher rates of net photosynthesis *at low light conditions* than canopy species."

Referee comment 56:

P17 L19-20: words missing

Author comment 56:

Yes indeed. The two sentences were rephrased and linked for clarification.

Author change in the text 56:

P18 L9: "Since net photosynthesis is the sum of simultaneously occurring photosynthesis and respiration processes, positive net photosynthesis rates may still be reached at higher temperatures in the light, as long as the photosynthetic capacity is high enough, whereas during the night, high temperatures could cause a major loss of carbon due to high respiration rates (Lange et al., 2000)."

Referee comment 57:

P17 L22: It may be worth mentioning that Wagner et al 2013 concluded that, although respiration losses may be high, this in itself does not explain low bryophyte growth in tropical lowlands, because respiration rates are adapted or acclimatized to the prevailing temperature conditions: in mosses growing at higher elevations the respiration rates are higher at the same temperatures, but still epiphytic bryophyte biomass is much higher here.

Author comment 57:

Indeed, this type of information can be added to the text.

Author change in the text 57:

P18 L14:"*However, Wagner et al. (2013) explained that the respiration losses themselves do not explain the low bryophyte growth in tropical lowlands, as respiration rates are adapted to the prevailing temperature conditions. At higher elevations they observed higher respiration rates at the same temperatures but still also higher biomass values.* "

Referee comment 58:

P18 L4: another example of a mismatch between cited literature and interpretation: you suggest that it is relevant that water contents in Zotz et al 1997 were measured during the same time of the year, but as this was a different region and a very different forest type, this temporal coincidence has no meaning whatsoever!

Author comment 58:

Yes, indeed the study of Zotz et al. 1997 was performed in a lower montane forest with an altitude of 1100 m, thus we decided to omit the comparison of the WC within the same season.

P18 L13-14 'whereas in the canopy, rain events, fog, and condensation seem to be equally important water sources for cryptogams.' What do you base this conclusion on??

Author comment 59:

Thanks for this comment, the text was revised accordingly.

Author change in the text 59:

P19 L3: "*In the understory, the WC of cryptogams seems to be predominantly regulated by rain events and the vegetation reduces the evaporation by its shadowing effect. An increased RH mostly slows down the drying in the understory, whereas in the canopy a nightly increase of RH also causes an increase of the bryophytes WC (Fig. 2). The effect of fog events is hard to distinguish from the influence of a high RH, as fog occurs when already a high RH persists. However, some events indicate an increase of WC upon fog (Fig. S8).*"

[Figure]

**Figure S8:** Two exemplary fog events and the reaction of the moisture sensors of the bryophytes (a and b). Each panel presents (A) a fog event defined by a visibility < 2000 m, (B) relative air humidity (RH), (C) rain, and (D) the water content (WC) of the bryophytes. In each panel, the fog event of interest is marked by a red box. For the WC sensors the number, height of installation, and division (M = Moss, L = Liverwort) are given.

Referee comment 60:

P18 L16: what does 'which' refer to? The reference seems strange here. (Figure 2: the wet season data are shown twice, the dry season data are missing! A legend is also missing.) → Already corrected by authors

Author comment60:

"Which" refers to the observation of Pardow and Lakatos (2013), where they describe that understory species are more sensitive to drought than canopy species.

However, the sentences was already omitted, due to reorganization of the section.

Referee comment 61:

Figure S2: in what way are these integrals? Do you mean interpolations?

Author comment 61:

The data with 30-minute time intervals are the average values of six 5-minute grid data. It indeed is better to say "*average*" instead of "integral".

Referee comment 62:

Supplement: P4 L7: looks like 2 replicates for *Octoblepharum*

Author comment 62:

Supplement Figure S5: Yes, indeed there were two replicates for *Octoblepharum*. This was already mentioned in your comment 16 and was corrected accordingly.

**References provided by the author:**

[revised manuscript text omitted]

**References provided by the referee:**

5    Gradstein, S.R., Churchill, S.P. & Salazar A., N. 2001. Guide to the Bryophytes of Tropical America. Memoirs New York Bot. Garden 86: 1-577.

Gradstein, S.R. 2006. The lowland cloud forest of French Guiana – a liverwort hotspot. Cryptogamie, Bryol. 27: 141-152.

Gradstein, S.R. & Ilkiu-Borges, A.-L. 2009. Guide to the Plants of Central French Guiana. Part IV. Liverworts and

10    Hornworts. Memoirs of the New York Botanical Garden 76, 4: 1- 140, 83 plates.

Gehrig-Downie C., Obregón A., Bendix J., Gradstein S.R. 2013. Diversity and vertical distribution of epiphytic liverworts in lowland rain forest and lowland cloud forest of French Guiana. Journal of Bryology 35: 243-254.

---

## Author Comment (AC3) · 12 Jun 2019

**Response to referee comments and suggestions on bg-2018-521 by N. Löbs et al.: "Microclimatic and ecophysiological conditions experienced by epiphytic bryophytes in an Amazonian rain forest"**

**Manuscript format description:**
Black text shows the original referee comment, and blue text shows the response of the authors and the explicit change in the text. The figure and table numbers refer to the revised manuscript.

**Anonymous Referee #2 submitted the comments RC2**

**General referee comment:**

The authors provide a description of bryophyte occurrence and microclimate in a tropical forest canopy. These data are scarce and therefore crucial for a variety of applications that the authors list at various times in the manuscript.

General author response: We would like to thank reviewer 2 for his/her appreciation of the microclimate data and the productive comments, which helped us to substantially improve our manuscript.

Referee comment 1:

First, amongst these are poor organization and a general lack of coherence. Facts about bryophytes (such as they are poikilohydric) are repeated often. No clear hypotheses or research questions are outlined. The introduction tells us that bryophytes are 'cool' and important to study but doesn't do a good job of setting up the study itself. Until the end of the methods section, I didn't realize that gas exchange measurements were not performed (something that is mentioned in abstract- If gas exchange in epiphytes is essential, why did the authors not make these measurements?).

Author comment 1:

In the introduction, our aim was to introduce the ecosystem and study site, the invested organisms and communities, and also the measurement approach. As this aim seems to be only partly fulfilled, we thoroughly checked and restructured the introduction. Specific changes were made to bring more clarity into the abstract and the methods section to facilitate an understanding of the study. $CO_2$ gas exchange measurements indeed would be interesting here, but go beyond the scope of the current study. They represent a major study by themselves, which should be conducted in the near future. For now, we used reliable literature data to investigate to activity patterns of bryophytes when respiration and photosynthesis potentially take place.

Author changes in the text 1:

P2 L7: "In this study, we present data on the microclimatic conditions, including water content, temperature, and light intensities experienced by epiphytic bryophytes along a vertical gradient and combine these with "above-canopy climate" data collected at the *Amazon Tall Tower Observatory* (*ATTO*) in the Amazonian rain forest between October 2014 and December 2016."

P2 L20: "For further investigations of the physiological activity patterns, $CO_2$ gas exchange measurements would be extremely helpful to characterize the response of key organisms to the environmental conditions and parameters."

Referee comment 2:

While I am quite satisfied by the measurement protocols and methodology (and that the epiphyte wetness-drying data are novel and important) the study ends up being merely a data reporting exercise with conclusions that often seem unsubstantiated by the data that are presented. Other times conclusions are trivial. For instance, Pg 18, lines 18-23 it is suggested that it is dark in the understory and therefore photosynthesis is light limited. I do not think that today one needs to go to the Amazon to make this conclusion, as this has been known for decades (for e.g. read classic reviews by Chazdon and Fetcher, 1984; Mooney et al., 1984). I seem facetious here, but the authors could use the same data to build upon these earlier findings, and find some nuance and/or insights. What is the knowledge gap that you are trying to fill with your measurements?

Author comment 2:

The main results and conclusions were revised by us to present the data in a more logical and substantiated way. We also utilized the literature offered by you, as it helped to arrange our results in a better framework. In our opinion, one major advantage of our study is, that we performed long-term measurements (running continuously over more than two years) at several heights along a trunk, thus obtaining a vertical profile of the conditions within the vegetation. We now highlight this aspect, apart from other some other, and with this new structure, we think we can emphasize and improve its significance.

Author changes in the text 2:

P3 L23: "*Studies in temperate zones address the importance of cryptogamic communities for this ecosystem (Gimeno et al., 2017; Rastogi et al., 2018). In contrast to that, there are only few data available for tropical regions. There is a gap of information regarding the functioning of those organisms in an environment with an almost constantly high relative humidity and temperature values. Thus, with these long-term continuous measurements we aim to provide data on seasonality patterns and also vertical gradients of microclimatic conditions within the canopy.*

*In the current study, we present the microclimatic conditions, including temperature, light, and water content of epiphytic bryophyte communities and an estimation of their activity patterns in response to annual and seasonal variations of climatic conditions, as well as along a vertical gradient from the understory to the canopy of the forest.*"

Referee comment 3:

I want to be clear that I do not think that this work is unpublishable, rather a considerable amount of work needs to be done, especially in the writing, to ensure that it is. The advantage of the study is that the authors have collected a vast amount of important data, and there are several questions that can be formulated and answered. For instance, Fig S.5 is very interesting, and one could speculate about the significance of Tair -TCryptogram relationship in different parts of the canopy, and its significance to physiology. Another question could be the importance of light flecks, since you have carefully measured PPFD within the canopy. Fog is also measured but these data seem largely ignored (I wonder if you had leaf wetness sensors, those data could bolster the study tremendously). I would recommend the corresponding author to read some of the classic literature on epiphyte distribution and abundance (e.g. Benzing, 1984). With some more data exploration and thought I think this could be a very significant contribution. In its current form however, the manuscript reads like an early draft of a thesis or a dissertation chapter, and I do not see it fit for publication in Biogeosciences, or a journal of similar repute.

Author comment 3:

We appreciate this criticism and now put more effort into the analysis and interpretation of the long-term data collected by us.

The relevance of fog was investigated in more detail and its relevance for the WC of the bryophytes is illustrated in the text and in Fig. S8 below. We also analyzed light flecks in the updated version. Unfortunately, no leaf wetness sensors have been installed, so this kind of data cannot be used.

Author change in the text 3:

P12 L16: "Furthermore, fog might serve as an additional water source for the epiphytic bryophytes, as the WC of the bryophytes increases upon fog events (Fig. S8)."

P19 L3:" *In the understory, the WC of cryptogams seems to be predominantly regulated by rain events and the vegetation reduces the evaporation by its shadowing effect. An increased RH mostly slows down the drying in the understory, whereas in the canopy a nightly increase of RH also causes an increase of the bryophyte WC (Fig. 2). The effect of fog events is hard to distinguish from the influence of high RH, as fog occurs when high RH persists. However, some events indicate an increase of the bryophyte WC upon fog (Fig. S8).*

[Figure]

**Figure S8:** Two exemplary fog events and the reaction of the moisture sensors of the bryophytes (a and b). Each panel presents (A) a fog event defined by a visibility < 2000 m, (B) relative air humidity (RH), (C) rain, and (D) the water content (WC) of the bryophytes. In each panel, the fog event of interest is marked by a red box. For the WC sensors the number, height of installation, and division (M = Moss, L = Liverwort) are given.

Referee comment 4:

Finally, authors should provide data access via a link with a doi to a data repository. I wonder if this is required of papers that are submitted to Biogeosciences.

Author comment 4:

5    The database on the water content, temperature, and light conditions of epiphytes is uploaded to the ATTO data portal (www.attoproject.org/).The data thus are maintained, obtain a doi and can be retrieved from that site.

**Specific/Minor comments below:**

Referee comment 5:

10    The abstract is a bit long with too many technical or field specific terms that should be introduced (in the introduction), since it makes it difficult to comprehend for the general reader. An example is "While the monthly average mesoclimatic ambient light intensities above the canopy revealed only minor variations: …" This is a well written but complicated sentence for the average reader. Please simplify.

Author comment 5:

15    The entire abstract was revised for better readability.

Author change in the text 5:

**"Abstract**. In the Amazonian rain forest, major parts of trees and shrubs are covered by epiphytic cryptogams of great taxonomic variety, but their relevance in biosphere-atmosphere exchange, climate processes, and nutrient cycling are largely unknown. As cryptogams are poikilohydric organisms, they

20    are physiologically active only under moist conditions. Thus, information on their water content, as well as temperature and light conditions experienced by them are essential to analyze their impact on local, regional, and even global biogeochemical processes.

In this study, we present long-term data on the microclimatic conditions, including water content, temperature, and light conditions experienced by epiphytic bryophytes along a vertical gradient, which

25    have been collected at the *Amazon Tall Tower Observatory* (*ATTO*) between October 2014 and December 2016. To put these data into perspective, we combine them with ambient "above canopy" climate data. While monthly averages of above-canopy light intensities revealed only minor variation over the course of the year, the light intensities prevailing at the bryophyte surfaces showed major variations depending on canopy height and foliation status of the surrounding vegetation. In the

30    understory (1.5 m), monthly average light intensities were similar throughout the year and individual values were extremely low, remaining below 5 $\mu$mol m$^{-2}$ s$^{-1}$ photosynthetic photon flux density over more than 92 % of the time. Temperatures showed only minor variation throughout the year with higher values and larger height-dependent differences during the dry season. The water contents of bryophytes varied depending on precipitation and air humidity. Whereas bryophytes at higher levels

35    were affected by frequent wetting and drying events, those close to the forest floor remained wet over longer time spans during the wet seasons. In general, bryophytes growing close to the forest floor were

limited by light availability, while those growing in the canopy had to withstand larger variations in microclimatic conditions, especially during the dry season. Utilizing literature data on the $CO_2$ gas exchange of lowland bryophytes, their potential physiological activity patterns could be determined. In follow-up studies, these data should be combined with on-site trace gas emission measurements to determine the role of bryophyte communities in climate-relevant trace cycling processes."

Referee comment 6:

Line 12: 1.5 m relative to what (i.e, please include canopy height). For the abstract something general, like 'near-surface' or 'in the understory' is more appropriate.

Author comment 6:

Yes, it is important to set the height of 1.5 m into relation, especially in the abstract. We exchanged "At 1.5 m height" by "in the understory" for more clarity.

Author change in the text 6:

P2 L12: "*In the understory (1.5 m), monthly average….*"

Referee comment 7:

Line 13: instead of saying "low, exceeding less than 8% …" you could say low, remaining below 5 µmol … more than 92% of the time.

Author comment 7:

Yes, it is easier to follow the way you proposed to revise this sentence. We changed it according to your advice.

Author change in the text 7:

P2 L14: "… individual values were extremely low, *remaining below* 5 µmol photosynthetic photon flux density for *more than 92 %* of the time."

Referee comment 8:

Lines 18-19: Dark respiration should occur independent of light (and unless temperatures are very low, which seems unlikely at your site). The references to photosynthesis and respiration are repeatedly incorrect. Photosynthesis and respiration are co-occurring biological processes (in the light), and therefore one may dominate over the other.

Author comment 8:

We agree with the revision, this sentence was expressed in a way, which could easily be misunderstood. As the entire abstract was restructured, this sentence was removed.

**Introduction**

5    Referee comment 9:

The first paragraph is a well written introduction to tropical forests, but has little do with the study. Either you should reframe it in the context of epiphytes or omit. Overall, the introduction does not set up the study satisfactorily.

Author comment 9:

10    The introduction was restructured and sentences were rewritten to better address the topic.

Author changes in the text 9: P2 L24 (Revised introduction):

"Epiphytic cryptogam communities comprise photoautotrophic bryophytes, algae, lichens, and cyanobacteria in varying proportions, growing together with heterotrophic fungi, other bacteria, and archaea. They can colonize plant surfaces in almost all habitats throughout the world (Büdel, 2002; 15    Elbert et al., 2012; Freiberg, 1999). Epiphytic bryophytes in the tropics play a prominent role in environmental nutrient cycling (Coxson et al., 1992; Zotz et al., 1997) and also influence the microclimate within the forest (Porada et al., 2018), thus contributing to the overall fitness of the host plants and the surrounding vegetation (Zartman, 2003). However, they are equally affected by deforestation and an increasing fragmentation (Zartman, 2003; Zotz et al., 1997).

20    In the Amazonian rain forest, cryptogamic communities mainly occur epiphytically on the stems, branches, and even leaves of trees, and in open forest fractions they may also occur on the soil (Richards, 1954). By 2013, 800 species of mosses and liverworts, 250 lichens species, and 1 800 fungal species have been reported for the Amazon region (Gradstein et al., 2001; Komposch and Hafellner, 2000; Normann et al., 2010; Piepenbring, 2007).

25    Tropical rain forests are characterized by humid conditions, high temperatures, minor annual fluctuations of temperature, and an immense species diversity of flora and fauna. They have been described to play important roles in the water cycle as well as for carbon, nitrogen, and phosphor fluxes on regional and global scales (Andreae et al., 2015). Up to now, ~ 16 000 tree species have been estimated for the Amazon (ter Steege et al., 2013), but the impact of anthropogenic activities on these 30    numbers is highly uncertain. Similarly, it is also hard to predict, to which extent the ongoing and envisioned changes will still ensure its ecological services as "green lung" and carbon sink of planet Earth (Soepadmo, 1993).

Physiologically, cryptogamic organisms in general and specifically also bryophytes are characterized by their poikilohydric nature, as they do not actively regulate their water status, but passively follow the 35    water conditions of their surrounding environment (Walter and Stadelmann, 1968). In a dry state, many bryophytes can outlast extreme weather conditions, being reactivated by water (Oliver et al., 2005;

Proctor, 2000; Proctor et al., 2007; Seel et al., 1992), and for several species also fog and dew can serve as a source of water (Lancaster et al., 1984; Lange et al., 2006; Lange and Kilian, 1985; Reiter et al., 2008). Accordingly, their physiological activity is primarily regulated by the presence of water and only secondarily by light and temperature (Green and Proctor, 2016).

Referee comment 10:

Pg 3 Line 13: 'By' not 'In' 2013.

Author comment 10:

Changed accordingly

Referee comment 11:

Pg 4 Line 5: Update references to carbonyl sulfide: (Gimeno et al., 2017; Rastogi et al., 2018).

Author comment 11:

Many thanks for the provision of these additional references, but due to a revision of the introduction,
15      the information on OCS has been omitted.

**Methods**

Referee comment 12:

Sec. 2.1. A greater description of the site is required. I would recommended starting with site characteristic and then describe the tower and measurements, not the other way around.

20      Author comment 12:

The advice to start with a description of the forest area and to subsequently characterize the study site itself is a good idea, and we reordered section 2.1. accordingly. However, we refrain from giving a more detailed description of the site, as this is nicely presented by Andreae et al. (2015), which belongs to the same special issue (on ATTO) as the current manuscript.

25      Author change in the text 12:

P4 L3:" The study site is located on a *terra firme* (plateau) forest area in the Amazonian rain forest, approx. 150 km northeast of Manaus, Brazil. The average annual rainfall is 2 540 mm year$^{-1}$ (de Ribeiro, 1984), reaching its monthly maximum of ~ 335 mm in the wet (February to May) and its minimum of ~ 47 mm in the dry season (August to November) (Pöhlker et al., 2018). These main seasons are linked
30      by transitional periods covering June and July after the wet and December and January after the dry season (Andreae et al., 2015; Martin et al., 2010; Pöhlker et al., 2016). The *terra firme* has an average growth height of ~ 21 meters, a tree density of around 598 trees ha$^{-1}$, and harbors around 4 590 tree

species on an area of ~ 3 784 000 km$^2$, thus comprising a very high species richness compared to other forest types (McWilliam et al., 1993; ter Steege et al., 2013). The measurements were conducted at the research site *ATTO* (*Amazon Tall Tower Observatory*; S 02° 08.602', W 59° 00.033', 130 m a. s. l.), which has been described in detail by Andreae and co-authors (2015). It comprises one walk-up tower and one mast of 80 m each, which have been operational since 2012, and a 325 m tower, which has been erected in 2015. The ATTO research platform has been established to investigate the functioning of tropical forests within the Earth system. It is operated to conduct basic research on greenhouse gas as well as reactive gas exchange between forests and the atmosphere and contributes to our understanding of climate interactions driven by carbon exchange, atmospheric chemistry, aerosol production, and cloud condensation."

Referee comment 13:

Pg 4 line 13: Remove "The".

Author comment 13:

The word was removed.

Author change in the text 13:

P4 L10: "*Measurements* were conducted at…"

Referee comment 14:

Pg 5 line 2: "were measured" not "are being measured".

Author comment 14:

In this case, we aimed to stress that the *measurements are still running*. Due to this we would like to keep the expression.

Referee comment 15:

P5 Line 5: this seems important for your study and you should describe why sensors were placed where they were placed, in addition to citing the Mota de Oliveira (2013) study.

Author comment 15:

We placed the sensors along a vertical gradient ranging from the understory to the canopy, in order to cover the range of microclimatic conditions experienced by the epiphytic bryophytes as thoroughly as possible. We included a phrase to express this intention.

Author change in the text 15:

P4 L20: "The sensors were placed along a vertical gradient at ~ 1.5, 8, 18, and 23 m above the ground, corresponding to the zones 1 to 4 used by Mota de Oliveira and ter Steege (2015) in order to cover the range of microclimatic conditions experienced by epiphytic bryophytes as thoroughly as possible."

Referee comment 16:

Pg 6: line 30: Why 60 C? Is this a temperature that these communities experience?

Author comment 16:

Drying the organisms at 60°C until weight consistency is a common procedure to determine the dry weight of these organisms. This is not related to the environmental field conditions, but is a standard method used to obtain a standardized dry weight.

Author change in the text 16:

P7 L12: "The dry weight (*DW*) was determined after drying at 60 °C *until weight consistency was reached (Caesar et al., 2018).*"

Referee comment 17:

Sec 2.5. Again, some information (a figure ideally) describing the vertical profile of the forest is necessary. That helps put the various sensor heights in perspective.

Author comment 17:

This is a good idea and we prepared a graphical scheme, which is presented in the supplementas Fig. S2 (see below).

Author changes in the text 17:

[Figure]

**Figure S2:** Schematic overview of the sensors installed at different height levels below, within, and above the canopy. The parameters water content (WC) and temperature (Temp) were measured within the bryophyte samples, the light sensors (PAR) were installed directly on top of the thalli. The average tree height of 21 m was determined for the Plateau forest in general.

Referee comment 18:

Lines 15-21: Why were rainfall values gap-filled? Also, isn't the sensor at 1.5 m the least well placed to record rain event. For instance, a small rain event might not even be recorded at 1.5m, as interception must be high in a high LAI forest. Alternatively, there could be time lags between when rainfall occurs at the canopy top and when it is measured by the 1.5 m ht rain gauge.

Author comment 18:

Unfortunately, over the course of the long-term rainfall measurements there are some measurement gaps, which needed to be filled in order to correctly analyze the data.

The rain gauge has been installed at 81 m height on the tower, while the canopy height in this forest is approximately 21 m. Consequently, the rain gauge is placed well above the canopy. As correctly assumed by you, the bryophytes in the understory (1.5 m) might not get watered by light rain events and there is also a delay of several minutes until they get wet compared to the canopy organisms.

Referee comment 19:

Pg 8 L32- Pg 9, L1: Rephrase sentence. Light intensity regulates the balance i.e. the net exchange between photosynthesis and respiration.

Author comment 19:

The sentences were rephrased to make clear that the light triggers, which process might dominate the carbon balance.

Author change in the text 19:

P9 L11: "While the availability of water determines the overall time of physiological activity, the light intensity regulates *whether* net photosynthesis (NP) or dark respiration (DR) *will dominate the overall metabolical balance.* "

Referee comment 20:

Line 14: Respiration takes place at all light levels. IT IS NOT A LIGHT DEPENDANT PROCESS (there can be significant inhibition of respiration at high light levels). Please check this basic tenet of biology.

Author comment 20:

We are aware of the fact that this was not expressed in a correct way. We meant to state that under these condition NET respiration is observed. We know that both photosynthesis and respiration often occur simultaneously and that the net balance is what is being measured. We changed that accordingly.

Author change in the text 20:

P9 L21: "At light intensities below the compensation point and water contents above $WCP_1$ net respiration *takes place*."

Referee comment 21:

Line 21: Based on *literature*, not literature *data*. Also, please cite the relevant literature.

**Author comment 21:**

We believe that literature data (i.e. data extracted from the literature) is the correct expression here and thus would like to keep this expression. We added the relevant literature citations.

**Author change in the text 21:**

P9 L28: "*Based on literature data (Frahm, 1990; Lösch et al., 1994; Wagner et al., 2013),* we calculated the timespans when these key points were passed, utilizing the *5-minute microclimate data.*"

**Results: Overall, I do not have issues with the content per se, but as I have stated before this section needs to be majorly revised/expanded. Some minor comments below.**

**Referee comment 22:**

Pg 10 Line 8: Micromet did not depend on years but varied amongst years.

**Author comment 22:**

Indeed we were not precise enough with our formulation. We rephrased the sentence accordingly.

**Author changes in the text 22:**

P10 L18: "Over the course of the two years of measurements, the monthly mean values of above-canopy meteorological conditions, the water content, temperature, and light of epiphytic bryophytes varied *between* seasons and years."

**Referee comment 23:**

Pg 10 Line 18: please define mesoclimate the first time you use this term.

**Author comment 23:**

Mesoclimate is a standing term describing the climate of a given habitat, covering a side length of some tens or hundreds of meters. We utilized this expression to distinguish the above-canopy climate measurements from the microclimate measurements conducted next to/within the bryophyte thalli. As this expression lead to some confusion, we replaced it by the term "above canopy climate" throughout the text.

**Referee comment 24:**

Pg 11 Line 30-31: What does this mean? Please elaborate.

**Author comment 24:**

At 1.5 m height the water content sensors always showed an increase after rain events, whereas this was not always observed for the sensors at the other height levels. This seems not fully logical at first sight, but we can imagine that the thalli growing on one side of the tree are sometimes not hit by the rain if it comes from one side. We could imagine that the rain intercepts with some inclination at the higher levels, whereas close to the ground this inclination is lost. This, however, is only one potential explanation which has not been verified by us.

Referee comment 25:

Pg 12 Sec 3.1.3 header: remove parenthesis

Author comment 25:

Yes, we can remove the parentheses in the header.

Author change in the text 25:

P12 L19: "3.1.3 Diel cycles in different seasons and years along a vertical gradient"

Referee comment 26:

Pg12 Line 13: which 'organisms'. Please specify.

Author comment 26:

Here, the epiphytes, which have been mentioned earlier in the sentence, are meant. We reformulated the sentence.

Author change in the text 26:

P13 L1: "The variability and the diel amplitudes tended to be higher for the epiphytes in the canopy than for *these* in the understory."

Referee comment 27:

Sec. 3.2. The scope of your inference is limited since you do not have replicates on different trees. I do not see this as limitation, but somewhere (probably in the discussion) you need to talk about heterogeneity in the microclimatic environment.

Author comment 27:

Yes, this is right, and we now describe this restriction of the measurements in an additional passage in the Material and Methods section.

Author change in the text 27:

P 4 L28: *"Due to constraints in accessibility, all sensors had to be installed on one tree. Thus, we expect that the replicate sensors at each height are not be fully independent and thus the variability could only partly be covered by our current setup."*

5 Referee comment 28:

P13 Lines 13-14. This has been mentioned previously (in Sec. 3.1.).

Author comment 28:

Whereas in the previous sections, we discuss the annual fluctuations of monthly mean values and the seasonal changes between the wet and dry season, the diel cycles are the focus of the current section.
10 Thus, the temperature needs to be discussed under these different aspects and this might give the impression of some repetition. However, we could not find the position where this statement has been made before. Perhaps you can give us a clearer hint on that.

Referee comment 29:

15 P13 Lines 25 to Pg 14 line 3: This is a well written paragraph but belongs in the discussion.

Author comment 29:

This section was rephrased, according to recalculations and was moved to the discussion section.

**Tables**

20 Referee comment 30:

Table 1: Why are these annual means presented? Why are light levels higher at 23 m than at 18m (again discuss heterogeneity)? I still don't have a good grasp of the canopy structure. I do not understand the sensor placement with respect to the canopy structure. Is the 23 m sensor above the canopy top (~ 21m)? Probably not, since light levels seem too low. Also, there seems to be some confusion between relative
25 humidity and water content.

Author comment 30:

The annual means are presented to show the differences between the years, to demonstrate that the climatic conditions change from one year to the next.

The light conditions at the different height levels of the canopy are discussed in the discussion section
30 (see below). The canopy structure and sensor positions have been already described above (comment 17). The tree is approximately 26 m high, which is now also mentioned in the methods section (P 5 L22),

thus the sensors from 1.5 to 23 m height are located on top of the bryophytes growing on the stem of the tree.

We clarified this by deleting the RH-data, as they do not really fit here.

Author change in the text 30:

Canopy structure and sensor position: P4 L32: "*Generally, the water content sensors have been placed in four different bryophyte types being heterogeneously distributed along the four height levels. At the height level of 1.5 m the water content sensors were installed in the moss species Sematophyllum subsimplex (5 sensors) and Leucobryum martianum (1 sensor), at 8 m in the species Octoblepharum cocuiense (3 sensors) and cf. Symbiezidium barbiflorum (3 sensors), at 18 and 23 m in the species cf. Symbiezidium barbiflorum (each all 6 sensors). The temperature sensors were installed in the same species at each height, and the light sensors were installed just next to measured species. Furthermore, the sensors were installed with the following orientations: at 1.5 and 8 m vertically along the trunk, at 18 m at the upper side of a slightly sloped branch, and at 23 m at the upper side of a vertical branch. Thus, also the orientation at the stem can influence the water content of the bryophytes, not only the species and the canopy structure.* "

Tree height: P 4 L23: "At each height level, six water content, two temperature, and two light sensors (except for 1.5 m with only one light sensor) were installed in different bryophyte species at the same tree, *measuring a height of approximately 26 m (Table S1b).*"

Relative humidity and water content: Table 1:" Annual mean values, standard deviation (± SD), and statistical significance of the difference between both years, listed for the following parameters: daytime average of photosynthetically active radiation ($PAR_{avg}$), daily maximum of photosynthetically active radiation ($PAR_{max}$), *temperature, and water content. For the first three parameters, both above-canopy and microclimatic data (assessed at the different height levels) are shown, whereas for water content only microclimatic data have been collected (a).*"

Referee comment 31:

Table 2. There are no significant differences between seasons for some variables (for e.g., temperature), even though this is alluded to in the results (Pg. 11, line 24).

Author comment 31:

The statement on P11 L24 refers to a significant difference of the temperature between 23 m and above canopy measurements assessed during the dry season, which is listed in Table S5. We changed the text to make this aspect clearer.

Author change in the text 31:

P 12 L3:" At 23 m height, temperatures within bryophytes were frequently higher than the above-canopy values, and during the dry season even the average seasonal temperature of the bryophytes was 0.5°C higher (p≤0.001) than the above canopy average temperature (Tab. S5 , Fig. S7)."

**Figures: Generally, the figures need to be clearer, and larger, since you have several subplots.**

Referee comment 32:

Fig 2. PAR and Temperature at different heights are very hard to see. Either summarize differently, or show a mean in this figure and direct to a figure in the supplemental with data from all heights.

10  Author comment 32:

We tried to adapt the figures for more clarity.

[revised manuscript text omitted]

Referee comment 33:

Figure 3. This also has too many sub-panels crammed in one figure. In the caption, why do you say ecophysiological, micrometeorological and ambient parameters (the same is actually true of Fig. 1 as well). Which ones are which? Why are they called parameters? What are you trying to parametrize? I make a point about this, because this is one of several instances where words are not chosen carefully. Was humidity not measured at all heights?

Author comment 33:

We are aware of the fact that figure 3 is quite complex and needs some attention to be fully understood. On the other hand, we think that it gives a lot of information and allows direct comparisons between the different parameters and thus we would like to keep it in the current way.

With the term "parameters" we refer to environmental parameters, like temperature, precipitation, light intensity, etc., also called climate parameters or climate factors. We changed the captions to clarify the parameters, which are presented. The relative air humidity was measured at 26 m, just above the canopy, while the water content was measured at all four height levels within the bryophyte cushions.

Author change in the text 33:

Figure 3: "*Mean diurnal cycles of water content, temperature, and light condition of bryophytes, and above-canopy* meteorological parameters …"

Referee comment 34:

Figure 4. The histograms are informative but the information provided in the various shaded regions is extremely hard to follow. In the end, I do not understand what the authors are trying to convey. Why is the y-axis broken in the histograms in the left most panel?

Author comment 34:

Figure 4 was adapted to make it clearer and easier comprehensible.

The y-axis in the left-hand panel (PAR) is broken, as the lowest light intensity was reached at a frequency between 50 and 70%. All the higher light intensities occurred at frequencies of only a few or even below 1%. To show both the high and low frequencies at good resolution, we decided to use a broken y-axis.

**References provided by the author:**

[revised manuscript text omitted]

Zotz, G., Büdel, B., Meyer, A., Zellner, H. and Lange, L.: Water relations and CO2 exchange of tropical bryophytes in a lower montane rain forest in Panama, Bot. Acta, 110, 9–17, doi:10.1111/j.1438-8677.1997.tb00605.x, 1997.
* * *
**References provided by the referee:**

Gradstein, S.R., Churchill, S.P. & Salazar A., N. 2001. Guide to the Bryophytes of Tropical America. Memoirs New York Bot. Garden 86: 1-577.

Gradstein, S.R. 2006. The lowland cloud forest of French Guiana – a liverwort hotspot. Cryptogamie, Bryol. 27: 141-152.

Gradstein, S.R. & Ilkiu-Borges, A.-L. 2009. Guide to the Plants of Central French Guiana. Part IV. Liverworts and Hornworts. Memoirs of the New York Botanical Garden 76, 4: 1- 140, 83 plates.

Gehrig-Downie C., Obregón A., Bendix J., Gradstein S.R. 2013. Diversity and vertical distribution of epiphytic liverworts in lowland rain forest and lowland cloud forest of French Guiana. Journal of Bryology 35: 243-254.

---

## Author Comment (AC5) · 12 Jun 2019

**Response to referee comments and suggestions on bg-2018-521 by N. Löbs et al.: "Microclimatic and ecophysiological conditions experienced by epiphytic bryophytes in an Amazonian rain forest"**

**Manuscript format description:**
Black text shows the original referee comment, and blue text shows the response of the authors and the explicit change in the text. The figure and table numbers refer to in the revised manuscript.

**Anonymous Referee #3 submitted the comments RC4**

**General referee comments:**

Dear Editor, dear authors

I have read with interest the manuscript entitled "Microclimatic and ecophysiological conditions experienced by epiphytic bryophytes in an Amazonian rain forest" by Löbs et al. submitted to Biogeosciences. Please find my comments related to it below:

I appreciate a strong point in this manuscript that is to contribute to raise the data availability regarding cryptogamic covers functional performance in tropical regions, and going further, the lack of data available in Central and South America. It seems that almost all the literature regarding this issue has been focused in Polar Regions some years ago and in drylands at the present. I also appreciate the novelty and the effort made to provide microclimatic data sets at those heights at the tree trunks. If we want to understand properly the relevance of these organisms in global cycles and their response under environmental changes a huge and very different biome as the tropics cannot be ignored. I think that authors do a complete revision of the literature available and try to contribute from there with their data. Mosses dominate cryptogamic covers in tropical regions in biodiversity, so the target organisms in the study seems to be quite correct.

General author response:

We would like to thank the reviewer for appreciating our work and for the efforts spent on our manuscript. His/her comments helped us to substantially improve it.

Referee comment 1:

But, at the same time, my opinion is that this lack of data availability in the region is an intrinsic weakness of the manuscript. My point here is that the manuscript is based in a double assumption rather than in strictly measured data sets. The first assumption would be the water content of the bryophytes through conductivity sensors.

I appreciate the effort made by the authors calibrating this methodology in the lab and this experimental testing gives higher credibility to the measurements. But then we see the big second assumption that is to extrapolate data taken from the literature to understand the functional performance of the bryophytes in the altitudinal gradient. I think that it is likely that possible inaccuracies could arise in this sense. Data available

in the literature is little, so, it must be difficult to find similar experimental designs that could help providing reliable extrapolations. I am not talking about finding same species with data available in the literature, but it would be interesting, in order to trust the ecophysiological data provided, to have data from a similar habitat following at least the light adaptation patterns of the species included in this work.

As I suppose that these data sets are very difficult to get, but I think that this manuscript is interesting and useful to the scientific community, I would make a proposal to the authors:

What about to include in your manuscript a few gas exchange checkpoints in the lab including relevant species inside the gradient. For example, one representative species in the understory and another one at the closer point of the canopy could serve as cardinal points to calibrate authors' predictions about net photosynthesis availability, time and amount of respiration and possible C losses, light cardinal points, adaptation strategies. This would improve the discussion substantially from my point of view.

I am not asking for a complete gas exchange profile of the species included in the study because I know how time consuming this technique is, just a few replicated checkpoints in the lab to see how close predictions are from reality. If they were far from each other, the real gas exchange parameters measured could work as a more reliable source of predictions than a very likely imprecise literature for the aim targeted. I would welcome further assumptions at this point, but based in some real measured values (I said in the lab because conditions are easier to control, but some field gas exchange data sound good for me also). I think that this could improve the manuscript and put it as a reference text in tropical epiphytic bryophytes functional performance due to the low amount of literature available.

Author comment 1:

Thank you very much for these constructive ideas on $CO_2$ gas exchange measurements. It indeed would be good to include some measurements conducted by ourselves. However, from past experiences we know that quick gas exchange measurements might deliver truly misleading results. Just as an example, it has been shown by colleagues, that after transport to the lab, tropical organisms showed only a fraction of the physiological activity previously assessed in the field. The samples had strongly suffered from the transport, as they had to be air-dried prior to the transport in order to avoid molding during that time. Thus, we think that $CO_2$ gas exchange measurements indeed make sense, but that they also need to be conducted with care. This indeed is planned for the future, but would go beyond the scope of the current study. For the present study, we found some very good data on lowland rain forest bryophytes, assessed by a group, which is well-experienced in $CO_2$ gas exchange measurements. Thus, for the current study we decided to use their results in order to assess potential physiological activity patterns, but we also stress the potential sources of error and inaccuracy of this approach. We hope that we could convince you of the validity of this approach.

During the review process, we conducted a complete revision of the calibration process for the water content sensors resulting in by far smaller inaccuracies.

**Some minor points also to comment:**

**INTRO**

Referee comment 2:

Page 3, Ls 20-25: I would focus in bryophytes functional properties rather than in general physiological features of cryptogamic covers because only bryophytes are included in the experimental design.

Author comment 2:

Thank you for this comment. The whole introduction was revised, with the aim to focus more on the epiphytic bryophyte communities.

Author changes in the text 2:

P2 L 24: "Epiphytic cryptogam communities comprise photoautotrophic bryophytes, algae, lichens, and cyanobacteria in varying compositions, growing together with heterotrophic fungi, other bacteria, and archaea. They can colonize plant surfaces in almost all habitats throughout the world (Büdel, 2002; Elbert et al., 2012; Freiberg, 1999). Epiphytic bryophytes in the tropics play a prominent role in environmental nutrient cycling (Coxson et al., 1992; Zotz et al., 1997) and also influence the microclimate within the forest (Porada et al., 2018), thus contributing to the overall fitness of the host plants and the surrounding vegetation (Zartman, 2003). However, they are equally affected by deforestation and an increasing fragmentation (Zartman, 2003; Zotz et al., 1997).

Physiologically, cryptogamic organisms in general and specifically also bryophytes are characterized by their poikilohydric nature, as they do not actively regulate their water status, but passively follow the water conditions of their surrounding environment (Walter and Stadelmann, 1968). In a dry state, many bryophytes can outlast extreme weather conditions, being reactivated by water (Oliver et al., 2005; Proctor, 2000; Proctor et al., 2007; Seel et al., 1992), and for several species also fog and dew can serve as a source of water (Lancaster et al., 1984; Lange et al., 2006; Lange and Kilian, 1985; Reiter et al., 2008). Accordingly, their physiological activity is primarily regulated by the presence of water and only secondarily by light and temperature (Green and Proctor, 2016)."

**METHODOLOGY**

Referee comment 3:

-Section 2.5. Could you please explain in more detail why some meteorological parameters are measured at 26m and light is measured at 75m?

Author comment 3:

Monitoring of the meteorological parameters is conducted in the course of the overall ATTO long-term measurements (for more details see Andreae et al., 2015). For this, different sensors have been installed at different heights in order to serve the needs. Ambient light is measured at 75 m in order to avoid shading of the canopy and also precipitation and fog need to be measured above the canopy (at 81 and 50 m height, respectively). The different height levels are also explained by the different amounts of space needed by the sensors. We see that as uncritical for these parameters, as ambient light intensity, fog, and precipitation should not vary between 50 and 81 m height. For relative ambient air humidity and ambient temperature we decided to use the data closest to the canopy, i.e. at 26 m height.

In the revised version of the manuscript, we provide the information that the meteorological parameters are assessed in the course of the long-term monitoring at the site and we also provide a scheme illustrating the different sensor locations below, within and above the canopy. We hope this will clarify the sensor setup in some more detail.

Author changes in the text 3:

P8 L14: "The meteorological parameters have been measured within the ATTO project *in the context of longterm monitoring* since 2012.

[Figure]

**Figure S2:** Schematic overview on the sensors installed at different height levels below, within, and above the canopy. The parameters water content (WC) and temperature (Temp) were measured within the bryophyte samples, the light sensors (PAR) were installed directly on top of the thalli. The average tree height of 21 m was determined for the plateau forest in general.

Referee comment 4:

-Section 2.6. I would establish the possible ranges for each ecophysiological parameter analyzed focusing more in tropical epiphytic bryophytes functional performance.

Author comment 4:

Yes, this entire estimation was revised to restrict the considered values to epiphytic bryophytes of tropical lowland forests.

Author change in the text 4:

Changes in Table S3.

**Table S3:** Parameters determining fractional time of photosynthesis and respiration. The lower water compensation point (WCP$_l$), the lower light compensation point (LCP$_l$), the temperature for optimal net photosynthesis (T$_{opt}$ NP), and the upper temperature compensation point (TCP) as relevant parameters have been extracted from published studies conducted at lowland sites of tropical rain forests.

| Parameter | Low | High | Unit | Reference | Study site |
|-----------|-----|------|------|-----------|------------|
| WCP$_l$ | 30 | 80 | % DW | Wagner et al 2013 | Panama, lowland rain forest, 0 m |
| LCP$_l$ | 3 | 12 | $\mu$mol m$^{-2}$ s$^{-1}$ PPFD | Lösch et al. 1994 | Zaire, lowland rain forest, 800 m |
| T$_{opt}$ NP | 24 | 27 | °C | Wagner et al 2013 | Panama, lowland rain forest, 0 m |
| TCP | 30 | 36 | °C | Wagner et al 2013 | Panama, lowland rain forest, 0 m |

**RESULTS**

Referee comment 5:

-Section 3.1. 2 consecutive years of microclimatic data availability is a good and interesting output provided by authors

Author comment 5:

Yes, this is a long term monitoring project and the database on the water content, temperature, and light conditions of epiphytes is uploaded to the ATTO data portal (www.attoproject.org/).The data thus are maintained, obtain a doi and can be retrieved from that site.

Author comment 6:

Indeed, it was not an easy task to structure the manuscript in a logical way and in the end we decided to use a structure to analyze the data according to different time frames (i.e., comparison of years, seasons, diel cycles, etc.). Thus, indeed, different climatic parameters are sometimes used within one paragraph to illustrate their interdependence. However, we also considered this comment and looked over the structure within the paragraphs again. We now avoid mixing different parameters wherever this is possible.

Author change in the text 6:

Some structural changes will be made throughout the manuscript in order to obtain an overall easier readability.

Referee comment 7:

P 10 L9, I think that authors missed a word after "35%", maybe "lower"?

Author comment 7:

Yes, "35 % *lower*", we added the word.

Author change in the text 7:

P10 L20: "Comparing the two consecutive years, the effect of an El Niño event was clearly detectable, as rainfall amounts were 35 % *lower* (525 mm versus 805 mm) and relative air humidity 11 % lower (81 % versus 92 %) between October 2015 and February 2016 than in the previous year (Fig. 1, Table 1)."

Referee comment 8:

How did authors compare climate statistically between years/seasons? Did you use a monthly basis? Daily basis?

Author comment 8:

The statistical comparison between the years was performed on the basis of 5-minute data points. The statistical tests and the data base being used will be explained in more detail in the Section "2.6 Statistical analysis" and in the header of each table, were results are presented.

Author changes in the text 8:

Table 1: Annual mean values…"Mean values and statistical tests were calculated from 5-minute intervals, except for $PAR_{max}$, where the daily maximums values were considered."

Table 2: Seasonal mean values…" Mean values for the respective seasons were calculated from 5-minute intervals of the years 2015 and 2016, except for $PAR_{max}$, where the daily maximum values were considered."

Referee comment 9:

P 10 L 25-26. If I understood ok, the idea is that the microclimatic T value at the moss level was higher than ambient T, and that this is a frequent pattern. What about the shading effect of the tree canopy over microclimatic T?

Author comment 9:

Yes, indeed there is some shading effect of the canopy, which could result in a reduced heating of the bryophytes, also at 23 m height within the canopy. However, also ambient T measurements are always performed in the shade to avoid a short-term impact of direct insulation. Thus, we do not think that there is a large difference in shading. However, we think that wind intensities are reduced within and below the canopy and that the bryophytes have a higher heat storage capacity, which both may cause higher temperatures measured within the bryophytes.

Referee comment 10:

Fig 1, legend. I would say estimated water content of the bryophytes rather than "ecophysiological conditions"

Author comment 10:

The expression "*ecophysiological*" was finally omitted and was changed throughout the text and figures and replaced by "*conditions of bryophytes*".

Author change in the text 10:

Figure 1: "*Water content, temperature, and light condition of bryophytes, and above-canopy meteorological conditions experienced* in the Amazonian rain forest."

DISCUSSION:

Referee comment 11:

P 14 Ls 22-24. I think that these patterns observed reinforces that measuring some gas exchange control points might be useful.

Author comment 11:

We completely agree that additional $CO_2$ gas exchange measurements would be of interest. Our hesitation to measure just some cardinal points is explained in the first section of this response letter. We also explain there, that, under the current conditions, we prefer to use a well-established study over quick measurements conducted by ourselves. We prefer to conduct an in-depth $CO_2$ gas exchange study in the near future, which, however, goes beyond the scope of the current manuscript at hand.

Referee comment 12:

P 4 line 13: Remove "The".

Author comment 12:

Yes, the word was removed.

Referee comment 13:

P 17 Ls 19-23. I do not understand this point properly.

Author comment 13:

This paragraph was adapted to clarify the information. The intention was to express that respiration is more sensitive to temperature than photosynthesis.

Author change in the text 13:

*P 18 L9:* "Temperature regulates the overall velocity of metabolic processes. Whereas it has a strong impact on respiration, the photosynthetic light reactions are by far less sensitive to temperature (Green and Proctor, 2016; Lange et al., 1998). Since net photosynthesis is the sum of simultaneously occurring photosynthesis and respiration processes, positive net photosynthesis rates may still be reached at higher temperatures in the light, as long as the photosynthetic capacity is high enough, whereas during the night, high temperatures could cause a major loss of carbon due to high respiration rates (Lange et al., 2000).

**References provided by the author:**

Büdel, B.: History of flora and vegetation during the quaternary, in Progress in Botany, edited by K. Esser, U. Lüttge, W. Beyschlag, and F. Hellwig, pp. 386–404, Springer-Verlag., 2002.

Coxson, D. S., McIntyre, D. D. and Vogel, H. J.: Pulse Release of Sugars and Polyols from Canopy Bryophytes in Tropical Montane Rain Pulse Release of Sugars and Polyols from Canopy Bryophytes in Tropical Montane Rain Forest (Guadeloupe, French West Indies), Biotropica, 24, 121–133, 1992.

Elbert, W., Weber, B., Burrows, S., Steinkamp, J., Büdel, B., Andreae, M. O. and Pöschl, U.: Contribution of cryptogamic covers to the global cycles of carbon and nitrogen, Nat. Geosci., 5, 459–462, doi:10.1038/ngeo1486, 2012.

Freiberg, E.: Influence of microclimate on the occurrence of Cyanobacteria in the Phyllosphere in a Premontane Rain Forest of Costa Rica, Plant Biol., 1, 244–252, 1999.

Green, T. G. A. and Proctor, M. C. F.: Physiology of Photosynthetic Organisms Within Biological Soil Crusts: Their Adaptation, Flexibility, and Plasticity, in Biological Soil Crusts: An Organizing Principle in Drylands, edited by B. Weber, B. Büdel, and J. Belnap, pp. 347–381, Springer International Publishing, Cham., 2016.

Lancaster, J., Lancaster, N. and Seely, M.: Climate of the Central Namib Desert, Madoqua, 14, 5–61, 1984.

Lange, O. L. and Kilian, E.: Reaktivierung der Photosynthese trockener Flechten durch Wasserdampfaufnahme aus dem Luftraum: Artspezifisch unterschiedliches Verhalten, Flora, 176, 7–23, doi:10.1016/S0367-2530(17)30100-7, 1985.

Lange, O. L., Belnap, J. and Reichenberger, H.: Photosynthesis of the cyanobacterial soil-crust lichen Collema tenax from arid lands in southern Utah, USA: Role of water content on light and temperature responses of CO2 exchange, Funct. Ecol., doi:10.1046/j.1365-2435.1998.00192.x, 1998.

Lange, O. L., Allan Green, T. G., Melzer, B., Meyer, A. and Zellner, H.: Water relations and CO2 exchange of the terrestrial lichen Teloschistes capensis in the Namib fog desert: Measurements during two seasons in the field and under controlled conditions, Flora - Morphol. Distrib. Funct. Ecol. Plants, 201(4), 268–280, doi:10.1016/J.FLORA.2005.08.003, 2006.

Oliver, M. J., Velten, J. and Mishler, B. D.: Desiccation Tolerance in Bryophytes: A Reflection of the Primitive Strategy for Plant Survival in Dehydrating Habitats?, INTEGR. COMP. BIOL, 45, 788–799, 2005.

Porada, P., Van Stan, J. T. and Kleidon, A.: Significant contribution of non-vascular vegetation to global rainfall interception, Nat. Geosci., 11(8), 563–567, doi:10.1038/s41561-018-0176-7, 2018.

Proctor, M. C. F.: The bryophyte paradox: Tolerance of desiccation, evasion of drought, Plant Ecol., 151, 41–49, doi:10.1023/A:1026517920852, 2000.

Proctor, M. C. F., Oliver, M. J., Wood, A. J., Alpert, P., Stark, L. R., Cleavitt, N. L. and Mishler, B. D.: Desiccation-tolerance in bryophytes: a review, Bryologist, 110, 595–621, 2007.

Reiter, R., Höftberger, M., G. Allan Green, T. and Türk, R.: Photosynthesis of lichens from lichen-dominated communities in the alpine/nival belt of the Alps – II: Laboratory and field measurements of CO2 exchange and water relations, Flora - Morphol. Distrib. Funct. Ecol. Plants, 203, 34–46, 2008.

Seel, W. E., Hendry, G. A. F. and Lee, J. A.: The combined effects of desiccation and irradiance on mosses from xeric and hydric habitats, J. Exp. Bot., doi:10.1093/jxb/43.8.1023, 1992.

Walter, H. and Stadelmann, E.: The Physiological Prerequisites for the Transition of Autotrophic Plants from Water to Terrestrial Life, Bioscience, 18(7), 694–701, 1968.

Zartman, C. E.: Habitat fragmentation impacts on epiphyllous bryophyte communities in central Amazonia, Ecology, 84(4), 948–954, doi:10.1890/0012-9658(2003)084[0948:HFIOEB]2.0.CO;2, 2003.

Zotz, G., Büdel, B., Meyer, A., Zellner, H. and Lange, L.: Water relations and CO2 exchange of tropical bryophytes in a lower montane rain forest in Panama, Bot. Acta, 110, 9–17, doi:10.1111/j.1438-8677.1997.tb00605.x, 1997.

---

## Author Response (AR1)

**Response to referee comments and suggestions on bg-2018-521 by N. Löbs et al.: "Microclimatic and ecophysiological conditions experienced by epiphytic bryophytes in an Amazonian rain forest"**

5      Dear Professor Bahn,

we would like to thank you and the reviewers for the initial manuscript evaluation and the comments, which helped to improve our manuscript. We appreciate the opportunity to revise our manuscript to address the constructive comments and suggestions from
10      the reviewers. Below we respond with a point-by-point explanation to the comments from each peer-reviewer with our responses in blue color following every comment. At the end of the comments we provide the manuscript and the supplement with all changes being marked.

15      Sincerely,

Nina Löbs, on behalf of the co-authors.
* * *
20      **Maaike Bader as Referee submitted the comments RC1 and RC3**

**Comments on the text:**
Black text shows the original referee comment, and blue text shows the response of the authors and the explicit changes in the revised text. The figure and table numbers refer to the revised manuscript.

**Maaike Bader RC1**

**General Referee comment:**
Dear authors,

The manuscript "Microclimatic and ecophysiological conditions experienced by epiphytic bryophytes in an Ama-
30      zonian rain forest" presents interesting data about the microclimate experienced by epiphytic bryophytes in a tropical rainforest, as well as unique measurements of the time these organisms stay wet. Such data is indeed very

valuable for understanding the distribution and ecophysiological behavior of such mosses and liverworts. The data

are well-presented graphically at different time scales, showing seasonal and diel patterns. There are some issues about the presentation of the interpretation that need addressing though, as explained below.

General author response:

5  We would like to thank Maaike Bader for the very constructive review and the helpful comments, which helped us to identify the critical aspects and to improve our manuscript.

Referee comment 1:

It is clear that is a great effort to measure such data in a rain forest environment and the difficulty of canopy access.

10  Because of this, and because of the absence of comparable data, the lack of replication (all samples were located close together on one stem or branch section per height on the tree) can be 'forgiven', but it should be mentioned and evaluated in the text!

Author comment 1:

We fully agree that it would be preferable to install sensors on several different trees in order to have fully inde-
15  pendent true replicates. However, as you stated correctly, it is a great effort to install and run microclimate meas-
urements in such a rain forest environment. Thus, we installed several sensors at each height in order to cover the variability at least to some extent. We added some more information about the incomplete replications in the meth-
ods section to explain the limitation of the measurement setup.

Author changes in the text 1:

20  P 7 L12: "The restriction of the measurements to one individual tree needs to be considered, as a complete inde-
pendence of the replicate sensors could not be assured. However, due to the large effort of such an installation within the rain forest, it was not possible to equip more trees with additional instruments. Thus, the data obtained from the measurements on this individual tree should be considered as exemplary. "

25  Referee comment 2:

I am also very aware of the almost complete lack of basic ecophysiological data on gas exchange in tropical low-
land bryophytes, data being available for only 6 species, presented in Wagner et al 2013. However, I do not think that this justifies using data from tropical montane forest species, especially not for temperature responses, which differ along elevation (as shown in the cited paper by Wagner et al), but also not for water content responses,
30  because montane species experience very different water regimes and are likely to employ different strategies concerning the preservation and use of their water contents – that is to say, the 'community weighted mean' of the strategies is likely to be different. I do think that it is a valuable exercise to estimate activity times for net photo-
synthesis and net respiration, but I think the lack of physiological data to base this estimation on needs to be dealt with differently. Some of the cited parameters (which are from montane species) are so unlikely (like a lower

activity level for water content of 225%...) or uncertain (note that in Wagner et al it is explicitly mentioned that the absolute carbon exchange values should be treated with caution because of uncertainty in the absolute carbon exchange rates measured. This is not a problem for the optimum ranges (T and WC), but it is a problem for the compensation points, to which your calculation is highly sensitive. I would recommend to use only the lowland data and to use these data more loosely, using them combined with your common sense to estimate (or select) likely parameter values and presenting only theoretical calculations like " if we assume that the LCP is 6 µmol/m2/s, the total A and Rd times would be x and x% of the time, whereas a LCP of 1 µmol/m2/s would allow net A x% of the time". This is not fundamentally different from your current presentation, but you could avoid having to present estimations of 0-100%, which are not very helpful, and it would acknowledge the fact that gas exchange data for lowland species are simply not sufficiently available to really allow the type of estimates you would like to make at this point.

Author comment 2:

Thank you very much for this helpful comment! It is correct that the data collected at montane rainforest sites are not suited for a comparison and thus we now refrain to the lowland forest data (location BT) given in Wagner et al. (2013).

For the light compensation point (LCP) we include another reference for bryophytes in lowland bamboo forests with values of 3-12 µmol s²s⁻¹ for the LCP (Lösch et al., 1994).

For the current manuscript we omitted the information on saturation points, as we found that the current data are not well-suited for an inclusion of these data.

Author changes in the text 2:

All tables, figures, and the values in the text have been adapted according to the revised calculations and the wording is now more careful according to the referee suggestions.

Referee comment 3:

Considering my previous point this one may be obsolete now, but it is not clear how the parameters in table 3 and S2 or those presented in L17-18 P9 were selected from Wagner et al 2013. Also, a 'water content compensation point' was not presented in Wagner et al although the paper is cited for it.

Author comment 3:

As described above, we now only consider the BT site located at sea level (see Tab. 3 and Tab S3). Accordingly, we cite the optimum temperature range with 24°C limiting the lower and 27°C limiting the upper end of the range as reported in Tab. 3 of Wagner et al. (2013).

The lower and upper temperature compensation points "TCP" of 30.0 and 36 °C were reported in Wagner et al. 2013 Tab. 3 as the temperatures when "$T_{NP=DR}$" for the site BT at sea level.

Data on the water compensation point were extracted from Fig. 1 in Wagner et al. 2013.

Author changes in the text 3:
The text and tables (Tab 3, Tab S3) were adapted accordingly.

Referee comment 4:

Also, a lot of the statements about 'tropical bryophytes' are supported by literature from montane forests, and a lot of the statements about 'epiphytic cryptogams' are based on literature on lichens. This is not wrong but it is a bit deceiving. There would be nothing wrong with emphasizing, not only at the end of the discussion but right up front, that very little data is available for tropical lowland bryophytes and that therefore you need to rely on quite a bit of rough guessing and extrapolation of results from other areas and other organisms. As long as you make clear what your limitations are, they can be dealt with.

- So: make clear what literature is about lichens and what is about mosses – although these organisms have eco-physiological similarities, they are not the same in all respects! For example, enthanolic fermentation and bioaer-osols have been observed for lichens but not for bryophytes, or am I wrong?

- And: be very careful, and be explicit about it, with using parameters and process knowledge based on montane forests and on lichens.

Author comment 4:

Thank you for your comment; we are aware that lichens and bryophytes do not behave identically in all respects, although there are quite some similarities. Thus, in the revised version we stress whenever we use information on lichens. When comparing our results with those from other studies, we specify the rainforest habitat and organism group.

Author changes in the text 4:

P 13 L2: "Furthermore, high nighttime temperatures cause increased carbon losses due to high respiration rates, as previously shown for lichens (Lange et al., 1998, 2000)."

P 19 L 33: "Bryophyte and lichen taxa in the understory are known to be adapted to these low light conditions and are able to make efficient use of the rather short periods of high light intensities (Lakatos et al., 2006; Lange et al., 2000; Wagner et al., 2014)."

Referee comment 5:

Water content can hardly be called 'ecophysiological conditions', I would recommend removing this term from the title. To make sure that the innovative data on water content are in the title, you could consider changing it to "Microclimatic conditions and water content fluctuations experienced by epiphytic bryophytes in an Amazonian rain forest"

Author comment 5:

We agree with your comment on the 'ecophysiological conditions'. Your recommended change of the title is a good solution and thus we adapted it.

Author changes in the text 5:

Title: "Microclimatic conditions and water content fluctuations experienced by epiphytic bryophytes in an Amazonian rain forest"

Referee comment 6:

The statement "Our data suggest that water contents are decisive for overall physiological activity, and light intensities determine whether net photosynthesis or dark respiration occurs, whereas temperature variations are only of minor relevance in this environment." In the abstract, and the statement that 'water content has turned out to be key' is not justified by your results. It is probably the case, but this is not suggested by your data – it could not be and was not addressed in your study, as realistic data about gas exchange is missing.

Author comment 6:

Thank you for your comment. Yes, we indeed do not have $CO_2$ gas exchange data in this study. Nevertheless, we think that already the microclimate data by themselves and the calculations on potential activity patterns support our statement that water contents are highly relevant whereas temperatures are of minor importance for physiological activity. Nevertheless, we try to clarify these issues in the revised version of the manuscript.

Author change in the text 6:

Changes were made in the abstract and the discussion sections to clarify these issues.

P 2 L 24: "In general, bryophytes growing close to the forest floor were limited by light availability, while those growing in the canopy had to withstand larger variations in microclimatic conditions, especially during the dry season. These data may be used as a starting point to investigate the role of bryophytes in various biosphere-atmosphere exchange processes, such as measurements of $CO_2$ gas exchange, and could be a tool to understand the functioning of the epiphytic community in greater detail."

Referee comment 7:

There is a lot of information in the methods section that is superfluous or irrelevant, whereas other information is missing. Superfluous/irrelevant: P4 L 24-26, 29-32; P5 L13-15; Equations 5-8; P6 L20 brand name of styrodur.

Author comment 7:

We agree to delete some information on the study site (P6 L 23-24) and on the neighboring forest types (P6 L 27-30).The styrodur brand (P9 L25) and the equations 5-8 (P 10 L 18ff) have been deleted due to overall revision of the calibration process.

Referee comment 8:

There is basically no information about the statistical analyses other than in what software they were performed…
Please explain what was tested, what were your units of replications, etc.

Author comment 8:

We agree with the comment, there was not enough information regarding statistical analyses. However, with the new calibration procedure and the situation of statistical replicates (e.g., light n=2, temperature n=2, above canopy light n=1, above canopy temperature n=1, above canopy RH n=1), we decided to delete the statistical tests and shortened this section considerably (P 14 L6-9). Thus, we also renamed this section, being now called "Data analysis".

Referee comment 9:

I am a bit afraid that you have used days as replications to compare climatic variables between years – is 26.6° really different from 26.4° C, or even 25.8° is different from 25.8° (Table 1)?? With enough (pseudo)replication any tiny difference can become 'significant', but that does not make it real…

Author comment 9:

Thank you very much for that good advice. We thought about this test again and decided to delete it, as the values indeed are not independent.

Referee comment 10:

Please present your experimental design (what species, what positions, justification for the pseudoreplication), preferably early in the methods section.

Author comment 10:

More details (that were previously in the Supplement) are now included in the text, section "2.2 Microclimatic conditions within epiphytic habitat". Furthermore we added the new Figures S2 and S3 to the Supplement showing the distribution of all the sensors along the vertical gradient and the morphological characteristics of the bryophyte species.

Author change in the text 10:

P7 L16-26: "Generally, the WC sensors were placed in four different bryophyte communities being heterogeneously distributed along the four height levels. At 1.5 m height, the WC sensors were installed in communities dominated by *Sematophyllum subsimplex* (5 sensors) and *Leucobryum martianum* (1 sensor), at 8 m in *Octoblepharum cocuiense* (3 sensors) and *Symbiezidium barbiflorum* (3 sensors), and at 18 and 23 m in *Symbiezidium barbiflorum* (6 sensors at each height level; Fig. S2, Fig. S3). The temperature sensors were installed in the same communities at each height, and the light sensors were installed adjacent to them on ~ 5 cm long sticks (Fig. S1). As the morphology of the different species affects their overall WC, different maximum WC and patterns of the drying process were observed (Tab. S1). The sensors were installed with the following orientations: at 1.5 and 8 m

vertically along the trunk, at 18 m at the upper side of a slightly sloped branch, and at 23 m at the upper side of a vertical branch. Thus, also the orientation at the stem may influence the WC of the bryophyte communities, not only the species and the canopy structure."

Supplement: Figure S2 and Figure S3 (see below) were added to the supplement.

[Figure]

**Figure S2:** Schematic overview on the sensors installed at different height levels below, within, and above the canopy. The parameters water content (WC) and temperature (Temp) were measured within the bryophyte samples, the light sensors (PAR) were installed directly on top of the thalli. The average tree height of 21 m was determined for the plateau forest in general.

[Figure]

**Figure S3:** The four bryophyte species being used for installation of the sensors of the microclimate station. (A, D, G, J, K) overview, (B, H, L) leaf, (C, F I) cell form, and (E, M) cross section of a leaf.

Referee comment 11:

It was not clear whether you used the 5-minute resolution data for calculating the times for A and Rd, or whether you only used the half-hour smoothed data. The smoothed data are fine for studying seasonal differences, but for the activity times and for quantifying the frequency of sun flecks (which would be interesting to do!) I would recommend using the 5-minute data.

Author comment 11:

For a calculation of A and Rd, the 5-minute data were used, as written in Table 3. We additionally provide this information in the methods section on P11 L31.

Author change in the text 11:

P11 L31:"Thus, for all calculations the 30-minute averages have been considered, except for the estimates of phys-iological activity. "

You mention that the conductivity showed 'short-time oscillations' - could these be explained physically? Were they regular fluctuations or just general instability?

Author comment 12:

The oscillations of the sensors represent a general instability of the system, as the measured values oscillated around the actual values. Accordingly, the 5-minute data set was only used for an estimation of the physiological activity, as here the information on short-term events (as e.g. light flecks) is needed. For all other calculations, the 30-minute averages were used.

Author changes in the text 12:

P11 L30: "The measured electrical conductivity values showed short-time oscillations, which could be removed with a 30-minute smoothing algorithm (Fig. S4). *Thus, for all calculations the 30-minute averages have been considered, except for the estimates of physiological activity*."

Referee comment 13:

Limitations should not only be acknowledged for the availability of gas-exchange parameters, but also, and early in the manuscript, for the measurements themselves. In particular, the quality of the WC calibration curves could be a problem. The calibration graphs show that there is indeed great variation between samples and between measurements, and that the models do not reflect the water contents very well even for the calibration data. As an example for the variability, the curves show that a conductivity of 800 mV (why is conductivity expressed in mV?? Should this not be in Ohm?) in Symbiezidium could be caused by a water content anywhere between 300 and 1700 %. What is the effect of this uncertainty on your results? For Octoblepharum the model underestimates the WC over much of the range (can this explain the low WC at 8 m?). For Sematophyllum the maximum conductivity measured in the field greatly surpasses the maximum values measured during calibration, which will, by the looks of it, results in a very high estimated water content even with the exponential correction. Why are these models not drawn for the whole range of measured conductivities? For example, the quadratic function for Leucobryum would mean that a very high conductivity, like the 1000 observed in the field, would indicate a lower WC than intermediate values. If you do not draw the whole curve, this potential artifact cannot be evaluated well.

Author comment 13:

After major considerations, we decided to thoroughly change the calibration routine.

Thus, also based on your reviewer comment 3, we decided to test and use an alternative approach for the calibration of the water content. For this, the maximum and minimum values of electrical conductivity reached in the field were assume that they are reached at the maximum and minimum water contents reached by the samples, whereas

the amplitude of the water contents was determined based on the laboratory measurements. In the new approach we assume, that the maximum electrical conductivity in the field is achieved at the maximum water content, as determined in the laboratory.

Author change in the text 13:

P11 L4: "A calibration was conducted for all the communities dominated by different bryophyte species. For this, samples of them were collected in the forest area surrounding the ATTO site. They were removed from the stem with a pocket knife and stored in paper bags in an air conditioned lab container until calibration (few hours after collection). Prior to the calibration, the samples were cleaned from adhering material using forceps. The weight of the bryophytes was determined when they were moistened until saturation (temperature 30° C, RH 100 %) and again after drying in a dryer overnight (temperature 40° C, RH 30 %) to simulate the natural range of the WC under controlled temperature and RH. The dry weight ($DW$) was determined after drying at 60° C until weight consistency was reached (Caesar et al., 2018). The WC of the sample was calculated according to the formula in Weber et al. (2016):

$$WC \ [\% \ DW] = \frac{(FW - DW)}{DW} * 100 \ \%, \tag{3}$$

with $FW$ as sample fresh weight [g] and $DW$ as sample dry weight [g].

The calibration of the water content was performed, based on the maximum and minimum values of electrical conductivity reached in the field and the amplitude of the WCs reached during the laboratory measurements. We assume, that the maximum electrical conductivity achieved in the field equals the maximum WC achieved in the laboratory due to water saturation of the samples during the laboratory measurement. Minimum electrical conductivity values reached in the field were assumed to correspond to air-dry samples, as we are confident that the samples dried out at least once during the dry season of the year. Accordingly, the water content (WC) was calculated as follows:

$$WC \ [\% \ DW] = \frac{(EC_i - EC_{min})}{(EC_{max} - EC_{\min})} * (WC_{max} - WC_{min}), \tag{4}$$

with $EC_i$ as electrical conductivity, $EC_{min}$ as minimum electrical conductivity, $EC_{max}$ as maximum electrical conductivity in the field, $WC_{max}$ as the maximum WC in the laboratory, and $WC_{min}$ as the minimum WC in the laboratory.

Referee comment 14:

Also, the observation that water saturation was never reached at the 3 higher levels seems to suggest that something was wrong either with your WC measurements or the literature parameters used… BUT, this statement (P13, L24) cannot be true based on your data, because *Symbiezidium* is present only in these three higher levels, and in the calibration curves you show that observed values go up to 1500% WC, which is well above the WSPs cited…

Author comment 14:

Maaike Bader RC1                                                                                              10

You are right, according to the original calibration the WSP of 349 % WC was almost never reached in the canopy (at 23, 18, and 8 m), whereas it was reached during 22% of the time at 1.5 m height. As described above, the original calibration has been replaced and we also restrict the calculations to the compensation points, which are more relevant than the saturation points in the current context.

**Referee comment 15:**

It was unclear to me what "upper three height levels the bryophyte taxa could not be securely determined. Thus, the bryophyte taxon with the highest abundance in the canopy communities, i.e., the liverwort *Symbiezidium barbiflorum* was used" means exactly. Did you install sensors only in this species, or did you do the calibration curve

10 only for this species and then use if for all the different (unidentified) species sampled at the higher height levels? This should be made clearer. I could imagine that you installed sensors in other liverworts looking similar to Symbiezidium and then assumed that the relationship between electrical conductivity and water content should not be more different between species than within species, due to the similar life form. This seems a reasonable assumption, but should be made explicit, and in table S1b the species should not be named if you do not know the

15 real name. Indicating if it was a moss or a liverwort, or the family it belongs to, would be useful though!

**Author comment 15:**

The sensors were installed in bryophytes morphologically similar to *Symbiezidium barbiflorum*. However, as the sensors were installed by a climber, it could not be completely reassured that always the same species was used. Nevertheless, we know that *Symbiezidium barbiflorum* was the most dominant species in the canopy of this tree,

20 and from all the information we have for each sensor, the identification of these samples should be correct. Furthermore, we clarified that bryophyte communities have been investigated and not single species, as in most cases a community of different species grows together. This was corrected accordingly in the text.

**Author change in the text 15:**

P7 L8: "It needs to be mentioned, that not only one single species was measured by one sensor, but usually several

25 bryophyte species and also other cryptogams, such as lichenized and non-lichenized fungi and algae, as well as heterotrophic fungi, bacteria and archaea, which grow together forming a cryptogamic community. Thus, the organisms mentioned throughout this paper were the dominating but not solitarily living species.."

**Referee comment 16:**

30 The use of different species at the different heights is a problem that also needs to be discussed earlier and more prominently and included in the analysis. It reads all through the manuscript as though differences in water content between height zones were caused by microclimatic differences, but of course a Leucobryum (cushion moss with specialized water-holding cells) is going to have very different water content dynamics that a *Symbiziedium* (prostrate leafy liverwort), even under the same environmental conditions. This is also obvious from your own data in

the calibration curves, the points for Leucobryum being much closer together, indicating that the drying was much slower than e.g. for *Symbiezidium*. For Oc*t*oblepharum the two (! Looks like they were only two though you write they were three) samples dried at quite different speeds, it looks like the slow sample was denser and thus had higher conductivity at similar water contents. At the moment, the whole manuscript reads a bit as though you consider all cryptogams are expected to respond more or less the same, but we know that there are big differences between species, in particular in terms of water-content dynamics as well as the responses to this water content. Although you do mention this briefly, I think it deserves a few more words at least.

Author comment 16:

Indeed, we mixed up the replicate numbers in the Supplement Figure S3, however, this information is obsolete, due to recalculations.

Regarding the behavior of different species during the drying process, this section was extended with more information.

Author change in the text 16:

P22 L3: "The high WC of the bryophyte samples in the canopy might be partly explained by the different water holding capacity of different bryophyte species (Lakatos et al., 2006; Romero et al., 2006; Williams and Flanagan, 1996)."

P22 L24: The microenvironmental conditions influence the WC of epiphytic bryophyte communities, but the ability to deal with these conditions differs among species (interspecific variability), being determined by morphological and physiological features. Apart from the long-term adaptation of the metabolic properties, the performance of species under differing microenvironmental conditions can also be modulated by acclimation processes (intraspecific variability), as, e.g., shown for bryophytes and lichens (Cornelissen et al., 2007; Pardow et al., 2010). "

Referee comment 17:

It would be really cool if you could detect a dew signal in the WC data, did you look for this? Mention this in the discussion to but the dew remarks into the context of your data.

Author comment 17:

This indeed is a relevant aspect, and we also considered if we could calculate these values. However, in order to calculate them we would need the temperature below the bark. Due to the lack of this information we cannot calculate dew formation.

Referee comment 18:

It would also be cool if you could detect relationship between cryptogam activity patterns and measured trace gas emissions – this tall canopy site would be one of the few places in the world where the needed data might be available, assuming that trace gases above the canopy are also monitored?

Author comment 18:

Yes indeed, different trace gases are monitored at this study site and investigations on this are planned for the near future. This, however, is beyond the scope of the current manuscript.

5    Referee comment 19:

The literature cited needs to be revised! Only few bryophyte papers are cited and often they are not the correct ones (see below)! Some examples:

p. 3, lines 15-16: Zotz et al 1997 is cited a lot but refers to a montane forest, and not to nutrient cycling, as suggested on this occasion.

10    p.8, lines 30-31: 'at least in the environment of the central Amazon' is followed by references out of which none are from the central Amazon, most are from cloud forest…(by the way, this sentence is more or less repeated on page 12, L 29-31)

p. 9, lines 5-6: 'For tropical species, values (of WCPl) in the range 5 between ~ 30 and ~ 225 % have been determined (Romero et al., 2006; Wagner et al., 2013; Zotz et al., 1997, 2003)' Again, these references are all from

15    montane species or do not mention WCPl values at all.

p. 6, lines 10 - 12: "Thus, the bryophyte taxon with the highest abundance in the canopy communities, i.e., the liverwort *Symbiezidium barbiflorum* was used (Gradstein and Allen, 1992; Mota de Oliveira et al., 2009; Mota de Oliveira and ter Steege, 2015; Pardow et al., 2012; Romanski et al., 2011; Sporn et al., 2010)." Of the 6 references cited here, *S. barbiflorum* is only mentioned in Gradstein and Allen (1992), the other 5 references do not cite this

20    species at all! (one of the papers cited, Sporn et al. 2010, even deals with Asia even though S. barbiflorum does not occur there, being restricted to America…). Interestingly, Gradstein and Allen (1992) state that S. barbiflorum is a characteristic shade epiphyte of forest understory communities, not canopy communities. Not-cited more recent publications on the habitat of *S. barbiflorum*, however, show that the species also occurs in the forest canopy (Gradstein et al. 2001, Gradstein 2006, Gradstein & Ilkiu-Borges 2009, Gehrig et al. 2013). These recent papers

25    show that S. barbiflorum is actually an ecological generalist, occurring in understory communities as well as in canopy communities. None of these non-cited papers document highest abundance of the species in canopy communities. Thus, the sentence on p. 6, lines 10-12, is rather wrong.

p. 3, line 12-13: "In 2013, 800 species of mosses and liverworts …,… have been reported for the Amazon region" (Mota de Oliveira & ter Steege 2013). The reference cited here is quite wrong, Mota de Oliveira & ter Steege did

30    not provide this number at all, instead they took it from Gradstein et al. (2001; correctly cited by Mota de Oliveira & ter Steege) who calculated 800 species in the Amazon region in their book based on a full-scale analysis of the bryophyte flora of the Neotropics. Thus, the correct reference here is Gradstein et al. (2001) and not Mota de Oliveira & ter Steege.

Author comment 19:

Regarding P3 L 15-16, (Now P3 L7): reference was removed

Regarding P8 L30-31, (Now P12 L30): reference was changed; the repeated sentence was removed from P20 L21

Regarding P9 L5-6, (Now P 13 L5): The references of montane cloud forests (Romero et al. 2006, Zotz et al. 1997, Zotz et al. 2003) were removed. Only the information related to a research site at sea level in Wagner et al. 2013 was considered. The information for the WCP there is was extracted from figure 1. Perhaps we could obtain the exact values from you?

Regarding P6 L10-12, (Now P9 L12): Indeed, this sentence in its reported version was wrong. Initially, this sentence intended to tell that in general in lowland rainforest liverworts are more abundant in the canopy than in the understory, which was observed by Pardow et al. 2012. However, due to internal revisions and changes in the text, the sense of this sentence changed, unfortunately resulting in a wrong statement. The sentence meanwhile was . Furthermore, in the Material and Method section we mention that we talk about dominating species.

Regarding P3 L12-13, (Now P3 L20): We corrected the reference according to your advice.

Author change in the text 19:

P12 L30: "The physiological activity of bryophytes – and of cryptogams in general – is primarily controlled by water and light, whereas temperature plays a secondary role – at least in the environment of the central Amazon (Lösch et al., 1994; Wagner et al., 2013)."

P13 L5: "For tropical species in lowlands near sea level in Panama, values in the range between ~30 and *~80 %* have been determined (Wagner et al., 2013; Table S3)."

P9 L12: deleted; equal information now provided in P7 L 8: "Thus, the organisms mentioned throughout this paper were the dominating but not solitarily living species."

P3 L20: "By 2013, 800 species of mosses and liverworts, 250 lichen species, and 1,800 fungal species have been reported for the Amazon region (Campos et al., 2015; Gradstein et al., 2001; Komposch and Hafellner, 2000; Normann et al., 2010; Piepenbring, 2007).

Referee comment 20:

Data availability: does this local database assure future data maintenance and retrieval? Please provide more details.

Author comment 20:

Yes, this is a long term monitoring project and the database on the water content, temperature, and light conditions of epiphytes is uploaded to the ATTO data portal (www.attoproject.org/).The data thus are maintained, obtain a doi and can be retrieved from that site.

Referee comment 21:

General: rather than 'mesoclimate', 'above-canopy climate' would be a more intuitive name for those measurements.

Author comment 21:

We agree with your advice to rename the "mesoclimate" to "above-canopy climate". Accordingly, this expression was changed throughout the text. Furthermore, the expression "ambient" was changed into "above-canopy".

Referee comment 22:

P3 L 9: instead of 'these' write 'such' (this is an example of the confusing mix of literature and statements about cryptogam communities in general (often based on soil crusts…) and on tropical lowland epiphytes.

Author comment 22:

P4 L31: We agree to substitute "these" by "such", however, this sentence was deleted in the meanwhile.

Referee comment 23:

P3 L 21: careful, not all bryophytes are desiccation tolerant, even if they are poikilohydric

Author comment 23:

Yes, we agree on that and added the expression "most species" and "for many species".

Author change in the text 23:

P 3 L12: "In a dry state, many of them can outlast extreme weather conditions, being reactivated by water (Oliver et al., 2005; Proctor, 2000; Proctor et al., 2007; Seel et al., 1992), and for several species even fog and dew can serve as a source of water (Lancaster et al., 1984; Lange et al., 2006; Lange and Kilian, 1985; Reiter et al., 2008).

Referee comment 24:

P4 L4-6: Add that most of this info is based on data from soil crusts and from temperate zones and that very little is known about biomass and functions of epiphytic cryptogam in tropical forests, especially in the lowlands.

Author comment 24:

That is right, that most of the fluxes were detected for soil communities. However, the information on VOC and aldehydes was performed on epiphytic lichens as well (Kesselmeier et al., 1999; Kuhn et al., 2000; Kuhn and Kesselmeier, 2000; Wilske and Kesselmeier, 1999).

This information was omitted in the meantime, due to reorganization of the whole section.

Referee comment 25:

P4 L 8: seasonal variation in what?

Author comment 25:

…the seasonal variation of climatic conditions.

The whole section was revised and this sentence was removed.

Referee comment 26:

P5 L2: why 'ecophysiological' water content? What other water content is there?

Author comment 26:

It is the "normal" water content of bryophytes, thus the word "ecophysiological" can be deleted.

Author change in the text 26:

P7 L2: "The parameters temperature and light within/on top of the bryophytes *and their water content* were measured with a microclimate station installed in September 2014 (Fig. S1)."

Referee comment 27:

P5 L3: use 'were' rather than 'are being', even if the measurements are continuing, because you are here presenting results of a specific period in the past. Same for P5 L 11: were taken (not have been taken)

Author comment 27:

We agree on that and changed the tenses accordingly.

Author change in the text 27:

P7 L3: "The parameters temperature and light within/on top of the bryophytes and their WC were measured with a microclimate station installed in September 2014 (Fig. S1)."

P7 L30: "Since the installation, automatic measurements at 5-minute intervals *were taken* with a data logger…"

Referee comment 28:

P5 L 5: instead of 'described by' use 'used by', because 'described' suggests that these zones were the output of a study, but it was the sampling design.

Author comment 28:

Done accordingly.

Author change in the text 28:

P7 L4: "…, corresponding to the zones 1 to 4 *used by* Mota de Oliveira and ter Steege (2015)."

Referee comment 29:

P5 L8: a cushion is a specific bryophyte life form, seeing your species the samples probably were not cushions in most cases…You could use 'bryophyte samples'.

Author comment 29:

The information is now provided in another sentence, due to revision. Overall, we talk about bryophyte communities, now.

Author change in the text 29:

P7 L20: "The temperature sensors were installed in the same communities at each height, and the light sensors were installed adjacent to them on ~ 5 cm long sticks (Fig. S1).

5    Referee comment 30:

P5 L 19: what do you mean with 'fluctuations'?

Author comment 30:

With "fluctuation" we meant to describe the oscillations of the measurement. This was changed accordingly.

Author change in the text 30:

10    P8 L10: "The WC values were *oscillating, causing an inaccuracy corresponding to* approximately 15 % dry weight (DW)."

Referee comment 31:

P6 L17: are nutrient content and temperature species-specific?

15    Author comment 31:

Yes, the nutrient content is species-specific. But the temperature cannot be actively regulated by the species, thus it is not species-specific, but dependent on the environment. However, both parameters influence the measurements of electrical conductivity, hence it is recommended to include an assessment of the species-specific nutrient contents and to do a correction by temperature, to receive the most accurate values.

Referee comment 32:

P7 L1: what is the sensor weight?

Author comment 32:

During the calibration the sensor is fixed in the bryophyte sample and both are lying on the balance. Accordingly,

25    the balance always reads and logs the total weight of 'sample plus sensor'. But as the weight of the sensor varies slightly depending on the tension of its wire, the weight of the sensor is not the same for each 'set of calibration measurements'. Thus, the weight of the dry' sample plus sensor' is recorded at the end of the measurement, as soon as weight consistence is given. Afterwards the sensor with its wire is removed from the sample and then only the weight of the sample can be recorded.

30    But the whole section was revised, and this sentence was removed.

Referee comment 33:

P7 L12: rather that presenting the models, which are very standard (except maybe for the exponential correction; if you want you could show the models in the appendix), a discussion about uncertainty propagation would be fitting here.

Author comment 33:

Due to recalculation of the WC the fits are omitted from the new version of the manuscript. A discussion on the inaccuracy of the measurements and the uncertainty of the resulting values is included in the material and methods section.

Author changes in the text 33:

P8 L10: "The WC values were oscillating, causing an inaccuracy corresponding to approximately 15 % dry weight (DW). Besides the specific position in the substrate, the WC also depended on the texture of the sample material, its ion concentration, and the temperature. Because of all these factors influencing the sensor readings, the provided values of the WC should be considered as the best possible estimates and not as exact values. "

P22 L21: "This variability of data, depending on the exact placement of the sensors, illustrates that calculated WCs could only be considered as approximate values"

Referee comment 34:

P8 L16: rainfall amounts would usually not be calculated by integration but by adding the rain amount (e.g. number of tipping events) per time period…

Author comment 34:

Yes, you are right, the rainfall amount should be summed up for certain time periods. However, sometimes rain detection was interrupted and data of short time gaps were missing. For these time periods, we decided to integrate the data not to underestimate the amount too much. We are aware of the fact, that these data gaps and the subsequent calculations may be a source of over- or underestimation. However, this to our knowledge, is the best way to deal with this problem. In addition, the data gaps were relatively small, thus not being a major source of error.

Referee comment 35:

P8 L26: explain 'UTC values'; and where are such times presented, and why not always use local time?

Author comment 35:

The UTC is the abbreviation of the universal coordinated time and it is used throughout this study. It allows the synchronization with other data sets. The local time (LT= UTC-4) is only used for the calculation and presentation of diurnal cycles, where it is explicitly marked. The UTC time is used for long data ranges, as monthly and seasonal data.

Author change in the text 35:

P12 L26:” Time readings are always presented as UTC (universal coordinated time) values, except for diurnal cycles, where local time (LT, i.e., UTC-4) is shown, as labeled in the figures.”

Referee comment 36:

P9 L23: This WCPl is not what you describe it to be (this would not be a compensation point), it is the point below which the WC is so low that photosynthesis cannot compensate respiration, respiration ceasing at lower WCs than photosynthesis.

Author comment 36:

Here indeed was a mistake. With the WCPl we wanted to explain the point, when net photosynthesis equals respiration, due to limited water availability. This was now changed in the text.

Author change in the text 36:

P13 L3: “The lower water compensation point (WCP) presents the minimum WC that allows positive *net photosynthesis.*”

Referee comment 37:

P9 L28: with ‘we found’ you mean ‘we assumed’?

Author comment 37:

Yes, “we assumed” is what we meant, but we found that we can delete this part of the sentence.

Author change in the text 37:

P13 L32:” The compensation points for the different parameters are also to some extent interrelated, e.g., the water compensation point of lichens has been shown to slightly increase with increasing temperature (Lange, 1980), but this can be neglected in such a first qualitative approach.”

Referee comment 38:

P10 L17: report the statistical results (test and test statistics)! This goes for all ‘significant’ (or non-significant) results.

Author comment 38:

Indeed, we missed to provide detailed information of the statistical test result. However, in the context of the revision, we decided to omit statistical tests.

Referee comment 39:

P11 L1: ‘The RH..’What RH? It is generally not always clear in the text what parameter you are talking about: daily means, monthly means, something else?

Author comment 39:

This section deals with the 'annual fluctuations of monthly mean values'. Accordingly the RH should be understood as the monthly mean. However, this sentences (P15L9) was omitted in the context of the revision.

Referee comment 40:

P12 L25: word missing

Author comment 40:

Done.

Author change in the text 40:

P17 L13: "At 23 m height, also the daily amplitudes tended to be higher during the dry compared to the wet seasons, whereas for the mosses at the lowest height levels the amplitudes tended to be higher during the wet season. For bryophyte communities at the other height levels the *amplitudes during the different seasons* were less clear.

Referee comment 41:

P13 L16-18: it would be relevant to mention whether such high temperatures were ever reached in wet bryophytes; I would expect that they would only occur while samples were dry.

Author comment 41: Your assumption indeed is right. We included Figure S8 (see below) showing the relation of temperature and water content at different heights along the tree and mentioned the temperature/WC relation in the text.

Author change in the text 41:

P24 L6: "Thus, the temperature did not seem to be a limiting factor for the physiological activity of epiphytic bryophytes in this environment (Fig. S8)."

[Figure]

Figure S8: Temperature condition of bryophytes related to their water content. The temperature was measured in bryophytes at different height levels along the tree. Data presented as 30-minute averages.

Referee comment 42:

P13 L 27: I guess you mean the LOWER end of the WCPl range?

Author comment 42:

Yes, indeed.

Author change in the text 42:

P25 L34:" Whereas the *lower* end of the WCP range (30 % DW) is reached during 100% of the time by the liverworts…

Referee comment 43:

P14 L6: you mean 'height', not 'altitude' here.

Author comment 43:

Yes, it should mean 'height'.

Author change in the text 43:

P19 L9: "The microclimatic conditions experienced by bryophyte communities along *a height* gradient at the ATTO site followed…"

Referee comment 44:

P14 L6-7: 'The microclimatic conditions experienced by bryophytes along an altitudinal gradient at the ATTO site follow the meteorological characteristics to some extent' - this needs some reference to time…

Author comment 44:

This was meant in a more general way and we think it makes sense to get somewhat more specific for the different sensor types and heights. Thus, we added a sentence as outlined below.

Author change in the text 44:

P19 L9: "The microclimatic conditions experienced by bryophytes along a height gradient at the ATTO site followed the meteorological parameters to some extent, but they also revealed microsite-specific properties regarding annual, seasonal, and diel microclimate patterns. Whereas water content and temperature readings mostly followed the patterns of the meteorological parameters precipitation and temperature, the light intensities were clearly altered, particularly at the lower levels of the canopy. "

Referee comment 45:

P14 L15-17: mention in methods

Author comment 45:

It is a good idea to provide this information in the methods section. We also rephrased this sentence in the discussion.

Author change in the text 45:

P7 L23: "*The sensors were installed at the following orientation: at 1.5 and 8 m vertically along the trunk, at 18 m at the upper side of a slightly sloped branch, and at 23 m at the upper side of a vertical branch.*"

P19 L19: "This was most probably an effect of the canopy structure, cushion orientation, and shading. The sensors *at 1.5 and 8 m were installed vertically along the trunk, at 18 m height they were placed on the upper side of a slightly sloped branch, and at 23 m they were positioned on the upper side of a vertical branch.*"

Referee comment 46:

P14 L18: 'may have periodically shaded the organisms': it seems to me that you can have observed whether this was the case or not: were any leaves situated close to these sensors? (Same for P16 L7-8)

Author comment 46:

As the sensors were located at 8, 18, and 36 m height, they were out of direct sight for us. One would have needed to install cameras to explore this over time. Maybe it is better to use the expression "may" instead "can".

Author change in the text 46:

P 19 L24: "As the light sensors at 23 m height were located within the canopy, newly growing leaves may have periodically shaded the organisms, which *may* explain the lower monthly PAR$_{avg}$ values at this height level compared to the values at the lower levels."

Referee comment 47:

P14 L20: was PARavg not the monthly average? Do you mean the monthly averages of the daily patterns?

Author comment 47:

We intended to differentiate between *PARavg* and *PARmax*. While PARavg is the average of a certain period, then specified for the duration, i.e. month or hour, the PARmax is the maximum PAR value reached per day. In the cited context it is the hourly average presented for the diel cycle.

Referee comment 48:

P15 L9-15: this could indeed be expected and is not very exciting. Your contribution here should be discussing the differences in temperature fluctuations quantitatively.

Author comment 48:

Yes, this is a good point and was considered by insertion of some more detail on this difference.

Author change in the text 48:

P15 L17: "*The daily amplitude of the temperature was about twice as large in the canopy as compared to the understory (Tab. S6).*"

Referee comment 49:

P15 L17-18: mention this reinstallation in the methods too.

Author comment 49:

The reinstallation is mentioned in the methods part according to your advice.

Author changes in the text 49:

P8 L5: "However, during stormy episodes and/or physical friction, some WC and temperature sensors fell out of the moss samples and required a reinstallation. Accordingly, the WC sensor 6 (1.5 m) was repositioned in January 2015, WC sensor 1 (1.5 m) in November 2015, WC sensor 1, 6 to 24 and all temperature sensors in November

2016. The periods when the sensors have not been installed in the bryophyte samples were excluded from the data set.*"*

Referee comment 50:

5    P15 L21-22: mention and discuss this earlier on.

Author comment 50:

This is a good point, and thus, a description of the uncertainty of the WC is now already provided in the material section.

Author change in the text 50:

10    P8 L12: "Besides the specific position in the substrate, the WC also depended on the texture of the sample material, its ion concentration, and the temperature. Because of all these factors influencing the sensor readings, the provided values of the WC should be considered as the best possible estimates and not as exact values."

Referee comment 51:

15    P15 L33-34: Is the canopy so open that the wind direction is noticed at 8 m height? Why did you choose the west side, I would expect you to select a side with good moss cover. Interesting if this happened to be the west side if this side receives less moisture. Can you explain this?

Author comment 51:

Yes, the west side indeed was chosen as we found the best bryophyte cover there. Although the wind intensity is

20    weaker inside the canopy, we still experienced wind below the canopy and think that this could have an impact on water and habitat conditions at the different expositions, which then could have an effect on the differential growth of bryophytes.

Author change in the text 51:

P21 L10: "Long-term climate data have shown that the winds during the wet season predominantly originated from

25    north and north-eastern directions, while during the dry season south- and south-easterly winds prevailed (Pöhlker et al., 2018). At 8 m height, the investigated bryophytes were exposed to the west, and thus were only sometimes directly influenced by precipitation."

Referee comment 52:

30    P16 L11: why does a light rain facilitate drying??

Author comment 52:

Maybe we expressed it in the wrong way. But we intended to say, that after a light rain event the bryophyte samples dried quicker again, as they got not completely saturated with water. We rephrased this sentences for clarification.

Author changes in the text 52:

P21 L24: "Most rain events in the Central Amazon occur in the early afternoon (12:00 – 14:00 LT) and more than 75 % of them are weak events of less than 10 mm (Cuartas et al., 2007), which cause no complete water saturation of the bryophytes. Consequently, the organisms dry much quicker than after a strong rain event that fully saturates the community."

Referee comment 53:

P16 L17: this has at best been estimated, and please specify what you mean by 4%: 4% of water input for bryophytes (or other epiphytes?), or just comprising (thus not 'providing') 4% of total precipitation?

Author comment 53:

This means, it was estimated that approximately 4 % of the total precipitation will reach the ground as stemflow water. Thus, 4 % of the rain water is directly available for epiphytic organisms. By our calculations, this means that 68 to 75 mm per year are available as stemflow water. We rephrased this sentence accordingly.

Author change in the text 53:

P21 L34: "It has been *estimated* that in tropical forests stemflow water could provide up to 4 % of the annual rainfall amount (Lloyd and Marques F, 1988; Marin et al., 2000; van Stan and Gordon, 2018), corresponding to maximum values of 68 and 75 mm for the years 2015 and 2016 at the ATTO site."

Referee comment 54:

P16 L22: the water holding capacity is not what you have been measuring…Otherwise, this sentence is very true: the high water contents may be due to the high water-holding capacities of these species.

Author comment 54:

Indeed we did not measure the water holding capacity or only indirectly during the calibrations. Instead, we found high water contents over prolonged times, which we wanted to describe here. We rephrased this sentence for clarification.

Author change in the text 54:

P22 L5: "The high WC of the bryophyte samples in the canopy might be partly explained by the different water holding capacity of different bryophyte species (Lakatos et al., 2006; Romero et al., 2006; Williams and Flanagan, 1996)."

Referee comment 55:

P17 L13-14: be careful with your wording: understorey species are probably more efficient at low light (lower LCP), but it would be weird if they had a higher potential photosynthesis.

Author comment 55:

We meant to say that understory species reach higher net photosynthesis rates at low light conditions. We changed this sentence accordingly.

Author change in the text 55:

P23 L15: "…and it has been reported that understory mosses and lichens indeed show higher rates of net photosynthesis *at low light conditions* as compared to canopy species…"

Referee comment 56:

P17 L19-20: words missing

Author comment 56:

Yes indeed. The two sentences were rephrased and linked for clarification.

Author change in the text 56:

P23 L30: "As the measured net photosynthesis rates are the sum of simultaneously occurring photosynthesis and respiration processes, positive net photosynthesis may still be reached at higher temperatures, if the photosynthetic capacity is high enough, whereas during the night, high temperatures could cause a major loss of carbon due to high respiration rates (Lange et al., 2000)."

Referee comment 57:

P17 L22: It may be worth mentioning that Wagner et al 2013 concluded that, although respiration losses may be high, this in itself does not explain low bryophyte growth in tropical lowlands, because respiration rates are adapted or acclimatized to the prevailing temperature conditions: in mosses growing at higher elevations the respiration rates are higher at the same temperatures, but still epiphytic bryophyte biomass is much higher here.

Author comment 57:

Indeed, this type of information can be added to the text.

Author change in the text 57:

P24 L10: "Similarly, Wagner and coauthors (Wagner et al., 2013) stated that the temperature likely was not a limiting factor for NP and growth of the bryophytes investigated by them in a lowland and highland rainforest in Panama."*"*

Referee comment 58:

P18 L4: another example of a mismatch between cited literature and interpretation: you suggest that it is relevant that water contents in Zotz et al 1997 were measured during the same time of the year, but as this was a different region and a very different forest type, this temporal coincidence has no meaning whatsoever!

Author comment 58:

Yes, indeed the study of Zotz et al. 1997 was performed in a lower montane forest at an altitude of 1100 m, thus we decided to omit the comparison of the WCs.

Referee comment 59:

P18 L13-14 'whereas in the canopy, rain events, fog, and condensation seem to be equally important water sources for cryptogams.' What do you base this conclusion on??

Author comment 59:

Based on the new calibration and calculation of the bryophyte water content (WC) the relation of the WC of bryophytes at different heights changed and required a new interpretation of WC levels. The conclusion, as it was in the previous version, has been removed.

However, we had a closer look at the fog events and can show some humidification of the bryophyte communities upon fog events.

Author change in the text 59:

P16 L15: "Nightly fog might serve as an additional source of water, as the WC of the bryophyte communities increased upon fog events (Fig. S7)."

[Figure]

**Figure S7:** Two exemplary fog events and the reaction of the moisture sensors of the bryophytes (a and b). Each panel presents (A) a fog event defined by a visibility < 2000 m, (B) relative air humidity (RH), (C) rain, and (D) the water content (WC) of the bryophytes. In each panel, the fog event of interest is marked by a red box. For the WC sensors the number, height of installation, and division (M = Moss, L = Liverwort) are given.

Referee comment 60:

P18 L16: what does 'which' refer to? The reference seems strange here. (Figure 2: the wet season data are shown twice, the dry season data are missing! A legend is also missing.) → Already corrected by authors

Author comment60:

"Which" refers to the observation of Pardow and Lakatos (2013), where they describe that understory species are more sensitive to drought than canopy species.

However, the sentences has been removed, due to reorganization of the section.

Referee comment 61:

Figure S2: in what way are these integrals? Do you mean interpolations?

Author comment 61:

The data with 30-minute time intervals are the average values of six 5-minute grid data. It indeed is better to say "*average*" instead of "integral" (Figure S4).

Referee comment 62:

Supplement: P4 L7: looks like 2 replicates for *Octoblepharum*

Author comment 62:

Supplement previous Figure S3: Yes, indeed there were two replicates for *Octoblepharum*. However, this calibration is not valid anymore and was deleted in the context of the revision.

**References provided by the author:**

[revised manuscript text omitted]

5  **References provided by the referee:**

Gradstein, S.R., Churchill, S.P. & Salazar A., N. 2001. Guide to the Bryophytes of Tropical America. Memoirs New York Bot. Garden 86: 1-577.

Gradstein, S.R. 2006. The lowland cloud forest of French Guiana – a liverwort hotspot. Cryptogamie, Bryol. 27: 141-152.

10  Gradstein, S.R. & Ilkiu-Borges, A.-L. 2009. Guide to the Plants of Central French Guiana. Part IV. Liverworts and Hornworts. Memoirs of the New York Botanical Garden 76, 4: 1- 140, 83 plates.

Gehrig-Downie C., Obregón A., Bendix J., Gradstein S.R. 2013. Diversity and vertical distribution of epiphytic liverworts in lowland rain forest and lowland cloud forest of French Guiana. Journal of Bryology 35: 243-254.

**Maaike Bader as referee submitted the comments RC1 and RC3**

**Maaike Bader RC3**

**Comments on the text:**

Black text shows the original referee comment, and blue text shows the response of the authors and the explicit changes in the revised text. The figure and table numbers refer to the revised manuscript.

Referee comment 1:

**An additional consideration: an alternative way to use the electrical resistance measurements**

Dear authors,

After some more thought and discussion with some colleagues, with whom we will be installing a similar system to measure moss wetness, I would like to suggest using more caution in the translation of the electrical resistance to moss water content and to propose an alternative way of interpreting the measurements. This is giving away the method we intend to use ourselves, which I think may be a good alternative for your study also. You are welcome to cite me for the idea if you think it appropriate.

It is clear that there is a very wide range of moss water-content (WC) values that may be indicated by any electrical resistance value measured. The values are more constrained for the cushion species (Leucobryum), which makes sense seeing that such a life form is denser and more homogenous than the other species, which are prostrate or consist of loosely scattered turf, if I am not mistaken. With such inhomogenous substrates, with different amounts of air and tissue between the probes for each sample, it is no wonder that the measured conductance is widely scattered within species. I think you should reconsider whether you should really try to deduct an absolute value of WC from these measurements. It looks like this is not really possible for most species.

It seems that the points within each calibration curves are nicely ordered, however. Therefore an alternative approach would be to only look at the changes in electrical conductivity, which should reliably indicate changes in water content. With this, you can deduct for any time period whether the samples were drying out or being wetted. When stable at low conductivity, this indicates that the samples are dry (in equilibrium with air humidity), when stable at high conductivity they must be completely wet during rain or fog events. If you have good data about the maximum water content of the species, you might even be able to interpolate between the stable low and the stable high, considering that drying tends to follow relatively smooth extinction curves, as you will see when plotting your calibration curves against time.

I hope this suggestion is of use.

Author comment 1:

Thank you very much for this good and helpful comment. After an intense re-analysis of our field and calibration data we decided to indeed use a calibration approach very similar to the suggested one. We explain this in our response to RC1 in comment 13: We performed a new approach for the calibration of the water content, based on

5      the maximum and minimum values of electrical conductivity reached in the field and the maximum and minimum of the water content reached during the laboratory measurements. With the new approach we assume that the maximum electrical conductivity achieved in the field corresponds to the maximum water content, which could be reached by the organism (and which had been determined during the laboratory experiments). The measurements of the electrical conductivity in the laboratory are not considered anymore. For that, the entire calibration process

10     and the subsequent results were re-calculated again.

**Anonymous Referee #2 submitted the comments RC2**

RECEIVED AND PUBLISHED: 15 FEBRUARY 2019

**Comments on the text:**

5    Black text shows the original referee comment, and blue text shows the response of the authors and the explicit changes in the revised text. The figure and table numbers refer to the revised

**General referee comment:**

The authors provide a description of bryophyte occurrence and microclimate in a tropical forest canopy. These
10    data are scarce and therefore crucial for a variety of applications that the authors list at various times in the manuscript.

General author response: We would like to thank reviewer 2 for his/her appreciation of the microclimate data and the productive comments, which helped us to substantially improve our manuscript.

15    Referee comment 1:

First, amongst these are poor organization and a general lack of coherence. Facts about bryophytes (such as they are poikilohydric) are repeated often. No clear hypotheses or research questions are outlined. The introduction tells us that bryophytes are 'cool' and important to study but doesn't do a good job of setting up the study itself. Until the end of the methods section, I didn't realize that gas exchange measurements were not performed (something
20    that is mentioned in abstract- If gas exchange in epiphytes is essential, why did the authors not make these measurements?).

Author comment 1:

In the introduction, our aim was to introduce the ecosystem and study site, the invested organisms and communities, and also the measurement approach. As this aim seems to be only partly fulfilled, we thoroughly checked and
25    restructured the introduction. Specific changes were made to bring more clarity into the abstract and the methods section to facilitate an understanding of the study. $CO_2$ gas exchange measurements indeed would have been interesting, but go beyond the scope of the current study. They make up a major study by themselves, which should be conducted in the near future. For now, we used reliable literature data to investigate the activity patterns of bryophytes when respiration and photosynthesis potentially take place.
30    Author changes in the text 1:

P2 L7: "In this study, we present data on the microclimatic conditions, including water content, temperature, and light intensities experienced by epiphytic bryophytes along a vertical gradient and combine these with "abovecanopy climate" data collected at the *Amazon Tall Tower Observatory* (*ATTO*) in the Amazonian rain forest between October 2014 and December 2016."

P2 L26: "These data may be used as a starting point to investigate the role of bryophytes in various biosphere-atmosphere exchange processes, such as measurements of $CO_2$ gas exchange, and could be a tool to understand the functioning of the epiphytic community in greater detail."

Referee comment 2:

While I am quite satisfied by the measurement protocols and methodology (and that the epiphyte wetness-drying data are novel and important) the study ends up being merely a data reporting exercise with conclusions that often seem unsubstantiated by the data that are presented. Other times conclusions are trivial. For instance, Pg 18, lines 18-23 it is suggested that it is dark in the understory and therefore photosynthesis is light limited. I do not think that today one needs to go to the Amazon to make this conclusion, as this has been known for decades (for e.g. read classic reviews by Chazdon and Fetcher, 1984; Mooney et al., 1984). I seem facetious here, but the authors could use the same data to build upon these earlier findings, and find some nuance and/or insights. What is the knowledge gap that you are trying to fill with your measurements?

Author comment 2:

The main results and conclusions were revised by us to present the data in a more logical and substantiated way. We also utilized the literature offered by you, as it helped to arrange our results in a better framework. In our opinion, one major advantage of our study is, that we performed long-term measurements (running continuously over more than two years) at several heights along a trunk, thus obtaining a vertical profile of the conditions within the vegetation. We now highlight this aspect, apart from other aspects, and with this new structure, we think we can emphasize and improve its significance.

Author changes in the text 2:

P4 L8: "Studies in temperate zones address the importance of cryptogamic communities for the ecosystem (Gimeno et al., 2017; Rastogi et al., 2018), but for the tropical area, few reports can be found in the literature. There is a lack of information regarding the functioning of such communities in an environment with an almost constant high relative humidity and temperature range. Thus, with the long-term continuous measurements presented here, we aim to provide data on seasonality patterns and the vertical profile of the microclimate within the canopy. In the current study, we present the microclimatic conditions, comprising the temperature, light, and WC of epiphytic bryophytes communities along a vertical gradient and an estimation of their activity patterns in response to annual and seasonal variations of climatic conditions."

Referee comment 3:

I want to be clear that I do not think that this work is unpublishable, rather a considerable amount of work needs to be done, especially in the writing, to ensure that it is. The advantage of the study is that the authors have collected a vast amount of important data, and there are several questions that can be formulated and answered. For instance, Fig S.5 is very interesting, and one could speculate about the significance of Tair -TCryptogram relationship in different parts of the canopy, and its significance to physiology. Another question could be the importance of light flecks, since you have carefully measured PPFD within the canopy. Fog is also measured but these data seem largely ignored (I wonder if you had leaf wetness sensors, those data could bolster the study tremendously). I would recommend the corresponding author to read some of the classic literature on epiphyte distribution and abundance (e.g. Benzing, 1984). With some more data exploration and thought I think this could be a very significant contribution. In its current form however, the manuscript reads like an early draft of a thesis or a dissertation chapter, and I do not see it fit for publication in Biogeosciences, or a journal of similar repute.

Author comment 3:

We appreciate this criticism and now put more effort into the analysis and interpretation of the long-term data collected by us.

The relevance of fog was investigated in more detail and its relevance for the WC of the bryophytes is illustrated in the text and in Fig. S7, as shown below. We also analyzed light flecks in the updated version. Unfortunately, no leaf wetness sensors have been installed, so this kind of data cannot be used.

Author change in the text 3:

P16 L15: "Nightly fog might serve as an additional source of water, as the WC of the bryophyte communities increased upon fog events (Fig. S7)."

P24 L21:" It is difficult to distinguish between the effect of fog and high RH, as fog occurs when high RH values persist already. However, some events indicate that the bryophyte WC could also increase upon fog (Fig. S7), which has also been shown in some other studies (León-Vargas et al., 2006). Also condensation needs to be considered as a water source for cryptogams, as demonstrated for epiphytic lichens (Lakatos et al., 2012)."

[Figure]

**Figure S7:** Two exemplary fog events and the reaction of the moisture sensors of the bryophytes (a and b). Each panel presents (A) a fog event defined by a visibility < 2000 m, (B) relative air humidity (RH), (C) rain, and (D) the water content (WC) of the bryophytes. In each panel, the fog event of interest is marked by a red box. For the WC sensors the number, height of installation, and division (M = Moss, L = Liverwort) are given.

**Referee comment 4:**

Finally, authors should provide data access via a link with a doi to a data repository. I wonder if this is required of papers that are submitted to Biogeosciences.

Author comment 4:

10    The database on the water content, temperature, and light conditions of epiphytes is uploaded to the ATTO data portal (www.attoproject.org/).The data thus are maintained, obtain a doi and can be retrieved from that site.

**Specific/Minor comments below:**

Referee comment 5:

The abstract is a bit long with too many technical or field specific terms that should be introduced (in the introduc-

15    tion), since it makes it difficult to comprehend for the general reader. An example is "While the monthly average mesoclimatic ambient light intensities above the canopy revealed only minor variations: …" This is a well written but complicated sentence for the average reader. Please simplify.

Author comment 5:

The entire abstract was revised for better readability.

20    Author change in the text 5:

**"Abstract**. In the Amazonian rain forest, major parts of trees and shrubs are covered by epiphytic cryptogams of great taxonomic variety, but their relevance in biosphere-atmosphere exchange, climate processes, and nutrient cycling are largely unknown. As cryptogams are poikilohydric organisms, they are physiologically active only under moist conditions. Thus, information on their water content, as well as temperature and light conditions ex-

25    perienced by them are essential to analyze their impact on local, regional, and even global biogeochemical processes. In this study, we present data on the microclimatic conditions, including water content, temperature, and light conditions experienced by epiphytic bryophytes along a vertical gradient and combine these with "above-canopy climate" data collected at the *Amazon Tall Tower Observatory* (*ATTO*) in the Amazonian rain forest between October 2014 and December 2016. While the monthly average of above-canopy light intensities revealed

30    only minor fluctuation over the course of the year, the light intensities experienced by the bryophytes varied depending on the location within the canopy, probably caused by individual shading by vegetation. In the understory (1.5 m), monthly average light intensities were similar throughout the year and individual values were extremely low, remaining below 3 $\mu$mol m$^{-2}$ s$^{-1}$ photosynthetic photon flux density during more than 98 % of the time. Temperatures showed only minor variations throughout the year with higher values and larger height-dependent differences during the dry season. The indirectly assessed water contents of bryophytes varied depending on precipitation, air humidity, and bryophyte type. Whereas bryophytes at higher levels were affected by frequent wetting and drying events, those close to the forest floor remained wet over longer time spans during the wet seasons. In general, bryophytes growing close to the forest floor were limited by light availability, while those growing in the canopy had to withstand larger variations in microclimatic conditions, especially during the dry season. These data may be used as a starting point to investigate the role of bryophytes in various biosphere-atmosphere exchange processes, such as measurements of $CO_2$ gas exchange, and could be a tool to understand the functioning of the epiphytic community in greater detail."

Referee comment 6:

Line 12: 1.5 m relative to what (i.e, please include canopy height). For the abstract something general, like 'near-surface' or 'in the understory' is more appropriate.

Author comment 6:

Yes, it is important to set the height of 1.5 m into relation, especially in the abstract. We exchanged "At 1.5 m height" by "in the understory" for more clarity.

Author change in the text 6:

P2 L15: "*In the understory (1.5 m), monthly average….*"

Referee comment 7:

Line 13: instead of saying "low, exceeding less than 8% …" you could say low, remaining below 5 µmol … more than 92% of the time.

Author comment 7:

Yes, it is easier to follow the way you proposed to revise this sentence. We changed it according to your advice.

Author change in the text 7:

P2 L16: "… individual values were extremely low, *remaining below* 3 µmol m$^{-2}$ s$^{-1}$ photosynthetic photon flux density during *more than 98 %* of the time."

Referee comment 8:

Lines 18-19: Dark respiration should occur independent of light (and unless temperatures are very low, which seems unlikely at your site). The references to photosynthesis and respiration are repeatedly incorrect. Photosynthesis and respiration are co-occurring biological processes (in the light), and therefore one may dominate over the other.

Author comment 8:

We agree with the revision, this sentence was expressed in a way, which could easily be misunderstood. As the entire abstract was restructured, this sentence was removed.

**Introduction**

Referee comment 9:

The first paragraph is a well written introduction to tropical forests, but has little do with the study. Either you should reframe it in the context of epiphytes or omit. Overall, the introduction does not set up the study satisfactorily.

Author comment 9:

The introduction was restructured and sentences were rewritten to better address the topic.

Author changes in the text 9: P3 L2 (Revised introduction):

"Cryptogamic communities comprise photosynthesizing organisms, i.e. cyanobacteria, algae, lichens, and bryophytes, which grow together with heterotrophic fungi, other bacteria, and archaea. They can colonize different substrates, such as soil, rock, and plant surfaces in almost all habitats throughout the world (Büdel, 2002; Elbert et al., 2012; Freiberg, 1999). In the tropics, epiphytic bryophyte communities widely cover the stems and branches of trees (Campos et al., 2015). Within that habitat, they may play a prominent role in environmental nutrient cycling (Coxson et al., 1992) and also influence the microclimate within the forest (Porada et al., 2019), thus contributing to the overall fitness of the host plants and the surrounding vegetation (Zartman, 2003). However, they are equally affected by deforestation and increasing forest fragmentation (Zartman, 2003; Zotz et al., 1997).

Physiologically, cryptogamic organisms are characterized by their poikilohydric nature, as they do not actively regulate their water status but passively follow the water conditions of their surrounding environment (Walter and Stadelmann, 1968). In a dry state, many of them can outlast extreme weather conditions, being reactivated by water (Oliver et al., 2005; Proctor, 2000; Proctor et al., 2007; Seel et al., 1992), and for several species even fog and dew can serve as a source of water (Lancaster et al., 1984; Lange et al., 2006; Lange and Kilian, 1985; Reiter et al., 2008). In contrast, high water contents (WC) may cause suprasaturation, when gas diffusion is restrained, causing reduced $CO_2$ gas exchange rates (Cowan et al., 1992; Lange and Tenhunen, 1981; Snelgar et al., 1981) and even ethanolic fermentation, as shown for lichens (Wilske et al., 2001). Accordingly, their physiological activity is primarily regulated by the presence of water and only secondarily by light and temperature (Lange et al., 1996, 1998, 2000; Rodriguez-Iturbe et al., 1999).

In the Amazonian rain forest, cryptogamic communities mainly occur epiphytically on the stems, branches, and even leaves of trees, and in open forest fractions they may also occur on the soil (Richards, 1954). By 2013, 800 species of mosses and liverworts, 250 lichen species, and 1,800 fungal species have been reported for the Amazon region (Campos et al., 2015; Gradstein et al., 2001; Komposch and Hafellner, 2000; Normann et al., 2010; Piepenbring, 2007). Tropical rain forests are characterized by humid conditions, high temperatures with minor

annual fluctuations, and an immense species diversity of flora and fauna. Currently, between 16 000 and 25 000 tree species have been estimated for the Amazonian rain forest (Hubbell et al., 2008; ter Steege et al., 2013). It has been described to play important roles in the water cycle, as well as for carbon, nitrogen, and phosphorus fluxes on regional and global scales (Andreae et al., 2015). However, it is also hard to predict, to which extent the ongoing and envisioned changes will still ensure its ecological services as "green lung" and carbon sink of planet Earth (Soepadmo, 1993).

Studies in temperate zones address the importance of cryptogamic communities for the ecosystem (Gimeno et al., 2017; Rastogi et al., 2018), but for the tropical area, few reports can be found in literature. There is a lack of information regarding the functioning of such communities in an environment with an almost constant high relative humidity and temperature range. Thus, with the long-term continuous measurements presented here, we aim to provide data on seasonality patterns and the vertical profile of the microclimate within the canopy. In the current study, we present the microclimatic conditions, comprising the temperature, light, and WC of epiphytic bryophytes communities along a vertical gradient and an estimation of their activity patterns in response to annual and seasonal variations of climatic conditions. "

Referee comment 10:

Pg 3 Line 13: 'By' not 'In' 2013.

Author comment 10:

Changed accordingly (P3 L21)

Referee comment 11:

Pg 4 Line 5: Update references to carbonyl sulfide: (Gimeno et al., 2017; Rastogi et al., 2018).

Author comment 11:

Many thanks for the provision of these additional references, but due to a revision of the introduction, the information on OCS has been omitted.

**Methods**

Referee comment 12:

Sec. 2.1. A greater description of the site is required. I would recommended starting with site characteristic and then describe the tower and measurements, not the other way around.

Author comment 12:

The advice to start with a description of the forest area and to subsequently characterize the study site itself is a good idea, and we reordered section 2.1 accordingly. However, we refrain from giving a more detailed description

of the site, as this is nicely presented by Andreae et al. (2015), which belongs to the same special issue (on ATTO) as the current manuscript.

Author change in the text 12:

P6 L3:"The study site is located within a *terra firme* (plateau) forest area in the Amazonian rain forest, approx. 150 km northeast of Manaus, Brazil. The average annual rainfall is 2,540 mm $a^{-1}$ (de Ribeiro, 1984), reaching its monthly maximum of ~ 335 mm in the wet (February to May) and its minimum of ~ 47 mm in the dry season (August to November) (Pöhlker et al., 2018). These main seasons are linked by transitional periods covering June and July after the wet and December and January after the dry season (Andreae et al., 2015; Martin et al., 2010; Pöhlker et al., 2016). The *terra firme* forest has an average growth height of ~ 21 meters, a tree density of ~ 598 trees $ha^{-1}$, and harbors around 4,590 tree species on an area of ~ 3,784,000 $km^2$, thus comprising a very high species richness compared to other forest types (McWilliam et al., 1993; ter Steege et al., 2013). Measurements were conducted at the research site *ATTO* (*Amazon Tall Tower Observatory*; S 02° 08.602', W 59° 00.033', 130 m a. s. l.), which has been fully described by Andreae and co-authors (2015). It comprises one walk-up tower and one mast of 80 m each, being operational since 2012, and a 325 m tower, which has been erected in 2015. The ATTO research platform has been established to investigate the functioning of tropical forests within the Earth system. It is operated to conduct basic research on greenhouse gas as well as reactive gas exchange between forests and the atmosphere and contributes to our understanding of climate interactions driven by carbon exchange, atmospheric chemistry, aerosol production, and cloud condensation. "

Referee comment 13:

Pg 4 line 13: Remove "The".

Author comment 13:

The word was removed.

Author change in the text 13:

P5 L11: "*Measurements* were conducted at…"

Referee comment 14:

Pg 7 line 2: "were measured" not "are being measured".

Author comment 14:

Done.

Referee comment 15:

P5 Line 5: this seems important for your study and you should describe why sensors were placed where they were placed, in addition to citing the Mota de Oliveira (2013) study.

Author comment 15:

We placed the sensors along a vertical gradient ranging from the understory to the canopy, in order to cover the range of microclimatic conditions experienced by the epiphytic bryophytes as thoroughly as possible. We included a phrase to express this intention.

Author change in the text 15:

P7 L4: "The sensors were placed along a vertical gradient at ~ 1.5, 8, 18, and 23 m above the ground, corresponding to the zones 1 to 4 used by Mota de Oliveira and ter Steege (2015), to investigate the variation within the story structure of the forest.."

Referee comment 16:

Pg 6: line 30: Why 60 C? Is this a temperature that these communities experience?

Author comment 16:

Drying the organisms at 60°C until weight consistency is a common procedure to determine the dry weight of these organisms. This is not related to the environmental field conditions, but is a standard method used to obtain a standardized dry weight.

Author change in the text 16:

P9 L4: "The dry weight (*DW*) was determined after drying at 60 °C *until weight consistency was reached (Caesar et al., 2018)*."

Referee comment 17:

Sec 2.5. Again, some information (a figure ideally) describing the vertical profile of the forest is necessary. That helps put the various sensor heights in perspective.

Author comment 17:

This is a good idea and we prepared a graphical scheme, which is presented in the supplement as Fig. S2 (see below).

Author changes in the text 17:

[Figure]

**Figure S2:** Schematic overview of the sensors installed at different height levels below, within, and above the canopy. The parameters water content (WC) and temperature (Temp) were measured within the bryophyte samples, the light sensors (PAR) were installed directly on top of the thalli. The average tree height of 21 m was determined for the Plateau forest in general.

Referee comment 18:

Lines 15-21: Why were rainfall values gap-filled? Also, isn't the sensor at 1.5 m the least well placed to record rain event. For instance, a small rain event might not even be recorded at 1.5m, as interception must be high in a high LAI forest. Alternatively, there could be time lags between when rainfall occurs at the canopy top and when it is measured by the 1.5 m ht rain gauge.

Author comment 18:

Unfortunately, over the course of the long-term rainfall measurements there are some measurement gaps, which needed to be filled in order to correctly analyze the data.

The rain gauge has been installed at 81 m height on the tower, while the canopy height in this forest is approximately 21 m. Consequently, the rain gauge is placed well above the canopy. As correctly assumed by you, the bryophytes in the understory (1.5 m) might not get watered by light rain events and there is also a delay of several minutes until they get wet compared to the canopy organisms.

Referee comment 19:

Pg 8 L32- Pg 9, L1: Rephrase sentence. Light intensity regulates the balance i.e. the net exchange between photosynthesis and respiration.

Author comment 19:

The sentence was rephrased to clary that light triggers, which process might dominate the carbon balance.

Author change in the text 19:

P12 L32: "While the availability of water determines the overall time of physiological activity, the light intensity regulates *whether* net photosynthesis (NP) or dark respiration (DR) *will dominate the overall metabolical balance*. "

Referee comment 20:

Line 14: Respiration takes place at all light levels. IT IS NOT A LIGHT DEPENDANT PROCESS (there can be significant inhibition of respiration at high light levels). Please check this basic tenet of biology.

Author comment 20:

We are aware of the fact that this was not expressed in a correct way. We meant to state that under these condition NET respiration is observed. We know that both photosynthesis and respiration often occur simultaneously and that the net balance is what is being measured. We changed that accordingly.

Author change in the text 20:

P13 L15: "At light intensities below the compensation point and WCs above WCP, respiration rates are higher than NP rates, causing overall net respiration to occur."

Referee comment 21:

Line 21: Based on *literature*, not literature *data*. Also, please cite the relevant literature.

Author comment 21:

We believe that literature data (i.e. data extracted from the literature) is the correct expression here, however, due to revision of the text this sentences was changed. We added the relevant literature citations.

Author change in the text 21:

P13 L22: "Unfortunately, literature data on the compensation points are rare, facilitating only a first approximate assessment of the physiological processes ( Lösch et al., 1994; Wagner et al., 2013)."

**Results: Overall, I do not have issues with the content per se, but as I have stated before this section needs to be majorly revised/expanded. Some minor comments below.**

Referee comment 22:

Pg 10 Line 8: Micromet did not depend on years but varied amongst years.

Author comment 22:

Indeed we were not precise enough with our formulation. We rephrased the sentence accordingly.

Author changes in the text 22:

P14 L14: "Over the course of the two years of measurements, the monthly mean values of the WC, temperature, and light conditions experienced by the epiphytic bryophyte communities, as well as the above-canopy meteorological conditions, varied between seasons and years."

Referee comment 23:

Pg 10 Line 18: please define mesoclimate the first time you use this term.

Author comment 23:

Mesoclimate is a standing term describing the climate of a given habitat, covering a side length of some tens or hundreds of meters. We utilized this expression to distinguish the above-canopy climate measurements from the microclimate measurements conducted next to/within the bryophyte thalli. As this expression lead to some confusion, we replaced it by the term "above-canopy climate" throughout the text.

Referee comment 24:

Pg 11 Line 30-31: What does this mean? Please elaborate.

Author comment 24:

(Now P16 L11) At 1.5 m height the water content sensors always showed an increase after rain events, whereas this was not always observed for the sensors at the other height levels. This seems not fully logical at first sight, but we can imagine that the thalli growing on one side of the tree are sometimes not reached by a rain event if it mainly comes from the other side. We could imagine that the rain intercepts with some inclination at the higher levels, whereas close to the ground this inclination is lost. This, however, is only one potential explanation which has not been verified by us.

Referee comment 25:

Pg 12 Sec 3.1.3 header: remove parenthesis

Author comment 25:

Yes, we can remove the parentheses in the header.

Author change in the text 25:

P16 L17: "3.1.3 Diel cycles in different seasons and years along a vertical gradient"

Referee comment 26:

Pg12 Line 13: which 'organisms'. Please specify.

Author comment 26:

Here, the epiphytes, which have been mentioned earlier in the sentence, are meant. In the context of revision this

10    sentence was omitted.

Referee comment 27:

Sec. 3.2. The scope of your inference is limited since you do not have replicates on different trees. I do not see this

as limitation, but somewhere (probably in the discussion) you need to talk about heterogeneity in the microclimatic

15    environment.

Author comment 27:

Yes, this is right, and we now describe this restriction of the measurements in an additional passage in the Material

and Methods section.

Author change in the text 27:

20    P 7 L12: "The restriction of the measurements to one individual tree needs to be considered, as a complete independence of the replicate sensors could not be assured. However, due to the large effort of such an installation

within the rain forest, it was not possible to equip more trees with additional instruments. Thus, the data obtained

from the measurements on this individual tree should be considered as exemplary."

25    Referee comment 28:

P13 Lines 13-14. This has been mentioned previously (in Sec. 3.1.).

Author comment 28:

Whereas in the previous sections, we discuss the annual fluctuations of monthly mean values and the seasonal

changes between the wet and dry season, the diel cycles are in the focus of the current section. Thus, the tempera-

30    ture needs to be discussed under these different aspects and this might give the impression of some repetition.

However, we could not find the position where this statement has been made before. Perhaps you can give us a

clearer hint on that.

Referee comment 29:

P13 Lines 25 to Pg 14 line 3: This is a well written paragraph but belongs in the discussion.

Author comment 29:

This section was rephrased, according to recalculations and was moved to the discussion section.

5   **Tables**

Referee comment 30:

Table 1: Why are these annual means presented? Why are light levels higher at 23 m than at 18m (again discuss heterogeneity)? I still don't have a good grasp of the canopy structure. I do not understand the sensor placement with respect to the canopy structure. Is the 23 m sensor above the canopy top (~ 21m)? Probably not, since light

10   levels seem too low. Also, there seems to be some confusion between relative humidity and water content.

Author comment 30:

The annual means are presented to show the differences between the years, i.e. to demonstrate that the climatic conditions change from one year to the next.

The light conditions at the different height levels of the canopy are discussed in the discussion section (see below).

15   The canopy structure and sensor positions have been already described above (comment 17). The tree is approximately 26 m high, which is now also mentioned in the methods section (P 7 L6), thus the sensors from 1.5 to 23 m height are located on top of the bryophytes growing on the stem of the tree.

We clarified this by deleting the RH-data, as they do not really fit here.

Author change in the text 30:

20   Canopy structure and sensor position: P7 L16: "Generally, the WC sensors were placed in four different bryophyte communities being heterogeneously distributed along the four height levels. At 1.5 m height, the WC sensors were installed in communities dominated by *Sematophyllum subsimplex* (5 sensors) and *Leucobryum martianum* (1 sensor), at 8 m in *Octoblepharum cocuiense* (3 sensors) and *Symbiezidium barbiflorum* (3 sensors), and at 18 and 23 m in *Symbiezidium barbiflorum* (6 sensors at each height level; Fig. S2, Fig. S3). The temperature sensors were

25   installed in the same communities at each height, and the light sensors were installed adjacent to them on ~ 5 cm long sticks (Fig. S1). As the morphology of the different species affects their overall WC, different maximum WC and patterns of the drying process were observed (Tab. S1). The sensors were installed with the following orientations: at 1.5 and 8 m vertically along the trunk, at 18 m at the upper side of a slightly sloped branch, and at 23 m at the upper side of a vertical branch. Thus, also the orientation at the stem may influence the WC of the bryophyte

30   communities, not only the species and the canopy structure. *"*

Tree height: P74 L6: "At each height level, six WC, two temperature, and two light sensors (except for 1.5 m with only one light sensor) were installed in/on top of different bryophyte communities located on an approximately 26 m high tree (Fig. S2, Table S1).

P36 L2: Relative humidity and water content: Table 1:

"Annual mean values and standard deviation (± SD) of mean daytime photosynthetically active radiation (PAR$_{avg}$), daily maxima of photosynthetically active radiation (PAR$_{max}$), temperature, and water contents (WC) of bryophytes at the four height levels and above the canopy (a)."

5    Referee comment 31:

Table 2. There are no significant differences between seasons for some variables (for e.g., temperature), even though this is alluded to in the results (Pg. 11, line 24).

Author comment 31:

The statement on P11 L24 refers to a significant difference of the temperature between 23 m and above canopy

10    measurements assessed during the dry season, which is listed in Table S5 (now Table 2). In the context of the revision, the statistical tests were omitted, thus no significant differences are presented anymore, but were replaced by "trends" and "tendencies".

Author change in the text 31:

P 16 L3:" At 23 m height, temperatures within the bryophyte communities were frequently higher than the above-

15    canopy values, and during the dry season even the seasonal average temperature was 0.5°C higher, probably due to surface heating (Tab. S2)."

**Figures: Generally, the figures need to be clearer, and larger, since you have several subplots.**

Referee comment 32:

20    Fig 2. PAR and Temperature at different heights are very hard to see. Either summarize differently, or show a mean in this figure and direct to a figure in the supplemental with data from all heights.

Author comment 32:

We tried to adapt the figures for more clarity. See the figures at the end of the revised manuscript below.

25    Referee comment 33:

Figure 3. This also has too many sub-panels crammed in one figure. In the caption, why do you say ecophysiolog- ical, micrometeorological and ambient parameters (the same is actually true of Fig. 1 as well). Which ones are which? Why are they called parameters? What are you trying to parametrize? I make a point about this, because this is one of several instances where words are not chosen carefully. Was humidity not measured at all heights?

30    Author comment 33:

We are aware of the fact that figure 3 is quite complex and needs some attention to be fully understood. On the other hand, we think that it gives a lot of information and allows direct comparisons between the different param- eters and thus we would like to keep it in the current way.

With the term "parameters" we refer to environmental parameters, like temperature, precipitation, light intensity, etc., also called climate parameters or climate factors. We changed the captions to clarify the parameters, which are presented. The relative air humidity was measured at 26 m, just above the canopy, while the water content was measured at all four height levels within the bryophyte cushions.

5   Author change in the text 33:

Figure 3 (P46 L3): "Mean diurnal cycles of water content (WC), temperature, and light conditions of bryophytes, and above-canopy meteorological parameters …"

Referee comment 34:

10  Figure 4. The histograms are informative but the information provided in the various shaded regions is extremely hard to follow. In the end, I do not understand what the authors are trying to convey. Why is the y-axis broken in the histograms in the left most panel?

Author comment 34:

Figure 4 was adapted to make it clearer and easier comprehensible.

15  The y-axis in the left-hand panel (PAR) is broken, as the lowest light intensity was reached at a frequency between 50 and 70%. All the higher light intensities occurred at frequencies of only a few or even below 1%. To show both the high and low frequencies at good resolution, we decided to use a broken y-axis.

**References provided by the author:**

[revised manuscript text omitted]

30    in a lower montane rain forest in Panama, Bot. Acta, 110, 9–17, doi:10.1111/j.1438-8677.1997.tb00605.x, 1997.
* * *
**References provided by the referee:**

Gradstein, S.R., Churchill, S.P. & Salazar A., N. 2001. Guide to the Bryophytes of Tropical America. Memoirs New York Bot. Garden 86: 1-577.

35    Gradstein, S.R. 2006. The lowland cloud forest of French Guiana – a liverwort hotspot. Cryptogamie, Bryol. 27: 141-152.

Gradstein, S.R. & Ilkiu-Borges, A.-L. 2009. Guide to the Plants of Central French Guiana. Part IV. Liverworts and Hornworts. Memoirs of the New York Botanical Garden 76, 4: 1- 140, 83 plates.

Gehrig-Downie C., Obregón A., Bendix J., Gradstein S.R. 2013. Diversity and vertical distribution of epiphytic

40    liverworts in lowland rain forest and lowland cloud forest of French Guiana. Journal of Bryology 35: 243-254.

**Anonymous Referee #3 submitted the comments RC4**

RECEIVED AND PUBLISHED: 25 FEBRUARY 2019

**Comments on the text:**
Black text shows the original referee comment, and blue text shows the response of the authors and the explicit changes in the revised text. The figure and table numbers refer to the revised manuscript.

**General referee comments:**

Dear Editor, dear authors

I have read with interest the manuscript entitled "Microclimatic and ecophysiological conditions experienced by epiphytic bryophytes in an Amazonian rain forest" by Löbs et al. submitted to Biogeosciences. Please find my comments related to it below:

I appreciate a strong point in this manuscript that is to contribute to raise the data availability regarding cryptogamic covers functional performance in tropical regions, and going further, the lack of data available in Central and South America. It seems that almost all the literature regarding this issue has been focused in Polar Regions some years ago and in drylands at the present. I also appreciate the novelty and the effort made to provide microclimatic data sets at those heights at the tree trunks. If we want to understand properly the relevance of these organisms in global cycles and their response under environmental changes a huge and very different biome as the tropics cannot be ignored. I think that authors do a complete revision of the literature available and try to contribute from there with their data. Mosses dominate cryptogamic covers in tropical regions in biodiversity, so the target organisms in the study seems to be quite correct.

General author response:

We would like to thank the reviewer for appreciating our work and for the efforts spent on our manuscript. His/her comments helped us to substantially improve it.

Referee comment 1:

But, at the same time, my opinion is that this lack of data availability in the region is an intrinsic weakness of the manuscript. My point here is that the manuscript is based in a double assumption rather than in strictly measured data sets. The first assumption would be the water content of the bryophytes through conductivity sensors.

I appreciate the effort made by the authors calibrating this methodology in the lab and this experimental testing gives higher credibility to the measurements. But then we see the big second assumption that is to extrapolate data taken from the literature to understand the functional performance of the bryophytes in the altitudinal gradient. I think that it is likely that possible inaccuracies could arise in this sense. Data available in the literature is little, so, it must be difficult to find similar experimental designs that could help providing reliable extrapolations. I am not talking about finding same species with data available in the literature, but it would be interesting, in order to trust the ecophysiological data provided, to have data from a similar habitat following at least the light adaptation patterns of the species included in this work.

As I suppose that these data sets are very difficult to get, but I think that this manuscript is interesting and useful to the scientific community, I would make a proposal to the authors:

What about to include in your manuscript a few gas exchange checkpoints in the lab including relevant species inside the gradient. For example, one representative species in the understory and another one at the closer point of the canopy could serve as cardinal points to calibrate authors' predictions about net photosynthesis availability, time and amount of respiration and possible C losses, light cardinal points, adaptation strategies. This would improve the discussion substantially from my point of view.

I am not asking for a complete gas exchange profile of the species included in the study because I know how time consuming this technique is, just a few replicated checkpoints in the lab to see how close predictions are from reality. If they were far from each other, the real gas exchange parameters measured could work as a more reliable source of predictions than a very likely imprecise literature for the aim targeted. I would welcome further assumptions at this point, but based in some real measured values (I said in the lab because conditions are easier to control, but some field gas exchange data sound good for me also). I think that this could improve the manuscript and put it as a reference text in tropical epiphytic bryophytes functional performance due to the low amount of literature available.

Author comment 1:

Thank you very much for these constructive ideas on $CO_2$ gas exchange measurements. It indeed would be good to include some measurements conducted by ourselves. However, from past experiences we know that quick gas exchange measurements might deliver truly misleading results. Just as an example, it has been shown by colleagues, that after transport to the lab, tropical organisms showed only a fraction of the physiological activity previously assessed in the field. The samples had strongly suffered from the transport, as they had to be air-dried prior to the transport in order to avoid molding during that time. Thus, we think that $CO_2$ gas exchange measurements indeed make sense, but that they also need to be conducted with care. This indeed is planned for the future, but would go beyond the scope of the current study. For the present study, we found some very good data on lowland rain forest bryophytes, assessed by a group, which is well-experienced in $CO_2$ gas exchange measurements. Thus, for the current study we decided to use their results in order to assess potential physiological activity patterns, but we also stress the potential sources of error and inaccuracy of this approach. We hope that we could convince you of the validity of this approach.

During the review process, we conducted a complete revision of the calibration process for the water content sensors resulting in by far smaller inaccuracies.

**Some minor points also to comment:**
**INTRO**

Referee comment 2:

Page 3, Ls 20-25: I would focus in bryophytes functional properties rather than in general physiological features of cryptogamic covers because only bryophytes are included in the experimental design.

Author comment 2:

Thank you for this comment. The whole introduction was revised, with the aim to focus more on the epiphytic bryophyte communities.

Author changes in the text 2:

P3 L 2: "Cryptogamic communities comprise photosynthesizing organisms, i.e. cyanobacteria, algae, lichens, and bryophytes, which grow together with heterotrophic fungi, other bacteria, and archaea. They can colonize different substrates, such as soil, rock, and plant surfaces in almost all habitats throughout the world (Büdel, 2002; Elbert et al., 2012; Freiberg, 1999). In the tropics, epiphytic bryophyte communities widely cover the stems and branches of trees (Campos et al., 2015). Within that habitat, they may play a prominent role in environmental nutrient cycling (Coxson et al., 1992) and also influence the microclimate within the forest (Porada et al., 2019), thus contributing to the overall fitness of the host plants and the surrounding vegetation (Zartman, 2003). However, they are equally affected by deforestation and increasing forest fragmentation (Zartman, 2003; Zotz et al., 1997).

Physiologically, cryptogamic organisms are characterized by their poikilohydric nature, as they do not actively regulate their water status but passively follow the water conditions of their surrounding environment (Walter and Stadelmann, 1968). In a dry state, many of them can outlast extreme weather conditions, being reactivated by water (Oliver et al., 2005; Proctor, 2000; Proctor et al., 2007; Seel et al., 1992), and for several species even fog and dew can serve as a source of water (Lancaster et al., 1984; Lange et al., 2006; Lange and Kilian, 1985; Reiter et al., 2008). In contrast, high water contents (WC) may cause suprasaturation, when gas diffusion is restrained, causing reduced $CO_2$ gas exchange rates (Cowan et al., 1992; Lange and Tenhunen, 1981; Snelgar et al., 1981) and even ethanolic fermentation, as shown for lichens (Wilske et al., 2001). Accordingly, their physiological activity is primarily regulated by the presence of water and only secondarily by light and temperature (Lange et al., 1996, 1998, 2000; Rodriguez-Iturbe et al., 1999).

In the Amazonian rain forest, cryptogamic communities mainly occur epiphytically on the stems, branches, and even leaves of trees, and in open forest fractions they may also occur on the soil (Richards, 1954). By 2013, 800 species of mosses and liverworts, 250 lichen species, and 1,800 fungal species have been reported for the Amazon region (Campos et al., 2015; Gradstein et al., 2001; Komposch and Hafellner, 2000; Normann et al., 2010; Piepenbring, 2007). Tropical rain forests are characterized by humid conditions, high temperatures with minor annual fluctuations, and an immense species diversity of flora and fauna. Currently, between 16 000 and 25 000 tree species have been estimated for the Amazonian rain forest (Hubbell et al., 2008; ter Steege et al., 2013). It has been described to play important roles in the water cycle, as well as for carbon, nitrogen, and phosphorus fluxes on regional and global scales (Andreae et al., 2015). However, it is also hard to predict, to which extent the ongoing and envisioned changes will still ensure its ecological services as "green lung" and carbon sink of planet Earth (Soepadmo, 1993).

Studies in temperate zones address the importance of cryptogamic communities for the ecosystem (Gimeno et al., 2017; Rastogi et al., 2018), but for the tropical area, few reports can be found in literature. There is a lack of information regarding the functioning of such communities in an environment with an almost constant high relative humidity and temperature range. Thus, with the long-term continuous measurements presented here, we aim to provide data on seasonality patterns and the vertical profile of the microclimate within the canopy. In the current study, we present the microclimatic conditions, comprising the temperature, light, and WC of epiphytic bryophytes communities along a vertical gradient and an estimation of their activity patterns in response to annual and seasonal variations of climatic conditions. "

**METHODOLOGY**

Referee comment 3:

-Section 2.5. Could you please explain in more detail why some meteorological parameters are measured at 26m and light is measured at 75m?

Author comment 3:

Monitoring of the meteorological parameters is conducted in the course of the overall ATTO long-term measurements (for more details see Andreae et al., 2015). For this, different sensors have been installed at different heights in order to serve the needs. Ambient (above-canopy) light is measured at 75 m in order to avoid shading of the canopy and also precipitation and fog need to be measured above the canopy (at 81 and 50 m height, respectively). The different height levels are also explained by the different amounts of space needed by the sensors. We see that as uncritical for these parameters, as ambient light intensity, fog, and precipitation should not vary between 50 and 81 m height. For relative ambient air humidity and ambient temperature we decided to use the data closest to the canopy, i.e. at 26 m height.

In the revised version of the manuscript, we provide the information that the meteorological parameters are assessed in the course of the long-term monitoring at the site and we also provide a scheme illustrating the different sensor locations below, within and above the canopy (Supplement Figure S2, see below). We hope this will clarify the sensor setup in some more detail.

Author changes in the text 3:

P12 L7: "For the purpose of long-term monitoring, a set of meteorological parameters is being measured within the ATTO project since 2012."

[Figure]

**Figure S2:** Schematic overview on the sensors installed at different height levels below, within, and above the canopy. The parameters water content (WC) and temperature (Temp) were measured within the bryophyte samples, the light sensors (PAR) were installed directly on top of the thalli. The average tree height of 21 m was determined for the plateau forest in general.

Referee comment 4:

-Section 2.6. I would establish the possible ranges for each ecophysiological parameter analyzed focusing more in tropical epiphytic bryophytes functional performance.

Author comment 4:

Yes, this entire estimation was revised to restrict the considered values to epiphytic bryophytes of tropical lowland forests. Also Table S3 in its revised version, only shows lowland studies.

Author change in the text 4:

Changes in Table S3.

**Table S3:** Parameters determining fractional time of photosynthesis and respiration. The lower water compensation point (WCP$_l$), the lower light compensation point (LCP$_l$), the temperature for optimal net photosynthesis (T$_{opt}$ NP),

and the upper temperature compensation point (TCP) as relevant parameters have been extracted from published studies conducted at lowland sites of tropical rain forests.

| Parameter | Low | High | Unit | Reference | Study site |
|---|---|---|---|---|---|
| $WCP_l$ | 30 | 80 | % DW | Wagner et al 2013 | Panama, lowland rain forest, 0 m |
| $LCP_l$ | 3 | 12 | $\mu$mol m$^{-2}$ s$^{-1}$ PPFD | Lösch et al. 1994 | Zaire, lowland rain forest, 800 m |
| $T_{opt}$ NP | 24 | 27 | °C | Wagner et al 2013 | Panama, lowland rain forest, 0 m |
| TCP | 30 | 36 | °C | Wagner et al 2013 | Panama, lowland rain forest, 0 m |

**RESULTS**

Referee comment 5:

-Section 3.1. 2 consecutive years of microclimatic data availability is a good and interesting output provided by authors

Author comment 5:

Yes, this is a long term monitoring project and the database on the water content, temperature, and light conditions of epiphytes is uploaded to the ATTO data portal (www.attoproject.org/).The data thus are maintained, obtain a doi and can be retrieved from that site.

Referee comment 6:

-All sections in general. I see that the headings do not correspond too much with what is written at each of the sections. Authors mix concepts in the same paragraphs such as microclimate, mesoclimate, water content, seasonal and daily analyses …Would it be possible to rethink the headings of the sections and write text more focused to each of the headings?

Author comment 6:

Indeed, it was not an easy task to structure the manuscript in a logical way and in the end we decided to use a structure to analyze the data according to different time frames (i.e., comparison of years, seasons, diel cycles, etc.). Thus, indeed, different climatic parameters are sometimes used within one paragraph to illustrate their interdependence. However, we also considered this comment and looked over the structure within the paragraphs again. We now avoid mixing different parameters wherever this is possible.

Author change in the text 6:

Some structural changes were made throughout the manuscript in order to obtain an overall better readability.

Referee comment 7:

P 10 L9, I think that authors missed a word after "35%", maybe "lower"?

Author comment 7:

Yes, "35 % *lower*", we added the word.

Author change in the text 7:

P14 L17: "Comparing the two consecutive years, the effect of an El Niño event was clearly detectable, as rainfall amounts were 35 % *lower* (525 mm versus 805 mm) and relative air humidity 11 % lower (81 % versus 92 %) between October 2015 and February 2016 as compared to the same time-span in the previous year (Fig. 1, Table S2)."

Referee comment 8:

How did authors compare climate statistically between years/seasons? Did you use a monthly basis? Daily basis?

Author comment 8:

The mean values were calculated from the 5-minute data points. However, in the revised version of the manuscript we decided to omit the statistical tests. Author changes in the text 8:

Table 1: Annual mean values…"Mean values were calculated from 5-minute intervals, except for $PAR_{max}$, where the daily maximum values were considered."

Table 2: Seasonal mean values…" Mean values for the respective seasons were calculated from 5-minute intervals of the years 2015 and 2016, except for $PAR_{max}$, where the daily maximum values were considered."

Referee comment 9:

P 10 L 25-26. If I understood ok, the idea is that the microclimatic T value at the moss level was higher than ambient T, and that this is a frequent pattern. What about the shading effect of the tree canopy over microclimatic T?

Author comment 9:

Yes, indeed there is some shading effect of the canopy, which could result in a reduced heating of the bryophytes, also at 23 m height within the canopy. However, also ambient T measurements are always performed in the shade to avoid a short-term impact of direct insulation. Thus, we do not think that there is a large difference in shading. However, we think that wind intensities are reduced within and below the canopy and that the bryophytes have a higher heat storage capacity, which both may cause higher temperatures measured within the bryophytes.

Referee comment 10:

Fig 1, legend. I would say estimated water content of the bryophytes rather than "ecophysiological conditions"

Author comment 10:

The expression "*ecophysiological*" was finally omitted and was changed throughout the text and figures and replaced by "*water content of bryophytes*".

Author change in the text 10:

Figure 1: "Water content (WC), temperature, and light conditions experienced by bryophyte communities, and above-canopy meteorological conditions in the Amazonian rain forest…."

DISCUSSION:

Referee comment 11:

P 14 Ls 22-24. I think that these patterns observed reinforces that measuring some gas exchange control points might be useful.

Author comment 11:

We completely agree that additional $CO_2$ gas exchange measurements would be of interest. Our hesitation to measure just some cardinal points is explained in the first section of this response letter. We also explain there, that, under the current conditions, we prefer to use a well-established study over quick measurements conducted by ourselves. We

prefer to conduct an in-depth $CO_2$ gas exchange study in the near future, which, however, goes beyond the scope of the current manuscript at hand.

Referee comment 12:

P 4 line 13: Remove "The".

Author comment 12:

Yes, the word was removed.

Referee comment 13:

P 17 Ls 19-23. I do not understand this point properly.

Author comment 13:

This paragraph was adapted to clarify the information. The intention was to express that respiration is more sensitive to temperature than photosynthesis.

Author change in the text 13:

*P 23 L27:* "The temperature regulates the velocity of metabolic processes, hence it has a strong impact on the respiration, while the photosynthetic light reaction is by far less sensitive (Elbert et al., 2012; Green and Proctor, 2016; Lange et al., 1998). As the measured net photosynthesis rates are the sum of simultaneously occurring photosynthesis and respiration processes, positive net photosynthesis may still be reached at higher temperatures, if the photosynthetic capacity is high enough, whereas during the night, high temperatures could cause a major loss of carbon due to high respiration rates (Lange et al., 2000)."

**References provided by the author:**

Büdel, B.: History of flora and vegetation during the quaternary, in Progress in Botany, edited by K. Esser, U. Lüttge, W. Beyschlag, and F. Hellwig, pp. 386–404, Springer-Verlag., 2002.

Coxson, D. S., McIntyre, D. D. and Vogel, H. J.: Pulse Release of Sugars and Polyols from Canopy Bryophytes in Tropical Montane Rain Pulse Release of Sugars and Polyols from Canopy Bryophytes in Tropical Montane Rain Forest (Guadeloupe, French West Indies), Biotropica, 24, 121–133, 1992.

Elbert, W., Weber, B., Burrows, S., Steinkamp, J., Büdel, B., Andreae, M. O. and Pöschl, U.: Contribution of cryptogamic covers to the global cycles of carbon and nitrogen, Nat. Geosci., 5, 459–462, doi:10.1038/ngeo1486, 2012.

Freiberg, E.: Influence of microclimate on the occurrence of Cyanobacteria in the Phyllosphere in a Premontane Rain Forest of Costa Rica, Plant Biol., 1, 244–252, 1999.

Green, T. G. A. and Proctor, M. C. F.: Physiology of Photosynthetic Organisms Within Biological Soil Crusts: Their Adaptation, Flexibility, and Plasticity, in Biological Soil Crusts: An Organizing Principle in Drylands, edited by B. Weber, B. Büdel, and J. Belnap, pp. 347–381, Springer International Publishing, Cham., 2016.

Lancaster, J., Lancaster, N. and Seely, M.: Climate of the Central Namib Desert, Madoqua, 14, 5–61, 1984.

Lange, O. L. and Kilian, E.: Reaktivierung der Photosynthese trockener Flechten durch Wasserdampfaufnahme aus dem Luftraum: Artspezifisch unterschiedliches Verhalten, Flora, 176, 7–23, doi:10.1016/S0367-2530(17)30100-7, 1985.

Lange, O. L., Belnap, J. and Reichenberger, H.: Photosynthesis of the cyanobacterial soil-crust lichen Collema tenax from arid lands in southern Utah, USA: Role of water content on light and temperature responses of CO2 exchange, Funct. Ecol., doi:10.1046/j.1365-2435.1998.00192.x, 1998.

Lange, O. L., Allan Green, T. G., Melzer, B., Meyer, A. and Zellner, H.: Water relations and CO2 exchange of the terrestrial lichen Teloschistes capensis in the Namib fog desert: Measurements during two seasons in the field and under controlled conditions, Flora - Morphol. Distrib. Funct. Ecol. Plants, 201(4), 268–280, doi:10.1016/J.FLORA.2005.08.003, 2006.

Oliver, M. J., Velten, J. and Mishler, B. D.: Desiccation Tolerance in Bryophytes: A Reflection of the Primitive Strategy for Plant Survival in Dehydrating Habitats?, INTEGR. COMP. BIOL, 45, 788–799, 2005.

Porada, P., Van Stan, J. T. and Kleidon, A.: Significant contribution of non-vascular vegetation to global rainfall interception, Nat. Geosci., 11(8), 563–567, doi:10.1038/s41561-018-0176-7, 2018.

[revised manuscript text omitted]
 bryophytes in the understoryat the lower levels (1.5 and 8 m;) were often still an increased WCwetnot completely dry when the next rain event started, while the WC of the liverworts bryophytes in the canopy had already dried outdecreased to the minimal level frequently reached during daytime (Fig. S56a).

Furthermore, the angle of the stem or branch colonized by the investigated bryophytes played a crucial role for rainwater absorption and the subsequent drying process (Table 2Fig. S2). The bryophytes at 1.5 and 8 m height were oriented vertically, those at 18 m were placed on the upper side of a slightly sloping branch, and those at 23 m were located on the upper side of a nearly horizontally oriented branch. Long-term climate data have shown that the winds during the wet season predominantly originated from north and north-eastern directions, while during the dry season south- and south-easterly winds prevailed (Pöhlker et al., 2018). At 8 m height, the investigated bryophytes were exposed to the west, and thus were only sometimes directly influenced by precipitation, as in most cases, due to the predominant wind directions, north , east , and south oriented tree fractions received the largest precipitation amounts. Long-term climate data have shown that the winds during the wet season predominantly originated from north and north-eastern directions, while during the dry season south- and south-easterly winds prevailed (Pöhlker et al., 2018). In contrast to that, the bryophytesAlso at 18 23 m height the bryophytes did not always showed a clear response to precipitation events, even ifalthough they weare oriented horizontally on a branch (Fig. 2, Fig. S56). Here, the bryophyte cushions were exposed to the south, which is more frequently influenced by rain events. Thus, the shift of the main wind direction from northeasterly to southeasterly might explain the fact that the bryophytes at 18 and 23 m height responded more strongly to rain events in the dry season than they did in the wet season. MoreoverIt can be expected that, besides the dominating wind direction, also the tree foliation and epiphytic vascular plants might shield the sensors from direct precipitation during the wet season.

[revised manuscript text omitted]

~~The optimum temperatures for net photosynthesis ($T_{opt}$) range from 25.0 to 27.5 °C for tropical bryophytes (Wagner et al., 2013), and these values were reached during 6 to 32 % of the time at all four height levels with no major differences among them. The upper temperature compensation point ($T_{CPu}$) between 30.0 and 36.0 °C (Wagner et al., 2013) was only rarely reached during our study (i.e., up to 17 % of the time). Thedoesplay a relevant role asFurthermore, it was indicated thatmightbeation, as the species are well adapted to the prevailing environmental condition~~s (Wagner et al., 2013).

[revised manuscript text omitted]
 2016, expeccept for PARmax, where the daily maximum values has beenwere considered.

| Height [m] | $PAR_{avg}$ daytime $[\mu mol\ m^{-2}\ s^{-1}]$ Mean | ± SD | $PAR_{max}$ $[\mu mol\ m^{-2}\ s^{-1}]$ Mean | ± SD | Temperature [°C] Mean | ± SD | RH (above canopy), WC [%] Mean | ± SD |
|---|---|---|---|---|---|---|---|---|
| **Wet season** | | | | | | | | |
| above-canopy | 738 | 566 | 2086 | 515 | 25.6 | 2.5 | 143 | 36 |
| 23 Liverwort | 30 | 3 | 248 | 194 | 25.3 | 2.0 | 125 | 33 |
| 18 Liverwort | 39 | 12 | 282 | 175 | 25.2 | 1.9 | 113 | 37 |
| 8 Liverwort | 31 | 26 | 144 | | 24.9 | 1.1 | 31 | 10 |
| 8 Moss | | | | | | | 64 | 29 |
| 1.5 Moss | 4 | 15 | 114 | 224 | 24.9 | 1.0 | 60 | 50 |
| **Transitional season Wet-Dry** | | | | | | | | |
| above-canopy | 861 | 649 | 2227 | 182 | 25.8 | 3.0 | 143 | 57 |
| 23 Liverwort | 41 | 72 | 414 | 252 | 25.7 | 2.8 | 128 | 41 |
| 18 Liverwort | 44 | 54 | 351 | 123 | 25.4 | 2.3 | 127 | 20 |
| 8 Liverwort | 66 | 88 | 165 | 218 | 24.9 | 1.4 | 25 | 5 |
| 8 Moss | | | | | | | 54 | 21 |
| 1.5 Moss | 2 | 12 | 61 | 102 | 24.6 | 1.1 | 24 | 15 |
| **Dry season** | | | | | | | | |
| above-canopy | 973 | 647 | 2100 | 609 | 26.7 | 3.4 | 119 | 52 |
| 23 Liverwort | 55 | 9 | 503 | 231 | 27.2 | 3.5 | 122 | 52 |
| 18 Liverwort | 41 | 13 | 412 | 190 | 26.5 | 2.9 | 107 | 52 |
| 8 Liverwort | 23 | 16 | 295 | 268 | 26.0 | 2.1 | 32 | 28 |
| 8 Moss | | | | | | | 51 | 33 |
| 1.5 Moss | 6 | 25 | 209 | 299 | 25.5 | 1.7 | 23 | 20 |
| **Transitional season Dry-Wet** | | | | | | | | |
| above-canopy | 785 | 617 | 1988 | 509 | 26.5 | 3.3 | 141 | 67 |
| 23 Liverwort | 55 | 91 | 530 | 297 | 27.2 | 3.7 | 130 | 48 |
| 18 Liverwort | 37 | 28 | 185 | 109 | 26.6 | 3.0 | 137 | 75 |
| 8 Liverwort | 21 | 47 | 269 | 178 | 26.3 | 2.5 | 61 | 49 |
| 8 Moss | | | | | | | 56 | 24 |
| 1.5 Moss | 4 | 20 | 107 | 113 | 26.0 | 2.1 | 35 | 33 |

Manuscript with track-changes 39

**Table 2:** Seasonal mean values, standard deviation (± SD), and statistically significant difference between seasons for the parameters photosynthetically active radiation ($PAR_{avg}$), daily maximum of photosynthetically active radiation ($PAR_{max}$), temperature, and above canopyambient relative humidity/water content (WC). Values measured as above canopyambient conditions and within/on top of bryophytes at four height levels. Mean values for the respective seasons were calculated from 5 minute intervals of the years 2015 and 2016, expect for the $PAR_{max}$, where the daily maximal value have been considered. Due to the absence of normal distribution and variance homogeneity, a non parametric Kruskal Wallis test with post hoc test was performed to compare values obtained for the different seasons. The statistical comparison among height levels for the different seasons is shown in Table S5.

| Season | $PAR_{avg}$ daytime [µmol m$^{-2}$·s$^{-1}$ PPFD] Mean | ± SD | sig. | $PAR_{max}$ [µmol m$^{-2}$·s$^{-1}$ PPFD] Mean | ± SD | sig. | Temperature [°C] Mean | ± | sig. | WC; above-canopy RH Mean | ± SD | sig. |
|---|---|---|---|---|---|---|---|---|---|---|---|---|
| **above-canopy** | | | | | | | | | | | | |
| Wet | 738 | 566 | a | 2086 | 515 | a | 25.6 | 2.5 | ab | 94 | 9 | a |
| Trans Wet-Dry | 861 | 649 | a | 2227 | 182 | a | 25.8 | 3.0 | ab | 91 | 11 | b |
| Dry | 973 | 647 | a | 2100 | 609 | a | 26.7 | 3.4 | bc | 87 | 14 | c |
| Trans Dry-Wet | 785 | 617 | a | 1988 | 509 | b | 26.5 | 3.3 | ca | 85 | 15 | d |
| Statistical test, p | 1.000 | | | ≤0.001 | | | ≤0.001 | | | ≤0.001 | | |
| **23 m** | - | - | - | - | - | - | - | - | - | - | - | - |
| Wet | 30 | 3 | a | 248 | 194 | a | 25.3 | 2.0 | a | 143 | 36 | |
| Trans Wet-Dry | 41 | 72 | a | 414 | 252 | b | 25.7 | 2.8 | b | 143 | 57 | |
| Dry | 55 | 9 | a | 503 | 231 | c | 27.2 | 3.5 | c | 119 | 52 | |
| Trans Dry-Wet | 55 | 91 | a | 530 | 297 | c | 27.2 | 3.7 | d | 141 | 67 | |
| Statistical test, p | 1.000 | | | ≤0.001 | | | ≤0.001 | | | | | |
| **18 m** | - | - | - | - | - | - | - | - | - | - | - | - |
| Wet | 39 | 12 | a | 282 | 175 | a | 25.2 | 1.9 | a | 110 | 37 | |
| Trans Wet-Dry | 44 | 54 | a | 351 | 123 | b | 25.4 | 2.3 | b | 133 | 20 | |
| Dry | 41 | 13 | a | 412 | 190 | b | 26.5 | 2.9 | c | 152 | 121 | |
| Trans Dry-Wet | 37 | 28 | a | 185 | 109 | c | 26.6 | 3.0 | d | 202 | 159 | |
| Statistical test, p | 1.000 | | | ≤0.001 | | | ≤0.001 | | | | | |
| **8 m Liverwort** | - | - | - | - | - | - | - | - | - | - | - | - |
| Wet | 31 | 26 | a | 144 | 194 | a | 24.9 | 1.1 | a | 32 | 11 | |
| Trans Wet-Dry | 66 | 88 | a | 165 | 218 | a | 24.9 | 1.4 | ab | 26 | 6 | |
| Dry | 23 | 16 | a | 295 | 268 | b | 26.0 | 2.1 | bc | 56 | 88 | |
| Trans Dry-Wet | 21 | 47 | a | 269 | 178 | b | 26.3 | 2.5 | cd | 72 | 100 | |
| Statistical test, p | 1.000 | | | ≤0.001 | | | ≤0.001 | | | | | |
| **8 m Moss** | - | - | - | - | - | - | - | - | - | - | - | - |
| Wet | | | | | | | | | | 41 | 21 | |
| Trans Wet-Dry | | | | | | | | | | 38 | 15 | |
| Dry | | | | | | | | | | 30 | 23 | |
| Trans Dry-Wet | | | | | | | | | | 42 | 23 | |
| **1.5 m** | - | - | - | - | - | - | - | - | - | - | - | - |
| Wet | 4 | 15 | a | 114 | 224 | a | 24.9 | 1.0 | a | 48 | 44 | |
| Trans Wet-Dry | 2 | 12 | a | 61 | 102 | a | 24.6 | 1.1 | b | 9 | 12 | |
| Dry | 6 | 25 | a | 209 | 299 | b | 25.5 | 1.7 | c | 13 | 11 | |

| Trans Dry Wet | 4 20 | a | 107 113 | b | 26.0 2.1 | d | 27 34 |
| Statistical test, p | 1.000 | | $\leq 0.001$ | | $\leq 0.001$ | | |

**Table 3:** The potential time ranges [%], during which the epiphytic bryophytes reached the lower compensation  points of light (LCP$_l$), the optimal temperature for net photosynthesis (T$_{opt}$), the upper compensation point for temperature (TCP$_u$),  the lower compensation  points  for water  (WCP). The conditions at which the compensation points were reached are listed, and the potential time range, during which NP and DR might occur were listed  Values are given for the different height levels and bryophyte divisions (M=moss, L=liverwort). Five-minute averages of measurements during the entire measurement period from October 2014 to December 2016 were considered. The ranges of the compensation points (CP) and optimal ranges (opt) were reported in  Lösch (1994) and Wagner et al. (2013) (see Table S3).

|  Height [m] | LCP$_l$ | T$_{opt}$ | TCP$_u$ | WCP$_l$ | Conditions  for NP and DR LCP$_l$/TCP$_{NP=DR}$/WCP$_{34}$  | NP WC > WCP$_l$ & & T < TCP$_u$ | DR WC > WCP$_l$ & PAR < LCP$_l$ or WC > WCP$_l$ & T > TCP$_u$ |
|---|---|---|---|---|---|---|---|
| | 3-12 µmol m$^{-2}$ s$^{-1}$ | 25.0-27.0 ° C | 30.0-36.0 ° C | 30-80 % DW |  | | |
| [m] | Time fraction when **cardinal points are reached** [% of time] | | | | µmol m$^{-2}$ s$^{-1}$/°C/%DW | Time fraction when **cardinal points are reached** [% of time] | |
| 23 L | 36-45 | 6-46 | 0-16 | 29 | 3-12/30-36/30-80 | 28-58 | 40-47 |
| 18 L | 39-47 | 6-51 | 0-13 | 47 | 3-12/30-36/30-80 | 27-59 | 30-33 |
| 8 L | 29-40 | 13-29 | 0-17 | 2-33 | 3-12/30-36/30-80 | 1-23 | 3-16 |
| 8 M | | | | 3-65 | 3-12/30-36/30-80 | 5-46 | 9-30 |
| 1.5 M | 2-15 | 14-30 | 0-11 | 8-291-36 | 3-12/30-36/30-80 | 0-65 | 10-26 |

**Figures**

[revised manuscript text omitted]

**Figure S3:** Calibration curves of water content sensors installed within different bryophyte species. The water content [% DW] is plotted against the electrical conductivity [mV] for the species a) *Leucobryum martianum*, b) *Sematophyllum subsimplex*, c) *Symbiezidium barbiflorum*, and d) *Octoblepharum cocuiense*. Of each bryophyte species three replicates (four for *Sematophyllum subsimplex*, two for *Symbiezidium barbiflorum*) were measured over the course of three subsequent wetting and drying cycles. The dots show the measured data points and the lines represent the statistical fit. Depending opn the data, a linear fit, quadratic fit or linear fit with exponential correction was used (see methods section for further details. The vertical grey bars indicate the data range covered during the field measurements. For each fit the R² and RMSE are given in the graphics.

[Figure]

**Figure S4:** Comparison of 5-minute (dots) and 30-minute (lines)  averages of exemplary sensors at each height level over a period of approx. one day in December 2016.

[Figure]

**Figure S45:** Representative periods during wet and dry season under the influence of El Niño,; data showing the water content (WC), temperature, and light conditions (PAR$_{avg}$) experience by bryophytes, and above-canopy meteorological conditions in the Amazonian rain forest. Shown are 8-day periods during a) the wet season 2016̶5 and b) the dry season 2015̶6. The micrometeorological parameters on top/within epiphytic cryptogamic communities represent (A) the

photosynthetically active radiation (PAR$_{avg}$) on top, (B) the temperature within, and (C) the water content of cryptogamic communities. The above-canopy meteorological parameters comprise (A) the above-canopy photosynthetically active radiation (PAR$_{avg}$ at 75 m), (B) the above-canopy temperature (at 26 m), (D) the relative air humidity (RH at 26 m), the presence of fog events, and (E) the rain. The data show 30-minute averages ± SD except for rain, which shows hourly sums. Data of replicate sensors installed within communities at the same height level were pooled, while above-canopy parameters were measured with one sensor each. The night time is shaded in grey (06:00 – 18:00 LT).

[Figure]

**Figure S56:** Temperature within bryophytes compared to the above-canopy  temperature. The temperature within bryophytes was measured at 1.5 m , 8 m , 18 m , and 23 m , while the above-canopy temperature was measured at 26 m height on the tower. Data present 30-minute averages with linear fits. For each height level the coefficients and the R² are given.

[Figure]

**Figure S7:** Two exemplary fog events and the reaction of the moisture sensors of the bryophytes (a and b). Each panel presents (A) a fog event with the parameters fog with visibility < 2000 m being defined as fog occurrence, (B) relative air humidity (RH), (C) rain, and (D) the water content (WC) of the bryophytes shown for some exemplary sensors. The fog event of interest is marked by a red box. For the WC sensors the number, height of installation, and division (M = Moss, L = Liverwort) are given.

[Figure]

**Figure S8:** Temperature conditions of bryophytes related to their water content. The temperature was measured in bryophytes at different height levels along the tree. Data presented as 30-minute averages.

**Table S1a:** Bryophyte species and calibration data of the water content sensors. Listed are the bryophyte species with their division (moss or liverwort)bryophyte type, their height and height zone of installation, the minimum ($Min_{Field}$) and the maximum ($Max_{Field}$) electrical conductivity assessed in the field, the maximum electrical conductivity measured during calibration ($Max_{Calib}$), the fit being used for the calibration (l = linear, lec = linear with exponential correction, sqrt = quadratic), the root mean square error (RMSE), and the determination coefficient R².., and the root mean square error (RMSE). The value of RMSE is given in % dry weight (% DW). Calculations are based on 30 minute integrals of data. *) The species name cannot be verified without any doubts, however, the information of the calibration done with this species was considered for further data analysis, due to morphological properties.

| Bryophyte species | Bryophyte type | Height [m] | Height zone | $Min_{Field}$ mV | $Max_{Field}$ mV | $Max_{Calib}$ mV | Fit | R² | RMSE % DW |
|---|---|---|---|---|---|---|---|---|---|
| *Symbiezidium barbiflorum* | Liverwort | 8, 18, 23 | 2, 3, 4 | 16 | 1857 | 2207 | l | 0.583 | 362 |
| *Octoblepharum cocuiense* | Moss | 8 | 2 | 13 | 750 | 992 | l | 0.662 | 124 |
| *Sematophyllum subsimplex* | Moss | 1.5 | 1 | 5 | 1940 | 290 | lec | 0.760 | 115 |
| *Leucobryum martianum* | Moss | 1.5 | 1 | 40 | 1005 | 656 | sqrt | 0.871 | 135 |

| Bryophyte species | Division | Height [m] | Height zone | RSME [% DW] | R² |
|---|---|---|---|---|---|
| *Sematophyllum subsimplex* | Moss | | | 44 | 0.95 |
| *Sematophyllum subsimplex* | Moss | | | 125 | 0.84 |
| *Sematophyllum subsimplex* | Moss | | | 35 | 0.95 |
| *Sematophyllum subsimplex* | Moss | | | 66 | 0.81 |
| *Octoblepharum cocuiense* | Moss | | | 66 | 0.89 |
| *Octoblepharum cocuiense* | Moss | | | | 0.8 |
| *Leucobryum martianum* | Moss | | | 162 | 0.89 |
| *Leucobryum martianum* | Moss | | | 135 | 0.72 |
| *Leucobryum martianum* | Moss | | | 159 | 0.76 |
| *Symbiezidium barbiflorum** | Liverwort | | | 146 | 0.89 |
| *Symbiezidium barbiflorum** | Liverwort | | | 306 | 0.78 |
| *Symbiezidium barbiflorum** | Liverwort | - | - | 235 | 0.8 |
| *Sematophyllum subsimplex* | Moss | 1.5 | 1 | 68 | 0.89 |
| *Octoblepharum cocuiense* | Moss | 8 | 2 | 85 | 0.84 |
| *Leucobryum martianum* | Moss | 1.5 | 1 | 152 | 0.79 |
| *Symbiezidium barbiflorum** | Liverwort | 8, 18, 23 | 2, 3, 4 | 229 | 0.82 |

**Table** S1bS1**:** Height of installation, and minimum and maximum values of the individual sensors of the microclimate station measuring water content, temperature, and light (PAR). For the water content sensors, also the bryophyte species are given. Based on 30-minute integralaverages.

| Water content | Height [m] | WC [% DW] min | WC [% DW] max | Bryophyte species |
|---|---|---|---|---|
| Sensor 01 | 1.5 | 187 | 6601512 | *Sematophyllum subsimplex* |
| Sensor 02 | 1.5 | 185 | 5431512 | *Sematophyllum subsimplex* |
| Sensor 03 | 1.5 | 0121 | 6401512 | *Sematophyllum subsimplex* |
| Sensor 04 | 1.5 | 3194 | 6391455 | *Leucobryum martianum* |
| Sensor 05 | 1.5 | 5124 | 6451512 | *Sematophyllum subsimplex* |
| Sensor 06 | 1.5 | 331 | 7301487 | *Sematophyllum subsimplex* |
| Sensor 07 | 8 | 114 | 14801286 | *Symbiezidium barbiflorum* |
| Sensor 08 | 8 | 218 | 1066798 | *Octoblepharum cocuiense* |
| Sensor 09 | 8 | 316 | 1223950 | *Octoblepharum cocuiense* |
| Sensor 10 | 8 | 218 | 1075789 | *Octoblepharum cocuiense* |
| Sensor 11 | 8 | 330 | 12621130 | *Symbiezidium barbiflorum* |
| Sensor 12 | 8 | 629 | 1355811 | *Symbiezidium barbiflorum* |
| Sensor 13 | 18 | 639 | 1584782 | *Symbiezidium barbiflorum* |
| Sensor 14 | 18 | 1738 | 1345295 | *Symbiezidium barbiflorum* |
| Sensor 15 | 18 | 1745 | 1552315 | *Symbiezidium barbiflorum* |
| Sensor 16 | 18 | 1344 | 1573327 | *Symbiezidium barbiflorum* |
| Sensor 17 | 18 | 1032 | 1342575 | *Symbiezidium barbiflorum* |
| Sensor 18 | 18 | 037 | 16421703 | *Symbiezidium barbiflorum* |
| Sensor 19 | 23 | 1443 | 1283536 | *Symbiezidium barbiflorum* |
| Sensor 20 | 23 | 45 | 393 | *Symbiezidium barbiflorum* |
| Sensor 21 | 23 | 1737 | 1252864 | *Symbiezidium barbiflorum* |
| Sensor 22 | 23 | 1348 | 1066774 | *Symbiezidium barbiflorum* |
| Sensor 23 | 23 | 2966 | 893514 | *Symbiezidium barbiflorum* |
| Sensor 24 | 23 | 068 | 1725492 | *Symbiezidium barbiflorum* |

| Temperature | Height [m] | Temperature [°C] min | Temperature [°C] max |
|---|---|---|---|
| Sensor 01 | 1.5 | 21.1 | 36.3 |
| Sensor 02 | 1.5 | 21.4 | 39.4 |
| Sensor 03 | 8 | 21.6 | 34.7 |
| Sensor 04 | 8 | 20.9 | 46.3 |
| Sensor 05 | 18 | 20.3 | 38.0 |
| Sensor 06 | 18 | 20.3 | 37.5 |
| Sensor 07 | 23 | 20.8 | 41.2 |
| Sensor 08 | 23 | 20.3 | 48.7 |

| Light | Height [m] | PAR [μmol m$^{-2}$ s$^{-1}$] min | PAR [μmol m$^{-2}$ s$^{-1}$] max |
|---|---|---|---|
| Sensor 01 | 1.5 | 0 | 6341546 |
| Sensor 02 | 8 | 0 | 5691461 |
| Sensor 03 | 8 | 0 | 11211502 |
| Sensor 04 | 18 | 0 | 5251386 |
| Sensor 05 | 18 | 0 | 6151080 |
| Sensor 06 | 23 | 0 | 6541326 |
| Sensor 07 | 23 | 0 | 7671351 |

**Table S1b:** Bryophyte species and calibration data of the water content sensors. Listed are the bryophyte species with their division (moss or liverwort), their height and height zone of installation, the root mean square error (RMSE), and the determination coefficient R². *) The species name cannot be verified without any doubts, however, the information of the calibration done with this species was considered for further data analysis, due to morphological properties.

| Bryophyte species | Division | Height [m] | Height zone | RSME [% DW] | R² |
|---|---|---|---|---|---|
| *Sematophyllum subsimplex* | Moss | | | 44 | 0.95 |
| *Sematophyllum subsimplex* | Moss | | | 125 | 0.84 |
| *Sematophyllum subsimplex* | Moss | | | 35 | 0.95 |
| *Sematophyllum subsimplex* | Moss | | | 66 | 0.81 |
| *Octoblepharum cocuiense* | Moss | | | 66 | 0.89 |
| *Octoblepharum cocuiense* | Moss | | | 103 | 0.8 |
| *Leucobryum martianum* | Moss | | | 162 | 0.89 |
| *Leucobryum martianum* | Moss | | | 135 | 0.72 |
| *Leucobryum martianum* | Moss | | | 159 | 0.76 |
| *Symbiezidium barbiflorum** | Liverwort | | | 146 | 0.89 |
| *Symbiezidium barbiflorum** | Liverwort | | | 306 | 0.78 |
| *Symbiezidium barbiflorum** | Liverwort | - | - | 235 | 0.8 |
| *Sematophyllum subsimplex* | Moss | 1.5 | 1 | 68 | 0.89 |
| *Octoblepharum cocuiense* | Moss | 8 | 2 | 85 | 0.84 |
| *Leucobryum martianum* | Moss | 1.5 | 1 | 152 | 0.79 |
| *Symbiezidium barbiflorum** | Liverwort | 8, 18, 23 | 2, 3, 4 | 229 | 0.82 |

**Table S2:** Monthly mean values and standard deviations (± SD) of photosynthetically active radiation (PAR$_{avg}$ daytime, measured at 75 m), daily maxima of photosynthetically active radiation (PAR$_{max}$), temperature (measured at 26 m), and relative humidity (RH, measured at 26 m). Rainfall is presented as the monthly amounts and the percentage of days with rain (measured at 81 m), and also the percentage of days when rain detection malfunctioned are listed. Fog events are given as the percentage of days. Due to data gaps in the measured rain data (shown in brackets) values for 21 days of rain were also extrapolated from existing data as described in methods section (values behind data in brackets). Values were calculated from 30-minute intervals. Fog has not being recorded in the time ranges of 31.05.  -20.10.2015, 30.04.  -06.07.2016, 01.09.  -31.12.2016 due to malfunction of the device.

| Month | PAR$_{avg}$ daytime [µmol m$^{-2}$ s$^{-1}$ ] | | PAR$_{max}$ [µmol m$^{-2}$ s$^{-1}$ ] | | Temperature [°C] | | RH [%] | | Rain [mm month$^{-1}$] | Rain [% days] | Defect on rain detection [% days] | Fog -[% days] |
|---|---|---|---|---|---|---|---|---|---|---|---|---|
| | Mean | ± SD | Mean | ± SD | Mean | ± SD | Mean | ± SD | | | | |
| Oct 2014 | 857 | 668 | 2201 | 509 | 26.0 | 2.8 | 90 | 11 | 212 | 58 | 0 | 55 |
| Nov 2014 | 832 | 624 | 2082 | 423 | 25.6 | 2.9 | 92 | 11 | 70 | 57 | 0 | 53 |
| Dec 2014 | 843 | 582 | 2140 | 346 | 26.3 | 2.7 | 90 | 11 | 123 | 42 | 0 | 42 |
| Jan 2015 | 637 | 525 | 1747 | 735 | 24.5 | 2.4 | 95 | 8 | 259 | 71 | 0 | 71 |
| Feb 2015 | 774 | 589 | 2058 | 600 | 25.4 | 2.6 | 92 | 10 | 140 | 64 | 0 | 46 |
| Mar 2015 | 680 | 534 | 2038 | 575 | 24.7 | 2.1 | 96 | 7 | 331 | 87 | 0 | 77 |
| Apr 2015 | 766 | 564 | 2155 | 463 | 25.3 | 2.5 | 93 | 10 | 189 | 80 | 0 | 40 |
| May 2015 | 725 | 559 | 2103 | 425 | 27.2 | n.a. | 93 | 6 | 320 | 90 | 0 | 58 |
| Jun 2015 | 804 | 562 | 2237 | 128 | 25.0 | 2.3 | 94 | 8 | 178 | 80 | 0 | 0* |
| Jul 2015 | 892 | 605 | 2238 | 188 | 25.7 | 3.0 | 91 | 11 | 74 | 65 | 0 | 0* |
| Aug 2015 | 1017 | 636 | 1722 | 957 | 27.1 | 3.3 | 83 | 13 | (23) 32* | 23 | 23 | 0* |
| Sep 2015 | 1148 | 687 | 2242 | 467 | 28.7 | 3.7 | 74 | 15 | 38 | 13 | 20 | 0* |
| Oct 2015 | 968 | 635 | 2072 | 514 | 28.4 | 3.6 | 78 | 16 | 55 | 35 | 3 | 13* |
| Nov 2015 | 887 | 624 | 1859 | 769 | 27.9 | 3.5 | 81 | 16 | (33) 37* | 30 | 17 | 23 |
| Dec 2015 | 862 | 575 | 2074 | 304 | 28.1 | 3.0 | 78 | 14 | 38 | 26 | 3 | 6 |
| Jan 2016 | 882 | 606 | 2175 | 270 | 28.2 | 3.4 | 78 | 16 | 52 | 48 | 0 | 13 |
| Feb 2016 | 743 | 550 | 1928 | 679 | 25.9 | 2.6 | 93 | 10 | (267) 341* | 79 | 52 | 48 |
| Mar 2016 | 692 | 545 | 2041 | 545 | 25.6 | 2.1 | 96 | 7 | 304 | 90 | 0 | 77 |
| Apr 2016 | 709 | 564 | 2088 | 443 | 25.6 | 2.3 | 96 | 7 | 277 | 87 | 0 | 73 |
| May 2016 | 817 | 603 | 2230 | 405 | 26.1 | 2.6 | 94 | 8 | 236 | 90 | 0 | 0* |
| Jun 2016 | 828 | 584 | 2178 | 261 | 25.6 | 2.8 | 92 | 10 | 105 | 57 | 0 | 0* |
| Jul 2016 | 917 | 629 | 2253 | 118 | 26.2 | 3.2 | 88 | 12 | 92 | 58 | 0 | 26* |
| Aug 2016 | 1016 | 648 | 2146 | 593 | 27.1 | 3.5 | 83 | 14 | 40 | 32 | 3 | 16 |
| Sep 2016 | 947 | 662 | 2230 | 543 | 26.5 | 3.1 | 89 | 12 | (77) 96* | 50 | 17 | 0* |
| Oct 2016 | 915 | 641 | 2323 | 192 | 27.1 | 3.3 | 86 | 14 | (1) 9* | 23 | 23 | 0* |
| Nov 2016 | 911 | 610 | 2227 | 217 | 27.1 | 3.3 | 87 | 13 | (30) 89* | 20 | 13 | 0* |
| Dec 2016 | 694 | 553 | 1955 | 503 | 25.4 | 2.7 | 94 | 10 | 223 | 71 | 0 | 0* |

*) Gaps in the data record due to malfunction of the device.

**Table S3:** Parameters determining time range of photosynthesis and respiration. The  lower water compensation point (WCP),  the lower light compensation point (LCP_l), the  temperature for optimal net photosynthesis (T_opt ), and the upper temperature compensation point (TCP as relevant parameters have been extracted from published studies conducted at various study sites in the tropical rain forest.

[revised manuscript text omitted]

**Table S65:** Daily maximum values of the photosynthetically active radiation (PAR$_{max}$), the temperature (Temp$_{max}$), and the water content (WC$_{max}$) of epiphytic bryophytes. Mean values and standard deviations (± SD) are shown for dry and wet seasons of the two years 2015 and 2016. For the above-canopy data maximum air humidity (RH) values measured at 26 m are shown, while for the bryophytes the water content was assessed. Above-canopy light intensity was measured at 75 m, above-canopy temperature and relative air humidity at 26 m.  Data of the sensors installed at the same height level were pooled, while the above-canopy parameters were measured with one sensor each.

| Season | PAR$_{max}$ [μmol m$^{-2}$ s$^{-1}$ ] Mean | ± SD | Temp$_{max}$ [°C] Mean | ± SD | RH$_{max}$ (above canopy), WC$_{max}$ [%] Mean | ± SD |
|---|---|---|---|---|---|---|
| above-canopy | | | | | | |
| Dry15 | 1966 | 730 | 33.5 | 2.1 | 96 | 5 |
| Dry16 | 2232 | 425 | 32.3 | 1.8 | 99 | 2 |
| Wet15 | 2089 | 515 | 28.9 | 2.4 | 99 | 3 |
| Wet16 | 2084 | 517 | 30.4 | 1.9 | 100 | 1 |
| 23 m Liverwort | | | | | | |
| Dry15 | 431 | 239 | 35.8 | 3.9 | 186 | 262 |
| Dry16 | 575 | 260 | 37.4 | 4.7 | 171 | 67 |
| Wet15 | 167 | 202 | 28.4 | 2.5 | 175 | 43 |
| Wet16 | 329 | 223 | 31.8 | 3.2 | 160 | 73 |
| 18 m Liverwort | | | | | | |
| Dry15 | 381 | 207 | 33.3 | 2.0 | 126 | 16 |
| Dry16 | 443 | 204 | 32.8 | 2.2 | 252 | 274 |
| Wet15 | 274 | 208 | 28.4 | 1.9 | 169 | 112 |
| Wet16 | 289 | 188 | 29.6 | 1.7 | 144 | 89 |
| 8 m Liverwort | | | | | | |
| Dry15 | | | | | 29 | 14 |
| Dry16 | | | | | 232 | 377 |
| Wet15 | | | | | 36 | 12 |
| Wet16 | | | | | 62 | 96 |
| 8 m Moss | | | | | | |
| Dry15 | 414 | 381 | 32.0 | 3.2 | 68 | 13 |
| Dry16 | 175 | 258 | 31.0 | 3.9 | 65 | 8 |
| Wet15 | 246 | 395 | 26.5 | 1.5 | 98 | 65 |
| Wet16 | 44 | 88 | 27.8 | 1.8 | 103 | 92 |
| 1.5 m Moss | | | | | | |
| Dry15 | 290 | 369 | 29.3 | 1.6 | 25 | 42 |
| Dry16 | 127 | 173 | 29.0 | 2.5 | 33 | 100 |
| Wet15 | 132 | 284 | 26.0 | 1.0 | 113 | 102 |
| Wet16 | 96 | 140 | 27.0 | 1.0 | 116 | 100 |

**Table S6:** Daily amplitudes of the photosynthetically active radiation (PAR_amp), the temperature (Temp_amp), and the water content (WC_amp) of epiphytic bryophytes. Mean values _and_ standard deviations (± SD) are shown for dry and wet seasons of the two years 2015 and 2016. For the a_bove-canopy_ data maximum air humidity (RH) values _measured at 26 m_ are shown, while for the bryophytes the water content was assessed. A_bove-canopy_ light intensity was measured at 75 m, a_bove-canopy_ temperature and relative air humidity at 26 m.  _Data of the sensors installed at the same height level were pooled, while the above-canopy parameters were measured with one sensor each._

| Season | PAR_amp [μmol m⁻² s⁻¹ ] | | Temp_amp [°C] | | RH_amp _(above canopy)_, WC_amp [%] | |
|---|---|---|---|---|---|---|
| | Mean | ± SD | Mean | ± SD | Mean | ± SD |
| **above-canopy** | | | | | | |
| Dry15 | 1966 | 730 | 9.8 | 2.1 | 39 | 10 |
| Dry16 | 2232 | 425 | 9.3 | 1.6 | 35 | 8 |
| Wet15 | 2089 | 515 | 6.3 | 2.8 | 18 | 13 |
| Wet16 | 2084 | 517 | 7.0 | 1.8 | 23 | 9 |
| **23 m Liverwort** | | | | | | |
| Dry15 | 431 | 239 | 11.2 | 3.2 | 132 | 258 |
| Dry16 | 575 | 260 | 13.2 | 4.5 | 109 | 65 |
| Wet15 | 167 | 202 | 5.3 | 2.4 | 73 | 43 |
| Wet16 | 329 | 223 | 8.1 | 3.1 | 95 | 71 |
| **18 m Liverwort** | | | | | __ | |
| Dry15 | 381 | 207 | 9.3 | 1.6 | 70 | 21 |
| Dry16 | 443 | 204 | 9.1 | 1.9 | 176 | 269 |
| Wet15 | 274 | 208 | 5.5 | 1.8 | 112 | 98 |
| Wet16 | 289 | 188 | 6.0 | 1.7 | 72 | 37 |
| **8 m Liverwort** | | | | | | |
| Dry15 | | | | | 18 | 13 |
| Dry16 | | | | | 240 | 396 |
| Wet15 | | | | | 11 | 6 |
| Wet16 | | | | | 41 | 56 |
| **8 m Moss** | | | | | | |
| Dry15 | 414 | 381 | 7.0 | 3.2 | 45 | 19 |
| Dry16 | 175 | 258 | 6.8 | 3.9 | 54 | 8 |
| Wet15 | 246 | 395 | 3.1 | 1.4 | 58 | 64 |
| Wet16 | 44 | 88 | 3.7 | 1.9 | 68 | 83 |
| **1.5 Moss** | | | | | __ | __ |
| Dry15 | 290 | 369 | 4.4 | 1.2 | 16 | 41 |
| Dry16 | 127 | 173 | 4.9 | 2.4 | 23 | 88 |
| Wet15 | 132 | 284 | 2.5 | 1.0 | 73 | 89 |
| Wet16 | 96 | 140 | 2.8 | 1.0 | 85 | 88 |

**Table S7:** Daily minimum values of the photosynthetically active radiation (PAR$_{min}$), the temperature (Temp$_{min}$), and the  water content (WC$_{min}$) of epiphytic bryophytes. Mean values and standard deviations (± SD) are shown for dry and wet seasons of the two years 2015 and 2016. For the above-canopy data maximum air humidity (RH) values

5 measured at 26 m are shown, while for the bryophytes the water content was assessed. Above-canopy light intensity was measured at 75 m, above-canopy temperature and relative air humidity at 26 m. Data of the sensors installed at the same height level were pooled, while the above-canopy

10 parameters were measured with one sensor each.

| Season | PAR$_{min}$ [µmol m$^{-2}$ s$^{-1}$ ] Mean | ± SD | Temp$_{min}$ [°C] Mean | ± SD | RH$_{min}$ (above canopy), WC$_{min}$ [%] Mean | ± SD |
|---|---|---|---|---|---|---|
| **above-canopy** | | | | | | |
| Dry15 | 0 | 0 | 23.7 | 1.1 | 57 | 11 |
| Dry16 | 0 | 0 | 23.1 | 0.9 | 65 | 8 |
| Wet15 | 0 | 0 | 22.6 | 1.7 | 81 | 12 |
| Wet16 | 0 | 0 | 23.5 | 0.7 | 77 | 9 |
| **23 m Liverwort** | | | | | | |
| Dry15 | 0 | 0 | 24.7 | 1.5 | 54 | 18 |
| Dry16 | 0 | 0 | 24.1 | 1.3 | 59 | 19 |
| Wet15 | 0 | 0 | 23.1 | 0.7 | 102 | 18 |
| Wet16 | 0 | 0 | 23.6 | 0.6 | 66 | 23 |
| **18 m Liverwort** | | | | | | |
| Dry15 | 0 | 0 | 24.0 | 1.0 | 55 | 23 |
| Dry16 | 0 | 0 | 23.7 | 1.0 | 89 | 82 |
| Wet15 | 0 | 0 | 22.9 | 0.6 | 57 | 34 |
| Wet16 | 0 | 0 | 23.6 | 0.6 | 72 | 25 |
| **8 m Liverwort** | | | | | | |
| Dry15 | | | | | 11 | 4 |
| Dry16 | | | | | 26 | 23 |
| Wet15 | | | | | 25 | 10 |
| Wet16 | | | | | 20 | 13 |
| **8 m Moss** | | | | | | |
| Dry15 | 0 | 0 | 25.0 | 1.0 | 24 | 13 |
| Dry16 | 0 | 0 | 24.2 | 0.9 | 10 | 7 |
| Wet15 | 0 | 0 | 23.4 | 0.5 | 41 | 18 |
| Wet16 | 0 | 0 | 24.0 | 0.5 | 35 | 21 |
| **1.5 m Moss** | | | | | | |
| Dry15 | 0 | 0 | 24.8 | 1.0 | 9 | 6 |
| Dry16 | 0 | 0 | 24.1 | 0.9 | 10 | 28 |
| Wet15 | 0 | 0 | 23.5 | 0.5 | 40 | 30 |
| Wet16 | 0 | 0 | 24.1 | 0.5 | 31 | 29 |

**Table S9:** ~~Daily maximum of the photosynthetically active radiation (PAR$_{max}$), the temperature (Temp$_{max}$), and the ambient relative humidity/water content (WC$_{max}$) of epiphytic bryophytes. Mean values, standard deviations (± SD), significance, and p-values are shown for dry and wet seasons of the two years 2015 and 2016. For the ambient data maximum air humidity (RH) values are shown, while for the bryophytes the water content was assessed. Ambient light intensity was measured at 75 m, ambient temperature and relative air humidity at 26 m. Due to the absence of normal distribution and variance homogeneity a non-parametric Kruskal Wallis test with post hoc test was performed to compare values obtained for different seasons.~~

| Height [m] | PAR$_{max}$ [μmol m$^{-2}$ s$^{-1}$ PPFD] Mean | ±SD | sig. | p | Temp$_{max}$ [°C] Mean | ±SD | sig. | p | WC$_{max}$ [%] Mean | ±SD | sig. | P |
|---|---|---|---|---|---|---|---|---|---|---|---|---|
| Dry season 2015 | | | | | - | | | - | | | | |
| above-canopy | 1966 | 730 | d | 0.000 | 33.5 | 2.1 | e | 0.000 | - | - | - | - |
| 23 Liverwort | 431 | 239 | e | | 35.8 | 3.9 | d | - | 195 | 275 | | |
| 18 Liverwort | 381 | 207 | be | | 33.3 | 2.0 | e | - | 132 | 16 | | |
| 8 Liverwort | | | | | | | | | 30 | 14 | | |
| 8 Moss | 414 | 381 | b | | 32.0 | 3.2 | b | - | 44 | 9 | | |
| 1.5 Moss | 290 | 369 | a | - | 29.3 | 1.6 | a | - | 21 | 36 | | |
| Dry season 2016 | | | | | - | | | - | | | | |
| above-canopy | 2232 | 425 | d | 0.000 | 32.3 | 1.8 | e | 0.000 | | | | |
| 23 Liverwort | 575 | 260 | e | | 37.4 | 4.7 | d | - | 179 | 70 | | |
| 18 Liverwort | 443 | 204 | b | | 32.8 | 2.2 | e | - | 265 | 287 | | |
| 8 Liverwort | | | | | | | | | 244 | 396 | | |
| 8 Moss | 175 | 258 | a | | 31.0 | 3.9 | b | - | 42 | 5 | | |
| 1.5 Moss | 127 | 173 | a | - | 29.0 | 2.5 | a | - | 27 | 82 | | |
| Wet season 2015 | | | | | - | | | - | | | | |
| above-canopy | 2089 | 515 | d | 0.000 | 28.9 | 2.4 | e | 0.000 | | | | |
| 23 Liverwort | 167 | 202 | e | | 28.4 | 2.5 | e | - | 183 | 46 | | |
| 18 Liverwort | 274 | 208 | b | | 28.5 | 1.9 | e | - | 178 | 118 | | |
| 8 Liverwort | | | | | | | | | 38 | 12 | | |
| 8 Moss | 246 | 395 | ac | | 26.5 | 1.5 | b | - | 63 | 42 | | |
| 1.5 Moss | 132 | 284 | a | - | 26.0 | 1.0 | a | - | 92 | 83 | | |
| Wet season 2016 | | | | | - | | | - | | | | |
| above-canopy | 2084 | 517 | d | 0.000 | 30.4 | 1.9 | ed | 0.000 | | | | |
| 23 Liverwort | 329 | 223 | e | | 31.8 | 3.2 | ed | - | 168 | 77 | | |
| 18 Liverwort | 289 | 188 | e | | 29.6 | 1.7 | e | - | 152 | 32 | | |
| 8 Liverwort | | | | | | | | | 65 | 65 | | |
| 8 Moss | 44 | 88 | b | | 27.8 | 1.8 | b | - | 66 | 59 | | |
| 1.5 Moss | 96 | 140 | a | - | 27.0 | 1.0 | a | - | 94 | 81 | | |

**Table S10:** Daily minimum of the photosynthetically active radiation ($PAR_{min}$), the temperature ($Temp_{min}$), and the ambient relative humidity/water content ($WC_{min}$) of epiphytic bryophytes. Mean values, standard deviations ($\pm$ SD), significance, and p-values are shown for dry and wet seasons of the two years 2015 and 2016. For the ambient data maximum air humidity (RH) values are shown, while for the bryophytes the water content was assessed. Ambient light intensity was measured at 75 m, ambient temperature and relative air humidity at 26 m. Due to the absence of normal distribution and variance homogeneity a non parametric Kruskal Wallis test with post hoc test was performed to compare values obtained for different seasons.

| - | $PAR_{min}$ [µmol m$^{-2}$ s$^{-1}$ PPFD] | | | | $Temp_{min}$ [°C] | | | | $WC_{min}$ [%] | | | |
|---|---|---|---|---|---|---|---|---|---|---|---|---|
| Height [m] | Mean | ± SD | sig. | p | Mean | ± SD | sig. | p | Mean | ± SD | sig. | p |
| Dry season 2015 | | | | | - | | | - | | | | |
| above canopy | 0 | 0 | a | 1.000 | 23.7 | 1.1 | c | 0.000 | - | - | - | - |
| 23 Liverwort | 0 | 0 | a | | 24.7 | 1.5 | b | - | 56 | 19 | | |
| 18 Liverwort | 0 | 0 | a | | 24.0 | 1.0 | c | - | 58 | 24 | | |
| 8 Liverwort | | | | | | | | | 11 | 5 | | |
| 8 Moss | 0 | 0 | a | | 25.0 | 1.0 | a | - | 15 | 8 | | |
| 1.5 Moss | 0 | 0 | a | - | 24.8 | 1.0 | ab | - | 7 | 5 | | |
| Dry season 2016 | | | | | - | | | - | | | | |
| above canopy | 0 | 0 | a | 1.000 | 23.1 | 0.9 | c | 0.000 | | | | |
| 23 Liverwort | 0 | 0 | a | | 24.1 | 1.3 | a | - | 60 | 20 | | |
| 18 Liverwort | 0 | 0 | a | | 23.7 | 1.0 | b | - | 93 | 86 | | |
| 8 Liverwort | | | | | | | | | 27 | 23 | | |
| 8 Moss | 0 | 0 | a | | 24.2 | 0.9 | a | - | 8 | 6 | | |
| 1.5 Moss | 0 | 0 | a | - | 24.1 | 0.9 | a | - | 8 | 23 | | |
| Wet season 2015 | | | | | - | | | - | | | | |
| above canopy | 0 | 0 | a | 1.000 | 22.6 | 1.7 | b | 0.000 | | | | |
| 23 Liverwort | 0 | 0 | a | | 23.1 | 0.7 | c | - | 107 | 19 | | |
| 18 Liverwort | 0 | 0 | a | | 22.9 | 0.6 | b | - | 60 | 36 | | |
| 8 Liverwort | | | | | | | | | 26 | 11 | | |
| 8 Moss | 0 | 0 | a | | 23.4 | 0.5 | a | - | 26 | 12 | | |
| 1.5 Moss | 0 | 0 | a | - | 23.5 | 0.5 | a | - | 32 | 24 | | |
| Wet season 2016 | | | | | - | | | - | | | | |
| above canopy | 0 | 0 | a | 1.000 | 23.5 | 0.7 | b | 0.000 | | | | |
| 23 Liverwort | 0 | 0 | a | | 23.6 | 0.6 | b | - | 69 | 25 | | |
| 18 Liverwort | 0 | 0 | a | | 23.6 | 0.6 | b | - | 76 | 26 | | |
| 8 Liverwort | | | | | | | | | 21 | 14 | | |
| 8 Moss | 0 | 0 | a | | 24.0 | 0.5 | a | - | 22 | 14 | | |
| 1.5 Moss | 0 | 0 | a | - | 24.1 | 0.5 | a | - | 26 | 23 | | |

Table S11: Daily amplitudes of the photosynthetically active radiation (PAR$_{amp}$), the temperature (Temp$_{amp}$), and the ambient relative humidity/water content (WC$_{amp}$) of epiphytic bryophytes. Mean values, standard deviations (± SD), significance, and p-values are shown for dry and wet seasons of the two years 2015 and 2016. For the ambient data maximum air humidity (RH) values are shown, while for the bryophytes the water content was assessed. Ambient light intensity was measured at 75 m, ambient temperature and relative air humidity at 26 m. Due to the absence of normal distribution and variance homogeneity a non-parametric Kruskal-Wallis test with post hoc test was performed to compare values obtained for different seasons.

| Height [m] | PAR$_{amp}$ [µmol m$^{-2}$ s$^{-1}$ PPFD] | | | | Temp$_{amp}$ [°C] | | | | WC$_{amp}$ [%] | | | |
| --- | --- | --- | --- | --- | --- | --- | --- | --- | --- | --- | --- | --- |
| | Mean | ± SD | sig. | p | Mean | ± SD | sig. | p | Mean | ± SD | sig. | p |
| **Dry season 2015** | | | | | - | | | - | | | | |
| above canopy | 1966 | 730 | d | 0.000 | 9.8 | 2.1 | dc | 0.000 | - | - | - | - |
| 23 Liverwort | 431 | 239 | c | | 11.2 | 3.2 | d | - | 139 | 271 | | |
| 18 Liverwort | 381 | 207 | bc | | 9.3 | 1.6 | c | | 74 | 22 | | |
| 8 Liverwort | | | | | | | | | 19 | 14 | | |
| 8 Moss | 414 | 381 | b | | 7.0 | 3.2 | b | - | 29 | 12 | | |
| 1.5 Moss | 290 | 369 | a | - | 4.4 | 1.2 | a | - | 13 | 35 | | |
| **Dry season 2016** | | | | | - | | | - | | | | |
| above canopy | 2232 | 425 | d | 0.000 | 9.3 | 1.6 | b | 0.000 | | | | |
| 23 Liverwort | 575 | 260 | c | | 13.2 | 4.5 | d | - | 118 | 66 | | |
| 18 Liverwort | 443 | 204 | b | | 9.1 | 1.9 | c | | 171 | 261 | | |
| 8 Liverwort | | | | | | | | | 216 | 391 | | |
| 8 Moss | 175 | 258 | a | | 6.8 | 3.9 | b | - | 35 | 5 | | |
| 1.5 Moss | 127 | 173 | a | - | 4.9 | 2.4 | a | - | 19 | 72 | | |
| **Wet season 2015** | | | | | - | | | - | | | | |
| above canopy | 2089 | 515 | d | 0.000 | 6.3 | 2.8 | c | 0.000 | | | | |
| 23 Liverwort | 167 | 202 | c | | 5.3 | 2.4 | c | - | 76 | 46 | | |
| 18 Liverwort | 274 | 208 | b | | 5.5 | 1.8 | c | - | 117 | 103 | | |
| 8 Liverwort | | | | | | | | | 12 | 6 | | |
| 8 Moss | 246 | 395 | ac | | 3.1 | 1.4 | b | - | 37 | 41 | | |
| 1.5 Moss | 132 | 284 | a | - | 2.5 | 1.0 | a | - | 60 | 73 | | |
| **Wet season 2016** | | | | | - | | | - | 69 | 72 | | |
| above canopy | 2084 | 517 | d | 0.000 | 7.0 | 1.8 | d | 0.000 | | | | |
| 23 Liverwort | 329 | 223 | c | | 8.1 | 3.1 | d | - | 99 | 75 | | |
| 18 Liverwort | 289 | 188 | c | | 6.0 | 1.7 | c | - | 76 | 39 | | |
| 8 Liverwort | | | | | | | | | 44 | 59 | | |
| 8 Moss | 44 | 88 | b | | 3.7 | 1.9 | b | - | 44 | 53 | | |
| 1.5 Moss | 96 | 140 | a | - | 2.8 | 1.0 | a | - | 69 | 72 | | |

---

## Referee Report (RR1)

1. Does the paper address relevant scientific questions within the scope of BG? yes
2. Does the paper present novel concepts, ideas, tools, or data? Yes (data)
3. Are substantial conclusions reached? Yes, although the most important conclusion, realistically, is that not enough ecophysiolgical data about tropical bryophytes are available to draw strong conclusions.
4. Are the scientific methods and assumptions valid and clearly outlined? Mostly yes, but the water content results cannot be right and need to be recalculated.
5. Are the results sufficient to support the interpretations and conclusions? Mostly yes, the conclusions are more cautious now than in the first version. Some details still need to be presented more cautiously.
6. Is the description of experiments and calculations sufficiently complete and precise to allow their reproduction by fellow scientists (traceability of results)? Of the experiments yes, of the calculations of water contents no.
7. Do the authors give proper credit to related work and clearly indicate their own new/original contribution? Yes.
8. Does the title clearly reflect the contents of the paper? Yes.
9. Does the abstract provide a concise and complete summary? Yes.
10. Is the overall presentation well structured and clear? Yes.
11. Is the language fluent and precise? Mostly yes.
12. Are mathematical formulae, symbols, abbreviations, and units correctly defined and used? Yes, but some are unnecessary.
13. Should any parts of the paper (text, formulae, figures, tables) be clarified, reduced, combined, or eliminated? Yes.
14. Are the number and quality of references appropriate? Yes.
15. Is the amount and quality of supplementary material appropriate? Yes.

The new version of your manuscript, now with the adjusted title "Microclimatic conditions and water content fluctuations experienced by epiphytic bryophytes in an Amazonian rain forest" has improved a lot, in particular in the more cautious discussion of the results. I now have only one major problem with the results, and that is in the very unlikely water content values.

I appreciate it that you tried out the method I suggested, and I still think that method makes sense, but the results are not very convincing as they are. It does not make any sense that the liverworts in the canopy should have a constantly high water content. Also, it does not make sense that the bryophytes in the understorey should never reach WCs above about 400% (Table S1, Fig S8 – even if this is for 30-min averages, during rain events the mosses could stay at their maximum capacity for half an hour), if their maximum WC is about 1500% for Leucobryum and 1000% for Sematophyllum (your data in Fig S3 in the previous version).

In any case, there must still be something wrong with the calculation. Perhaps it would help to not take the absolute min and max mV signal ever measured by the sensors, but something like the 5% and 95% Quartiles, to avoid using spurious signals. Supporting the assumption that this has happened is that it does not make sense that any of the mosses and liverworts should go down to values as low as 0 or 1% WC - this is what one would reach in a good drying oven, not at >70% RH… So the real minimum WC should be higher. Also, there may be a meaning in the fact that the maximum electrical conductivity measured in the field was much higher than those measured during

calibration for Leucobryum (Lm) and Sematophyllum (Ss) (Fig S3 of previous version). The result of this is that the new function used estimates much lower WC values than those estimated by the calibration curves. I agree that the calibration curves (not presented in this version of the paper as they were not used for the new WC calculations) were problematic due to the huge variability (especially for Symbiezidium, not so much for Lm). Still, they might be used to constrain the range of relationships between electrical conductivity and water contents that can be considered acceptable, and for Lm and Ss the current functions use would fall outside this range, with systematic underestimations of the water content. If the calibration curves have a meaning, I would expect electrical conductivity in the field to be usually between near-0 and 600 (Lm) or 300 (Ss). By plotting the histogram of the conductivity values it may be possible to identify a more realistic value corresponding to maximum moss wetness.

For Symbiezidium I suspect that the water content is, on the other hand, systematically over-estimated. It seems rather impossible to me that a canopy bryophyte would maintain a WC of around 100% all the time. It would be really good if the quantitative translation of sensor output to water content can be managed, because the data do show that there is a signal in the data, e.g. by the strong response of at least some of the sensors to rain events. I think there are some interesting patterns in the data, in particular the diel fluctuation of the WC in the upper canopy, apparently following RH fluctuations. It does make sense that this diel fluctuation is largest at the upper two heights. So it is mainly the determination of the absolute WC that is problematic, not so much the fluctuations.

There may be a way to deal with this though. Apart from trying not using the absolute minimum and maximum ever measured but the 5% / 95% quartiles, you may also need to use the 30-minute averages, so as not to use the lower end of the short-term fluctuations as a minimum and upper end of those fluctuations as a maximum. Judging from Figure S4 those fluctuations may be a problem especially at 23 m, which could explain the high mean values there if you use the 5-min data for the calibration. I could imagine that after rain, they may also become more pronounced at e.g. 1.5 m. I might have to take back the recommendation to use 5-min values for the estimation of activity times, at least for WC.

A second point: I wonder how interesting it is, in the context of predicting activity of the mosses, to put emphasis on the seasonal patterns in the mean values of the climatic variables. What could be more interesting is to analyse the bryophyte activity patterns separately for the seasons.

**Some smaller points:**

Even if you do not use the calibration any more in this version, please do describe the fact that variability was high in these experiments, so that the uncertainty of the WC values is expected to be very high. Discuss why the calibration curves could not be used, so why the current approach was chosen. Also, on the positive side, you can describe how single measurement series showed a more or less linear relationship between mV and WC (except at very high WC, where the measurement tended to became saturated and mV did not longer change), justifying your linear approach in calculating WC from mV.

It would be good to present the min and max WC values (from your calibration curves) to give readers more insight into the WC calculations.

The discussion could present a stronger line in the points being made. As it is, some paragraphs seem a bit lost, rather than incorporated into a story. The story could, for example, be centred on the activity patterns, presenting the microclimatic data in that context. At the moment, context is missing a bit for some sections.

Marks, R.A., B.D. Pike and D. Nicholas McLetchie, 2019. Water stress tolerance tracks environmental exposure and exhibits a fluctuating sexual dimorphism in a tropical liverwort. Oecologia. Available from https://www.ncbi.nlm.nih.gov/pubmed/31664577. DOI 10.1007/s00442-019-04538-2

**Detailed suggestions:**

Abstract (p2), L16 I would not call the diel fluctuations in WC ´frequent wetting and drying events´ (although my expectation would indeed be that they would go through more than in the understorey, but form your data this is not obvious), how about calling them diel fluctuations? It looks like for your samples, the lower ones actually lived through more wetting and drying events as they responded to rainfall more directly / consistently (would be nice to quantify ´consistently´..). In any case, the WC calculations need to be revised so this sentence may still change, although the fluctuations will probably be little affected by the recalculation of the absolute values.

P2 L7-8 why the quotation marks?

P2 L21: measurements of $CO_2$ gas exchange would be necessary, but this study is not a starting point for such measurements. Maybe change to "supported by measurements of $CO_2$ gas exchange"

P2 L30 I do not understand the ´equally´ here

P3 L13 Have been reported

P3 L17 Clarify what ´It´ is

P3 L25 The ´Thus´ does not fit well. You introduce ecosystem functions but present data on microclimate…

P3 L29 variations in climatic conditions

P4 L23 What exactly do you mean by the ´story structure´? Is this more than just the vertical structure?

P5 L1 Why ´Generally´?

P5 L10-11 not only….. and height.

P5 L21 were not installed

P5 L28-30 Did you check/correct for drifts in the measured light levels due to e.g. algal growth on the light sensors?

P6 L11 would the voltage not be proportional to the conductivity if the sensor pins did not have a fixed distance?

P6 L22-25 This calculation of the water content does not really need a formula or a citation. It is simply the water content expressed per dry mass.

P6 L27-30 Like explained above, I would try using not the absolute min and max but something like the 5% and 95% quartiles, or some other reasonable min and max based on a histogram of the mV signals.

P6 L5 remove 8

P6 L6 were considered

P6 L12-15 try using a parallel structure (why were some data collected, others measured and others assessed?

P6 L20 I am still not convinced the ´integration´ is what you mean here.

P8 L7 We have already shown that high respiration loss due to high temperatures are probably not so important, as respiration rates are adapted/acclimatized to the elevation at which bryophytes grow (Wagner et al 2013, Annals of Botany) This could already be acknowledged here. And also in P12 L3-4, and in P16 L12, rather than suggesting that there is still all reason to think that night T is important to then, surprise surprise, conclude that it is not..

P8 L15 WCs BELOW the WCP

P8 L23 if light intensity is above and temperature below the compensation point

P9 L16-19 I would suggest to make clear here already that these measurements may be strongly influenced by local canopy cover, thus not necessarily reflecting the conditions for that stratus of the forest in general, and should therefore be taken with caution.

P9 L24-26 why is this interesting?

P9 L27 similar patterns to what?

P10 L2 RH where?

P10 L3 I really do not belive this result (RH highest at 23 m)

P10 L8-9 Suggestion to not talk about this result as ´showing high WC´ but rather as ´showing high conductivity, but this could not be related to the WC because of reinstallation… Or if you have enough data you could adjust the function by taking a new max (95%) and min (5%) mV signal.

P10 L21 Temperatures showed (not reflected)

P10 L28 Any way of knowing whether the fog touched the canopy? That would

P11 L2-3 This is a very interesting observation. It would be great if you could show (calculate) how consistent this response is. The examples shown in Fig S7 are not necessarily convincing, given all the fluctuations in the WC…

P11 L10-11 these amplitudes are caused by fluctuations at different scales (rain events at 1.5 m, diel humidity fluctuations at 23-m, therefore I am not convinced that it is useful to compare them. Also, this sentence seems repeated in L 27-30.

P11 L21 what ´mean´ are your referring to?

P11 L9 make clear here whether ´reported time´ considers 24-h or only daytime

P12 L12 Instead of ´microclimatic temperatures´ I would use ´temperatures inside the moss stands´

P13 Because of the reasons described in L12-13 and L17-18, I would not present the differences in light levels between seasons (L6-8) in that context, though it is probably worth discussing the artefacts because of the use of these data for estimating activity patterns.

P13 L31-34 What is the interest in knowing the difference in mean temperature, which is physiologically meaningless..?

P14 L1 This sentence can be removed

P14 L5-8 Example of a ´lost´ paragraph with no clear function in the story

P14 L9 it is not the response that changes between seasons, but the conditions.

P15 L4 What is ´stepwise´ about this drying?

P15 L5-7 You know what the water holding capacity (WHC) of your species is, and you even have information about their drying speed (both from your calibration curves), so there is no need to speculate here. Unfortunately, I think the WHCs of your species do not explain the pattern at all, if anything, they would cause a reverse pattern, with Leucobryum staying moist longer than Symbiezidium…

P15 L16 I agree with this, but it could be elaborated upon a bit more, and I think you need to aim at making these values indeed approximate but no longer biased.

P15 L21-22 It is not correct to equal acclimation processes to intraspecific variation. Intraspecific variation can also be a result of adaptation, see e.g. Marks et al 2019. The references cited here also do not refer to acclimation.

P15 L24-25 What is the function of this sentence? It breaks up the flow of the story.

P16 L1-4 reacting rapidly and efficiently to light flecks is not at all the same as being efficient at low light levels, this sentence thus does not make much sense:

P16 first paragraph. The ´so what´ of this paragraph is unclear.

P16 L7-8 is photosynthesis not a metabolic process?

P16 L17 I do not agree that this has been shown at all, I think this sentence and those that follow will be removed after revising the WC method.

P16 L24-25 Really? What would be the mechanisms for this?

P17 L3 was "exceeded" during.. Except that it was not ;-)

P17 L13 I agree! Except from questioning the generality of the published compensation points, this would also be a good place to critically evaluate your microclimatic data!

P18 L1-2 revise after recalculations. And I would rise the diurnal variation in WC in the upper levels to the conclusion section. This could be very relevant, but this depends on the exact WC values and the exact WCPs, neither of which, unfortunately, we know…

P18 L4 … minor variation relative to the physiological tolerances of the mosses, as far as these are known, …

**Tables**

Table 3 It is not totally clear what is meant by ´are reached´. It looks like you mean ´exceeded´, which has the problem that, depending whether it is a lower or upper CP, the shown values can be the time net photosynthesis could be positive (LCP, WCP) or the time that it could not (TCP)… While for Topt it is not so clear.

Table3 : why a column for showing the ´conditions´ which are the same everywhere?

P29 L4 occur are listed

**Figures**

P30 L5 remove ´epiphytic´ or ´the´

Fig 2: it is hard to see all the information here, could it be printed larger?

Fig 3 match order in caption to order in graph.

Fig 3b: The green for WC at 18m is a different colour than the rest

**Supplementary material**

Fig S4 This figure is cool it shows nicely how the T at 23m is most variable and mostly higher than at lower heights. The model for the 23-m data does not seem to fit well, though this is hard to see well because of the superposition of the points… Maybe also provide some panels with the data and models per height separately?

& Digits in R2…, Above canopy AT 26 m

Table S1 The 0 and 1% minimum WC values are probably just values due to some unexplained fluctuations in the sensors, so I would not report these extremes but rather the 5% quartile or something like that. The same goes for the ´maximum´.

Table S3 Explain what low and high mean

Table S7 Is showing PARmin really necessary?

**References**

Marks, R.A., B.D. Pike and D. Nicholas McLetchie, 2019. Water stress tolerance tracks environmental exposure and exhibits a fluctuating sexual dimorphism in a tropical liverwort. Oecologia. Available from https://www.ncbi.nlm.nih.gov/pubmed/31664577. DOI 10.1007/s00442-019-04538-2

---

## Author Response (AR2)

Response to referee comments and suggestions on bg-2018-521 by N. Löbs et al.: "Microclimatic conditions and water content fluctuations experienced by epiphytic bryophytes in an Amazonian rain forest"

**Dear Professor Bahn,**

we would like to thank you and the reviewers for the manuscript evaluation and the comments, which helped to improve our manuscript. We appreciate the opportunity to revise our manuscript to address the constructive comments and suggestions from the reviewers. Below we respond with a point-by-point explanation to the comments from you and each peer-reviewer with our responses in blue color following every comment. At the end of the comments we provide the manuscript and the supplement with all changes being marked. The figure and table numbers refer to the revised manuscript. Furthermore, some two supplemental files (*"Exemplary results for the percentiles 5 and 95", "Exemplary results for the percentiles 1 and 99"*) are attached. In this revised version, Rodrigo Alves has been added to the authors' list, as he has helped considerably with data analysis and improving the manuscript.

Sincerely,

Nina Löbs, on behalf of the co-authors.

**Comments of the editor:**

your manuscript has been re-assessed by two of the earlier reviewers, who both think that the manuscript does not live up to its potential concerning an assessment of the functioning of tropical epiphytic bryophytes in the Amazon. However, they also think that it should be in principle publishable because it provides new and useful microclimatic data with functional relevance. I am therefore inclined to accept our manuscript for publication in Biogeosciences, in case you manage to address the concerns the reviewers have raised in a satisfactory manner. Most importantly, reviewer #1 has had some serious doubts concerning the values you provide for water contents and thinks that there might have been an issue with your calculations. As good estimates of water contents are key to any functional interpretation it will be essential to follow up on this concern very thoroughly.

**Author comments:**

Thank you very much for the chance to improve and resubmit our manuscript once again! During the revision process, we carefully reinvestigated the calculations of the water content values once again, which is documented in the response letter and the supplementary material of the manuscript. We believe that we now identified the most appropriate method to determine the water content data, based on the electrical conductivity measurements and we carefully discuss the meaning and the limits of these results. We feel the manuscript has improved a lot and each suggestion and comment has been addressed. We hope that this version will be suitable for publication in Biogeosciences.

**Referee report #1**

Submitted on 07 Nov 2019, Referee Maaike Bader

Referee comment general:

The new version of your manuscript, now with the adjusted title "Microclimatic conditions and water content fluctuations experienced by epiphytic bryophytes in an Amazonian rain forest" has improved a lot, in particular in the more cautious discussion of the results. I now have only one major problem with the results, and that is in the very unlikely water content values.

I appreciate it that you tried out the method I suggested, and I still think that method makes sense, but the results are not very convincing as they are. It does not make any sense that the liverworts in the canopy should have a constantly high water content. Also, it does not make sense that the bryophytes in the understorey should never reach WCs above about 400% (Table S1, Fig S8 – even if this is for 30-min averages, during rain events the mosses could stay at their maximum capacity for half an hour), if their maximum WC is about 1500% for Leucobryum and 1000% for Sematophyllum (your data in Fig S3 in the previous version).

In any case, there must still be something wrong with the calculation. Perhaps it would help to not take the absolute min and max mV signal ever measured by the sensors, but something like the 5% and 95% Quartiles, to avoid using spurious signals. Supporting the assumption that this has happened is that it does not make sense that any of the mosses and liverworts should go down to values as low as 0 or 1% WC - this is what one would reach in a good drying oven, not at >70% RH... So the real minimum WC should be higher. Also, there may be a meaning in the fact that the maximum electrical conductivity measured in the field was much higher than those measured during calibration for Leucobryum (Lm) and Sematophyllum (Ss) (Fig S3 of previous version). The result of this is that the new function used estimates much lower WC values than those estimated by the calibration curves. I agree that the calibration curves (not presented in this version of the paper as they were not used for the new WC calculations) were problematic due to the huge variability (especially for Symbiezidium, not so much for Lm). Still, they might be used to constrain the range of relationships between electrical conductivity and water contents that can be considered acceptable, and for Lm and Ss the current functions use would fall outside this range, with systematic underestimations of the water content. If the calibration curves have a meaning, I would expect electrical conductivity in the field to be usually between near-0 and 600 (Lm) or 300 (Ss). By plotting the histogram of the conductivity values it may be possible to identify a more realistic value corresponding to maximum moss wetness.

For Symbiezidium I suspect that the water content is, on the other hand, systematically overestimated. It seems rather impossible to me that a canopy bryophyte would maintain a WC of around 100% all the time. It would be really good if the quantitative translation of sensor output to water content can be managed, because the data do show that there is a signal in the data, e.g. by the strong response of at least some of the sensors to rain events. I think there are some interesting patterns in the data, in

particular the diel fluctuation of the WC in the upper canopy, apparently following RH fluctuations. It does make sense that this diel fluctuation is largest at the upper two heights. So it is mainly the determination of the absolute WC that is problematic, not so much the fluctuations.

There may be a way to deal with this though. Apart from trying not using the absolute minimum and maximum ever measured but the 5% / 95% quartiles, you may also need to use the 30-minute averages, so as not to use the lower end of the short-term fluctuations as a minimum and upper end of those fluctuations as a maximum. Judging from Figure S4 those fluctuations may be a problem especially at 23 m, which could explain the high mean values there if you use the 5-min data for the calibration. I could imagine that after rain, they may also become more pronounced at e.g. 1.5 m. I might have to take back the recommendation to use 5-min values for the estimation of activity times, at least for WC.

A second point: I wonder how interesting it is, in the context of predicting activity of the mosses, to put emphasis on the seasonal patterns in the mean values of the climatic variables. What could be more interesting is to analyse the bryophyte activity patterns separately for the seasons.

**Author comment:**

Dear Maaike Bader, we would like to thank you for your critical and very constructive comments on our manuscript. We appreciate your feedback a lot!

Many thanks for your idea to calculate the quartiles in order to obtain more reliable water content values. For the correction/ limitation of the considered data range, we calculated three percentile options, i.e. 5 and 95%, 1 and 99%, and 0.1 and 99.9%, to eliminate potential outliers. The resulting data range for the different sensors and percentiles are listed in the *Table S3 (see below)*.

Table S3: Electrical conductivity data and the resulting range of water content data. Besides the original minimum and maximum values of electrical conductivity (Min\_total, Max\_total), the ranges after subtraction of 0.1, 1 and 5% of the data from the upper and lower end are shown (Min\_0.1, Min\_1, Min\_5, Max\_5, Max\_1, Max\_0.1). Calculations are based on the field measured electrical conductivity data at 5-minute intervals, given for the 24 sensors. The percentiles chosen: 0.1 and 99.9 are marked in red.

|          |                          |           | Percentiles of the electrical conductivity (EC) of the 5-min interval |         |       |       |       |       | erval       |           |
|----------|--------------------------|-----------|-----------------------------------------------------------------------|---------|-------|-------|-------|-------|-------------|-----------|
| SensorNr | Species                  | Division  | Min_total                                                             | Min_0.1 | Min_1 | Min_5 | Max_5 | Max_1 | Max_0.1     | Max_total |
|          |                          |           | [mV]                                                                  | [mV]    | [mV]  | [mV]  | [mV]  | [mV]  | [mV]        | [mV]      |
| 1        | Sematophyllum subsimplex | Moss      | 24                                                                    | 27      | 32    | 39    | 408   | 783   | 1223        | 1935      |
| 2        | Sematophyllum subsimplex | Moss      | 23                                                                    | 27      | 33    | 41    | 303   | 450   | 670         | 1392      |
| 3        | Sematophyllum subsimplex | Moss      | 35                                                                    | 36      | 38    | 40    | 372   | 759   | 1100        | 1615      |
| 4        | Leucobryum martianum     | Moss      | 35                                                                    | 38      | 39    | 41    | 72    | 174   | 3 91 | 1039      |
| 5        | Sematophyllum subsimplex | Moss      | 24                                                                    | 37      | 38    | 41    | 352   | 721   | 1076        | 1741      |
| 6        | Sematophyllum subsimplex | Moss      | 5                                                                     | 6       | 15    | 37    | 236   | 406   | 542         | 965       |
| 7        | Symbiezidium barbiflorum | Liverwort | 14                                                                    | 16      | 17    | 20    | 77    | 571   | 1004        | 1427      |
| 8        | Octoblepharum cocuiense  | Moss      | 14                                                                    | 15      | 16    | 19    | 55    | 66    | 155         | 662       |
| 9        | Octoblepharum cocuiense  | Moss      | 12                                                                    | 15      | 17    | 20    | 77    | 172   | 356         | 787       |
| 10       | Octoblepharum cocuiense  | Moss      | 14                                                                    | 16      | 18    | 21    | 103   | 189   | 411         | 654       |
| 11       | Symbiezidium barbiflorum | Liverwort | 32                                                                    | 35      | 37    | 38    | 86    | 264   | 578         | 1255      |
| 12       | Symbiezidium barbiflorum | Liverwort | 29                                                                    | 33      | 35    | 36    | 54    | 218   | 429         | 900       |
| 13       | Symbiezidium barbiflorum | Liverwort | 40                                                                    | 42      | 44    | 48    | 495   | 646   | 803         | 868       |
| 14       | Symbiezidium barbiflorum | Liverwort | 39                                                                    | 42      | 44    | 47    | 147   | 199   | 239         | 328       |
| 15       | Symbiezidium barbiflorum | Liverwort | 46                                                                    | 50      | 52    | 54    | 177   | 228   | 312         | 350       |
| 16       | Symbiezidium barbiflorum | Liverwort | 46                                                                    | 50      | 53    | 57    | 88    | 167   | 237         | 363       |
| 17       | Symbiezidium barbiflorum | Liverwort | 32                                                                    | 37      | 39    | 43    | 156   | 235   | 315         | 638       |
| 18       | Symbiezidium barbiflorum | Liverwort | 41                                                                    | 41      | 44    | 47    | 107   | 313   | 555         | 1890      |
| 19       | Symbiezidium barbiflorum | Liverwort | 43                                                                    | 50      | 54    | 60    | 141   | 190   | 244         | 595       |
| 20       | Symbiezidium barbiflorum | Liverwort |                                                                       |         |       |       |       |       |             |           |
| 21       | Symbiezidium barbiflorum | Liverwort | 31                                                                    | 39      | 44    | 48    | 152   | 285   | 543         | 959       |
| 22       | Symbiezidium barbiflorum | Liverwort | 47                                                                    | 52      | 56    | 61    | 139   | 206   | 485         | 859       |
| 23       | Symbiezidium barbiflorum | Liverwort | 65                                                                    | 74      | 79    | 84    | 117   | 136   | 220         | 571       |
| 24       | Symbiezidium barbiflorum | Liverwort | 69                                                                    | 83      | 89    | 94    | 123   | 198   | 297         | 546       |

Additionally, the results obtained by utilization of all these percentiles are listed in the supplementary files (*"Exemplary results for the percentiles 5 and 95", "Exemplary results for the percentiles 1 and 99",* Percentiles 0.1 and 99.9 are presented in the revised *manuscript"*). In these supplementary files the translation from 'measured electrical conductivity' via 'by percentiles cleaned electrical conductivity' to 'calculated water content' can be seen in the first figure each.

After careful comparison and evaluation we decided to consider 0.1 % and 99.9 % percentiles, because the 99.9 % percentile already shows a large difference to the total data range (Max\_total), which might be explained by outliers, while the steps to the next percentiles are not that big anymore and might present reasonable measurements. For an equal correction at both ends, we decided to also consider 0.1 % for the lower data range; however, there are not as extreme outliers as observed for the upper data range.

In addition, we also looked into the electrical conductivity data of the lab versus field data once again. As we see a large variation of EC-values for the different samples of one species and we couldn't use the exact samples of field measurements during lab calibrations, we believe that it is better to only use the maximum and minimum water content values from these lab measurements but not the electrical conductivity values or the form of the curve obtained in the lab. The minimum WC corresponds to the weight of the sample after drying at 40°C and 30 % relative humidity in order to use a realistic data range. As you can see in the table above, this minimum water content is ranging between 13 and 16% for the different species.

The water contents reached by the sample replicates during measurements in the laboratory are shown in an additional table (*Table S2, see below*). As you can see here, *Symbiezidium* consistently reached high water content values for all replicate samples.

Thus, we believe that the high water contents reached by the samples at the upper canopy levels and the relatively quick drying of the samples close to ground levels is not an artefact but a real feature, which is caused by the exact habitat occupied by these samples in the field. At 1.5 m height, *Sematophyllum subsimplex* and *Leucobryum martianum* grow on the vertical stem. During rain events they get wet, but after that they dry rather quickly again as the water effectively drains from the samples. At 18 and 23 m height, *Symbiezidium barbiflorum* grows on an inclined branch and on the upper side of a branch (*see Fig. S4*). Here, the samples keep increased water contents over longer time spans. This is a general pattern, which could already be observed in the original electrical conductivity values (see Fig. S7 in the revised *manuscript*). We explain this also in the manuscript on page 17 line22ff. and Page 21 line 17ff.

We also happily took up your suggestion to analyze the physiological activity during the different seasons. The results of these calculations are shown in *Fig. 4* of the main manuscript *(see below)*, were we plotted the results for the wet and the dry season in a separate manner.

**Author changes in the text:**

P17 L22: "The high WC of the bryophyte samples in the canopy can be explained by the higher water holding capacity of the liverwort *Symbiezidium*, which dominated in the canopy, and by its growth on inclined or vertical stems, where water drainage is less effective as compared to the vertical stem at the lower two levels. The relevance of the water holding capacity for the water content of different bryophyte species has already been described in several other studies (Lakatos et al., 2006; Romero et al., 2006; Williams and Flanagan, 1996)."

P21 L17:" In the canopy, the dominating liverworts responded to the nightly increase of RH, which was not observed for the mosses in the understory. Thus, the relevant water source for bryophytes in the understory might be rain, while for the bryophytes in the canopy the nightly increase of the RH might be relevant for an activation of the physiological processes."

---

## Author Response (AR3)

Response to referee comments and suggestions on bg-2018-521 by N. Löbs et al.: "Microclimatic conditions and water content fluctuations experienced by epiphytic bryophytes in an Amazonian rain forest"

**Dear Professor Bahn,**

we would like to thank you and the reviewers for the manuscript evaluation and the comments, which helped once again to improve our manuscript. We appreciate the opportunity to revise our manuscript one last time to address the constructive comments and suggestions from the reviewers. We once again worked intensively with the data, and believe that we now could solve all the remaining issues. Below we respond with a point-by-point explanation to the comments from the peer-reviewer with our responses in blue color following every comment. At the end of the comments we provide the manuscript and the supplement with all changes being marked.

Sincerely,

Nina Löbs, on behalf of the co-authors.

**Comments on the text:**

Black text shows the original referee comment, and blue text shows the response of the authors and the explicit changes in the revised text. The figure and table numbers refer to the revised manuscript with the marked changes.

**Referee report #2**

Maaike Bader, 06 April 2020

**Dear authors,**

In the new revised version a lot of the issues in the previous one have been addressed. I still think that the water-content data for the bryophytes are a unique and interesting data set. My main concern remains however, but I hope it can still be solved:

The impossibly high water contents of the air-dry liverworts in the canopy are still presented, and although the strangeness of the values is discussed (not convincingly), the values are not discarded or further corrected. I really think that data about the course of the water content of tropical bryophytes is very valuable and needs to be published, but these values must clearly be wrong, so they should NOT be presented as they are. Figure S5 gives a hint about what the cause of the strange values may be: Due to the unexplained short-term fluctuations, the water content, and thus apparently the electrical conductivity (EC) values, for the samples at 23m vary a lot around the half-hour mean. If we assume that the half-hour mean is the most realistic real EC value and that the short-term variations are noise, it is not a good idea to take the 5-minutes-based minimum and maximum (even not using a 1 or 5%, 95% or 99%

or whatever percentile) values for the calibration, because that way the noise is going to determine the calculated water contents. I hope I made the problem clear.

Dear Maaike Bader, following your suggestion, we carefully checked the data of all sensors again and decided not to use the data of some sensors and to exclude the data of the 18 m level from the analysis, as the larger fluctuations there could not be completely explained by us. We now included a new Figure S5, which shows the information used for data evaluation and calibration, i.e., the rainfall data, the raw EC values, and the calculated WC data. With this information, one could confirm the sensor's response to rain, and the overall response of each sample. In order to minimize the noise, only the 30-min averages were used for all new calculations and figures, as stated in the text.

For sensor #23 at the 23 m level we present a close-up view, which illustrates that the remaining "scatter" is caused by a pattern of daily fluctuations of the water content.

Another possible problem is that the very high WCmax of Symbiezidium that you determined in your calibration measurements (which does NOT mean that they would have a WC of 400% when air dry!) does cause a high calculated WC if a linear relationship between EC and WC is assumed, which is not realistic for the very high WCs. So here, you might need to start the linear part of the calibration curve at a lower WC.

We agree with your comment, and we altered the calibration accordingly. We now assume that all samples get completely dry at least once during the measurement period and this minimum electrical conductivity now corresponds to a water content of 0%. The material and methods section has been adapted accordingly and the new values are presented in Table S1.

I do not know if it will be possible to get a value for the water content that is reliable in absolute terms. Perhaps your best bet is to not even try and to only report on the fluctuations, i.e. wetting after rain (which I consider a clear and undisputed signal) and daily fluctuations at 23 m (which in my mind may reflect the equilibration with fluctuating air humidity, already indicating that it cannot be true that the WC is around 400, because at that value they would be wetter than in equilibrium with moist air. Information about such fluctuations is already interesting as a first bit of data, as it shows e.g. how long mosses stay wet after a rain. Because of the lack of replication and different positions of the bryophytes on the trunk it is indeed hard to attribute these patterns to the height on the tree, but even without that, it gives an indication of the patterns that are possible in rain forest epiphytes. In summary: either correct the values based on a new calibration (which should of course still be consistent and rational and not just a fine-tuning to get the desired result), or present the WC fluctuations without quantifying the absolute values.

As described above, the calibration was recalculated using only the 30-min averaged data and an adapted calibration method. In the figures 1, 2 and S6 the general WC patterns are shown for the different heights and seasons.

You can really emphasize harder that the type of data you present is unique. Microclimate has been measured in vertical profiles in rain forests before, but data on the temperature and water content fluctuations inside mosses, which are very relevant for estimating or modelling their productivity, have

never been measured for so long or along a height profile, as far as I know. In the last paragraph of the introduction, make clearer why these data are exciting.

The last paragraph of the introduction was modified accordingly, as follows: (P3, L30-P4 L4 in document with marked changes) "In the current study, long-term continuous measurements of temperature, light and water content inside bryophyte communities have been conducted along a vertical gradient. To our knowledge, our study is the first one measuring microclimatic parameters and the water status inside bryophyte communities in a rainforest environment. With these data on the microclimate along a vertical profile and during different seasons, we believe to provide a unique dataset, combined with an estimation of the activity patterns of bryophyte communities in a tropical rainforest. "

In the discussion (par 4.1) make clearer what the previous studies did and did not measure.

The text was modified according to the suggestion, as follows: (P14, L17-24) "In the current study we measured the microclimatic conditions experienced by epiphytic bryophyte communities along a vertical gradient over the course of more than two years. In previous studies, microclimatic data on the light, temperature, and air humidity have been assessed at different height levels within the forest (Chazdon and Fetcher, 1984; Lösch et al., 1994; Romero et al., 2006), but long-term measurements of the water content and the light and temperature on top and inside the cryptogamic communities and have been missing up to now."

Another major point is that the presentation of the estimated potential physiological activity (Par 3.2 and 4.2) needs some reformulation in the sense that the compensation points should be consequently presented as rough proposals, not as values that you know to be valid for your environment or species. So try to use formulations like "If we assume a WCP of X (REF), the duration of activity would be Y". Adjust in the caption of Fig 4 too. Also, 4.2. is quite long considering how little data you have to really be able to discuss this.

The sections 3.2 and 4.2 were reformulated, following the suggestion. The caption of the Fig 4 was also changed accordingly.

Some smaller remarks:

P2 L24-27 These introductory sentences are not really necessary.

Thanks for the suggestion, the introductory sentences were removed from the text (P2, L24-27).

P3 L15 a paragraph break is needed here. L15-17 are not necessary.

The alterations were made accordingly.

P5 L23 Expressing the oscillations in terms of WC would assume that the WC is correctly determined. Better express it as EC, and take it into consideration when calculating WC (see above)

**The alteration was done accordingly (P5, L33).**

**P6 L10 what is this conversion factor base on?**

Measurements of electrical conductivity also show a temperature-dependency, which needs to be considered during data analysis. For this, an exponential model type has been established by Sheets & Hendrickx (1995), as presented in Corwin & Lesch (2005). Ma et al. (2011) evaluated different models for electrical conductivity measurements and determined one as being best suited for the temperature correction of electrical conductivity measurements. This correction factor also proved to work fine on our data. This is explained in detail in Weber et al. (2016).

**P6 L15-25 This section does not yet describe the calibration. This is described on P7 L1-14**

This is correct. We now write "To determine the maximum water content of the different bryophyte communities, samples of them were collected in the forest area sur-rounding the ATTO site." (P6, L26-27).

**P7 L15 Explain the origin of the oscillations and take them into account for determining the WC**

This was done accordingly (P7, L28-32): "The measured electrical conductivity values showed short-time oscillations, which might be caused by the fact that the bryophytes cushions have some air spaces inbetween, as we observed that these oscillations are less pronounced in denser substrate. Nevertheless, the overall functionality of the sensors is still ensured also in less dense material, and the short-term fluctuations could be removed with a 30-minute smoothing algorithm (Fig. S5). Thus, for all calculations the 30-minute averages were used.

P7 L18-19 Huge error bars suggest that this averaging may not be justified. Can you argue that it is? How are different species at one height a uniform group?

The dataset was double checked, and sensors with large fluctuations that could not be justified by the meteorological parameters were excluded from the calculations. The Figure S5 presents the remaining sensors with their raw EC values. After these recalculations the error bars are much smaller now.

P13 L13-16 You could use these results to your advantage by pointing out how they show the larger smallscale variation in environmental conditions du to shading and tree topography, which may cause conditions to be more variable within centimetres than the 'microclimate' along the larger height gradient along the tree.

The suggestion was added to the discussion section: (P14, L30 – P15, L2) "Within one height level, the small-scale environmental conditions, such as radiation and shading, water conditions, and wind velocity vary, depending on the specific habitat conditions, as e.g. exposition, tree foliage and inclination of the substrate (Barkman, 1958; Campos et al., 2019; Cornelissen and ter Steege, 1989; de Oliveira and de

Oliveira, 2016; Sierra et al., 2018). These small-scale patterns also explain the variability within one height level. "

P15 L12 After reinstallation, without recalibration, these values are no longer usable, I would just not present them.

This was done accordingly and the data are not presented anymore.

**Literature citations:**

Barkman, J. J.: Phytosociology and ecology of cryptogamic epiphytes., 1958.

Campos, L. V., Mota de Oliveira, S., Benavides, J. C., Uribe-M, J. and ter Steege, H.: Vertical distribution and diversity of epiphytic bryophytes in the Colombian Amazon, J. Bryol., 41(4), 328–340, doi:10.1080/03736687.2019.1641898, 2019.

Chazdon, R. L. and Fetcher, N.: Light Environments of Tropical Forests, in Physiological ecology of plants of the wet tropics: Proceedings of an International Symposium Held in Oxatepec and Los Tuxtlas, Mexico, June 29 to July 6, 1983, edited by E. Medina, H. A. Mooney, and C. Vázquez-Yánes, pp. 27–36, Springer Netherlands, Dordrecht., 1984.

Cornelissen, J. H. C. and ter Steege, H.: Distribution and ecology of epiphytic bryophytes and lichens in dry evergreen forest of guyana, J. Trop. Ecol., 5(2), 131–150, doi:10.1017/S0266467400003400, 1989.

Corwin,D.L.& Lesch, S.M. (2005) Apparent soil electrical conductivity measurements in agriculture. Computers and Electronics in Agriculture, 46, 11–43.

Lösch, R., Mülders, P., Fischer, E. and Frahm, J. P.: Scientific Results of the BRYOTROP Expedition to Zaire and 3 . Photosynthetic gas exchange of bryophytes from different forest types in eastern Central Africa ., Trop. Bryol., 9, 169–185, 1994.

Ma, R., McBratney, A., Whelan, B., Minasny, B. & Short, M. (2011) Comparative temperature correction models for soil electrical conductivity measurement. Precision Agriculture, 12, 55–66.

de Oliveira, H. C. and de Oliveira, S. M.: Vertical distribution of epiphytic bryophytes in Atlantic forest fragments in Northeastern Brazil, Acta Bot. Brasilica, 30(4), 609–617, doi:10.1590/0102-33062016abb0303, 2016.

Romero, C., Putz, F. E. and Kitajima, K.: Ecophysiology in relation to exposure of pendant epiphytic bryophytes in the canopy of a tropical montane oak forest, Biotropica, doi:10.1111/j.1744-7429.2006.00099.x, 2006.

Sheets, K.R. & Hendrickx, J.M.H. (1995) Non-invasive soil water content measurement using electromagnetic induction. Water Resource Research, 31, 2401–2409.

Sierra, A. M., Vanderpoorten, A., Gradstein, S. R., Pereira, M. R., Bastos, C. J. P. and Zartman, C. E.: Bryophytes of Jaú National Park (Amazonas, Brazil): Estimating species detectability and richness in a lowland Amazonian megareserve, Bryologist, 121(4), 571–588, doi:10.1639/0007-2745-121.4.571, 2018.

**Microclimatic conditions and water content fluctuations experienced by epiphytic bryophytes in an Amazonian rain forest**

Nina Löbs1\*, David Walter1,2, Cybelli G. G. Barbosa1, Sebastian Brill1, Rodrigo P. Alves1, Gabriela R. Cerqueira3,
Marta de Oliveira Sá3, Alessandro C. de Araújo4, Leonardo R. de Oliveira3, Florian Ditas1, Daniel Moran-Zuloaga1,
Ana Paula Pires Florentino1, Stefan Wolff1, Ricardo H. M. Godoi5, Jürgen Kesselmeier1, Sylvia Mota de Oliveira6,
Meinrat O. Andreae1,7, Christopher Pöhlker1, Bettina Weber1,8\*

[revised manuscript text omitted]

- 5 jeuneaceae) as mat (Batista and Santos, 2016; Valente et al., 2017). Those forms determine the storage capacity and water loss rate, whereas prevails the capillarity conduction or retention (Proctor, 1990). As a result, lower values of WC can be associated with turf life form due to their shoots with larger crowded leaves, dense foliage and the frequently occurring weft rhizoid, showing particularly high values for capillary water conduction, i.e., lower values of water retention. The opposite occurs with wefts and mats, where the capillarity retention of water
- 10 is higher (Mägdefrau, 1982). Thereby, mats developed an increased drought tolerance, being more adapted to dry conditions as well as to extreme changes, whereas wefts are life forms characteristic of humidity areas (Gimingham and Birse, 1957).

[revised manuscript text omitted]

(b)

| Parameter     | 2015        | 2016        |  |  |
|---------------|-------------|-------------|--|--|
|               | Sum         | Sum         |  |  |
| Rain (days)   | (199) 202   | (197) 215   |  |  |
| ( mm ) | (1680) 1693 | (1702) 1863 |  |  |
| Fog (days)    | 21*         | 28*         |  |  |

\*: Gaps in the data record due to malfunction of fog sensor during time window of 31.05. - 20.10.2015, 30.04. -

06.07.2016, and 01.09. - 31.12.2016.

**Table 2** Seasonal mean values and standard deviations ( $\pm$  SD) of the mean photosynthetically active radiation (PARavg), the daily maximum of photosynthetically active radiation (PARmax), the temperature, and the abovecanopy relative humidity (RH) or water content (WC) of bryophytes determined at different height levels and above the canopy. Mean values for the respective seasons were calculated from 305-minute intervals of the years 2015 and from October 2014 to November 2016, except for PARmax, where the daily maximum values were considered. Values for PARmax can be found in Table S7.

RH (above-canopy) [%], PARavg daytime Temperature Height  $[\mu mol m^{-2} s^{-1}]$ [°C] WC [%] [m] Mean  $\pm$  SD Mean ± SD Mean  $\pm$  SD Wet season 2 738 46 25.7 0.7 94 above-canopy 25.3 3 27 17 23 m 0.6 41 8m 41 24 24.9 0.4 93 21 24.9 1.5 3 1 0.4 83 26 **Transitional season Wet-Dry** 2 53 25.6 0.5 91 above-canopy 860 38 29 25.7 0.7 49 23 m 4 8 m 63 14 24.9 0.4 72 27 1.5 m 2 1 24.6 0.2 31 6 **Dry season** 950 93 27.2 84 above-canopy 1.0 6 21 27.8 23 m 54 1.2 45 10 24 17 26.6 0.9 58 20 8 m 1.5 m 5 4 26.0 0.8 30 31 **Transitional season Dry-Wet** 8 26.5 87 above-canopy 784 111 1.6 23 m 52 34 27.1 2.2 37 2 23 5 26.2 1.7 58 13 8 m 25.9 53 1.5 m 4 1 1.4 52

| Height         | <del>PAR</del>
[μn | <del>avg daytime
10l m-2 s-1]</del> | PAR max
[µmol-m -2 -s -1 ] |                | <del>Temperature</del>
[° <del>C]</del> |                | RH (above-canopy) [%],
₩C [%] |               |
|----------------|-----------------------|----------------------------------------------------------------------|----------------------------------------------------------------|----------------|--------------------------------------------|----------------|----------------------------------|---------------|
| <del>[m]</del> | Mean                  | ± SD                                                                 | Mean                                                           | ± SD           | Mean                                       | ŧ              | Mean                             | ± SD          |
| Wet season-    |                       |                                                                      |                                                                |                |                                            |                |                                  |               |
| above-canopy   | 738                   | <del>566</del>                                                       | <del>2086</del>                                                | <del>515</del> | <del>25.6</del>                            | 2.5            | <del>95</del>                    | 9             |
| 23 Liverwort   | <del>30</del>         | 3                                                                    | <del>248</del>                                                 | <del>194</del> | <del>25.3</del>                            | 2.0            | <del>283</del>                   | <del>83</del> |
| 18 Liverwort   | <del>39</del>         | <del>12</del>                                                        | 282                                                            | <del>175</del> | 25.2                                       | <del>1.9</del> | <del>197</del>                   | <del>66</del> |
| 8 Liverwort    | 31                    | <del>26</del>                                                        | <del>144</del>                                                 |                | <del>24.9</del>                            | 1.1            | <del>66</del>                    | <del>22</del> |
| 8 Moss         |                       |                                                                      |                                                                |                |                                            |                | <del>182</del>                   | <del>63</del> |
| 1.5 Moss       | 4                     | <del>15</del>                                                        | 114                                                            | <del>224</del> | <del>24.9</del>                            | 1.0            | <del>121</del>                   | <del>91</del> |

Manuscript with marked changes

| Transitional season | Transitional season Wet-Dry   |                                |                                |                               |  |  |  |  |  |
|----------------------------|-------------------------------|--------------------------------|--------------------------------|-------------------------------|--|--|--|--|--|
| above canopy               | <del>861</del> 649            | <del>2227</del> <del>182</del> | <del>25.8</del> <del>3.0</del> | <del>91</del> <del>11</del>   |  |  |  |  |  |
| 23 Liverwort               | 41 72                         | 414 252                        | <del>25.7</del> <del>2.8</del> | <del>308</del> <del>109</del> |  |  |  |  |  |
| 18 Liverwort               | 44 <del>5</del> 4             | <del>351</del> <del>123</del>  | <del>25.4</del> <del>2.3</del> | <del>200</del> <del>34</del>  |  |  |  |  |  |
| 8 Liverwort                | <del>66</del> <del>88</del>   | <del>165</del> <del>218</del>  | <del>24.9</del> <del>1.4</del> | <del>53</del> <del>10</del>   |  |  |  |  |  |
| 8 Moss                     |                               |                                |                                | <del>161</del> <del>56</del>  |  |  |  |  |  |
| 1.5 Moss                   | $\frac{2}{12}$                | <del>61</del> <del>102</del>   | 24.6 1.1                       | <del>55</del> <del>28</del>   |  |  |  |  |  |
| Dry season                 |                               |                                |                                |                               |  |  |  |  |  |
| above-canopy               | <del>973</del> 647            | <del>2100</del> 609            | <del>26.7</del> <del>3.4</del> | <del>87</del> <del>1</del> 4  |  |  |  |  |  |
| 23 Liverwort               | <del>55</del> 9               | <del>503</del> <del>231</del>  | <del>27.2</del> <del>3.5</del> | <del>273</del> <del>125</del> |  |  |  |  |  |
| 18 Liverwort               | 41 13                         | 4 12 190         | <del>26.5</del> <del>2.9</del> | <del>188</del> <del>89</del>  |  |  |  |  |  |
| 8 Liverwort                | <del>23</del> <del>16</del>   | <del>295</del> <del>268</del>  | 26.0 2.1                       | <del>63</del> 45              |  |  |  |  |  |
| 8 Moss                     |                               |                                |                                | <del>166</del> <del>70</del>  |  |  |  |  |  |
| 1.5 Moss                   | <del>6</del> <del>25</del>    | <del>209</del> <del>299</del>  | <del>25.5</del> <del>1.7</del> | <del>53</del> <del>37</del>   |  |  |  |  |  |
| Transitional season | - <del>Dry-Wet</del>          |                                |                                |                               |  |  |  |  |  |
| above-canopy               | <del>785</del> <del>617</del> | <del>1988</del> <del>509</del> | $\frac{26.5}{3.3}$             | <del>85</del> <del>15</del>   |  |  |  |  |  |
| 23 Liverwort               | <del>55</del> <del>91</del>   | <del>530</del> <del>297</del>  | <del>27.2</del> <del>3.7</del> | <del>289</del> <del>113</del> |  |  |  |  |  |
| 18 Liverwort               | <del>37</del> <del>28</del>   | <del>185</del> <del>109</del>  | <del>26.6</del> <del>3.0</del> | <del>227</del> <del>121</del> |  |  |  |  |  |
| 8 Liverwort                | <del>21</del> <del>47</del>   | <del>269</del> <del>178</del>  | $\frac{26.3}{2.5}$             | <del>112</del> <del>84</del>  |  |  |  |  |  |
| 8 Moss                     |                               |                                |                                | <del>180</del> <del>67</del>  |  |  |  |  |  |
| 1.5 Moss                   | 4 20                          | <del>107</del> <del>113</del>  | $\frac{26.0}{2.1}$             | <del>74</del> <del>60</del>   |  |  |  |  |  |

**Table 3:** The potential time fractions [%], during which the epiphytic bryophytes at the different height levels exceeded the lower compensation points of light (LCP1), the upper compensation points for temperature (TCP), the lower compensation points for water (WCP), and reached the optimal temperature for net photosynthesis ( $T_{opt}$ ). The results are shown separately for a) the wet season (February-May) and b) the dry season (August-November).

5 Values are given for the different height levels (1.5, 8, <del>18,</del> 23 m) and bryophyte divisions (M=moss, L=liverwort). For the net photosynthesis (NP) it is required that WC > WCP,  $PAR > LCP_1$  and T > TCP, for the dark respiration (DR) it is necessary that WC > WCP and PAR < LCP1 or WC > WCP and T > TCP. Five Thirty30-minute averages of measurements during the entire measurement period from October 2014 to November December 2016 were considered. The ranges of the compensation points (CP) and the optimum temperature (opt) were reported in Lösch (1994) and Wagner et al. (2013) (see Table S4).

**10**

**a) Wet season**

| Height | Division  | LCP 1                   | Topt | TCP                | WCP            | NP     | DR           |
|---------------|------------------|-------------------------------------------|------------------------|--------------------|----------------|---------------|--------------|
|               |                  | ≥ 3-12                             | 24.0-27.0       | ≥ 30.0-36.0 | ≥ 30-80 |               |              |
|               |                  | $\mu$ mol m -2 s -1 | ° C             | ° C                | % DW    |               |              |
| [m]    | L/M       | Time fra                                  | action when            | cardinal points    | s are reache   | d/exceeded [% | of time]     |
| 23     | L                | 33-43                              | 4-54            | 0-3         | 3-80    | 1-30   | 2-52  |
| 8      | M & L | 24-31                              | 2-74            | 0           | 42-94          | 14-35  | 29-59 |
| 1.5    | M                | 2-19                               | 2-77            | 0           | 32-80   | 1-13   | 32-67 |

**b) Dry season**

| Height | Division  | LCP 1                   | Topt | TCP         | WCP     | NP     | DR          |
|---------------|------------------|-------------------------------------------|------------------------|--------------------|----------------|---------------|-------------|
|               |                  | ≥ 3-12                             | 24.0-27.0       | ≥ 30.0-36.0 | ≥ 30-80 |               |             |
|               |                  | $\mu$ mol m -2 s -1 | ° C             | ° C         | % DW    |               |             |
| [ m ]  | L/M       | Time fra                                  | action when            | cardinal points    | s are reache   | d/exceeded [% | of time]    |
| 23     | L                | 40-46                              | 6-35            | 0-27               | 6-64    | 1-24   | 4-45 |
| 8      | M & L | 18-35                              | 8-51            | 0-11        | 5-84    | 2-34   | 7-51 |
| 1.5           | M                | 3-16                                      | 9-59            | 0-4                | 2-21           | 0-5           | 10-26       |

**15 a) Wet season**

| Height         | Division | LCP 1                      | <del>Topt</del> | TCP                        | WCP                         | NP                  | ÐR               |
|----------------|-----------------|---------------------------------------|----------------------------|----------------------------|-----------------------------|---------------------|------------------|
|                |                 | ≥3-12                          | <del>24.0-27.0</del>       | ≥ 30.0-36.0         | ≥ 30-80              |                     |                  |
|                |                 | µmol m -2 -s -1 | °- C                | ° C                        | <del>% DW</del>             |                     |                  |
| <del>[m]</del> | <del>L/M</del>  | Ŧi                                    | <del>me fraction w</del> l | <del>ten cardinal po</del> | <del>ints are reach</del> o | ed/exceeded [%-of t | time]            |
| 23             | Ł               | <del>34-43</del>                      | 4 5 4               | 0-3                        | <del>98-100</del>           | <del>27-38</del>    | <del>62-66</del> |
| <del>18</del>  | Ł               | 4 <del>0</del> -4 <del>6</del>        | 4 <del>.55</del>           | 0-2                        | <del>96-100</del>           | <del>32-43</del>    | <del>56-57</del> |
| 8              | Ł               | <del>25-31</del>                      | <del>2-74</del>            | θ                          | <del>18-98</del>            | <del>5-40</del>     | <del>11 56</del> |
| 8              | M               |                                       |                            |                            | <del>88-100</del>           | <del>26-36</del>    | <del>54-63</del> |
| 1.5            | M               | <del>2-19</del>                       | 2 77                       | θ                          | <del>53 95</del>            | 1-15                | <del>53-79</del> |

| b) Dry season  |                 |                                       |                              |                            |                   |                     |                  |  |  |
|----------------|-----------------|---------------------------------------|------------------------------|----------------------------|-------------------|---------------------|------------------|--|--|
| Height         | Division | LCP                                   | T opt             | TCP                        | WCP               | NP                  | DR               |  |  |
|                |                 | ≥ 3-12                         | 24.0-27.0                    | ≥ 30.0-36.0         | ≥ 30-80    |                     |                  |  |  |
|                |                 | µmol m -2 -s -1 | ◦ C                   | °-C                        | <del>% DW</del>   |                     |                  |  |  |
| <del>[m]</del> | L/M             | Ŧi                                    | i <del>me fraction w</del> l | <del>hen cardinal po</del> | ints are reach    | ed/exceeded [% of i | time]            |  |  |
| 23             | Ł               | 4 <del>2</del> 47                     | <del>6-34</del>              | <del>3-26</del>            | <del>96-100</del> | <del>18-41</del>    | <del>57-59</del> |  |  |
| <del>18</del>  | Ł               | <del>38-46</del>                      | <del>5-40</del>              | 0-23                       | <del>93-100</del> | <del>21-43</del>    | <del>51-54</del> |  |  |
| 8              | F               | <del>19-36</del>                      | <del>8-52</del>              | 0-10                       | <del>5-86</del>   | <del>3-34</del>     | <del>15-55</del> |  |  |
| 8              | M               |                                       |                              |                            | <del>84-98</del>  | <del>14-39</del>    | <del>52-56</del> |  |  |
| 1.5            | M               | 4-16                                  | <del>9-60</del>              | 0-3                        | 4 54              | 0-10                | <del>16-52</del> |  |  |

**Figures**

---

## Author Response (AR4)

**Response to referee comments and suggestions on bg-2018-521 by N. Löbs et al.: "Microclimatic conditions and water content fluctuations experienced by epiphytic bryophytes in an Amazonian rain forest"**

Dear Professor Bahn,

we would like to thank you and the reviewer for the manuscript evaluation and the comments, which helped to improve it once again. We indeed highly appreciate the efforts made by Maaike Bader and have already included her in the acknowledgement section. Below we respond with a point-by-point explanation to the comments from the peer-reviewer with our responses in blue color following every comment. At the end of the comments we provide the manuscript and the supplement with all changes being marked.

Sincerely,

Nina Löbs, on behalf of the co-authors.
* * *
**Comments on the text:**
Black text shows the original referee comment, and blue text shows the response of the authors and the explicit changes in the revised text. The figure and table numbers refer to the revised manuscript.
* * *
**Referee report #1**

**Maaike Bader, 04 August 2020**

Dear authors,

Thank you for once again reviewing your manuscript and for seriously and effectively addressing my concerns. I think that you have now solved the main problem in the previous versions, the plausibility of the water content values. I think that the calibration method and the orders of magnitude of the results now make sense, and I am really pleased about this. Of course, it still does not imply that they necessarily present the correct absolute values, but together with the other microclimatic data they do show nicely how the dynamics of the water contents in these bryophytes may respond to environmental conditions. I think this is an important contribution of this paper.

Now that the main issue is solved, I think the manuscript just needs one more round of polishing, highlighting and discussing the exciting findings more, correcting some mistakes, optimising some graphs and formulations and checking for style and grammar.

The WC values now look more plausible to me and in the examples shown (Figs 2 and S6) they show a wonderful reaction to rain events. Could you corroborate this impression by a correlation between the WC and the rain amount in the preceding 12 h (or something like that)?

Dear Maaike Bader, thank you very much for your positive review! We are glad that you also consider the main issue to be solved now. As suggested by you, we plotted a correlation graph between the WC and the rain amount of the preceding 24 h, and a linear regression line shows that over 30% of the variability of the mean water content could be explained by the precipitation within the preceding 24 hours. These results are now shown in Fig. S8. The figure is referred to in the manuscript (P 12, L31-32: "The WC of the bryophytes reacted quite reliably upon rain that had fallen in the preceding hours (Fig. S5; Fig. S8) with some differences between the different height levels. …").

It is interesting that the WC of the samples as 23 m sometimes do (especially on 10-09-2016) and sometimes do not at all (March 2015) react to rain. Apart from possible umbrella effects of the canopy, could it also be due to a reinstallation and recalibration of the sensors around June 2015? There seems to be a large and systematic difference between the EC (and WC) values before and after this maintenance period for the sensors at 1.5 m (Fig S5). I guess this may be partly due to the drier season after this period, but I am not convinced that this explains the pattern completely. It may be worth elaborating a bit on this point.

As mentioned by you, some of the EC data indeed look different after as compared to before the maintenance period around June 2015. This definitely is partly caused by the different climatic conditions and the resulting growth of new foliage. But we indeed cannot completely exclude that also the repositioning of the sensors had some effect. We now explain that in the discussion in the section on the sensors at 23 m height (P16, L12-14: "Apart from the effect of growing foliage, one has to keep in mind that there was a re-installation of the sensors around July 2015, which could cause differences in the sensor readings before and after this event.") and in the section on the understory sensors (P16, L22-23: "Also here, a potential effect of the sensor re-installation around July 2015 has to be kept in mind.").

The diel fluctuations (mentioned P17, L1) in WC are not mentioned clearly enough in the results. They can be seen a little bit in Fig S8 and better in S5c, but the former figure is referred to only in the context of condensation and the latter not at all in results or discussion. I think these daily fluctuations deserve more attention (especially as you use it as a main conclusion), so I would discuss them with reference to figure S5c, or add another figure similar to Fig S8 but showing more days (only on days without much rain so that the scale of the WC can be precise enough to show the fluctuations - these are hard to see in Fig 2 and S6).

Following your suggestion, information on the diel fluctuations in the results section was made clearer and a reference to figure S5c was added. In addition, we also created a new Fig. S11, which shows these fluctuations during the dry and the wet season. Indeed, we observed that the fluctuations occur during all seasons, but are more regular and thus obvious during the dry season, due to the regular climate with only rare precipitation events. This information is now presented in the results section (P13, L2-5: "Overall, the bryophytes at 8 m and 23 m showed a regular and pronounced daily fluctuation of the WC, which occurred during all seasons, but was particularly regular during the dry season, due to the rare interfering rain events (Fig. 2, Fig. S5c, Fig. S11).").

In the discussion section, we also added the following text and a link to figure S5c: (P16 L33 - P17L3: "The water content data at 8 m and 23 m height showed diel fluctuations, which were particularly regular during the dry season, due to rare interfering rain events (Fig. S5c). They showed a parallel behavior to the RH data with the highest values reached during the morning hours (Fig. S11). It is well known from the

literature that moist bryophytes and many cryptogams could utilize high air humidity as a source of water (Lange et al., 2001; Raggio et al., 2017) and this likely also occurs here.")

We also included this information in the conclusions (P19L15-17: "The bryophytes at 8 m and 23 m height showed regular daily fluctuations of the WC contents, which went in parallel to RH and reached highest values during the morning hours.").

I like Figure S9, but it does not show whether reaching the dew point is related to the fluctuations in the WC. An interesting analysis would be to calculate a correlation between the distance below the dew point reached and the response in WC on that morning. The same could be done for the fog events. The difference in daily fluctuations between the seasons (mentioned in the conclusions) is also interesting and worth some more discussion (P15 L2-3: is this phenomenon more frequent in the dry season?). In fact, I think this point could perhaps make for an additional publication by itself!

As suggested by you, we looked at the events when the dew point was reached in some more detail. We plotted a characteristic sequence during the wet and the dry season (Fig. S11), and here we saw, that the temperature in the bryophytes dropped below the dewpoint of the ambient air mostly during the morning hours. This happened more frequently during the wet as compared to the dry season, which might be caused by a shading effect of the foliage, leading to a lower bryophyte temperature and causing a drop below dewpoint temperature. We also show exemplary dewpoint events at 1.5 and 23 m, where negative dewpoint spread values seem to cause increased water content values (Fig. 12). To analyze this potential effect, we extracted the minimum dewfall spread value per day, and for all negative daily values we calculated the slope of the water content data of the last 4 hours prior the negative dewpoint spread. Relating these parameters with each other, we obtained small negative Pearson's R values for the three WC sensors at 23 m height (i.e., $R_{WC21}$: -0.071; $R_{WC23}$: -0.076; $R_{WC24}$: -0.040), suggesting that bryophyte temperatures below the dewpoint temperature of the ambient air cause an increase in water content and thus a condensation of water. For fog, such a relationship could not be observed.

We adapted the text accordingly (P13L5-17: "A potential condensation, when the temperature of the bryophytes drops below the dewpoint of the ambient air, was mostly reached during the morning hours (Fig. S9-S12). This occurred during ~50 % of the wet season and ~30% of the dry season days at the surface of bryophytes at 23 m height (Fig. S10, Fig. S11). Contrastingly, at 1.5 m height dew point temperatures were only surpassed during ~9 % of the days, independently of season. Plots of exemplary dewpoint events at 1.5 m and 23 m height suggest that negative dewpoint spread values (i.e., bryophytes temperature below dewpoint of ambient air) cause increased water content values (Fig. 12). To analyze this potential effect, we extracted the largest dewfall spread value per day, and for all negative daily values we calculated the slope of the water content data of the last 4 hours prior the negative dewpoint spread. Relating these parameters with each other, we obtained small negative Pearson's R values for the three WC sensors at 23 m height (i.e., $R_{WC21}$: -0.071; $R_{WC23}$: -0.076; $R_{WC24}$: -0.040), suggesting that bryophyte temperatures below the dewpoint temperature of the ambient air caused an increase in water content and thus a condensation of water.").

The new analysis is described in the methods section (P9L12-17:" To analyze if condensation might influence the WC in the bryophytes, a correlation between events with negative $\Delta T_d$ and the change of the WC in the 4h before these events were calculated. The potential effect of fog was analyzed by calculating

the average change in water content (ΔWC) from the beginning of the fog event until 1h later. The dew point and fog calculations and correlations were performed with R version 3.6.1 (2019-07-05)…”).

The discussion section was adapted accordingly (P17L3-10: “In addition, also condensation and fog need to be considered as potential additional sources of water for epiphytic covers as well as for near-stem vegetation at the forest floor (Lakatos et al., 2012; van Stan and Gordon, 2018; León-Vargas et al., 2006). Our data show that the necessary conditions for condensation were regularly met and occurred most frequently during the wet season at 23 m height (Fig. S11). Cases with a negative dewpoint spread, when condensation could occur, were related to increasing water contents of the bryophytes, supporting the calculated condensation data. During the occurrence of fog, an increase in water contents could not be directly proven, suggesting that fog does not represent a major water source for the bryophytes.”).

We also looked in the daily fluctuations in some more detail and found that they indeed occur in all seasons but are just more obvious during the dry season, as the daily patterns are more regular, due to the rare rain events. We treated this topic in the previous comment.

I think Figure S5 is useful, but I recommend making the graphs larger. In the supplement I do not think that there is the need to save space, so I would go for clarity here. I also like the magnification shown in S5c, though judging from the rain amounts it does not really correspond to the box in the coarse-scale graph, which I would recommend stretching vertically to make clear that the entire graph (all parameters) are shown in the magnification.

Many thanks for your suggestion; Figure S5 was magnified to show all the parameters with clarity, and the box shown in S5c was changed accordingly.

The way in which stemflow is discussed could be improved. In my view, this is not ´additional´ water in the sense that moisture input from condensation may be (so I would not merge these two points into one sentence like you now do, P16 L 30), but it is redistributed rain water (mostly, as condensation and fog would hardly lead to stemflow) that benefits epiphytes on the tree stem (i.e. in your case those at 1.5 m in particular). As stemflow concentrates rainfall from a larger area to the stem, this could explain why the bryophytes on the stem respond so reliably to rainfall. For a recent review about this see Mendieta-Leiva, G., P. Porada and M.Y. Bader, 2020. Interactions of epiphytes with precipitation partitioning. In: Precipitation partitioning by vegetation. A global synthesis, J. T. Van Stan IIE. Gutmann and J. Friesen, (Eds.). Springer Nature Switzerland: pp: 133-145.

The text was altered following your suggestion: (P16 L24-32: “In this rain forest environment, epiphytes growing in different parts of the tree and along the stem can benefit from different sources of water. The gross precipitation, as the main water source, can be converted into throughfall, stemflow, water storage and water vapor (Mendieta-Leiva et al., 2020). Thus, rainwater can influence the bryophytes in variable ways depending on its redistribution and the microenvironmental conditions: at the canopy level, direct interception of precipitation can be used for water storage, whereas in the understory, stemflow is more pronounced and contributes to the water supply of the bryophytes. It has been estimated that in tropical forests up to 4 % of the annual rainfall amount could be converted into stemflow (Lloyd and Marques F, 1988; Marin et al., 2000; van Stan and Gordon, 2018), corresponding to maximum values of 68 and 75 mm in the years 2015 and 2016, respectively, at the ATTO site.”).

And I think I found one important mistake in your calculation of activity times, or possibly two (sorry I did not notice this in the previous round):

1. You assume that "For the net photosynthesis (NP) it is required that … T > TCP". However, the TCP is an upper limit, so for a positive NP you need T to be below this compensation point!

This is correct, the calculation was done considering T < TCP, it was only a typo and it is now correct: (P31 L6-7) "For the net photosynthesis (NP) it is required that WC > WCP, PAR > $LCP_l$ and T < TCP…"

2. The range for Topt can be considered a range within (not above or below!) which NP is optimal. Therefore I would expect only one value for the % of time where the bryophytes at a particular height in the tree are within this range, not a range of %. This is in contrast to the other ranges which are ranges of estimates for which we do not know which applies so that indeed it is useful to report on the estimated % of time if we assume one or the other value.

Thank you for this valuable comment. We adopted the data accordingly.

Small typos / grammar problems (not exhaustively reviewed) and needs for short clarifications:

P3 L31 …communities were conducted …

Following your suggestion, this alteration was done (P3 L26).

P4 L25: add the tree species and some information about tis phenology (it seems from the discussion that it is a semideciduous tree..?). This is quite relevant for extrapolating the results.

The required information was added to the text: (P4 L19-23) "The parameters temperature and light within/on top of the bryophyte communities and their WC were measured with a microclimate station installed along one evergreen tree of the species *Buchenavia parvifolia* Ducke (Combretaceae) in September 2014 (Fig. S1). The family regularly occurs in the Amazon rainforest, and represents a common genus in tropical America, growing on clayey soil in plateau environments. It presents flowers during the dry and fruits during the wet season (Stace, 2007)."

P5 L12: 26-m-high

The text was modified accordingly (P5 L26).

P5 29: data that could be used…

The text was changed accordingly (P6 L8).

P5 L33: The electrical conductivity (EC) values on which the WC calculations were based (see Paragraph 2.3 below) showed some unexplained oscillation, causing …

The text was changed accordingly: (P6 L9-10) "The electrical conductivity (EC) values, on which the WC calculations were based (see Paragraph 2.3 below) showed some unexplained oscillation, causing an inaccuracy…"

P6 L9: please explain this ´fluctuated´

The values obtained from the light sensors varied in a range of 10 µmol m$^{-2}$ s$^{-1}$ photosynthetic photon flux density (PPFD), and all the 5-min data were averaged to present the same period for all the parameters (30-min). The sentence was rewritten to make this clearer: (P6 L19-24: "The average daily PAR values were calculated from the data collected during daytime, i.e., 6:00 to 18:00, while PAR$_{max}$ represents the daily maximum value. The 5-minute readings obtained from the light sensors fluctuated by approximately ± 10 µmol m$^{-2}$ s$^{-1}$ photosynthetic photon flux density (PPFD).  To smoothen the microclimate data (i.e. the PPFD values obtained from light sensors, the temperature values measured within the bryophytes and water content values obtained from electrical conductivity sensors), 30-minute averages were calculated and used for all further calculations.")

P7 L17: I would add here that 0% was an approximation (simplification), as no moss would really reach 0% unless in a good drying oven… But they would dry to a minimum WC of a few %, so near enough to 0% for this model assumption.

The text was modified according to the suggestion: (P7 L26-32: "The minimum electrical conductivity achieved in the field was used as an approximation (simplification) of a water content of 0%. We are aware of the fact that bryophytes do not reach a water content of 0% under field conditions, but they can be expected to dry to a water content of a few %, which we assume as close enough for this model assumption in a tropical rainforest environment.")

P8 L24: Please start this point by explaining why you calculated this (to estimate the likelihood of water input by dew?). It also would not hurt to introduce this idea in the introduction. I think it is a very important and exciting point!

A short explanation was inserted in the text: (P8 L29-31: "In order to assess the potential water input by condensation, we calculated the dew point temperature, at which saturated air humidity levels are reached. ").

The idea was also added in the Introduction: (P2 L31 – P3 L4: "In a dry state, many of them can outlast extreme weather conditions, being reactivated by water (Oliver et al., 2005; Proctor, 2000; Proctor et al., 2007; Seel et al., 1992). This water can be supplied by precipitation, either directly intercepted or taken up from stemflow. For several species, also condensation of fog and dew can serve as a source of water (Lancaster et al., 1984; Lange et al., 2006; Lange and Kilian, 1985; Reiter et al., 2008).").

P9 L7: Fig S5 does not show PAR values…

This is correct, the Fig S5 does not show PAR values, but the text (P11 L7) refers to Table S5 where the values for monthly mean values and standard deviations of PAR (average and maximum) are shown.

P13 L8: remove in before 90%

The word was replaced by "during" (P13 L23)

P13 L10: relatively similar or similarly high

The sentence was corrected (P13 L24-25): "maximum light intensities were similarly high …."

P16 L7: those at 23 m

The sentence was modified accordingly (P16 L8)

P16 L23 remove the second ´during the dry season´

The phrase was removed accordingly (P16 L18)

P17 L8: we observed (instead of ´observed by us´)

The sentence was removed over the course of the editing process (as suggested by you).

P17 L14-16: stating that Leucobryum may have a low water retention seems strange, as this genus has special water-retaining cells… I would think that it is not particularly good at capturing water (e.g. from fog), but then very good at retaining it.

This is correct, the sentence has been removed over the course of the reviewing process (as suggested by you).

P17: I would consider removing L5-32, as it does not really add much to your story. The life-from discussion seems too much discussion for the amount of data you have per life form and does not sound like you have profoundly researched it yet. Similarly, with the number of samples you studied it seems unjustified to draw any conclusions about the pattern of distributions among height zones. You could mention this in the methods section where you present the species studied, to justify your choice by showing that you chose typical species for the ecosystem.

This is correct, the life form was not profoundly researched, and the life-form description was moved to the Methods section: (P5 L3-24: "The WC sensors were placed in four different bryophyte communities being heterogeneously distributed along three height levels. At 1.5 m height, the WC sensors were installed in communities dominated by *Sematophyllum subsimplex* (5 sensors) and *Leucobryum*

*martianum* (1 sensor), at 8 m in *Octoblepharum cocuiense* (2 sensors) and *Symbiezidium barbiflorum* (1 sensor), and at 23 m in *Symbiezidium barbiflorum* (3 sensors; Fig. S2, Fig. S3). The communities used for a placement of the sensors reflect the distribution of bryophytes among height zones in the Amazon rainforest (Cornelissen and ter Steege, 1989; Mota de Oliveira, 2010; Mota de Oliveira and ter Steege, 2015; Pantoja et al., 2015). Studies describe that Lejeuneaceae (common liverwort family of the Amazon region comprising the genus *Symbiezidium*) are more diverse and abundant in the canopy area, while mosses are mainly concentrated at the tree base and trunk in a plateau ecosystem (Campos et al. 2019; Mota de Oliveira 2010, 2018). The species identified by us (Table S1) have also been reported as being frequent at other tropical rain forest sites (Campos et al., 2015; Dislich et al., 2018; Gradstein and Salazar Allen, 1992; Mota de Oliveira et al., 2009; Pinheiro da Costa, 1999). They show different water holding capacities, which are influenced by their life form (Lakatos et al., 2006; Romero et al., 2006; Williams and Flanagan, 1996; Proctor, 1990). The liverwort *Symbiezidium barbiflorum* (Lejeuneaceae) has been described to have the life-form of mats (Batista and Santos, 2016; Mägdefrau, 1982; Valente et al., 2017), which are characterized by a high capillarity retention of water, supporting the storage of condensed water. Mats also have an increased drought-tolerance, being more adapted to dry conditions as well as to extreme changes (Gimingham and Birse, 1957). *Sematophyllum subsimplex* (Sematophyllaceae) and *Leucobryum martianum* (Dicranaceae) belong to the life-forms of wefts and turfs, respectively (Mägdefrau, 1982, Batista and Santos, 2016; Valente et al., 2017). Turfs show high capillary water conduction and are well known for special water-retaining cells, whereas wefts show high values of capillary water conduction but lower values of water retention (Mägdefrau, 1982), being characteristic for humid areas (Gimingham and Birse, 1957).")

P22 L 1: I would not say that the WC has "turned out to be" the key parameter controlling the overall physiological activity of the organisms, because you did not measure this activity. Better use a more cautious formulation like "appears to be"

The text was modified accordingly (P19L8)

P22 L3 remove the comma after major rain events

The comma was removed (P19 L10)

P22 L8: here also remove the speculative remark about thallus morphology. This is, by the way, not a seasonal feature anyway…

As suggested, the remark regarding the thallus morphology has been removed.

P22 L13-14: do you really know whether this nightly condensation activates physiological processes? Is it enough for photosynthesis, for example? I do not think that we know this, and I do think it would be very important to know it, so perhaps formulate it as a research need rather than a finding.

The sentence was rewritten for clarity (P 19 L18-22: "Thus, our data suggest that the relevant water source for bryophytes in the understory is rain, while for the bryophytes in the canopy RH fluctuations and dew

condensation might be relevant. With the current data at hand, however, it cannot be answered if the daily fluctuations and the dew condensation events are large enough to activate physiological processes; this topic, indeed, would deserve to be investigated in a separate in-depth study.").

Table 1b: please help the reader by providing the number of days for which the fog sensor was operational (in addition to the sensor outage times)

The requested information was included in Table 1 and Table S3.

Table 3: please make clearer what data were used: the averages of the different samples per height zone? Or the individual samples (i.e. is the range shown a function only of the range of possible parameter values for the cardinal points, or also a function of the differences between samples?).

The data show the averages of the different samples per height zone. This is now also written in the legend (P31 Table 3: "The data show the averages of the different samples per height zone.").

Fig 4 caption: I do not think that the ´Estimated´ is appropriate here.

The word was removed from the caption (P39 L2): "Frequency of mean photosynthetically active radiation…"

Figure S8 caption: Exemplary

The spelling was corrected: "Exemplary daily ….".

[revised manuscript text omitted]

**Sensors at 23 m height level (inner canopy)**

[Figure]

**Sensors at 8 m height level (stem)**

**Sensors at 1.5 m height level (understory)**

[Figure]

[Figure]

[Figure]

[Figure]

**Figure S5.** Long-term measurements of preciepitation, electrical conductivity, and the calculated water content. All the sensors utilized for further calculations are shown: (a, b, c, d) at 23 m height, (d, e, f, g) at the 8 m height, and (g, h, i, j, k, l, m) at 1.5 m height. Gaps in the dataset correspond to maintenance periods.

[Figure]

**Figure S6:** Temperature within bryophytes compared to the above-canopy temperature. The temperature within bryophytes was measured at 1.5 m, 8 m, and 23 m, while the above-canopy temperature was measured at 26 m height on the tower. The data are presented per height zone and also pooled together in the lowest panel. Data present 30-minute averages with linear fits, of the function y = a + bx, with the coefficients (± 1 std. dev.) and the R² are given in the figure for each height level.

[Figure]

a) Wet season (El Niño)

Time [day/month/year]

[Figure]

**Figure S7:** Representative periods during wet and dry season under the influence of El Niño, showing light conditions (PAR), temperature, and water content (WC) experienced by bryophytes, and above-canopy meteorological conditions in the Amazonian rain forest. Shown are 8-day periods during a) the wet season 2016 and b) the dry season 2015. The micrometeorological parameters on top/within epiphytic cryptogamic communities represent (A) the photosynthetically active radiation (PAR) on top, (B) the temperature within, and (C) the water content of cryptogamic communities. The above-canopy meteorological parameters comprise (A) the above-canopy photosynthetically active radiation (PAR at 75 m), (B) the above-canopy temperature (at 26 m), (D) the relative air humidity (RH at 26 m), the presence of fog events, and (E) the rain amount. The data show 30-minute averages ± SD except for rain, which shows hourly sums. Data of replicate sensors installed within communities at the same height level were pooled, while above-canopy parameters were measured with one sensor each. The nighttime is shaded in grey color (06:00 – 18:00 LT).

[Figure]

**Figure S8:** Correlation between the water content (WC) and the rain amount in the preceding 24 h. Linear trendline with formula added to illustrate the relationship between both parameters.

[Figure]

**Figure S98:** Exemplary daily (micro-)climatic conditions at the canopy level, showing the WC values of the 3 sensors at 23m [%] (A), the dew point spread at 23 m [°C] (B), and the environmental factors relative humidity RH[%], temperature T [°C] and direct normal irradiance DNI [W m$^{-2}$] measured at 26 m (C).

[Figure]

**Figure S109**: Diel dew point spread at 1.5 m and 23 m height levels in a 24h cycle, illustrating the difference between the temperature of the substrate (T$_s$) and the dew point of the surrounding air (T$_d$). If the suface temperature is lower than the dew point of the surrounding air (values below red line), condensation might occur.

[Figure]

**Figure S11:** Characteristic sequence of (micro)climatic conditions during the dry and the wet season at (A) 23 m and (B) 1.5 m height. In the plots the relative air humidity [%], ambient and bryophyte temperature [°C], water content of the bryophytes [% dry weight], and dew point spread [°C] are shown.

[Figure]

[Figure]

**Figure S12:** Exemplary (micro)meteorological data comprising dewpoint events at (A) 23 m and (B) 1.5 m height. In the plots, the dew point spread [°C], precipitation [mm * h-1], relative humidity [%], and the water content [% dry weight] are shown.

[Figure]

**Figure S13̶0̶:** Temperature conditions of bryophytes related to their water content. The temperature was measured in bryophytes at different height levels along the tree. Data presented as 30-minute averages.

**Table S1:** Height of installation, minimum and maximum values of the individual sensors of the microclimate station measuring water content, temperature, and light. For the water content sensors, also the bryophyte species are given. Based on 30-minute averages.

| Sensor No | Height [m] | Water content [% DW] | | Bryophyte species | Sensor No | Height [m] | Temperature [°C] | |
|---|---|---|---|---|---|---|---|---|
| | | min | max | | | | min | max |
| Sensor 01 | 1.5 | 0 | 763 | *Sematophyllum subsimplex* | Sensor 01 | 1.5 | 21.1 | 36.3 |
| Sensor 02 | 1.5 | 0 | 763 | *Sematophyllum subsimplex* | Sensor 02 | 1.5 | 21.4 | 39.4 |
| Sensor 03 | 1.5 | 0 | 763 | *Sematophyllum subsimplex* | Sensor 03 | 8 | 21.6 | 34.7 |
| Sensor 04 | 1.5 | 0 | 1373 | *Leucobryum martianum* | Sensor 04 | 8 | 20.9 | 46.3 |
| Sensor 05 | 1.5 | 0 | 763 | *Sematophyllum subsimplex* | Sensor 07 | 23 | 20.8 | 41.2 |
| Sensor 06 | 1.5 | 0 | 763 | *Sematophyllum subsimplex* | Sensor 08 | 23 | 20.3 | 48.7 |
| Sensor 09 | 8 | 0 | 1318 | *Octoblepharum cocuiense* | | **Height [m]** | **PAR [μmol m$^{-2}$ s$^{-1}$]** | |
| Sensor 10 | 8 | 0 | 1318 | *Octoblepharum cocuiense* | | | | |
| Sensor 11 | 8 | 0 | 1658 | *Symbiezidium barbiflorum* | | | min | max |
| Sensor 21 | 23 | 0 | 1658 | *Symbiezidium barbiflorum* | Sensor 01 | 1.5 | 0 | 634 |
| Sensor 23 | 23 | 0 | 1658 | *Symbiezidium barbiflorum* | Sensor 02 | 8 | 0 | 569 |
| Sensor 24 | 23 | 0 | 1658 | *Symbiezidium barbiflorum* | Sensor 03 | 8 | 0 | 1121 |
| | | | | | Sensor 06 | 23 | 0 | 654 |
| | | | | | Sensor 07 | 23 | 0 | 767 |

**Table S2**: Water content range measured during the calibration in the laboratory for the different replicates of the four bryophyte species. Listed are the minimum and maximum water content values (WC) measured at full water saturation ($WC_{max}$) and in the end of drying when weight stability was reached over more than 5 minutes ($WC_{min}$). Data shown for each replicate (1–4) and the species average (all).

| Species | Replicate sample | $WC_{min}$ | $WC_{max}$ |
|---|---|---|---|
| *Leucobryum martianum* | 1 | 32 | 1487 |
| *Leucobryum martianum* | 2 | 10 | 931 |
| *Leucobryum martianum* | 3 | 10 | 1241 |
| *Leucobryum martianum* | 4 | 7 | 1834 |
| *Sematophyllum subsimplex* | 1 | 14 | 614 |
| *Sematophyllum subsimplex* | 2 | 14 | 698 |
| *Sematophyllum subsimplex* | 3 | 14 | 468 |
| *Sematophyllum subsimplex* | 4 | 14 | 459 |
| *Sematophyllum subsimplex* | 5 | 7 | 1576 |
| *Symbiezidium barbiflorum* | 1 | 15 | 1657 |
| *Symbiezidium barbiflorum* | 2 | 15 | 1982 |
| *Symbiezidium barbiflorum* | 3 | 15 | 1581 |
| *Symbiezidium barbiflorum* | 4 | 22 | 1412 |
| *Octoblepharum cocuiense* | 1 | 23 | 742 |
| *Octoblepharum cocuiense* | 2 | 16 | 870 |
| *Octoblepharum cocuiense* | 3 | 6 | 2342 |
| *Leucobryum martianum* | all | 15 | 1373 |
| *Sematophyllum subsimplex* | all | 13 | 763 |
| *Symbiezidium barbiflorum* | all | 16 | 1658 |
| *Octoblepharum cocuiense* | all | 15 | 1318 |

**Table S3:** Monthly mean values and standard deviations (± SD) of photosynthetically active radiation (PAR$_{avg}$ daytime, measured at 75 m), daily maxima of photosynthetically active radiation (PAR$_{max}$), temperature (measured at 26 m), and relative humidity (RH, measured at 26 m). Rainfall is presented as the monthly amounts and the percentage of days with rain (measured at 81 m), and also the percentage of days when rain detection malfunctioned are listed. Fog events are given as the percentage of days. Dry season data are shaded in red, wet season data in blue and transitional periods are unshaded. Due to data gaps in the measured rain data (shown in brackets) values for 21 days of rain were also extrapolated from existing data as described in methods section (values behind data in brackets). Values were calculated from 30-minute intervals. Fog has not being recorded in the time ranges of 31.05. – 20.10.2015, 30.04. – 06.07.2016, 01.09. – 31.12.2016 due to malfunction of the device, thus the last column provides data about the number of operational days per month.

| Month | PAR$_{avg}$ daytime [µmol m$^{-2}$ s$^{-1}$] | | PAR$_{max}$ [µmol m$^{-2}$ s$^{-1}$] | | Temperature [°C] | | RH [%] | | Rain [mm month$^{-1}$] | Rain [% days] | Defect on rain detection [% days] | Fog [% days] | Days with fog data |
|---|---|---|---|---|---|---|---|---|---|---|---|---|---|
| | Mean | ± SD | Mean | ± SD | Mean | ± SD | Mean | ± SD | | | | | |
| Oct 2014 | 857 | 668 | 2201 | 509 | 26.0 | 2.8 | 90 | 11 | 212 | 58 | 0 | 55 | 30 |
| Nov 2014 | 832 | 624 | 2082 | 423 | 25.6 | 2.9 | 92 | 11 | 70 | 57 | 0 | 53 | 30 |
| Dec 2014 | 843 | 582 | 2140 | 346 | 26.3 | 2.7 | 90 | 11 | 123 | 42 | 0 | 42 | 30 |
| Jan 2015 | 637 | 525 | 1747 | 735 | 24.5 | 2.4 | 95 | 8 | 259 | 71 | 0 | 71 | 29 |
| Feb 2015 | 774 | 589 | 2058 | 600 | 25.4 | 2.6 | 92 | 10 | 140 | 64 | 0 | 46 | 28 |
| Mar 2015 | 680 | 534 | 2038 | 575 | 24.7 | 2.1 | 96 | 7 | 331 | 87 | 0 | 77 | 31 |
| Apr 2015 | 766 | 564 | 2155 | 463 | 25.3 | 2.5 | 93 | 10 | 189 | 80 | 0 | 40 | 29 |
| May 2015 | 725 | 559 | 2103 | 425 | 27.2 | n.a. | 93 | 6 | 320 | 90 | 0 | 58 | 30 |
| Jun 2015 | 804 | 562 | 2237 | 128 | 25.0 | 2.3 | 94 | 8 | 178 | 80 | 0 | 0* | 0 |
| Jul 2015 | 892 | 605 | 2238 | 188 | 25.7 | 3.0 | 91 | 11 | 74 | 65 | 0 | 0* | 0 |
| Aug 2015 | 1017 | 636 | 1722 | 957 | 27.1 | 3.3 | 83 | 13 | (23) 32* | 23 | 23 | 0* | 0 |
| Sep 2015 | 1148 | 687 | 2242 | 467 | 28.7 | 3.7 | 74 | 15 | 38 | 13 | 20 | 0* | 0 |
| Oct 2015 | 968 | 635 | 2072 | 514 | 28.4 | 3.6 | 78 | 16 | 55 | 35 | 3 | 13* | 11 |
| Nov 2015 | 887 | 624 | 1859 | 769 | 27.9 | 3.5 | 81 | 16 | (33) 37* | 30 | 17 | 23 | 28 |
| Dec 2015 | 862 | 575 | 2074 | 304 | 28.1 | 3.0 | 78 | 14 | 38 | 26 | 3 | 6 | 31 |
| Jan 2016 | 882 | 606 | 2175 | 270 | 28.2 | 3.4 | 78 | 16 | 52 | 48 | 0 | 13 | 31 |
| Feb 2016 | 743 | 550 | 1928 | 679 | 25.9 | 2.6 | 93 | 10 | (267) 341* | 79 | 52 | 48 | 29 |
| Mar 2016 | 692 | 545 | 2041 | 545 | 25.6 | 2.1 | 96 | 7 | 304 | 90 | 0 | 77 | 31 |
| Apr 2016 | 709 | 564 | 2088 | 443 | 25.6 | 2.3 | 96 | 7 | 277 | 87 | 0 | 73 | 28 |
| May 2016 | 817 | 603 | 2230 | 405 | 26.1 | 2.6 | 94 | 8 | 236 | 90 | 0 | 0* | 0 |
| Jun 2016 | 828 | 584 | 2178 | 261 | 25.6 | 2.8 | 92 | 10 | 105 | 57 | 0 | 0* | 0 |
| Jul 2016 | 917 | 629 | 2253 | 118 | 26.2 | 3.2 | 88 | 12 | 92 | 58 | 0 | 26* | 26 |
| Aug 2016 | 1016 | 648 | 2146 | 593 | 27.1 | 3.5 | 83 | 14 | 40 | 32 | 3 | 16 | 31 |
| Sep 2016 | 947 | 662 | 2230 | 543 | 26.5 | 3.1 | 89 | 12 | (77) 96* | 50 | 17 | 0* | 0 |
| Oct 2016 | 915 | 641 | 2323 | 192 | 27.1 | 3.3 | 86 | 14 | (1) 9* | 23 | 23 | 0* | 0 |
| Nov 2016 | 911 | 610 | 2227 | 217 | 27.1 | 3.3 | 87 | 13 | (30) 89* | 20 | 13 | 0* | 0 |
| Dec 2016 | 694 | 553 | 1955 | 503 | 25.4 | 2.7 | 94 | 10 | 223 | 71 | 0 | 0* | 0 |

(*) Gaps in the data record due to malfunction of the device.

**Table S4:** Parameters determining the time range of photosynthesis and respiration. The ranges of values defining the lower water compensation point (WCP), the lower light compensation point (LCP$_l$), the temperature for optimal net photosynthesis (T$_{opt}$), and the upper temperature compensation point (TCP) as relevant parameters have been extracted from published studies conducted at various study sites in the tropical rain forest.

| Parameter | Range of values | Reference | Study site |
|---|---|---|---|
| WCP | 30–80 % DW | Wagner et al 2013 | Panama, lowland rain forest, 0 m |
| LCP$_l$ | 3–12 µmol m$^{-2}$ s$^{-1}$ | Lösch et al. 1994 | Zaire, lowland rain forest, 800 m |
| T$_{opt}$ | 24–27 °C | Wagner et al 2013 | Panama, lowland rain forest, 0 m |
| TCP | 30–36 °C | Wagner et al 2013 | Panama, lowland rain forest, 0 m |

**Table S5:** Monthly mean values and standard deviations (± SD) of the photosynthetically active radiation (PAR$_{avg}$ daytime), the daily maxima of photosynthetically active radiation (PAR$_{max}$), temperature, and water content of bryophytes at four height levels. Dry season data are shaded in red, wet season data in blue and transitional periods are unshaded. Values were calculated from 30-minute intervals. N.a.: data not available.

| Month | PAR$_{avg}$ daytime[µmol m$^{-2}$ s$^{-1}$] | | | | | | PAR$_{max}$ [µmol m$^{-2}$ s$^{-1}$] | | | | | |
| | 1.5 m | | 8 m | | 23 m | | 1.5 m | | 8 m | | 23 m | |
| | Mean | ± SD | Mean | ± SD | Mean | ± SD | Mean | ± SD | Mean | ± SD | Mean | ± SD |
|---|---|---|---|---|---|---|---|---|---|---|---|---|
| Oct 2014 | 4 | 8 | 30 | 31 | 88 | 90 | 75 | 105 | 285 | 231 | 624 | 286 |
| Nov 2014 | 4 | 11 | 23 | 32 | 24 | 37 | 142 | 131 | 396 | 321 | 378 | 275 |
| Dec 2014 | 6 | 18 | 31 | 50 | 25 | 33 | 236 | 172 | 435 | 228 | 346 | 235 |
| Jan 2015 | 3 | 8 | 22 | 28 | 20 | 27 | 155 | 96 | 341 | 219 | 341 | 246 |
| Feb 2015 | 2 | 3 | 31 | 21 | 16 | 17 | 46 | 33 | 173 | 183 | 234 | 244 |
| Mar 2015 | 3 | 4 | 43 | 35 | 16 | 15 | 45 | 55 | 292 | 159 | 128 | 117 |
| Apr 2015 | 6 | 20 | 80 | 105 | 16 | 18 | 346 | 310 | 480 | 231 | 241 | 231 |
| May 2015 | 6 | 32 | 66 | 71 | 16 | 17 | 634 | 428 | 282 | 236 | 146 | 137 |
| Jun 2015 | 2 | 3 | 73 | 64 | 18 | 20 | 42 | 51 | 214 | 125 | 177 | 141 |
| Jul 2015 | 3 | 12 | 54 | 73 | 15 | 18 | 168 | 178 | 727 | 301 | 152 | 144 |
| Aug 2015 | 13 | 56 | 66 | 115 | 24 | 23 | 601 | 414 | 746 | 193 | 227 | 170 |
| Sep 2015 | 9 | 21 | 28 | 47 | 65 | 66 | 248 | 204 | 403 | 224 | 492 | 229 |
| Oct 2015 | 3 | 4 | 15 | 15 | 44 | 30 | 53 | 47 | 128 | 99 | 221 | 157 |
| Nov 2015 | 4 | 7 | 16 | 25 | 61 | 64 | 82 | 95 | 315 | 151 | 475 | 208 |
| Dec 2015 | 5 | 11 | 22 | 35 | 88 | 103 | 112 | 116 | 308 | 171 | 645 | 250 |
| Jan 2016 | 4 | 7 | 16 | 21 | 88 | 103 | 72 | 91 | 177 | 143 | 692 | 294 |
| Feb 2016 | 3 | 4 | 13 | 11 | 57 | 46 | 46 | 54 | 79 | 76 | 388 | 237 |
| Mar 2016 | 3 | 7 | 28 | 15 | 37 | 33 | 102 | 125 | 107 | 80 | 268 | 215 |
| Apr 2016 | 5 | 15 | 27 | 19 | 38 | 31 | 192 | 199 | 59 | 27 | 270 | 203 |
| May 2016 | 3 | 7 | n.a. | n.a. | 45 | 41 | 114 | 109 | n.a. | n.a. | 286 | 209 |
| Jun 2016 | 2 | 2 | n.a. | n.a. | 58 | 68 | 25 | 34 | n.a. | n.a. | 416 | 199 |
| Jul 2016 | 2 | 4 | n.a. | n.a. | 72 | 86 | 30 | 44 | n.a. | n.a. | 527 | 204 |
| Aug 2016 | 9 | 34 | 31 | 52 | 71 | 94 | 319 | 216 | 340 | 241 | 614 | 256 |
| Sep 2016 | 3 | 7 | 13 | 24 | 55 | 69 | 102 | 84 | 250 | 137 | 508 | 244 |
| Oct 2016 | 2 | 3 | 7 | 9 | 47 | 54 | 35 | 28 | 106 | 71 | 421 | 219 |
| Nov 2016 | 3 | 5 | 9 | 13 | 73 | 85 | 59 | 51 | 172 | 114 | 606 | 251 |
| Dec 2016 | 4 | 12 | 24 | 38 | 52 | 56 | 156 | 131 | 361 | 282 | 457 | 274 |

**Continuation of Table S5**

| Month | Temperature [°C] | | | | | | Water content [% DW] | | | | | |
|---|---|---|---|---|---|---|---|---|---|---|---|---|
| | 1.5 m | | 8 m | | 23 m | | 1.5 m Moss | | 8 m M&L | | 23 m Liverwort | |
| | Mean | ± SD | Mean | ± SD | Mean | ± SD | Mean | ± SD | Mean | ± SD | Mean | ± SD |
| Oct 14 | 25.0 | 1.3 | 25.2 | 1.6 | 26.3 | 2.9 | 115 | 107 | 110 | 111 | 42 | 20 |
| Nov 14 | 25.3 | 1.2 | 25.7 | 1.4 | 26.2 | 2.3 | 38 | 30 | 53 | 16 | 42 | 20 |
| Dec 14 | 25.4 | 1.1 | 25.8 | 1.3 | 26.6 | 2.1 | 49 | 48 | 56 | 20 | 35 | 7 |
| Jan 15 | 24.2 | 1.1 | 24.3 | 1.3 | 24.6 | 1.8 | 129 | 113 | 76 | 32 | 39 | 12 |
| Feb 15 | 24.5 | 1.0 | 24.5 | 1.1 | 25.0 | 1.8 | 87 | 67 | 69 | 33 | 38 | 8 |
| Mar 15 | 24.4 | 0.9 | 24.3 | 0.9 | 24.5 | 1.3 | 106 | 65 | 102 | 64 | 38 | 6 |
| Apr 15 | 24.6 | 0.9 | 24.7 | 1.1 | 24.9 | 1.8 | 79 | 65 | 73 | 31 | 39 | 8 |
| May 15 | 24.6 | 0.9 | 24.5 | 0.9 | 24.8 | 1.7 | 130 | 90 | 103 | 58 | 39 | 8 |
| Jun 15 | 24.5 | 0.9 | 24.5 | 1.0 | 25.0 | 1.9 | | | | | | |
| Jul 15 | 24.5 | 1.1 | 25.0 | 1.5 | 25.5 | 2.5 | 37 | 24 | 103 | 45 | 49 | 23 |
| Aug 15 | 25.4 | 1.2 | 26.3 | 2.0 | 27.0 | 2.8 | 20 | 10 | 67 | 17 | 42 | 21 |
| Sep 15 | 27.0 | 1.7 | 27.8 | 2.2 | 29.0 | 3.4 | 13 | 17 | 60 | 15 | 36 | 18 |
| Oct 15 | 27.2 | 1.8 | 28.0 | 2.2 | 29.4 | 3.2 | 13 | 16 | 36 | 21 | 43 | 52 |
| Nov 15 | 27.2 | 1.9 | 27.6 | 2.3 | 29.2 | 3.6 | 16 | 14 | 51 | 30 | 37 | 32 |
| Dec 15 | 27.3 | 1.6 | 27.9 | 2.0 | 29.4 | 3.4 | 15 | 11 | 48 | 24 | 35 | 18 |
| Jan 16 | 27.4 | 1.8 | 28.0 | 2.2 | 29.4 | 3.8 | 16 | 14 | 51 | 31 | 37 | 16 |
| Feb 16 | 25.2 | 1.0 | 25.4 | 1.2 | 26.2 | 2.5 | 80 | 93 | 99 | 80 | 43 | 16 |
| Mar 16 | 25.2 | 0.9 | 25.1 | 0.9 | 25.6 | 1.8 | 74 | 68 | 91 | 49 | 41 | 13 |
| Apr 16 | 25.2 | 1.0 | 25.2 | 1.1 | 25.7 | 2.0 | 63 | 45 | 131 | 85 | 43 | 14 |
| May 16 | 25.3 | 1.0 | 25.3 | 1.2 | 26.1 | 2.3 | 42 | 33 | 75 | 39 | 44 | 16 |
| Jun 16 | 24.6 | 1.1 | 24.6 | 1.3 | 25.8 | 2.8 | 31 | 18 | 61 | 30 | 45 | 13 |
| Jul 16 | 24.8 | 1.2 | 25.3 | 1.7 | 26.7 | 3.4 | 24 | 22 | 52 | 21 | 53 | 36 |
| Aug 16 | 25.7 | 1.8 | 26.3 | 2.4 | 28.0 | 4.1 | 22 | 28 | 59 | 99 | 59 | 130 |
| Sep 16 | 25.5 | 1.3 | 25.9 | 1.7 | 27.1 | 3.3 | 28 | 40 | 52 | 39 | 67 | 111 |
| Oct 16 | 26.2 | 1.6 | 26.8 | 1.9 | 28.0 | 3.4 | 17 | 9 | 45 | 18 | 43 | 38 |
| Nov 16 | 25.9 | 1.7 | 26.5 | 2.1 | 28.0 | 3.4 | 18 | 20 | 49 | 49 | 44 | 37 |
| Dec 16 | 25.4 | 1.3 | 25.0 | 1.7 | 25.6 | 2.5 | | | | | | |

**Table S6:** Mean values and standard deviations (± SD) of the daily maxima of photosynthetically active radiation ($PAR_{max}$) for each height level shown for 2015 and 2016, considering that 2015 was an El Niño year (additional information to Table 1).

| Height | PAR$_{max}$ [µmol m$^{-2}$ s$^{-1}$] | | | |
| --- | --- | --- | --- | --- |
| | **2015** | | **2016** | |
| | **Mean** | **± SD** | **Mean** | **± SD** |
| above canopy | 1766 | 415 | 1842 | 364 |
| 23 m | 125 | 123 | 226 | 140 |
| 8 m | 186 | 195 | 68 | 90 |
| 1.5 m | 49 | 89 | 29 | 45 |

**Table S7:** Mean values and standard deviations (± SD) of the daily maxima of photosynthetically active radiation (PAR$_{max}$) for each height level shown for the different seasons (additional information to Table 2).

| Height | PAR$_{max}$ [$\mu$mol m$^{-2}$ s$^{-1}$] | |
|---|---|---|
| [m] | Mean | ± SD |
| **Wet season** | | |
| above-canopy | 1687 | 431 |
| 23 m | 245 | 82 |
| 8 m | 210 | 151 |
| 1.5 m | 191 | 206 |
| **Transitional season Wet-Dry** | | |
| above-canopy | 1855 | 233 |
| 23 m | 318 | 183 |
| 8 m | 471 | 363 |
| 1.5 m | 66 | 68 |
| **Dry season** | | |
| above-canopy | 1924 | 370 |
| 23 m | 457 | 147 |
| 8 m | 314 | 184 |
| 1.5 m | 172 | 177 |
| **Transitional season Dry-Wet** | | |
| above-canopy | 1691 | 407 |
| 23 m | 496 | 165 |
| 8 m | 324 | 95 |
| 1.5 m | 146 | 61 |